# Learning the Linear Quadratic Regulator from Nonlinear Observations

**Zakaria Mhammedi**
ANU and Data61
zak.mhammedi@anu.edu.au

**Dylan J. Foster**
MIT
dylanf@mit.edu

**Max Simchowitz**
UC Berkeley
msimchow@berkeley.edu

**Dipendra Misra**
Microsoft Research NYC
dimisra@microsoft.com

**Wen Sun**
Microsoft Research NYC
sun.wen@microsoft.com

**Akshay Krishnamurthy**
Microsoft Research NYC
akshaykr@microsoft.com

**Alexander Rakhlin**
MIT
rakhlin@mit.edu

**John Langford**
Microsoft Research NYC
jcl@microsoft.com

## Abstract

We introduce a new problem setting for continuous control called the LQR with Rich Observations, or RichLQR. In our setting, the environment is summarized by a low-dimensional continuous latent state with linear dynamics and quadratic costs, but the agent operates on high-dimensional, nonlinear observations such as images from a camera. To enable sample-efficient learning, we assume the learner has access to a class of *decoder functions* (e.g., neural networks) that is flexible enough to capture the mapping from observations to latent states. We introduce a new algorithm, RichID, which learns a near-optimal policy for the RichLQR with sample complexity scaling only with the dimension of the latent state space and the capacity of the decoder function class. RichID is oracle-efficient and accesses the decoder class only through calls to a least-squares regression oracle. Our results constitute the first provable sample complexity guarantee for continuous control with an *unknown* nonlinearity in the system model.

## 1 Introduction

In reinforcement learning and control, an agent must learn to minimize its overall cost in a unknown dynamic environment that responds to its actions. In recent years, the field has developed a comprehensive understanding of the non-asymptotic sample complexity of *linear control*, where the dynamics of the environment are determined by a noisy linear system of equations. While studying linear models has led to a number of new theoretical insights, most practical control tasks are nonlinear. In this paper, we develop efficient algorithms with provable sample complexity guarantees for nonlinear control with rich, flexible function approximation.

For some control applications, the dynamics themselves are truly nonlinear, but another case—which is particularly relevant to real-world systems—is where there are (unknown-before-learning) latent linear dynamics which are identifiable through a nonlinear observation process. For example, cameras watching a robot may capture enough information to control its actuators, but the optimal control law is unlikely to be a simple linear function of the pixels. More broadly, with the decrease in costs of sensing hardware, it is now common to instrument complex control tasks with high-throughput measurement apparatus such as cameras, lidar, contact sensors, or other alternatives.

These measurements constitute *rich observations* which often capture relevant information about the system state. However, deriving a control policy from these complex, high-dimensional sources remains a significant challenge in both theory and practice.

**The RichLQR setting.** We propose a learning-theoretic framework for rich observation continuous control in which the environment is summarized by a low dimensional continuous latent state (such as joint angles), while the agent operates on high-dimensional observations (such as images from a camera). While this setup is more general, we focus our technical developments on perhaps the simplest instantiation: the *rich observation linear quadratic regulator* (RichLQR). The RichLQR posits that latent states evolve according to noisy linear equations and that each observation can be associated with a latent state by an unknown nonlinear mapping.

We assume that every possible high-dimensional observation of the system corresponds to a unique latent system state, a property we term *decodability*. This assumption is natural in applications where the observations contain significantly more information than needed to control the system. However, decoding the latent state may require a highly nonlinear mapping, in which case linear control on the raw observations will perform poorly. Our aim is to learn such a mapping from data and use it for optimal control in the latent space.

## 1.1 LQR with Rich Observations

RichLQR is a continuous control problem described by the following dynamics:

$$\mathbf{x}_{t+1} = A\mathbf{x}_t + B\mathbf{u}_t + \mathbf{w}_t, \quad \mathbf{y}_t \sim q(\cdot \mid \mathbf{x}_t). \tag{1}$$

Starting from $\mathbf{x}_0$, the system state $\mathbf{x}_t \in \mathbb{R}^{d_\mathbf{x}}$ evolves as a linear combination of the previous state, a control input $\mathbf{u}_t \in \mathbb{R}^{d_\mathbf{u}}$ selected by the learner, and zero-mean i.i.d. process noise $\mathbf{w}_t \in \mathbb{R}^{d_\mathbf{x}}$. The learner does not directly observe the state, and instead sees an *observation* $\mathbf{y}_t \in \mathbb{R}^{d_\mathbf{y}}$ drawn from the observation distribution $q(\cdot \mid \mathbf{x}_t)$.[1] Here $d_\mathbf{y} \gg d_\mathbf{x}$; $\mathbf{x}_t$ might represent the state of a robot's joints, while $\mathbf{y}_t$ might represent an image of the robot in a scene. Given a policy $\pi_t(\mathbf{y}_0, \ldots, \mathbf{y}_t)$ that selects control inputs $\mathbf{u}_t$ based on past and current observations, we measure performance as

$$J_T(\pi) := \mathbb{E}_\pi \left[ \frac{1}{T} \sum_{t=1}^{T} \mathbf{x}_t^\top Q \mathbf{x}_t + \mathbf{u}_t^\top R \mathbf{u}_t \right], \tag{2}$$

where $Q, R > 0$ are quadratic state and control cost matrices and $\mathbb{E}_\pi$ denotes the expectation when the system's dynamics (1) evolve under $\mathbf{u}_t = \pi_t(\mathbf{y}_0, \ldots, \mathbf{y}_t)$.

In our model, the dynamics matrices $(A, B)$ and the observation distribution $q(\cdot \mid \mathbf{x})$ are unknown to the learner. We assume that the control cost matrix $R > 0$ is known, but the state cost matrix $Q > 0$ is unknown (so as not to tie the cost matrices to the system representation). We also assume the instantaneous costs $\mathbf{c}_t := \mathbf{x}_t^\top Q \mathbf{x}_t + \mathbf{u}_t^\top R \mathbf{u}_t$, are revealed on each trajectory at time $t$ (this facilitates learning $Q$, but not $A$ or $B$). The learner's goal is to PAC-learn an $\varepsilon$-optimal policy: given access to $n$ trajectories from the dynamics (1), produce a policy $\widehat{\pi}$ such that $J_T(\widehat{\pi}) - J_T(\pi_\infty) \le \varepsilon$, where $\pi_\infty$ is the optimal infinite-horizon policy. If the dynamics matrices $(A, B)$ were known and the state $\mathbf{x}_t$ were directly observed, the RichLQR would reduce to the classical LQR problem [18], and we could compute an optimal policy for (2) using dynamic programming. Indeed, the optimal policy has the form $\pi_\infty(\mathbf{x}_t) = K_\infty \mathbf{x}_t$, where $K_\infty$ is the optimal infinite-horizon state-feedback matrix given by the Discrete Algebraic Riccati Equation (Eq. (A.1) in Appendix A.2). To facilitate the use of optimal control tools in our nonlinear observation model, we make the following assumption, which asserts the state $\mathbf{x}_t$ can be uniquely recovered from the observation $\mathbf{y}_t$.

**Assumption 1** (Perfect decodability)**.** There exists a *decoder function* $f_\star : \mathbb{R}^{d_\mathbf{y}} \to \mathbb{R}^{d_\mathbf{x}}$ such that $f_\star(y) = x$ for all $y \in \operatorname{supp} q(\cdot \mid x)$.[2]

While a perfect decoder $f_\star$ is guaranteed to exist under Assumption 1 (and thus the optimal LQR policy can be executed from observations), $f_\star$ is *not known* to the learner in advance. Instead, we assume that the learner has access to a class of functions $\mathscr{F}$ (e.g., neural networks) that is rich enough to express the perfect decoder. Our statistical rates depend on the capacity of this class.

**Assumption 2** (Realizability). The learner's decoder class $\mathscr{F}$ contains the true decoder $f_\star$.

While these assumptions—especially decodability—may seem strong at first glance, we show that without strong assumptions on the observation distribution, the problem quickly becomes statistically intractable. Consider the following variant of the model (1):

$$\mathbf{x}_{t+1} = A\mathbf{x}_t + B\mathbf{u}_t + \mathbf{w}_t, \qquad \mathbf{y}_t = f_\star^{-1}(\mathbf{x}_t) + \varepsilon_t, \tag{3}$$

where $\varepsilon_t$ is an independent *output noise* variable with $\mathbb{E}[\varepsilon_t] = 0$. In the absence of the noise $\varepsilon_t$, the system (3) is a special case of (1) for which $f_\star$ is the true decoder, but in general the noise breaks perfect decodability. Unfortunately, our first theorem shows that in general, output noise can lead to exponential sample complexity for learning nonlinear decoders, even under very benign conditions.

**Theorem (informal).** *Consider the dynamics (3) with $d_\mathbf{x} = d_\mathbf{y} = d_\mathbf{u} = T = 1$ and unit Gaussian noise. For every $\varepsilon > 0$ there exists an $\mathcal{O}(\varepsilon^{-1})$-Lipschitz decoder $f_\star$ and realizable function class $\mathscr{F}$ with $|\mathscr{F}| = 2$ such that any algorithm requires $\Omega(2^{\left(\frac{1}{\varepsilon}\right)^{2/3}})$ trajectories to learn an $\varepsilon$-optimal decoder.*

A full statement and proof for this lower bound is deferred to Appendix D for space.

**Our Algorithm: RichID.** Our main contribution is a new algorithmic principle, *Rich Iterative Decoding*, or RichID, which solves the RichLQR problem with sample complexity scaling polynomially in the latent dimension $d_\mathbf{x}$ and the decoder class capacity $\ln|\mathscr{F}|$. We analyze an algorithm based on this principle called RichID-CE ("RichID with Certainty Equivalence"), which solves the RichLQR by learning an off-policy estimator for the decoder, using the off-policy decoder to approximately recover the dynamics $(A, B)$, and then using these estimates to iteratively learn a sequence of on-policy decoders along the trajectory of a near-optimal policy. Our main theorem is as follows.

**Theorem 1** (Main theorem). *Under appropriate regularity conditions on the system parameters and noise process (Assumptions 1-8), RichID-CE learns an $\varepsilon$-optimal policy for horizon $T$ using $C \cdot \frac{(d_\mathbf{x}+d_\mathbf{u})^{16} T^4 \ln|\mathscr{F}|}{\varepsilon^6}$ trajectories, where $C$ is a problem-dependent constant.*[3]

Theorem 1 shows that it is possible to learn the RichLQR with complexity polynomial in the latent dimension and decoder class capacity $\ln|\mathscr{F}|$, and independent of the observation space. To our knowledge, this is the first polynomial-in-dimension sample complexity guarantee for continuous control with an unknown system nonlinearity and general function classes. The main challenge we overcome in attaining Theorem 1 is trajectory mismatch; a learned decoder $\hat{f}$ which accurately approximates the true decoder $f_\star$ well on one trajectory may significantly deviate from $f_\star$ on another. Our algorithm addresses this issue using a carefully designed *iterative decoding* procedure to learn a sequence of decoders on-policy. We present our main theorem for finite classes $\mathscr{F}$ for simplicity, but this quantity arises only through standard generalization bounds for least squares, and can trivially be replaced by learning-theoretic complexity measures such as Rademacher complexity (in fact, local Rademacher complexity). For example, if $\mathscr{F}$ has pseudodimension $d$, one can replace $\ln|\mathscr{F}|$ with $\widetilde{\mathcal{O}}(d)$.

Theorem 1 requires relatively strong assumptions on the dynamical system—in particular, we require that the system matrix $A$ is stable, and that the process noise is Gaussian. Nonetheless, we believe that our results represent an important first step toward developing provable and practical sample-efficient algorithms for continuous control beyond the linear setting, and we are excited to see technical improvements addressing these issues in future research.

## 1.2 Technical Preliminaries

In the interest of brevity, we present an abridged discussion of technical preliminaries; all omitted formal definitions, further assumptions, and additional notation are deferred to Appendix A. The main assumptions used by RichID are as follows.

**Assumption 3** (Gaussian initial state and process noise). The initial state satisfies $\mathbf{x}_0 \sim \mathcal{N}(0, \Sigma_0)$, and process noise is i.i.d. $\mathbf{w}_t \sim \mathcal{N}(0, \Sigma_w)$. Here, $\Sigma_0, \Sigma_w$ are *unknown* to the learner, with $\Sigma_w > 0$.

**Assumption 4** (Controllability). For each $k \geq 1$, define $\mathcal{C}_k := [A^{k-1}B \mid \ldots \mid B] \in \mathbb{R}^{d_\mathbf{x} \times k d_\mathbf{u}}$. We assume that $(A, B)$ is *controllable*, meaning that $\mathcal{C}_{\kappa_\star}$ has full column rank for some $\kappa_\star \in \mathbb{N}$.

Note that Assumption 4 imposes the constraint $d_{\mathbf{u}}\kappa_\star \geq d_{\mathbf{x}}$, which we use to simplify expressions.

**Assumption 5** (Growth Condition)**.** There exists $L \geq 1$ such that $\|f(y)\| \leq L \max\{1, \|f_\star(y)\|\}$ for all $y \in \mathcal{Y}$ and $f \in \mathscr{F}$.

**Assumption 6** (Stability)**.** $A$ is stable; that is, $\rho(A) < 1$, where $\rho(\cdot)$ denotes the spectral radius.

Our algorithms and analysis make heavy use of the Gaussian process noise assumption, which we use to calculate closed-form expressions for certain conditional expectations that arise under the dynamics model (1). We view relaxing this assumption as an important direction for future work. Controllability is somewhat more standard [24], and the growth condition ensures predictions do not behave too erratically. Stability ensures the state remains bounded without an initial stabilizing controller. While assuming access to an initial stabilizing controller is fairly standard in the recent literature on linear control, this issue is more subtle in our nonlinear observation setting. These assumptions can be relaxed somewhat; see Appendix B.4. We make the stability assumption quantitative via the notion of "strong stability" (Appendix A). Finally, we assume access to bounds on various system parameters.

**Assumption 7.** We assume that the learner has access to parameter upper bounds $\Psi_\star \geq 1$, $\alpha_\star \geq 1$, $\gamma_\star \in (0,1)$, and $\kappa \in \mathbb{N}$ such that **(I)** $\kappa \geq \kappa_\star$, **(II)** $A$ and $(A + BK_\infty)$ are both $(\alpha_\star, \gamma_\star)$-strongly stable, and **(III)** $\Psi_\star$ is an upper bound on the operator norms of $A, B, Q, R, \Sigma_w, \Sigma_w^{-1}, \Sigma_0, K_\infty$, and $P_\infty$.[4]

We use $\mathcal{O}_\star(\cdot)$ to suppress polynomial factors in $\alpha_\star, \gamma_\star^{-1}, (1-\gamma_\star)^{-1}, \Psi_\star, L$, and $\sigma_{\min}^{-1}(\mathcal{C}_\kappa)$, and all logarithmic factors except for $\ln|\mathscr{F}|$ and $\ln(1/\delta)$. We also write $f = \check{\mathcal{O}}(g)$ if $f(x) \leq cg(x)$ for all $x \in \mathcal{X}$, where $c = \mathrm{poly}(\gamma_\star(1-\gamma_\star), \alpha_\star^{-1}, \Psi_\star^{-1}, L^{-1}, \sigma_{\min}(\mathcal{C}_\kappa))$ is a *sufficiently small* constant.

## 2 An Algorithm for LQR with Rich Observations

---
**Algorithm 1** RichID-CE
---
1: **Inputs:**
    $\varepsilon$ (suboptimality), $T$ (horizon), $\mathscr{F}$ (decoder class), $d_{\mathbf{x}}, d_{\mathbf{u}}$ (latent dimensions),
    $\Psi_\star, \kappa, \alpha_\star, \gamma_\star$ (system parameter upper bounds), $R$ (control cost).
2: **Parameters:** `// see Appendix J for values.`
    $n_{\mathrm{id}}, n_{\mathrm{op}}$ `// sample size for Phase/Phase II and Phase III, respectively.`
    $\kappa_0$ `// burn-in time index.`
    $r_{\mathrm{id}}, r_{\mathrm{op}}$ `// radius for sets` $\mathscr{H}_{\mathrm{id}}$ `and` $\mathscr{H}_{\mathrm{op}}$`.`
    $\sigma^2$ `// exploration variance.`
    $\bar{b}$ `// clipping parameter for decoders.`
3: **Phase I** `// learn a coarse decoder (see Section 2.1)`
4: Set $\hat{f}_{\mathrm{id}} \leftarrow \mathrm{GETCOARSEDECODER}(n_{\mathrm{id}}, \kappa_0, \kappa, r_{\mathrm{id}})$. `// Algorithm 3`
5: **Phase II** `// learn system's dynamics and cost (see Section 2.2)`
6: Set $(\widehat{A}_{\mathrm{id}}, \widehat{B}_{\mathrm{id}}, \widehat{\Sigma}_{w,\mathrm{id}}, \widehat{Q}_{\mathrm{id}}) \leftarrow \mathrm{SYSID}(\hat{f}_{\mathrm{id}}, n_{\mathrm{id}}, \kappa_0, \kappa)$. `// Algorithm 4`
7: **Phase III** `// compute optimal policy (see Section 2.3 and Appendix H)`
8: Set $\widehat{\pi} \leftarrow \mathrm{COMPUTEPOLICY}(\widehat{A}_{\mathrm{id}}, \widehat{B}_{\mathrm{id}}, \widehat{\Sigma}_{w,\mathrm{id}}, \widehat{Q}_{\mathrm{id}}, R, n_{\mathrm{op}}, \kappa, \sigma^2, T, \bar{b}, r_{\mathrm{op}})$. `// Algorithm 5`
9: **Return:** $\widehat{\pi}$.
---

We now present our main algorithm, RichID-CE (Algorithm 1), which attains a polynomial sample complexity guarantee for the RichLQR.

**Algorithm overview.** Algorithm 1 consists of three phases. In Phase I (Algorithm 3), we roll in with Gaussian control inputs and learn a good decoder under this roll-in distribution by solving a certain regression problem involving our decoder class $\mathscr{F}$. In Phase II (Algorithm 4), we leverage this decoder to learn a *model* $(\widehat{A}, \widehat{B})$ for the system dynamics (up to a similarity transform). Due to linearity of the dynamics, this model is valid on any trajectory. Moreover, we can synthesize a controller $\widehat{K}$ so that the controller $\mathbf{u}_t = \widehat{K}\mathbf{x}_t$ is optimal for $(\widehat{A}, \widehat{B})$, and thus near-optimal for $(A, B)$.

To actually implement this feedback controller, we still need a good decoder for the state. Unfortunately, our decoder from Phase I may be inaccurate along the optimal (or near-optimal) trajectory. Thus, in Phase III (Algorithm 5) we inductively solve a sequence of regression problems—one for each time $t = 0, \ldots, T$—to learn a sequence of state decoders $(\hat{f}_t)$, such that for each $t$, $\hat{f}_t \approx f_\star$ under the roll-in distribution induced by playing $\widehat{K}\hat{f}_s(\mathbf{y}_s)$ for $s < t$. We do this by rolling in with this near-optimal policy until $t$, but rolling out with purely Gaussian inputs. The former ensures that the decoder is accurate along the desired trajectory. The latter ensures that the regression at time $t$ is essentially "independent" of approximation errors incurred by steps $0, \ldots, t - 1$, avoiding an accumulation of errors which would otherwise compound exponentially in the horizon $T$.

In what follows, we walk through each phase in detail and explain the motivation, the technical assumptions required, and the key performance guarantees.

## 2.1 Phase I: Learning a Coarse Decoder

In Phase I (Algorithm 3), we gather $2n_{\mathrm{id}}$ trajectories by selecting independent standard Gaussian inputs $\mathbf{u}_t \sim \mathcal{N}(0, I_{d_{\mathbf{u}}})$ for each $0 \le t \le \kappa_1 := \kappa_0 + \kappa$, where we recall that $\kappa$ is an upper-bound on the controllability index $\kappa_\star$, and where $\kappa_0$ is a "burn-in" time used to ensure mixing to a near-stationary distribution (this is useful for learning $(A, B)$ in (9), ensuring $\hat{f}_{\mathrm{id}}$ is accurate at both time $\kappa_1$ and $\kappa_1 + 1$). $\kappa_0$ is given by:

$$\kappa_0 := \left\lceil (1 - \gamma_\star)^{-1} \ln \left( 84 \Psi_\star^5 \alpha_\star^4 d_{\mathbf{x}} (1 - \gamma_\star)^{-2} \ln(10^3 \cdot n_{\mathrm{id}}) \right) \right\rceil. \tag{4}$$

Let $(\mathbf{u}_0^{(i)}, \mathbf{y}_0^{(i)}, \mathbf{c}_0^{(i)}), \ldots, (\mathbf{u}_{\kappa_1}^{(i)}, \mathbf{y}_{\kappa_1}^{(i)}, \mathbf{c}_{\kappa_1}^{(i)}), \mathbf{y}_{\kappa_1+1}^{(i)}$ denote the $i$th trajectory gathered in this fashion. We now show that for the state distribution induced the control inputs above, the true decoder $f_\star$ can be recovered up to a linear transformation by solving a regression problem whose goal is to predict a sequence of control inputs from the observations at time $\kappa_1$. Define $\mathbf{v} := (\mathbf{u}_{\kappa_0}^\top, \ldots, \mathbf{u}_{\kappa_1-1}^\top)^\top$. Our key lemma (Lemma G.3) shows that

$$\forall y \in \mathbb{R}^{d_{\mathbf{y}}}, \quad h_\star(y) := \mathbb{E}[\mathbf{v} \mid \mathbf{y}_{\kappa_1} = y] = \mathcal{C}_\kappa^\top \Sigma_{\kappa_1}^{-1} f_\star(y), \tag{5}$$

$\mathcal{C}_\kappa := [A^{\kappa-1}B \mid \ldots \mid B]$; and $\Sigma_{\kappa_1} := A^{\kappa_1} \Sigma_0 (A^{\kappa_1})^\top + \sum_{t=0}^{\kappa_1} A^{t-1}(\Sigma_w + BB^\top)(A^{t-1})^\top$. where we recall that $\mathcal{C}_\kappa = [A^{\kappa-1}B \mid \ldots \mid B]$ and define

$$\Sigma_{\kappa_1} := A^{\kappa_1} \Sigma_0 (A^{\kappa_1})^\top + \sum_{t=0}^{\kappa_1} A^{t-1}(\Sigma_w + BB^\top)(A^{t-1})^\top.$$

This lemma relies on perfect decodability and the fact that $\mathbf{v}$ and $\mathbf{x}_{\kappa_1}$ are jointly Gaussian. In particular, by verifying $\|\mathcal{C}_\kappa^\top \Sigma_{\kappa_1}^{-1}\|_{\mathrm{op}} \le \sqrt{\Psi_\star}$, the expression (5) ensures that $h_\star$ belongs to the class $\mathscr{H}_{\mathrm{id}} := \{ Mf(\cdot) \mid f \in \mathscr{F}, \ M \in \mathbb{R}^{\kappa d_{\mathbf{u}} \times d_{\mathbf{x}}}, \ \|M\|_{\mathrm{op}} \le \sqrt{\Psi_\star} \}$(that is, we can take $r_{\mathrm{id}} = \sqrt{\Psi_\star}$). The main step of Phase I solves the well-specified regression problem:

$$\hat{h}_{\mathrm{id}} \in \arg\min_{h \in \mathscr{H}_{\mathrm{id}}} \sum_{i=1}^{n_{\mathrm{id}}} \|h(\mathbf{y}_{\kappa_1}^{(i)}) - \mathbf{v}^{(i)}\|^2. \tag{6}$$

Phase I is computationally efficient whenever we have a *regression oracle* for the induced function class $\mathscr{H}_{\mathrm{id}}$. For many function classes of interest, such as linear functions and neural networks, solving regression over this class is no harder than regression over the original decoder class $\mathscr{F}$, so we believe this is a reasonably practical assumption. For $n_{\mathrm{id}}$ sufficiently large, a standard analysis for least squares shows that the regressor $\hat{h}_{\mathrm{id}}$ has low prediction error relative to $h_\star$ in (5). However, this representation is overparameterized and takes values in $\mathbb{R}^{\kappa d_{\mathbf{x}}}$ even though the true state lies in only $d_{\mathbf{x}}$ dimensions. For the second part of Phase I, we perform principle component analysis to reduce the dimension to $d_{\mathbf{x}}$. Specifically, we compute a dimension-reduced decoder via

$$\hat{f}_{\mathrm{id}}(y) := \widehat{V}_{\mathrm{id}}^\top \cdot \hat{h}_{\mathrm{id}}(y) \in \mathbb{R}^{d_{\mathbf{x}}}, \tag{7}$$

where $\widehat{V}_{\mathrm{id}} \in \mathbb{R}^{\kappa d_{\mathbf{u}} \times d_{\mathbf{x}}}$ is an arbitrary orthonormal basis for the top $d_{\mathbf{x}}$ eigenvectors of the empirical second moment matrix $\sum_{i=n_{\mathrm{id}}+1}^{2n_{\mathrm{id}}} \hat{h}_{\mathrm{id}}(\mathbf{y}_{\kappa_1}^{(i)}) \hat{h}_{\mathrm{id}}(\mathbf{y}_{\kappa_1}^{(i)})^\top / n_{\mathrm{id}}$. This approach exploits that the output of the Bayes regressor $h_\star$—being a linear function of the $d_{\mathbf{x}}$-dimensional system state—lies in a $d_{\mathbf{x}}$-dimensional subspace. Having reviewed the two components of Phase I, we can now state the main guarantee for this phase. In light of (5), the result essentially follows from standard tools for least-squares regression with a well-specified model, plus an analysis for PCA with errors in variables.

**Theorem 2** (Guarantee for Phase I). *If $n_{\mathrm{id}} = \Omega_\star(d_\mathbf{x} d_\mathbf{u} \kappa(\ln|\mathscr{F}| + d_\mathbf{u} d_\mathbf{x} \kappa))$, then with probability at least $1 - 3\delta$, there exists an invertible matrix $S_{\mathrm{id}} \in \mathbb{R}^{d_\mathbf{x} \times d_\mathbf{x}}$ such that*

$$\mathbb{E}\|\hat{f}_{\mathrm{id}}(\mathbf{y}_{\kappa_1}) - S_{\mathrm{id}} f_\star(\mathbf{y}_{\kappa_1})\|^2 \leq \mathcal{O}_\star\left(\frac{d_\mathbf{u} \kappa(\ln|\mathscr{F}| + d_\mathbf{u} d_\mathbf{x} \kappa)\ln^3(n_{\mathrm{id}}/\delta)}{n_{\mathrm{id}}}\right),$$

*and for which $\sigma_{\min}(S_{\mathrm{id}}) \geq \sigma_{\min,\mathrm{id}} := \sigma_{\min}(\mathcal{C}_\kappa)(1 - \gamma_\star)(4\Psi_\star^2 \alpha_\star^2)^{-1}$ and $\|S_{\mathrm{id}}\|_{\mathrm{op}} \leq \sigma_{\max,\mathrm{id}} := \sqrt{\Psi_\star}$.*

### 2.2 Phase II: System Identification

In Phase II, we use the decoder from Phase I to learn the system dynamics, state cost, and process noise covariance up to the basis induced by the transformation $S_{\mathrm{id}}$. Our targets are:

$$A_{\mathrm{id}} := S_{\mathrm{id}} A S_{\mathrm{id}}^{-1}, \quad B_{\mathrm{id}} := S_{\mathrm{id}} B, \quad \Sigma_{w,\mathrm{id}} := S_{\mathrm{id}} \Sigma_w S_{\mathrm{id}}^\top, \quad Q_{\mathrm{id}} := S_{\mathrm{id}}^{-\top} Q S_{\mathrm{id}}^{-1}. \tag{8}$$

The key technique we use is to pretend that the decoder's output $\hat{f}_{\mathrm{id}}(\mathbf{y}_{\kappa_1})$ is the true state $\mathbf{x}_{\kappa_1}$, then perform regressions which mimic the dynamics in (1):

$$(\widehat{A}_{\mathrm{id}}, \widehat{B}_{\mathrm{id}}) \in \underset{(A,B)}{\arg\min} \sum_{i=2n_{\mathrm{id}}+1}^{3n_{\mathrm{id}}} \|\hat{f}_{\mathrm{id}}(\mathbf{y}_{\kappa_1+1}^{(i)}) - A\hat{f}_{\mathrm{id}}(\mathbf{y}_{\kappa_1}^{(i)}) - B\mathbf{u}_{\kappa_1}^{(i)}\|^2, \quad \text{and} \tag{9}$$

$$\widehat{\Sigma}_{w,\mathrm{id}} = \frac{1}{n_{\mathrm{id}}} \sum_{i=2n_{\mathrm{id}}+1}^{3n_{\mathrm{id}}} (\hat{f}_{\mathrm{id}}(\mathbf{y}_{\kappa_1+1}^{(i)}) - \widehat{A}_{\mathrm{id}}\hat{f}_{\mathrm{id}}(\mathbf{y}_{\kappa_1}^{(i)}) - \widehat{B}_{\mathrm{id}}\mathbf{u}_{\kappa_1}^{(i)})^{\otimes 2}, \quad \text{where } v^{\otimes 2} := vv^\top. \tag{10}$$

Similarly, we recover the state cost $Q$ by fitting a quadratic function to observed costs via

$$\widetilde{Q}_{\mathrm{id}} \in \underset{Q}{\arg\min} \sum_{i=2n_{\mathrm{id}}+1}^{3n_{\mathrm{id}}} \left(\mathbf{c}_{\kappa_1}^{(i)} - (\mathbf{u}_{\kappa_1}^{(i)})^\top R\mathbf{u}_{\kappa_1}^{(i)} - \hat{f}_{\mathrm{id}}(\mathbf{y}_{\kappa_1}^{(i)})^\top Q\hat{f}_{\mathrm{id}}(\mathbf{y}_{\kappa_1}^{(i)})\right)^2, \tag{11}$$

and then setting $\widehat{Q}_{\mathrm{id}} = \left(\frac{1}{2}\widetilde{Q}_{\mathrm{id}} + \frac{1}{2}\widetilde{Q}_{\mathrm{id}}^\top\right)_+$ as the final estimator, where $(\cdot)_+$ truncates non-positive eignvalues to zero. This is the only place where the algorithm uses the cost oracle.

Since Theorem 2 ensures that $\hat{f}_{\mathrm{id}}(\mathbf{y}_{\kappa_1})$ is not far from $S_{\mathrm{id}}\mathbf{x}_{\kappa_1}$, the regression problems (9)–(11) are all nearly-well-specified, and we have the following guarantee.

**Theorem 3** (Guarantee for Phase II). *If $n_{\mathrm{id}} = \Omega_\star\left(d_\mathbf{x}^2 d_\mathbf{u} \kappa(\ln|\mathscr{F}| + d_\mathbf{u} d_\mathbf{x} \kappa) \max\{1, \sigma_{\min}(\mathcal{C}_\kappa)^{-4}\}\right)$, then with probability at least $1 - 11\delta$ over Phases I and II,*

$$\|[\widehat{A}_{\mathrm{id}}; \widehat{B}_{\mathrm{id}}] - [A_{\mathrm{id}}; B_{\mathrm{id}}]\|_{\mathrm{op}} \vee \|\widehat{Q}_{\mathrm{id}} - Q_{\mathrm{id}}\|_{\mathrm{op}} \vee \|\widehat{\Sigma}_{w,\mathrm{id}} - \Sigma_{w,\mathrm{id}}\|_{\mathrm{op}} \leq \varepsilon_{\mathrm{id}}, \tag{12}$$

*where $\varepsilon_{\mathrm{id}} \leq \mathcal{O}_\star\left(n_{\mathrm{id}}^{-1/2} \ln^2(n_{\mathrm{id}}/\delta)\sqrt{d_\mathbf{x} d_\mathbf{u} \kappa(\ln|\mathscr{F}| + d_\mathbf{u} d_\mathbf{x} \kappa)}\right)$.*

To simplify presentation, we assume going forward that $S_{\mathrm{id}} = I_{d_\mathbf{x}}$, which is without loss of generality (at the cost of increasing parameters such as $\Psi_\star$ and $\alpha_\star$ by a factor of $\|S_{\mathrm{id}}\|_{\mathrm{op}} \vee \|S_{\mathrm{id}}^{-1}\|_{\mathrm{op}}$),[5] and drop the "id" subscript on the estimators $\widehat{A}_{\mathrm{id}}, \widehat{B}_{\mathrm{id}}$, and so forth to reflect this.[6]

### 2.3 Phase III: Decoding Observations Along the Optimal Path

Given the estimates $(\widehat{A}, \widehat{B}, \widehat{Q})$ from Theorem 3, we can use certainty equivalence to synthesize an optimal controller matrix $\widehat{K}$ for the estimated dynamics. As long as $\varepsilon_{\mathrm{id}}$ in (12) is sufficiently small, the policy $\mathbf{u}_t = \widehat{K}\mathbf{x}_t$ is stabilizing and near optimal.

To (approximately) implement this policy from rich observations, it remains to accurately estimate the latent state. The decoder learned in Phase I does not suffice; it only ensures low error on trajectories generated with random Gaussian inputs, and not on the trajectory induced by the near-optimal policy. Indeed, while it is tempting to imagine that the initial decoder $\hat{f}$ might generalize across different trajectories, this is not the case in unless we place strong structural assumptions on $\mathscr{F}$.

Instead, we iteratively learn a sequence of decoders $\hat{f}_t$—one per timestep $t = 1, \ldots, T$. Assuming $\widehat{K} \approx K_\infty$ is near optimal, the suboptimality $J_T(\pi) - J_T(\pi_\infty)$ of the policy $\pi(\mathbf{y}_{0:t}) \coloneqq \widehat{K}\hat{f}_t(\mathbf{y}_{0:t})$ is controlled by the sum $\sum_{t=1}^T \mathbb{E}_\pi \|\hat{f}_t(\mathbf{y}_{0:t}) - f_\star(\mathbf{y}_t)\|^2$ (note that the regret does not take into account step 0). Thus, to ensure low regret, we ensure that, for all $t \geq 1$, the decoder $\hat{f}_t$ has low prediction error *on the distribution induced by running $\pi$ with previous decoders* $(\hat{f}_\tau)_{1 \leq \tau < t}$ and $\widehat{K}$. This motivates the following iterative decoding procedure, executed for each time step $t = 1, \ldots, T$:

**Step 1.** Collect $2n_{\mathrm{op}}$ trajectories by executing the randomized control input $\mathbf{u}_\tau = \widehat{K}\hat{f}_\tau(\mathbf{y}_{0:\tau}) + \boldsymbol{\nu}_\tau$, for $0 \leq \tau \leq t$, and $\mathbf{u}_\tau = \boldsymbol{\nu}_\tau$, for $t < \tau < t + \kappa$, where $\boldsymbol{\nu}_\tau \sim \mathcal{N}(0, \sigma^2 I_{d_{\mathbf{u}}})$; here, $n_{\mathrm{op}} \in \mathbb{N}$ and $\sigma^2 \leq 1$ are algorithm parameters to be specified later.

**Step 2.** Obtain a *residual decoder* $\hat{h}_t$ satisfying (13) by solving regressions (21) and (22) using a regression oracle.

**Step 3.** Form a state decoder $\hat{f}_{t+1}$ from $\hat{h}_t$ and $\hat{f}_t$ using the update equation (14).

Forming the decoder $\hat{f}_1$ requires additional regression steps (described in Appendix B.3) which account for the uncertainty in the initial state $\mathbf{x}_0$. At each subsequent time $t$, the most important part of the procedure above is **Step 2**, which aims to produce a regressor $\hat{h}_t$ such that

$$\hat{h}_t(\mathbf{y}_{t+1}) - A\hat{h}_t(\mathbf{y}_t) \approx B\mathbf{u}_t + \mathbf{w}_t = \mathbf{x}_{t+1} - A\mathbf{x}_t. \tag{13}$$

As we shall see, enforcing accuracy on the *increments* $\mathbf{x}_{t+1} - A\mathbf{x}_t$ allows us to set up regression problems which do not depend on, and thus *do not* propagate forward, the errors in $\hat{f}_t$. In contrast, a naive regression—say, $\arg\min_f \mathbb{E}\left[\|f(\mathbf{y}_{t+1}) - (A + B\widehat{K})\hat{f}_t(\mathbf{y}_t) - B\boldsymbol{\nu}_t\|^2\right]$—could compound decoding errors exponentially in $t$.

Luckily, the increments in (13) are sufficient for recovery of the state by unfolding a recursion; this comprises **Step 3**. Let $\bar{b} > 0$ be an algorithm parameter. Given a regressor $\hat{h}_t$ satisfying (13) and the current decoder $\hat{f}_t$, we form next state decoder $\hat{f}_{t+1}$ via

$$\hat{f}_{t+1}(\cdot) \coloneqq \tilde{f}_{t+1}(\cdot)\mathbb{I}\{\|\tilde{f}_{t+1}(\cdot)\| \leq \bar{b}\}; \quad \tilde{f}_{t+1}(\mathbf{y}_{0:t+1}) \coloneqq \left(\hat{h}_t(\mathbf{y}_{t+1}) - \widehat{A} \cdot \hat{h}_t(\mathbf{y}_t)\right) + \widehat{A} \cdot \hat{f}_t(\mathbf{y}_{0:t}), \tag{14}$$

where we set $\tilde{f}_0 \equiv \hat{f}_0 \equiv 0$. By clipping $\tilde{f}_t$, we ensure states remain bounded, which simplifies the analysis. Crucially, by building our decoders $(\hat{f}_\tau)$ this way, we ensure that the decoding error grows at most linearly in $t$—as opposed to exponentially—as long as the system is stable (i.e. $\rho(A) < 1$).

It remains to describe how to obtain a regressor $\hat{h}_t$ satisfying (13). To this end, we use the added Gaussian noise $\boldsymbol{\nu}_t$ to set up the regression.

**Warm-up: Invertible $B$.** As a warm-up, suppose that $B$ is invertible. Then, for the matrix $M_1 \coloneqq B^\top (BB^\top + \sigma^{-2}\Sigma_w)^{-1}$, one can compute

$$\mathbb{E}[\boldsymbol{\nu}_t \mid \mathbf{y}_{0:t+1}] \overset{(*)}{=} \mathbb{E}[\boldsymbol{\nu}_t \mid \mathbf{w}_t + B\boldsymbol{\nu}_t] \overset{(**)}{=} M_1(\mathbf{w}_t + B\boldsymbol{\nu}_t) = M_1(\mathbf{x}_{t+1} - A\mathbf{x}_t - B\widehat{K}\hat{f}_t(\mathbf{y}_{0:t})). \tag{15}$$

The identity $(*)$ uses the fact that conditioning on $\mathbf{y}_{0:t+1}$ is equivalent to conditioning on $\mathbf{x}_{0:t+1}$, due to perfect decodability. We then use that the conditional distribution of $\boldsymbol{\nu}_t \mid \mathbf{x}_{0:t+1}$ is equivalent to $\boldsymbol{\nu}_t \mid \mathbf{x}_t, \mathbf{x}_{t+1}$, which is in turn equivalent to $\boldsymbol{\nu}_t \mid \mathbf{w}_t + B\boldsymbol{\nu}_t$ due to the linear dynamics Eq. (1).

Since conditional expectations minimize the square loss, learning a residual regressor $\hat{h}_t$ which approximately minimizes

$$h \mapsto \mathbb{E}\left[\|\boldsymbol{\nu}_t - M_1(h(\mathbf{y}_{t+1}) - Ah(\mathbf{y}_t) - B\widehat{K}\hat{f}_t(\mathbf{y}_{0:t}))\|^2\right] \tag{16}$$

produces a decoder $\hat{h}_{t+1}$ approximately satisfying (13):

$$M_1(\hat{h}_t(\mathbf{y}_{t+1}) - A\hat{h}_t(\mathbf{y}_{t+1})) \approx M_1(\mathbf{x}_{t+1} - A\mathbf{x}_{t+1}), \tag{17}$$

$$\text{since} \quad M_1(\hat{h}_t(\mathbf{y}_{t+1}) - A\hat{h}_t(\mathbf{y}_{t+1}) - B\widehat{K}\hat{f}_t(\mathbf{y}_{0:t})) \approx M_1(\mathbf{x}_{t+1} - A\mathbf{x}_{t+1} - B\widehat{K}\hat{f}_t(\mathbf{y}_{0:t})).$$

For invertible $B$, the matrix $M_1$ is invertible, and so from (17), our state decoder $\hat{h}_{t+1}$ indeed satisfies (13): $\hat{h}_t(\mathbf{y}_{t+1}) - A\hat{h}_t(\mathbf{y}_t) \approx \mathbf{x}_{t+1} - A\mathbf{x}_t$. We emphasize that regressing to purely Gaussian inputs $\boldsymbol{\nu}_t$ is instrumental in ensuring the conditional expectation equality in (15) holds. The noise variance $\sigma^2$ trades off between the conditioning of the regression, and the excess suboptimality caused by noise injection; we choose it so that the final suboptimality is $\mathcal{O}_\star(\epsilon)$.

**Extension to general controllable systems.** For non-invertible $B$, we aggregate more regressions. For $k \in [\kappa]$, let $M_k \coloneqq \mathcal{C}_k^\top (\mathcal{C}_k \mathcal{C}_k^\top + \sigma^{-2} \sum_{i=0}^{k} A^{i-1} \Sigma_w (A^{i-1})^\top)^{-1}$, where we recall $\mathcal{C}_k$ from Assumption 4. Generalizing (15), we show (Lemma I.7 in Appendix H) that the outputs $(\mathbf{y}_\tau)$ and the Gaussian perturbation vector $\boldsymbol{\nu}_{t:t+k-1} \coloneqq (\boldsymbol{\nu}_t^\top, \ldots, \boldsymbol{\nu}_{t+k-1}^\top)^\top$ generated according to **Step 1** above satisfy, for all $k \in [\kappa]$,

$$\mathbb{E}[\boldsymbol{\nu}_{t:t+k-1} \mid \mathbf{y}_{0:t}, \mathbf{y}_{t+k}] = M_k(\mathbf{x}_{t+k} - A^k \mathbf{x}_t - A^{k-1} B \hat{f}_t(\mathbf{y}_{0:t})) =: \phi_{t,k}^\star(\mathbf{y}_{0:t+k}). \tag{18}$$

Defining concatentations $\phi_t^\star \coloneqq (\phi_{t,1}^\star, \ldots, \phi_{t,\kappa}^\star)$ and $\mathcal{M} \coloneqq [M_1^\top \mid (M_2 A)^\top \mid \cdots \mid (M_\kappa A^{\kappa-1})^\top]^\top$ and stacking the conditional expectations gives:

$$\mathbb{E}[\phi_t^\star(\mathbf{y}_{0:t+\kappa}) \mid \mathbf{y}_{0:t+1}] = \mathcal{M}(B \boldsymbol{\nu}_t + \mathbf{w}_t) = \mathcal{M}(f_\star(\mathbf{y}_{t+1}) - A f_\star(\mathbf{y}_t) - B \widehat{K} \hat{f}_t(\mathbf{y}_{0:t})). \tag{19}$$

Hence, with infinite samples (and knowledge of $B$), we are able to recover the residual quantity $\mathcal{M}(f_\star(\mathbf{y}_{t+1}) - A f_\star(\mathbf{y}_t))$. Again, the Gaussian inputs enable the conditional expectations (18) and (19). The crucial insight for the stacked regression is that by rolling in and switching to pure Gaussian noise only *after* time $t$, we maintain Gaussianity, while still yielding decoders that are valid on-trajectory *up to* time $t$. To ensure that we accurately recover the increment $f_\star(\mathbf{y}_{t+1}) - A f_\star(\mathbf{y}_t)$, we require the overdetermined matrix $\mathcal{M}$ to be invertible. To facilitate this, let $\mathcal{M}_{\sigma^2}$ denote the value of $\mathcal{M}$ as a function of $\sigma^2$, and let

$$\overline{\mathcal{M}} \coloneqq \lim_{\sigma \to 0} \mathcal{M}_{\sigma^2} / \sigma^2 \tag{20}$$

be the (normalized) limiting matrix as noise tends to zero, which is an intrinsic problem parameter.

**Assumption 8.** The limiting matrix $\overline{\mathcal{M}}$ satisfies $\lambda_{\mathcal{M}} \coloneqq \lambda_{\min}^{1/2}(\overline{\mathcal{M}}^\top \overline{\mathcal{M}}) > 0$.

This assumption is central to the analysis, and we believe it is reasonable: we are guaranteed that it holds if the system is controllable and either $A$ or $B$ has full column rank—see Appendix B.5.

To approximate the conditional expectations (18), (19) from finite samples, we define another expanded function class

$$\mathscr{H}_{\mathrm{op}} \coloneqq \{M f(\cdot) \mid f \in \mathscr{F}, M \in \mathbb{R}^{d_\mathbf{x} \times d_\mathbf{x}}, \|M\|_{\mathrm{op}} \leq \Psi_\star^3\},$$

and use $(\widehat{M}_k)$ and $\widehat{\mathcal{M}}$ to denote plugin estimates of $(M_k)$ and $\mathcal{M}$, respectively, constructed from $\widehat{A}$ and $\widehat{B}$. Here, the subscript "op" subscript on $\mathscr{H}_{\mathrm{op}}$ abbreviates "on-policy".

Next, given a state decoder $\hat{f}_t$ for time $t$ and $k \in [\kappa]$, we define

$$\widehat{\phi}_{t,k}(h, \mathbf{y}_{0:t}, \mathbf{y}_{t+k}) \coloneqq \widehat{M}_k\left(h(\mathbf{y}_{t+k}) - \widehat{A}^k h(\mathbf{y}_t) - \widehat{A}^{k-1} \widehat{B} \widehat{K} \hat{f}_t(\mathbf{y}_{0:t})\right) \quad \text{for } h \in \mathscr{H}_{\mathrm{op}}.$$

With this and the $2 n_{\mathrm{op}}$ trajectories $\{(\mathbf{y}_\tau^{(i)}, \boldsymbol{\nu}_\tau^{(i)})\}_{1 \leq i \leq 2 n_{\mathrm{op}}}$ gathered in **Step 1** above, we obtain $\hat{h}_t$ by solving the following two-step regression:

$$\hat{h}_{t,k} \in \underset{h \in \mathscr{H}_{\mathrm{op}}}{\arg\min} \sum_{i=1}^{n_{\mathrm{op}}} \left\| \widehat{\phi}_{t,k}(h, \mathbf{y}_{0:t}^{(i)}, \mathbf{y}_{t+k}^{(i)}) - \boldsymbol{\nu}_{t:t+k-1}^{(i)} \right\|^2, \quad \forall k \in [\kappa], \tag{21}$$

$$\hat{h}_t \in \underset{h \in \mathscr{H}_{\mathrm{op}}}{\arg\min} \sum_{i=n_{\mathrm{op}}+1}^{2 n_{\mathrm{op}}} \left\| \widehat{\mathcal{M}} \cdot \left(h(\mathbf{y}_{t+1}^{(i)}) - \widehat{A} \cdot h(\mathbf{y}_t^{(i)}) - \widehat{B} \widehat{K} \cdot \hat{f}_t(\mathbf{y}_{0:t}^{(i)})\right) - \widehat{\Phi}_t(\mathbf{y}_{0:t+\kappa}^{(i)}) \right\|^2, \tag{22}$$

$$\text{where} \quad \widehat{\Phi}_t(\mathbf{y}_{0:t+\kappa}) \coloneqq [\widehat{\phi}_{t,1}(\hat{h}_{t,1}, \mathbf{y}_{0:t}, \mathbf{y}_{t+1})^\top, \ldots, \widehat{\phi}_{t,\kappa}(\hat{h}_{t,\kappa}, \mathbf{y}_{0:t}, \mathbf{y}_{t+\kappa})^\top]^\top \in \mathbb{R}^{(1+\kappa)\kappa d_\mathbf{u}/2}. \tag{23}$$

We see that the first regression approximates (18), while the second approximates (19). We can now state the guarantee for Phase III.

**Theorem 4.** *Suppose* $\varepsilon_{\mathrm{id}}^2 \leq \breve{\mathcal{O}}((\ln|\mathscr{F}| + d_\mathbf{x}^2) n_{\mathrm{op}}^{-1})$. *If we set* $\bar{b}^2 = \Theta_\star((d_\mathbf{x} + d_\mathbf{u}) \ln(n_{\mathrm{op}}))$, $r_{\mathrm{op}} = \Psi_\star^3$, *and* $\sigma^2 = \breve{\mathcal{O}}(\lambda_{\mathcal{M}})$, *we are guaranteed that for any* $\delta \in (0, 1/e]$, *with probability at least* $1 - \mathcal{O}(\kappa T \delta)$,

$$\mathbb{E}_{\widehat{\pi}}\left[ \max_{1 \leq t \leq T} \|\hat{f}_t(\mathbf{y}_{0:t}) - f_\star(\mathbf{y}_t)\|^2 \right] \leq \mathcal{O}_\star\left( \frac{\lambda_{\mathcal{M}}^{-2}}{\sigma^4} \cdot T^3 \kappa^2 (d_\mathbf{x} + d_\mathbf{u})^4 \cdot \frac{(d_\mathbf{x}^2 + \ln|\mathscr{F}|) \ln^5(n_{\mathrm{op}}/\delta)}{n_{\mathrm{op}}} \right). \tag{24}$$

To obtain Theorem 1, we combine Theorem 3 and Theorem 4, then appeal to Theorem 7 (Appendix F), which bounds the policy suboptimailty in terms of regression errors. Finally, we set $\sigma \propto \varepsilon$ so that the suboptimality due to adding the Gaussian noise $(\boldsymbol{\nu}_t)_{0 \leq t \leq T}$ is low. See Appendix J for details.

# 3 Discussion

We introduced RichID, a new algorithm for sample-efficient continuous control with rich observations. We hope that our work will serve as a starting point for further research into sample-efficient continuous control with nonlinear observations, and we are excited to develop the techniques we have presented further, both in theory and practice. To this end, we list a few interesting directions and open questions for future work.

- While our results constitute the first polynomial sample complexity guarantee for the RichLQR, the sample complexity can certainly be improved. An important problem is to characterize the fundamental limits of learning in the RichLQR and design algorithms to achieve these limits, which may require new techniques. Of more practical importance, however, is to remove various technical assumptions used by RichID. We believe the most important assumptions to remove are (I) the assumption that the open-loop system is stable (Assumption 6), which is rarely satisfied in practice; and (II) the assumption that process noise is Gaussian, which is currently used in a rather strong sense to characterize the Bayes optimal solutions to the regression problems solved in RichID.

- RichID-CE is a model-based reinforcement learning algorithm. We are excited at the prospect of expanding the family of algorithms for RichLQR to include provable model-free and direct policy search-based algorithms. It may also be interesting to develop algorithms with guarantees for more challenging variants of the RichLQR, including regret rather than PAC-RL, and learning from a single trajectory rather than multiple episodes.

- Can we extend our guarantees to more rich classes of latent dynamical systems? For example, in practice, rather than assuming the latent system is linear, it is common to assume that it is *locally linear*, and apply techniques such as iterative LQR [44].

**Related work.** Our model and approach are related to the literature on Embedding to Control (E2C), and related techniques [44, 2, 13, 22, 36, 7] (see also [23]). At a high level, these approaches learn a decoder that maps images down to a latent space, then performs simple control techniques such as iterative LQR (iLQR) in the latent space (Watter et al. [44] is a canonical example). These approaches are based on heuristics, and do not offer provable sample complexity guarantees to learn the decoder in our setting.

Our work is also related to recent results on rich observation reinforcement learning with discrete actions [16]. We view our model as the control-theoretic analog of the *Block MDP* model studied by Du et al. [8], Misra et al. [28], in which a latent state space associated with a discrete Markov Decision Process is decodable from rich observations. However, our RichLQR setting is quite different technically due to the continuous nature of the latent space, and the results and techniques are incomparable. In particular, discretization approaches immediately face a curse-of-dimensionality phenomenon and do not yield tractable algorithms. Interestingly, our setting does not appear to have low Bellman rank in the sense of Jiang et al. [16].

A recent line of work [29, 33, 11] gives non-asymptotic system identification guarantees for a simple class of "generalized linear" dynamical systems. These results address a non-linear dynamic system, but are incomparable to our own as the non-linearity is known and the state is directly observed. Our results also are related to the LQG problem, which is a special case of (3) with linear observations; recent work provides non-asymptotic guarantees [24, 39, 20]. These results show that linear classes do not encounter the sample complexity barrier exhibited by Theorem 5.

Finally, we mention two concurrent works which consider similar settings. First, [12] give guarantees for a simpler problem in which we observe a linear combination of the latent state and a nonlinear nuisance parameter, and where there is no noise. Second, Dean and Recht [6] (see also Dean et al. [7]) give sample complexity guarantees for a variant of the our setting in which there is no system noise, and where $\mathbf{y}_t = g_\star(C\mathbf{x}_t)$, where $C \in \mathbb{R}^{p \times d_\times}$ and $g_\star : \mathbb{R}^p \to \mathbb{R}^{d_\mathbf{y}}$ is a smooth function. They provide a nonparametric approach which scales exponentially in the dimension $p$. Compared to this result, the main advantage of our approach is that it allows for general function approximation; that is, we allow for arbitrary function classes $\mathscr{F}$, and our results depend only on the capacity of the class under consideration. In terms of assumptions, the addition of the $C$ matrix allows for maps that (weakly) violate the perfect decodability assumption; we suspect that our results can be generalized in this fashion. Likewise, we believe that our stability assumption (i.e. $\rho(A) < 1$) can be removed in the absence of system noise (system noise is one of the primary technical challenges we overcome).

## Acknowledgments and Disclosure of Funding

This work was done while ZM was an intern at Microsoft Research. DF acknowledges the support of NSF TRIPODS grant #1740751. MS was supported by an Open Philanthropy AI Fellowship. AR acknowledges the support of ONR awards #N00014-20-1-2336 and #N00014-20-1-2394

## Broader Impact

There is potential for research into the RichLQR setting, or more generally perception-based control, to have significant societal impact. Perception-based control systems are already being deployed in applications such as autonomous driving and aerial vehicles where algorithmic errors can have catastrophic consequences. Unfortunately, there has been little research into the theoretical foundations of such systems, and so the methods being deployed do not enjoy the formal guarantees that we should demand for high-stakes applications. Thus, we are hopeful that with a principled understanding of the foundations of perception-based control, which we pursue here, we will develop the tools and techniques to make these systems safe, robust, and reliable.

## Footnotes

[1] Our results do not depend on $d_\mathbf{y}$, and in fact do not even require that $\mathbf{y}$ belongs to a vector space.

[2] We remark that $f_\star$ is typically referred to as an *encoder* rather than a decoder in the autoencoding literature.

[3]See Theorem 1a in Appendix J for the full theorem statement.

[4]Here, $P_\infty$ solves the DARE ((DARE) in Appendix A.2), and $K_\infty$ is the optimal infinite horizon controller.

[5]The controller $S_{\mathrm{id}} K_\infty$ attains the same performance on $(A_{\mathrm{id}}, B_{\mathrm{id}})$ as $K_\infty$ on $(A, B)$

[6]We make this reasoning precise in the proof of Theorem 1.

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
