[Supplementary Material]

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

_{\text{id}}, r_{\text{op}}$ `// radius for sets` $\mathscr{H}_{\text{id}}$ `and` $\mathscr{H}_{\text{op}}$`.`
    $\sigma^2$ `// exploration variance.`
    $\bar{b}$ `// clipping parameter for decoders.`
3: **Phase I** `// learn a coarse decoder (see Section 2.1)`
4: Set $\hat{f}_{\text{id}} \leftarrow \text{GetCoarseDecoder}(n_{\text{id}}, \kappa_0, \kappa, r_{\text{id}})$. `// Algorithm 3`
5: **Phase II** `// learn system's dynamics and cost (see Section 2.2)`
6: Set $(\widehat{A}_{\text{id}}, \widehat{B}_{\text{id}}, \widehat{\Sigma}_{w,\text{id}}, \widehat{Q}_{\text{id}}) \leftarrow \text{SysID}(\hat{f}_{\text{id}}, n_{\text{id}}, \kappa_0, \kappa)$. `// Algorithm 4`
7: **Phase III** `// compute optimal policy (see Section 2.3 and Appendix H)`
8: Set $\widehat{\pi} \leftarrow \text{ComputePolicy}(\widehat{A}_{\text{id}}, \widehat{B}_{\text{id}}, \widehat{\Sigma}_{w,\text{id}}, \widehat{Q}_{\text{id}}, R, n_{\text{op}}, \kappa, \sigma^2, T, \bar{b}, r_{\text{

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

[7]http://pytorch.org/

[8]https://scikit-learn.org/stable/

[9]Proposition F.1 and Proposition F.2 are immediate consequences of Lemma F.1, proven in Appendix F.

[10]To apply the proposition as stated in their paper, we use that $\rho(A_{\mathrm{cl},\infty}) \le \gamma_\infty$ by Gelfand's formula, and that their parameter $\tau(A_{\mathrm{cl},\infty}, \gamma_\infty)$ is bounded by $\alpha_\infty$.

[11]In the notation of Mania et al. [24], we can take $\rho \le \gamma_A$ and $\tau(A, \gamma_A) \le \alpha_A$.

[12] It is possible to get better dependence on $T$ by choosing different values for $\tau$ based on $\varepsilon$, but for the sake of simplicity we do not pursue this here.

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

# Contents of Appendix

# A  Organization, Notation, and Preliminaries

## A.1  Appendix Organization

This appendix is organized as follows.

- This section (Appendix A)—beyond this overview—contains additional notation and technical preliminaries omitted from the main body for space.
- Appendix B contains omitted details for the main algorithm (RichID-CE), including pseudocode with parameter values instantiated precisely (Appendix B.2), an overview of the initial state learning phase (Appendix B.3), and extensions and additional discussion of assumptions (Appendix B.4, Appendix B.5).
- Appendix C contains a validation experiment for Phases I and II of RichID-CE.
- Appendix D contains a formal statement and proof for the lower bound (Theorem 5).
- Appendix E and Appendix F contain basic technical tools for learning theory and linear control theory, respectively, which are invoked within the proof of the main theorem.
- All subsequent sections are devoted to proving our main theorem (Theorem 1):
  - Appendix G contains statements and proofs for all results concerning Phases I and II of RichID-CE, including Theorem 2 and Theorem 3.
  - Appendix H contains statements and proofs for Phase III, including Theorem 4.
  - Appendix J states the full version of the Theorem 1 (Theorem 1a), and shows how to deduce the proof from the results of Appendix G and Appendix H.

We remind the reader that each numbered theorem from the main body has an corresponding "full" version in the appendix, which we denote using the "a" suffix (e.g., the full version of Theorem 1 is Theorem 1a).

## A.2  Additional Preliminaries

**Policies, interaction model, and sample complexity.**  Formally, a policy $\pi$ for the setup (1) is a sequence of mappings $(\pi_t)_{t=0}^{T}$, where $\pi_t$ maps the observations $\mathbf{y}_0, \ldots, \mathbf{y}_t$ to an output control signal $\mathbf{u}_t$. In each round of interaction, the learner proposes a policy $\pi$ and observes a trajectory $\mathbf{u}_0, \mathbf{y}_0, \ldots, \mathbf{u}_T, \mathbf{y}_T$ where $\mathbf{u}_t = \pi_t(\mathbf{y}_0, \ldots, \mathbf{y}_t)$. We measure the sample complexity to learn an $\varepsilon$-optimal policy for $J_T$ in terms of the number of trajectories observed in this model. However, to simplify the description of our algorithm, we allow the learner to execute trajectories of length $2T + \mathcal{O}_\star(1)$ during the learning process, even though the objective is $J_T$. To avoid trivial issues caused by unidentifiability of the initial state $\mathbf{x}_0$, we define $J_T$ to measure cost on times $1, \ldots, T$. On the other hand, our rollouts begin at time $0$: the initial state is $\mathbf{x}_0$, and the first control input executed is $\mathbf{u}_0$.

**Cost functions.**  We assume that the control cost matrix $R > 0$ is known but, to avoid tying costs to the unknown latent representation $\mathbf{x}$, we assume that the state cost matrix $Q > 0$ is unknown. Instead, we assume that the learner has access to an additional *cost oracle* which on each trajectory at time $t$ reveals $\mathbf{c}_t \coloneqq \mathbf{x}_t^\top Q \mathbf{x}_t + \mathbf{u}_t^\top R \mathbf{u}_t$. For simplicity, we place the following mild regularity conditions on the cost matrices.

**Assumption 9.** The cost matrices $Q$ and $R$ satisfy $\lambda_{\min}(Q), \lambda_{\min}(R) \geq 1$.

This assumption can be made to hold without loss of generality whenever $Q, R > 0$ via rescaling.

**The DARE and infinite-horizon optimal control.**  Controllability (and more generally stabilizability) implies that there is a unique positive definite solution $P_\infty > 0$ to the *discrete algebraic Riccati equation* (DARE),

$$P = A^\top P A + Q - A^\top P B (R + B^\top P B)^{-1} B^\top P A, \qquad \text{(DARE)}$$

which characterizes the optimal cost function for the LQR problem in the infinite-horizon setting. Our analysis uses $P_\infty$, and our algorithms use the optimal infinite-horizon state feedback controller

$$K_\infty \coloneqq -(R + B^\top P_\infty B)^{-1} B^\top P_\infty A. \qquad \text{(A.1)}$$

When the state $\mathbf{x}_t$ is directly observed, the optimal infinite-horizon controller is the time-invariant feedback policy $u = K_\infty x$. Thus, the optimal infinite-horizon policy for RichLQR, given the exact decoder, is $\pi_\infty(y) = K_\infty f_\star(y)$. We use this controller as our benchmark. Our analysis also uses the closely related infinite-horizon covariance matrix

$$\Sigma_\infty := R + B^\top P_\infty B.$$

Our algorithm relies on *certainty equivalence*, in which we estimate $K_\infty$ by solving the DARE with plug-in estimates $\widehat{A}, \widehat{B}$ of $(A, B)$ to obtain a matrix $\widehat{P}$, and take $\widehat{K} := -(R + \widehat{B}^\top \widehat{P} \widehat{B})^{-1} \widehat{B}^\top \widehat{P} \widehat{A}$. For this, it will be convenient to define the following operator.

**Definition 1** (DARE operator). *We define the* DARE *operator as the operator which takes in matrices* $(A_0, B_0, R_0, Q_0)$ *with* $R_0, Q_0 \geq 0$, *and returns* $(P, K)$ *such that*

$$P = A_0^\top P A_0 + Q - A_0^\top P B_0 (R + B_0^\top P B_0)^{-1} B_0^\top P A_0,$$
$$K = -(R + B_0^\top P B_0)^{-1} B_0^\top P A_0.$$

**Strong stability.** We quantify stability via *strong stability* [4]. Intuitively, a matrix $X$ is *strongly stable* if its powers $X^n$ decay geometrically in a quantitative sense.

**Definition 2** (Strong stability). *A matrix* $X \in \mathbb{R}^{d_\mathsf{x} \times d_\mathsf{x}}$ *is said to be* $(\alpha, \gamma)$-*strongly stable if there exists* $S \in \mathbb{R}^{d_\mathsf{x} \times d_\mathsf{x}}$ *such that* $\|S\|_{\mathrm{op}} \|S^{-1}\|_{\mathrm{op}} \leq \alpha$ *and* $\|S^{-1} X S\|_{\mathrm{op}} \leq \gamma < 1$.

We frequently make use of the fact that if $X$ is $(\alpha, \gamma)$-strongly stable, then

$$\|X^n\|_{\mathrm{op}} = \|S(S^{-1} X S)^n S^{-1}\|_{\mathrm{op}} \leq \|S^{-1}\|_{\mathrm{op}} \|S\|_{\mathrm{op}} \|S^{-1} X S\|_{\mathrm{op}}^n \leq \alpha \gamma^n.$$

We let $(\alpha_A, \gamma_A)$ and $(\alpha_\infty, \gamma_\infty)$ be the strong parameters for $A$ and $A_{\mathrm{cl},\infty} := A + B K_\infty$ respectively. Under Assumption 4 and Assumption 6, we are guaranteed that $\gamma_A, \gamma_\infty < 1$ (see Proposition F.1 and Proposition F.2 for quantitative bounds). Finally, we recall from Assumption 7 that we assume the learner knows upper bounds $(\alpha_\star, \gamma_\star)$ such that $\alpha_A \vee \alpha_\infty \leq \alpha_\star$ and $\gamma_A \vee \gamma_\infty \leq \gamma_\star$.

### A.3 Additional Notation

**Asymptotic notation.** Lastly, we adopt standard non-asymptotic big-oh notation. For functions $f, g : \mathcal{X} \to \mathbb{R}_+$, we write $f = \mathcal{O}(g)$ if there exists some universal constant $C > 0$, which does not depend on problem parameters, such that $f(x) \leq C g(x)$ for all $x \in \mathcal{X}$. Our proofs also use the shorthand $f(x) \lesssim g(x)$ to denote $f = \mathcal{O}(g)$. We use $\widetilde{\mathcal{O}}(\cdot)$ so suppress logarithmic dependence on system parameters, time horizon, and dimension. We use $\mathcal{O}_\star(\cdot)$ to suppress polynomial factors in $\alpha_\star, \gamma_\star^{-1}, (1 - \gamma_\star)^{-1}, \Psi_\star, L$, and $\sigma_{\min}^{-1}(\mathcal{C}_\kappa)$, and all logarithmic factors except for $\ln|\mathscr{F}|$ and $\ln(1/\delta)$. We write $f = \Omega_\star(g)$ if $f(x) \geq C g(x)$ for all $x \in \mathcal{X}$, where $C$ is a *sufficiently large* constant whose value is polynomial in the same parameters. Lastly, we write $f = \breve{\mathcal{O}}(g)$ if $f(x) \leq c g(x)$ for all $x \in \mathcal{X}$, where $c = \mathrm{poly}(\gamma_\star(1 - \gamma_\star), \alpha_\star^{-1}, \Psi_\star^{-1}, L^{-1}, \sigma_{\min}(\mathcal{C}_\kappa))$ is a *sufficiently small* constant.

**General notation.** For a vector $x \in \mathbb{R}^d$, we let $\|x\|$ denote the euclidean norm and $\|x\|_\infty$ denote the element-wise $\ell_\infty$ norm. We let $\|x\|_A = \sqrt{x^\top A x}$ for $A \geq 0$. For a matrix $A$, we let $\|A\|_{\mathrm{op}}$ denote the operator norm. If $A$ is symmetric, we let $\lambda_{\min}(A)$ denote the minimum eigenvalue. For a potentially asymmetric matrix $A \in \mathbb{R}^{d \times d}$, we let $\rho(A) := \max\{|\lambda_1(A)|, \ldots, |\lambda_d(A)|\}$ denote the spectral radius. For a symmetric matrix $M \in \mathbb{R}^d$, $(M)_+$ denotes the result of thresholding all negative eigenvalues to zero, and we let $\lambda_1(M), \ldots, \lambda_d(M)$ denote the eigenvalues of $M$, sorted in decreasing order. Similarly, for a matrix $A \in \mathbb{R}^{d_1 \times d_2}$, we let $\sigma_1(A), \ldots, \sigma_{d_1 \wedge d_2}(A)$ denote the singular values of $A$, sorted in decreasing order, and use the shorthand $\sigma_{\min}(A) = \sigma_{d_1 \wedge d_2}(A)$. We let $\mathrm{vec}(A) \in \mathbb{R}^{d_1 d_2}$ be the vectorization of $A$. For matrices $A$ and $B$, we use $[A \,|\, B]$ or $[A; B]$ to denote their horizontal concatenation.

| Notation | Definition |
|---|---|
| **Basic Definitions** | |
| $(A, B)$ | system matrices |
| $q(\cdot\|x)$ | emissions model |
| $f_\star$ | true decoder |
| $\mathscr{F}$ | function class |
| $\mathbf{w}_t$ | process noise, which follows $\mathbf{w}_t \sim \mathcal{N}(0, \Sigma_w)$ |
| $\mathbf{x}_t$ | system state, which has initial state $\mathbf{x}_0 \sim \mathcal{N}(0, \Sigma_0)$ |
| $\mathbf{u}_t$ | control input |
| $\mathbf{y}_t$ | observations |
| $\mathbf{c}_t$ | observed cost, with $\mathbf{c}_t = \mathbf{x}_t^\top Q \mathbf{x}_t + \mathbf{u}_t^\top R \mathbf{u}_t$ |
| $R, Q$ | control cost, state cost |
| $T$ | horizon |
| $J_T$ | cost functional |
| $\pi_\infty$ | infinite horizon optimal policy |
| $\mathcal{C}_k$ | controllability matrix $\left(\left[ A^{k-1}B \mid \ldots \mid B \right]\right)$ |
| $K_\infty, P_\infty, \Sigma_\infty$ | infinite-horizon optimal controller, Lyapunov matrix, covariance matrix (DARE) |
| **System Parameter Bounds** | |
| $\Psi_\star$ | upper bound on system parameter norms (Assumption 7) |
| $(\alpha_\star, \gamma_\star)$ | upper bound on strong stability parameter (Assumption 7) |
| $\kappa$ | controllability index upper bound (Assumption 7) |
| $L$ | growth condition on $\mathscr{F}$ (Assumption 5) |
| $(\alpha_A, \gamma_A)$ | strong stability parameters for $A$ (Proposition F.2) |
| $(\alpha_\infty, \gamma_\infty)$ | strong stability parameters for $A + BK_\infty$ (Proposition F.1) |

Table 1: Summary of notation.

## B  Full Algorithm Description

Our algorithm is broken into three phases. In the first phase, we excite the system with Gaussian inputs, then solve a carefully designed regression problem to recover a decoder $\hat{f}$ whose performance is near-optimal under the steady state distribution. The choice of *what* regression problem to solve is rather subtle, and we show in Appendix B.1 that many naive approaches (e.g., predicting observations from inputs) fail. Our first key contribution is to show that an approach based on predicting *inputs from observations* succeeds under appropriate assumptions.

The second phase of our algorithm estimates the dynamics matrices $(A, B)$ and certain other system parameters using our learned decoder's prediction $\hat{f}(\mathbf{y}_t)$ as a plug-in estimate for the system state $\mathbf{x}_t$. We then use these estimates to synthesize a near-optimal linear controller $\widehat{K}$. The analysis here is rather straightforward, albeit somewhat technical due to the misspecification error caused by the inexact state estimates.

**Key challenge: Trajectory mismatch.**  The third phase of our algorithm solves a major issue we call *trajectory mismatch*. Suppose for simplicity that $\widehat{K} = K_\infty$, i.e. we exactly recover the optimal controller in the second phase (in reality, we must account for approximation error). A tempting approach is to select $\mathbf{u}_t = K_\infty \hat{f}(\mathbf{x}_t)$, where $\hat{f}$ is the decoder learned in the first phase. Unfortunately, this decoder is only guaranteed to be accurate on the steady state distribution induced by the Gaussian inputs we use for the first phase; there is no guarantee that this decoder will be accurate on the state distribution induced by the policy above. Indeed, this is an instance of a common technical issue in statistical learning: In general, given a function $\hat{f}$ such that $\mathbb{E}_P \| \hat{f}(x) - f^\star(x) \|^2 \leq \varepsilon$ for a distribution $P$, we have no guarantee that $\mathbb{E}_Q \| \hat{f}(x) - f^\star(x) \|^2 \leq \varepsilon$ for a different distribution $Q$ unless we put strong structural assumptions on either $P/Q$ or the function class $\mathcal{F}$. Since we do not make such assumptions, we solve this problem by learning a new decoder. This is where our work departs from recent efforts such as [6], who—by working with nonparametric classes which incur exponential sample complexity—learn a decoder which *uniformly* approximates $f_\star$; such an approach does not succeed in the general setting we consider here.

At this point, the challenge we face is how to learn a new decoder $\hat{f}$ that approximates $f_\star$ on trajectories induced by playing $\widehat{\pi}(\mathbf{y}_t) = K_\infty \hat{f}(\mathbf{y}_t)$. In particular, the foundational performance difference lemma [17] implies that it suffices to ensure that

$$\sum_{t=1}^{T} \mathbb{E}_{\widehat{\pi}}\left[\|\hat{f}(\mathbf{y}_t) - f_\star(\mathbf{y}_t)\|^2\right] \le \varepsilon. \tag{B.1}$$

This presents a clear chicken-and-egg problem: how do we ensure that $\hat{f}$ enjoys (B.1) on its *own* induced policy $\widehat{\pi}$?

**Our solution: Iterative decoding.** We address this issue by iteratively learning a sequence of time-dependent decoders $\{\hat{f}_t\}_{t=1}^T$. For each iteration $1 \le t \le T$ we predict $\mathbf{x}_t$ with $\hat{f}_t(\mathbf{y}_t)$, where $\hat{f}_t$ is a decoder learned at the previous iteration, and follow the induced policy $\widehat{\pi}_t(\mathbf{y}_t) = K_\infty \hat{f}_t(\mathbf{y}_t)$. We then estimate $\hat{f}_{t+1}$ by learning to predict $\mathbf{x}_{t+1}$ under the trajectory induced by playing $\mathbf{u}_1 = \widehat{\pi}_1(\mathbf{y}_1), \dots, \mathbf{u}_t = \widehat{\pi}_t(\mathbf{y}_t)$. The important point here is that the induced distribution for $\mathbf{x}_t$ does not depend on $\hat{f}_t$, only on $\hat{f}_1, \dots, \hat{f}_{t-1}$, thereby solving the chicken-and-egg problem.

A major technical challenge is ensuring that this iterative decoding procedure does not lead to errors which compound exponentially in the horizon $T$; this issue can easily arise if the misspecification error for the regression problem we solve to learn $\hat{f}_t$ depends on the quality of the previous decoders $\hat{f}_1, \dots, \hat{f}_{t-1}$. To solve this issue, we work with another carefully designed regression problem. The key idea is to roll in with the policies $\widehat{\pi}_1, \dots, \widehat{\pi}_{t-1}$, but roll out with purely Gaussian inputs for steps $\tau = t, t + 1, \dots$. This allows us to set up a regression problem which is *well-specified* and enjoys the advantages of Gaussianity, while remaining valid under the trajectory induced by $\{\widehat{\pi}_t\}$. The analysis for this phase is quite technical due to the inexact estimates from the first two phases. Showing that the indirect regression problems we solve eventually lead to a good predictor for the state requires substantial effort.

Before presenting the full RichID algorithm, we first provide additional details on the high-level idea behind our method and show where many naive approaches fail.

## B.1 Predicting Inputs from Outputs: The Bayes Regression Function

At the core of our algorithm is a simple but indispensible identity for the Bayes predictor that arises when we aim to predict control inputs $\mathbf{u}$ from observations $\mathbf{y}$ in the RichLQR model. As a motivating example, let a time $\tau \ge 1$ be fixed, suppose we take Gaussian control inputs $\mathbf{u}_{1:\tau} := (\mathbf{u}_1^\top, \dots, \mathbf{u}_\tau^\top)^\top \sim \mathcal{N}(0, I_{\tau d_\mathbf{u}})$, and consider the resulting state $\mathbf{x}_{\tau+1}$. Suppose that our goal is to estimate $f_\star$ with expected $L_2$ error under the marginal distribution of $\mathbf{x}_{\tau+1}$. That is, we wish to ensure

$$\mathbb{E}\left[\|\hat{f}(\mathbf{y}_{\tau+1}) - f_\star(\mathbf{y}_{\tau+1})\|^2\right] \le (\text{something small}), \tag{B.2}$$

where $\mathbb{E}[\cdot]$ denotes the expectation under the Gaussian inputs above.

**Attempt 1.** The natural strategy to attain (B.2) is to regress $\mathbf{y}_{\tau+1}$ to $\mathbf{u}_{1:\tau}$. For example, note that linearity of the dynamics ensures that there exists a matrix $M_\star \in \mathbb{R}^{d_\mathbf{x} \times \tau d_\mathbf{u}}$ such that $\mathbb{E}[\mathbf{x}_{\tau+1} \mid \mathbf{u}_{1:\tau}] = M_\star \mathbf{u}_{1:\tau}$. Thus, one could attempt the regression

$$\min_{f \in \mathscr{F}, M \in \mathbb{R}^{d_\mathbf{x} \times \tau d_\mathbf{u}}} \mathbb{E}\left[\|M\mathbf{u}_{1:\tau} - f(\mathbf{y}_{\tau+1})\|^2\right].$$

Unfortunately, there are too many degrees of freedom in this minimization problem: if $0 \in \mathscr{F}$, then the above is minimized with $f = 0$ and $M = 0$.

**Attempt 2.** A second attempt might be to hope that all $f \in \mathscr{F}$ are invertible, and try to solve a regression problem based on reconstructing the observations:

$$\min_{f^{-1}: f \in \mathscr{F}, M \in \mathbb{R}^{d_\mathbf{x} \times \tau d_\mathbf{u}}} \mathbb{E}\left[\|f^{-1}(M\mathbf{u}_{1:\tau}) - \mathbf{y}_{\tau+1}\|^2\right].$$

However, since $\mathbf{x}_{\tau+1} = M_\star \mathbf{u}_{1:\tau} + (\text{noise})$, passing through the nonlinearity $f^{-1}$ obviates any clear guarantees. In particular, this setup does not satisfy the usual first-order condition for regression with a well-specified model. A secondary issue is that even in the absence of system noise, this approach would likely incur dependence on the *observation* dimension $d_\mathbf{y}$.

**Our approach.** Our approach is to flip the input and target and regress $\mathbf{u}_{1:\tau}$ to $\mathbf{y}_{\tau+1}$. Specifically, we consider the regression:

$$\min_{g=Mf:f\in\mathscr{F},M\in\mathbb{R}^{\tau d_{\mathbf{u}}\times d_{\mathbf{x}}}} \mathbb{E}\left[\|g(\mathbf{y}_{\tau+1}) - \mathbf{u}_{1:\tau}\|^2\right]. \tag{B.3}$$

Let us motivate this approach and shed some light on the properties of the solution to this problem. Leveraging the perfect decodability assumption, one can show that $\mathbb{E}[\mathbf{u}_{1:\tau} \mid \mathbf{y}_{\tau+1}] = \mathbb{E}[\mathbf{u}_{1:\tau} \mid \mathbf{x}_{\tau+1} = f_\star(\mathbf{y}_{\tau+1})]$. Moreover, since $\mathbf{x}_{\tau+1}$ and $\mathbf{u}_{1:\tau}$ are *jointly Gaussian* (due to linearity of the dynamics and Gaussianity of the process noise), a simple calculation reveals that there exists a matrix $\widetilde{M}$ such that $\mathbb{E}[\mathbf{u}_{1:\tau} \mid \mathbf{x}_{\tau+1} = x] = \widetilde{M}x$. Hence,

$$\mathbb{E}[\mathbf{u}_{1:\tau} \mid \mathbf{y}_{\tau+1}] = \widetilde{M}f_\star(\mathbf{y}_{\tau+1}).$$

In particular, this implies that the unconstrained minimizer (i.e., over all measurable functions $g$) in (B.3) lies in the set $\{Mf : f \in \mathscr{F}\}$. Hence, since conditional expectations minimize square loss, we find:

> *Up to a set of measure zero, any minimizer of* (B.3) *must have the form* $g = \widetilde{M}f_\star$. *In other words, the population risk minimizer recovers* $f_\star$ *up to a linear transformation.*

Note that this crucially relies on Gaussianity, because while $\mathbb{E}[\mathbf{x}_{\tau+1} \mid \mathbf{u}_{1:\tau}]$ is linear in $\mathbf{u}_{1:\tau}$ for arbitrary mean-zero process noise, the same is not generally true when considering $\mathbb{E}[\mathbf{u}_{1:\tau} \mid \mathbf{x}_{\tau+1}]$. But with this strong assumption, we find that (B.3) allows us to recover $f_\star$ up to a global linear transformation. Of course, there are numerous remaining subtleties, including:

- Inverting $\widetilde{M}$ to recover $f_\star$.

- Identifying $\widetilde{M}$, especially since the learner does not know the system dynamics or noise covariance at first.

- Passing from population risk to empirical risk from finite samples.

How we address the above issues varies in different phases of the RichID-CE, and the remainder of this section supplies these details. But the fundamental principle—that we can solve empirical versions of (B.3) to recover linear transformations of $f_\star$—remains the core workhorse of RichID-CE.

**Remark 1** (Oracle Efficiency). *Consider the empirical version of* (B.3) *in which we gather $n$ trajectories and solve*

$$\min_{g=Mf,f\in\mathscr{F}} \sum_{i=1}^{n} \|g(\mathbf{y}_{\tau+1}^{(i)}) - \mathbf{u}_{1:\tau}^{(i)}\|^2,$$

*where the superscript $i$ denotes the $i$-th trajectory. Solving problems of this form is computationally efficient whenever we have a* regression oracle *for the induced class* $\{g = Mf \mid f \in \mathscr{F}, M \in \mathbb{R}^{d_{\mathbf{u}}\tau \times d_{\mathbf{x}}}\}$. *For many function classes of interest, such as linear functions and neural networks, solving regression over this class is no harder than regression over the original decoder class $\mathscr{F}$. We believe this is a reasonable and practical assumption.*

## B.2 Pseudocode

---

**Algorithm 2** RichID-CE (Full version with explicit parameter values)

---

1: **Inputs:**

$\varepsilon$ (sub-optimality), $T$ (horizon), $\mathscr{F}$ (decoder class), $(d_{\mathbf{x}}, d_{\mathbf{u}})$ (latent dimensions), $(\Psi_\star, \kappa, \alpha_\star, \gamma_\star)$ (system parameter upper-bounds), $\sigma^2$ (exploration parameter), $\lambda_{\mathcal{M}}$ (as in Assumption 8).

2: **Initialize:**

$n_{\mathrm{id}} = \Omega_\star\Big(\lambda_{\mathcal{M}}^{-2}\kappa^4(d_{\mathbf{x}} + d_{\mathbf{u}})^{16} T^3 \ln^{14}(\varepsilon^{-1}/\delta) \cdot \frac{\ln|\mathscr{F}|}{\varepsilon^6}\Big).$

$n_{\mathrm{op}} = \Omega_\star\Big(\lambda_{\mathcal{M}}^{-2}\kappa^3(d_{\mathbf{x}} + d_{\mathbf{u}})^{14} T^3 \ln^{13}(\varepsilon^{-1}/\delta) \cdot \frac{\ln|\mathscr{F}|}{\varepsilon^6}\Big).$

$\kappa_0 = \big\lceil (1 - \gamma_\star)^{-1} \ln\big(84 \Psi_\star^5 \alpha_\star^4 d_{\mathbf{x}}(1 - \gamma_\star)^{-2} \ln(10^3 \cdot n_{\mathrm{id}})\big)\big\rceil.$ **// burn-in time**

$r_\star = \sigma_{\min}(\mathcal{C}_\kappa)^{-3}(1 - \gamma_\star)^{-3}(4^3|\Psi_\star|^{21/2}\alpha_\star^6).$ **// upper-bound on** $\big(\Psi_\star\|S_{\mathrm{id}}\|_{\mathrm{op}}\|S_{\mathrm{id}}^{-1}\|_{\mathrm{op}}\big)^3$

$\bar{b} = \Theta_\star\big((d_{\mathbf{x}} + d_{\mathbf{u}})\ln(\varepsilon^{-1}/\delta)\big).$ **// clipping parameter for the decoders**

$\sigma^2 = \varepsilon^2/\bar{b}^2.$ **// exploration parameter**

3: **Phases I // learn a coarse decoder (see Section 2.1)**

4: Set $\hat{f}_{\mathrm{id}} \leftarrow \mathrm{GETCOARSEDECODER}(n_{\mathrm{id}}, \kappa_0, \kappa, \sqrt{\Psi_\star}).$ **// Algorithm 3**

5: **Phases II // learn system's dynamics and cost (see Section 2.2)**

6: Set $(\widehat{A}_{\mathrm{id}}, \widehat{B}_{\mathrm{id}}, \widehat{\Sigma}_{w,\mathrm{id}}, \widehat{Q}_{\mathrm{id}}) \leftarrow \mathrm{SYSID}(\hat{f}_{\mathrm{id}}, n_{\mathrm{id}}, \kappa_0, \kappa, \sqrt{\Psi_\star}).$ **// Algorithm 4**

7: **Phase III // compute optimal policy (see Section 2.3 and Appendix H)**

8: Set $\widehat{\pi} \leftarrow \mathrm{COMPUTEPOLICY}(\widehat{A}_{\mathrm{id}}, \widehat{B}_{\mathrm{id}}, \widehat{\Sigma}_{w,\mathrm{id}}, \widehat{Q}_{\mathrm{id}}, R, n_{\mathrm{op}}, \kappa, \sigma^2, T, \bar{b}, r_\star).$ **// Algorithm 5**

9: **return:** $\widehat{\pi}.$

---

**Algorithm 3** GETCOARSEDECODER: Phase I of RichID-CE (Section 2.1).

---

1: **Inputs:**

$n_{\mathrm{id}}$ **// sample size.**

$\kappa_0$ **// burn-in time index.**

$\kappa$ **// upper bound on the controllability index** $\kappa_\star$.

$r_{\mathrm{id}}$ **// upper bound on the matrix** $M$ **in the definition of** $\mathscr{H}_{\mathrm{id}}$.

2: Set $\mathscr{H}_{\mathrm{id}} \coloneqq \big\{ Mf(\cdot) \mid f \in \mathscr{F}, \; M \in \mathbb{R}^{\kappa d_{\mathbf{u}} \times d_{\mathbf{x}}}, \; \|M\|_{\mathrm{op}} \le r_{\mathrm{id}} \big\}.$

3: Set $\kappa_1 = \kappa_0 + \kappa.$

4: Gather $2n_{\mathrm{id}}$ trajectories by sampling control inputs $\mathbf{u}_0, \ldots, \mathbf{u}_{\kappa_1-1} \sim \mathcal{N}(0, I_{d_{\mathbf{u}}}).$

5: **Phase I: // Learn coarse decoder (see Section 2.1).**

6: Set $\hat{h}_{\mathrm{id}} = \arg\min_{h \in \mathscr{H}_{\mathrm{id}}} \sum_{i=1}^{n_{\mathrm{id}}} \|h(\mathbf{y}_{\kappa_1}^{(i)}) - \mathbf{v}^{(i)}\|^2,$ where $\mathbf{v} \coloneqq (\mathbf{u}_{\kappa_0}^\top, \ldots, \mathbf{u}_{\kappa_1-1}^\top)^\top.$

7: Set $\widehat{V}_{\kappa,\mathrm{id}}$ to be an orthonormal basis for top $d_{\mathbf{x}}$-eigenvectors of $\frac{1}{n_{\mathrm{id}}} \sum_{i=n_{\mathrm{id}}+1}^{2n_{\mathrm{id}}} \hat{h}_{\mathrm{id}}(\mathbf{y}_{\kappa_1}^{(i)}) \hat{h}_{\mathrm{id}}(\mathbf{y}_{\kappa_1}^{(i)})^\top$

8: Set $\hat{f}_{\mathrm{id}}(\cdot) \coloneqq \widehat{V}_{\kappa,\mathrm{id}}^\top \hat{h}_{\mathrm{id}}(\cdot).$ **// coarse decoder.**

9: **Return:** Coarse decoder $\hat{f}_{\mathrm{id}}.$

---

**Algorithm 4** SYSID: Phase II of RichID-CE (Section 2.2).

---

1: **Require:**
2:    Cost oracle to access the cost $\mathbf{c}_t$ at time $t \geq 1$.
3: **Inputs:**
    $\hat{f}_{\mathrm{id}}$ `// coarse decoder.`
    $n_{\mathrm{id}}$ `// sample size.`
    $\kappa_0$ `// burn-in time index.`
    $\kappa$ `// upper bound on the controllability index` $\kappa_\star$`.`

4: Set $\kappa_1 = \kappa_0 + \kappa$.
5: Gather $n_{\mathrm{id}}$ trajectories by sampling control inputs $\mathbf{u}_0, \ldots, \mathbf{u}_{\kappa_1-1} \sim \mathcal{N}(0, I_{d_\mathbf{u}})$.
6: **Phase II:** `// Recover system dynamics and cost (see Section 2.2).`
7: Set $(\widehat{A}_{\mathrm{id}}, \widehat{B}_{\mathrm{id}}) \in \arg\min_{(A,B)} \sum_{i=2n_{\mathrm{id}}+1}^{3n_{\mathrm{id}}} \| \hat{f}_{\mathrm{id}}(\mathbf{y}_{\kappa_1+1}^{(i)}) - A\hat{f}_{\mathrm{id}}(\mathbf{y}_{\kappa_1}^{(i)}) - B\mathbf{u}_{\kappa_1}^{(i)} \|^2$.
8: Set $\widehat{\Sigma}_{w,\mathrm{id}} = \frac{1}{n_{\mathrm{id}}} \sum_{i=2n_{\mathrm{id}}+1}^{3n_{\mathrm{id}}} (\hat{f}_{\mathrm{id}}(\mathbf{y}_{\kappa_1+1}^{(i)}) - \widehat{A}_{\mathrm{id}}\hat{f}_{\mathrm{id}}(\mathbf{y}_{\kappa_1}^{(i)}) - \widehat{B}_{\mathrm{id}}\mathbf{u}_{\kappa_1}^{(i)})^{\otimes 2}$, where $v^{\otimes 2} := vv^\top$.
9: Set $\widetilde{Q}_{\mathrm{id}} = \min_Q \sum_{i=2n_{\mathrm{id}}+1}^{3n_{\mathrm{id}}} \left( \mathbf{c}_{\kappa_1}^{(i)} - (\mathbf{u}_{\kappa_1}^{(i)})^\top R\mathbf{u}_{\kappa_1}^{(i)} - \hat{f}_{\mathrm{id}}(\mathbf{y}_{\kappa_1}^{(i)})^\top Q\hat{f}_{\mathrm{id}}(\mathbf{y}_{\kappa_1}^{(i)}) \right)^2$.
10: Set $\widehat{Q}_{\mathrm{id}} = \left( \frac{1}{2}\widetilde{Q}_{\mathrm{id}} + \frac{1}{2}\widetilde{Q}_{\mathrm{id}}^\top \right)_+$, where $(\cdot)_+$ truncates all negative eigenvalues to zero.
11: **Return:** System and cost matrices $(\widehat{A}_{\mathrm{id}}, \widehat{B}_{\mathrm{id}}, \widehat{\Sigma}_{w,\mathrm{id}}, \widehat{Q}_{\mathrm{id}})$.

---

### B.3 Overview for Learning the Initial State

In this section, we give an overview for how Phase III of RichID (Algorithm 5) learns a predictor for the initial state $\mathbf{x}_0$; this is an edge case which is not discussed in the main body due to space limitation, and comprises Line 19 through Line 25 in Algorithm 5. This discussion supplements Section 2.3 of the main body, and together these sections give constitute our high-level overview of Phase III.

If we ignore the clipping in (14), the state decoders $(\hat{f}_\tau)_{t \geq 2}$ follow the recursion

$$\hat{f}_{t+1}(\mathbf{y}_{0:t+1}) = \hat{h}_t(\mathbf{y}_{t+1}) - \widehat{A}\hat{h}_t(\mathbf{y}_t) + \widehat{A}\hat{f}_t(\mathbf{y}_{0:t}), \quad \text{for } t \geq 1,$$

which means that all the decoding error for any $t$ will depend on the error of the decoder $\hat{f}_1$ for the state $\mathbf{x}_1$. To ensure that $\hat{f}_1$ is accurate, we need to somehow learn to decode the inital state $\mathbf{x}_0$, which we recall is assumed to be distributed as $\mathcal{N}(0, \Sigma_0)$. The challenge here is that the covariance matrix $\Sigma_0$ is unknown, and we need to estimate it in order to "back out" the initial state through the approach in Appendix B.1. This is achieved by Line 19 through Line 25 of Algorithm 5, which we explain in detail below. Briefly, the idea is that since $\mathbf{x}_1 = A\mathbf{x}_0 + B\mathbf{u}_0 + \mathbf{w}_0$, to accurately predict $\mathbf{x}_1$ it suffices to have good predictors for $\mathbf{w}_0$ and $A\mathbf{x}_0$. We can learn a predictor for $\mathbf{w}_0$ in the same fashion as for all the other timesteps, and most of the work in Line 19 through Line 25 is to learn a regression function $\hat{f}_{A,0}$ that accurately predicts $A\mathbf{x}_0$.

To begin, in Line 19 we execute Gaussian control inputs $\mathbf{v}_\tau$ for $0 \leq \tau < \kappa$. We then proceed as follows.

**Line 21.** As we show in Theorem 8 (Appendix H), $\hat{h}_t(\mathbf{y}_{t+1}) - \widehat{A}\hat{h}_t(\mathbf{y}_t) - \widehat{B}(\widehat{K}\hat{f}_t(\mathbf{y}_{0:t}) + \mathbf{v}_t)$ approximates the system's noise $\mathbf{w}_t$, for $t \geq 0$. In particular, since $\hat{f}_0 \equiv 0$ by definition (Line 11), $\hat{h}_0(\mathbf{y}_1) - \widehat{A}\hat{h}_0(\mathbf{y}_0) - \widehat{B}\mathbf{v}_0$ approximates the noise $\mathbf{w}_0$. Since we have $\mathbf{x}_1 = A\mathbf{x}_0 + B\mathbf{u}_0 + \mathbf{w}_0$, it remains to get a good estimator for $A\mathbf{x}_0$. To this end, we observe that the predictor $\hat{h}_{\mathrm{ol},1}$ in Line 21 is (up to a generalization bound) equal to

$$\arg\min_{h \in \mathscr{H}_{\mathrm{op}}} \mathbb{E}_{\widehat{\pi}} \left[ \| h(\mathbf{y}_1) - \mathbf{w}_0 \|^2 \right],$$

which we show—under the realizability assumption—is given by

$$\mathbb{E}\left[ \mathbf{w}_0 \mid \mathbf{y}_1 \right] = \mathbb{E}\left[ \mathbf{w}_0 \mid \mathbf{x}_1 \right] = \Sigma_w(\sigma^2 BB^\top + \Sigma_w + A\Sigma_0 A^\top)^{-1}(A\mathbf{x}_0 + B\mathbf{v}_0 + \mathbf{w}_0), \tag{B.4}$$

where the last equality—like the rest of our Bayes characterizations—follows by Fact G.2.

---

**Algorithm 5** COMPUTEPOLICY: Phase III of RichID-CE

---

1: **Inputs:** $(\widehat{A}, \widehat{B}, \widehat{\Sigma}_w, \widehat{Q}, R)$ `// estimates for the system parameters and cost matrices.`

2: **Parameters:**

       $n_{\mathrm{op}}$ `// proportional to the sample size.`

       $\kappa$ `// upper-bound on the controllability index` $\kappa_*$`.`

       $\sigma^2$ `// exploration parameter.`

       $\bar{b}$ `// clipping parameter for the decoders.`

       $r_{\mathrm{op}}$ `// parameter to define the function class.`

3: Set $\mathscr{H}_{\mathrm{op}} = \left\{ Mf(\cdot) \mid f \in \mathscr{F}, \ M \in \mathbb{R}^{d_{\mathbf{x}} \times d_{\mathbf{x}}}, \ \|M\|_{\mathrm{op}} \le r_{\mathrm{op}} \right\}.$

4: Set $n_{\mathrm{init}} = n_{\mathrm{op}}.$

5: **Phase III:** `// Learn on-policy decoders (see Section 2.3 and Appendix B.3).`

6: Set $(\widehat{P}, \widehat{K}) \coloneqq \mathrm{DARE}(\widehat{A}, \widehat{B}, \widehat{Q}, R)$ (Definition 1).

7: **for** $k = 1, \ldots, \kappa$ **do**

8:     Set $\widehat{\mathcal{C}}_k = \left[ \widehat{A}^{k-1}\widehat{B} \mid \cdots \mid \widehat{B} \right].$

9:     Set $\widehat{M}_k \coloneqq \widehat{\mathcal{C}}_k^{\top} (\widehat{\mathcal{C}}_k \widehat{\mathcal{C}}_k^{\top} + \sigma^{-2} \sum_{i=1}^{k} \widehat{A}^{i-1} \widehat{\Sigma}_w (\widehat{A}^{i-1})^{\top})^{-1}.$

10: Set $\widehat{\mathcal{M}} = \left[ \widehat{M}_1^{\top} \mid \cdots \mid (\widehat{M}_\kappa \widehat{A}^{\kappa-1})^{\top} \right]^{\top}.$

11: Define $\hat{f}_0(y_0) = 0$ for all $y_0 \in \mathcal{Y}.$

12: **for** $t = 0, \ldots, T-1$ **do**

13:     Collect $2n_{\mathrm{op}}$ trajectories by executing the randomized control input $\mathbf{u}_\tau = \widehat{K}\hat{f}_\tau(\mathbf{y}_{0:\tau}) + \mathbf{v}_\tau,$

14:       for $0 \le \tau \le t$, and $\mathbf{u}_\tau = \mathbf{v}_\tau$, for $t < \tau < t + \kappa$, where $\mathbf{v}_\tau \sim \mathcal{N}(0, \sigma^2 I_{d_{\mathbf{u}}}).$

15:     **for** $k = 1, \ldots, \kappa$ **do**

16:       Set $\hat{h}_{t,k} \in \arg\min_{h \in \mathscr{H}_{\mathrm{op}}} \sum_{i=1}^{n_{\mathrm{op}}} \left\| \widehat{\phi}_{t,k}(h, \mathbf{y}_{0:t}^{(i)}, \mathbf{y}_{t+k}^{(i)}) - \mathbf{v}_{t:t+k-1}^{(i)} \right\|^2,$

         where $\widehat{\phi}_{t,k}(h, \mathbf{y}_{0:t}, \mathbf{y}_{t+k}) \coloneqq \widehat{M}_k \left( h(\mathbf{y}_{t+k}) - \widehat{A}^k h(\mathbf{y}_t) - \widehat{A}^{k-1}\widehat{B}\widehat{K}\hat{f}_t(\mathbf{y}_{0:t}) \right).$

17:     Set $\hat{h}_t \in \arg\min_{h \in \mathscr{H}_{\mathrm{op}}} \sum_{i=n_{\mathrm{op}}+1}^{2n_{\mathrm{op}}} \left\| \widehat{\mathcal{M}} \left( h(\mathbf{y}_{t+1}^{(i)}) - \widehat{A}h(\mathbf{y}_t^{(i)}) - \widehat{B}\widehat{K}\hat{f}_t(\mathbf{y}_{0:t}^{(i)}) \right) - \widehat{\phi}_t(\mathbf{y}_{0:t+\kappa}^{(i)}) \right\|^2,$

         where $\widehat{\phi}_t(\mathbf{y}_{0:t+\kappa}) \coloneqq \left[ \widehat{\phi}_{t,1}(\hat{h}_{t,1}, \mathbf{y}_{0:t}, \mathbf{y}_{t+1})^{\top}, \ldots, \widehat{\phi}_{t,\kappa}(\hat{h}_{t,\kappa}, \mathbf{y}_{0:t}, \mathbf{y}_{t+\kappa})^{\top} \right]^{\top}.$

18:     **if** $t = 0$ **then** `// Initial state learning phase (Appendix B.3).`

19:       Collect $2n_{\mathrm{init}}$ trajectories by executing the control input $\mathbf{u}_\tau = \mathbf{v}_\tau$, for $0 \le \tau < \kappa,$

20:         where $\mathbf{v}_\tau \sim \mathcal{N}(0, \sigma^2 I_{d_{\mathbf{u}}}).$

21:       Set $\hat{h}_{\mathrm{ol},1} \in \arg\min_{h \in \mathscr{H}_{\mathrm{op}}} \sum_{i=1}^{n_{\mathrm{init}}} \left\| h(\mathbf{y}_1^{(i)}) - \left( \hat{h}_0(\mathbf{y}_1^{(i)}) - \widehat{A}\hat{h}_0(\mathbf{y}_0^{(i)}) - \widehat{B}\mathbf{v}_0^{(i)} \right) \right\|^2.$

22:       Set $\widehat{\Sigma}_{\mathrm{cov}} \coloneqq \frac{1}{n_{\mathrm{init}}} \sum_{i=n_{\mathrm{init}}+1}^{2n_{\mathrm{init}}} \hat{h}_{\mathrm{ol},1}(\mathbf{y}_1^{(i)}) \hat{h}_{\mathrm{ol},1}(\mathbf{y}_1^{(i)})^{\top}.$

23:       Set $\tilde{h}_{\mathrm{ol},0} \in \arg\min_{h \in \mathscr{H}_{\mathrm{op}}} \sum_{i=n_{\mathrm{init}}+1}^{2n_{\mathrm{init}}} \left\| h(\mathbf{y}_0^{(i)}) - \hat{h}_{\mathrm{ol},1}(\mathbf{y}_1^{(i)}) \right\|^2.$

24:       Set $\hat{f}_{A,0}(\mathbf{y}_0) = \widehat{\Sigma}_w \widehat{\Sigma}_{\mathrm{cov}}^{-1} \tilde{h}_{\mathrm{ol},0}(\mathbf{y}_0).$

25:       Set $\tilde{f}_1(\mathbf{y}_{0:1}) = \hat{h}_0(\mathbf{y}_1) - \widehat{A}\hat{h}_0(\mathbf{y}_0) + \hat{f}_{A,0}(\mathbf{y}_0).$

26:     **else**

27:       Set $\tilde{f}_{t+1}(\mathbf{y}_{0:t+1}) = \hat{h}_t(\mathbf{y}_{t+1}) - \widehat{A}\hat{h}_t(\mathbf{y}_t) + \widehat{A}\hat{f}_t(\mathbf{y}_{0:t}).$

28:     Set $\hat{f}_{t+1}(\mathbf{y}_{0:t+1}) = \tilde{f}_{t+1}(\mathbf{y}_{0:t+1}) \mathbb{I}\{ \|\tilde{f}_{t+1}(\mathbf{y}_{0:t+1})\| \le \bar{b} \}.$

29:     Set controller $\widehat{\pi}_{t+1}(\mathbf{y}_{0:t+1}) = \widehat{K}\hat{f}_{t+1}(\mathbf{y}_{0:t+1}) + \mathbf{v}_{t+1}$, with $\mathbf{v}_{t+1} \sim \mathcal{N}(0, \sigma^2 I_{d_{\mathbf{u}}}).$

30: **Return:** Controller $\widehat{\pi} = (\widehat{\pi}_t)_{t=1}^{T}.$

**Line 22.** Now, given that $\hat{h}_{\mathrm{ol},1}(\mathbf{y}_1) \approx \mathbb{E}[\mathbf{w}_0 \mid \mathbf{y}_1]$, one can recognize that the matrix $\widehat{\Sigma}_{\mathrm{cov}}$ in Line 22 is an estimator for the matrix

$$\Sigma_w(\sigma^2 BB^\top + \Sigma_w + A\Sigma_0 A^\top)^{-1}\Sigma_w. \tag{B.5}$$

In particular, even though we cannot recover the covariance matrix $\Sigma_0$, the estimator $\widehat{\Sigma}_{\mathrm{cov}}$ gives a means to predict $A\mathbf{x}_0$, leading to an accurate decoder $\hat{f}_1$.

**Line 23.** Since $\hat{h}_{\mathrm{ol},1}(\mathbf{y}_1)$ accurately predicts $\mathbb{E}[\mathbf{w}_0 \mid \mathbf{y}_1]$ (whose closed form expression we recall is given by the RHS of (B.4)), the predictor $\tilde{h}_{\mathrm{ol},0}$ in Line 23 can be seen to approximate

$$\underset{h\in\mathscr{H}_{\mathrm{op}}}{\arg\min} \, \mathbb{E}_{\widehat{\pi}}\left[\|h(\mathbf{y}_0) - \Sigma_w(\sigma^2 BB^\top + \Sigma_w + A\Sigma_0 A^\top)^{-1}(A\mathbf{x}_0 + B\mathbf{v}_0 + \mathbf{w}_0)\|^2\right],$$

which (under realizability) is simply

$$\mathbb{E}[\Sigma_w(\sigma^2 BB^\top + \Sigma_w + A\Sigma_0 A^\top)^{-1}(A\mathbf{x}_0 + B\mathbf{v}_0 + \mathbf{w}_0) \mid \mathbf{x}_0] = \Sigma_w(\sigma^2 BB^\top + \Sigma_w + A\Sigma_0 A^\top)^{-1}A\mathbf{x}_0. \tag{B.6}$$

**Lines 24 and 25.** In light of (B.6) and the fact that $\widehat{\Sigma}_{\mathrm{cov}}$ is an estimator of the matrix in (B.5), we are guaranteed that $\widehat{\Sigma}_w \widehat{\Sigma}_{\mathrm{cov}}^{-1} \hat{h}_{\mathrm{ol},0}(\mathbf{y}_0)$ accurately predicts $A\mathbf{x}_0$, which motivates the updates in Lines 24 and 25.

## B.4 Extensions

**Relaxing the stability assumption.** We believe that our algorithm can be extended to so-called *marginally stable* systems, where $\rho(A)$ can be as large as 1 (rather than strictly less than 1). In such systems, there exist system-dependent constants $c_1, c_2 > 0$ for which $\|A^n\|_{\mathrm{op}} \le c_1 n^{c_2}$ for all $n$. In general, these constants may be large, and in the worst case $c_2$ may be as large as $d_{\mathbf{x}}$ (or, more generally, the largest Jordan block of $A$); see, e.g., Simchowitz et al. [37] for discussion. Nevertheless, if $c_1, c_2$ are treated as problem dependent constants, we can attain polynomial sample complexity. The majority of Algorithm 1 can remain as-is, but the analysis will replace the geometric decay of $A$ with the polynomial growth bound above. This will increase our sample complexity by a $\mathrm{poly}(c_1 T^{c_2})$ factor, where $T$ is the time horizon.

The only difficulty is that we can no longer directly identify the matrices $A$ and $B$ in Phase II. This is because our current analysis uses the *mixing* property of $A$, which entails that if $\rho(A) < 1$, then for $t$ sufficiently large, under purely Gaussian inputs $\mathbf{x}_t$ and $\mathbf{x}_{t+1}$ have similar distributions. This ensures that predictors learned at time $t$ are similar to those at time $t + 1$. However, this is no longer true if $\rho(A) = 1$. To remedy this, we observe that it is still possible to recover the controllability matrix $[B; AB; A^2 B; \ldots; A^{k-1}B]$ from the regression problem in Phase I up to a change of basis (see, e.g., Simchowitz et al. [38] for guarantees for learning such a matrix in the marginally stable setting). We can then recover the matrices $A$ and $B$ from the controllability matrix up to orthogonal transformation using the Ho-Kalman procedure (see Oymak and Ozay [30] or Sarkar et al. [32] for refined guarantees).

**Relaxing the controllability assumption.** If the system is not controllable, then we may not be able to recover the state exactly. Instead, we can recover the state up to the limiting-column space of the matrices $(\mathcal{C}_k)$, which is always attained for $k \le d_{\mathbf{x}}$. We can then use this to run a weaker controller (e.g., an observer-feedback controller) based on observations of the projection of the state onto this subspace.

**Other extensions.** The assumption on the growth rate for $\mathscr{F}$ can be replaced with the bound $\|f(y)\| \le L\max\{1, \|f_\star(y)\|^p\} \, \forall f \in \mathscr{F}$ for any $p \ge 1$, at the expense of degrading the final sample complexity.

## B.5 Invertibility of the $\mathcal{M}_0$-matrix

As discussed in the main body, the matrix $\mathcal{M}^\top \mathcal{M}$ is full rank whenever either $A$ or $B$ is full rank and the system is controllable. Indeed, if $\mathrm{rank}(B) = d_{\mathbf{x}}$, then $\overline{M}_1$, the first block of $\overline{\mathcal{M}}$, can be

checked to be invertible. If $(A, B)$ are controllable, then $\overline{M}_\kappa$ is full rank. Thus, if in addition, if $A$ is invertible, then the last block of $\overline{\mathcal{M}}$, $\overline{M}_\kappa A^{\kappa-1}$, is full rank, ensuring our desired assumption holds. We are interested to understand if there are other more transparent conditions under which our recovery guarantees hold. Assumption 8 has the following immediate implication:

**Lemma B.1.** Suppose Assumption 8 holds and let $\mathcal{M}_{\sigma^2}$ denote the value of the matrix $\mathcal{M}$ in (19) for noise parameter $\sigma^2$. Then, for all $\sigma^2$ sufficiently small, $\lambda_{\min}^{1/2}(\mathcal{M}_{\sigma^2}^\top \mathcal{M}_{\sigma^2}) \geq \lambda_{\mathcal{M}} \cdot \sigma^2/2 > 0$.

*Proof.* By continuity of the matrix inverse [31, Theorem 2.2], we have

$$\|\mathcal{M}_{\sigma^2}^\top \mathcal{M}_{\sigma^2}/\sigma^4 - \overline{\mathcal{M}}^\top \overline{\mathcal{M}}\|_{\mathrm{F}} \overset{\sigma \to 0}{\to} 0. \tag{B.7}$$

On the other hand, by [14, Corollary 6.3.8], we have

$$|\lambda_{\min}(\mathcal{M}_{\sigma^2}^\top \mathcal{M}_{\sigma^2}/\sigma^4) - \lambda_{\min}(\overline{\mathcal{M}}^\top \overline{\mathcal{M}})| \leq \|\mathcal{M}_{\sigma^2}^\top \mathcal{M}_{\sigma^2}/\sigma^4 - \overline{\mathcal{M}}^\top \overline{\mathcal{M}}\|_{\mathrm{F}}.$$

Combining this with (I.8) implies that $\lambda_{\min}^{1/2}(\mathcal{M}_{\sigma^2}^\top \mathcal{M}_{\sigma^2})/\sigma^2 \overset{\sigma \to 0}{\to} \lambda_{\min}^{1/2}(\overline{\mathcal{M}}^\top \overline{\mathcal{M}}) = \lambda_{\mathcal{M}}$, and so for all sufficiently small $\sigma$, we have $\lambda_{\min}^{1/2}(\mathcal{M}_{\sigma^2}^\top \mathcal{M}_{\sigma^2}) \geq \lambda_{\mathcal{M}} \cdot \sigma^2/2$. $\qquad \square$

## C  A Validation Experiment

In this section we provide a basic validation experiment for RichID using an instance of the RichLQR problem with four-dimensional latent linear dynamics and images as observations. Code for the experiment is available at `https://github.com/cereb-rl`.

**Dynamics and observation model.**  We consider a system in which the agent is a point mass in 2D space obeying modified Newtonian dynamics. The state $\mathbf{x} = [\mathbf{x}^{(p)}; \mathbf{x}^{(v)}] \in \mathbb{R}^4$ consists of concatenation of a 2D position vector $\mathbf{x}^{(p)}$ and a 2D velocity vector $\mathbf{x}^{(v)}$. The control $\mathbf{u} \in \mathbb{R}^2$ determines the agent's 2D-acceleration. We use an absolute frame of reference to measure the agent's position and velocity. Dynamics of the agent are given by

$$\underbrace{\begin{bmatrix} \mathbf{x}_{t+1}^{(p)} \\ \mathbf{x}_{t+1}^{(v)} \end{bmatrix}}_{\mathbf{x}_{t+1}} = \underbrace{\begin{bmatrix} 0.9I_2 & I_2 \\ 0 & 0.3I_2 \end{bmatrix}}_{A} \underbrace{\begin{bmatrix} \mathbf{x}_t^{(p)} \\ \mathbf{x}_t^{(v)} \end{bmatrix}}_{\mathbf{x}_t} + \underbrace{\begin{bmatrix} 0.5I_2 \\ I_2 \end{bmatrix}}_{B} \mathbf{u}_t + \mathbf{w}_t, \qquad \mathbf{w}_t \sim \mathcal{N}(0, I_4),$$

where $I_n$ is the identity matrix with $n$ rows and columns. These dynamics represent *dampened* Newtonian motion. We observe that $A$ is invertible and $\rho(A) = 0.9$. For the cost function, we choose

$$c(x, u) = x^\top x + \lambda u^\top u,$$

which corresponds to the case where $Q = I_4$ and $R = \lambda I_2$; here, $\lambda > 0$ controls the penalty for acceleration. Lastly, we initialize the agent in a randomly chosen starting state $\mathbf{x}_0 \sim \mathcal{N}(0, I_4)$ (which corresponds to $\Sigma_0 = I_4$).

The observations $\mathbf{y}$ consist of two RGB images of size $40 \times 40$ encoding the agent's position and velocity, respectively. We first restrict the position and velocity vectors to a $[-20, 20]^2$ grid by mapping $\mathbf{x}^{(p)}[1]$ to $\max(-20, \min(20, \mathbf{x}^{(p)}[1]))$ and so on. We then map these restricted position and velocity vectors to a green pixel in a $40 \times 40$ grid. This allows us to generate two images of size $40 \times 40 \times 3$, one for the position and one fo the velocity. We then add noise independently to each pixel by sampling from $\mathcal{N}(0, 0.1)$. To obtain the observation $\mathbf{y}$, the two images are concatenated along the channel dimension to generate a single observation of size $d_{\mathbf{y}} = 40 \times 40 \times 6$, which is 2400 times larger than the state dimension. We visualize the velocity component for an image generated in this fashion in Figure 1.

Figure 1: Randomly sampled image corresponding to the velocity component of the observation.

**Experimental setup and results.**  We evaluate the performance of Phase I and Phase II of RichID-CE in recovering the dynamics $(A, B)$. We set $\kappa = 5 \times d_{\mathbf{x}}$ and $\kappa_0 = 1$ in Algorithm 3 and Algorithm 4. We collect $3n_{\mathrm{id}}$ episodes by taking random actions from $\mathcal{N}(0, I_2)$. Using the first $n_{\mathrm{id}}$ episodes, we train a neural network model to predict actions $\mathbf{u}_{\kappa_0:\kappa_1-1}$ from $\mathbf{x}_{\kappa_1}$. For the regression model $\hat{h}_{\mathrm{id}}$, we use a two-layer convolutional neural network with Leaky ReLU nonlinearities. In the first layer, we apply 16 eight-by-eight kernels with stride 4. In the second layer, we apply 16 eight-by-eight kernels with stride 2. After the last convolution layer, we flatten the output image to project it to the required dimension $\kappa d_{\mathbf{u}}$ using a single linear layer. We train the model using Adam optimization with learning rate 0.001 and mini-batches of size 32 [19]. We use PyTorch 1.5 to implement this model, and initialize using the default PyTorch initialization.[7] We then perform dimensionality reduction as in Phase I to transform $\hat{h}_{\mathrm{id}}$ into a $d_{\mathbf{x}}$-dimensional decoder $\hat{f}_{\mathrm{id}}$. Finally, we recover $\widehat{A}_{\mathrm{id}}$ and $\widehat{B}_{\mathrm{id}}$ by solving the regression

problem on Line 7 in Phase II. We use the linear regression toolkit in the `scikit-learn` library to perform the this regression.[8]

Our main result for Phase II (Theorem 3) asserts that $\widehat{A}_{\mathrm{id}}$ and $\widehat{B}_{\mathrm{id}}$ approximately recover $A$ and $B$ up to a similarity transformation. While this similarity transformation is unknown to the algorithm, for the purposes of evaluation we compute the optimal similarity transformation using knowledge of the ground truth state. We find that using $n_{\mathrm{id}} = 30000$, the algorithm recovers the system matrices $A$ and $B$ up to element-wise absolute error at most $< 0.07$ (after similarity transformation).

# D Lower Bound for `RichLQR` Without Perfect Decodability

## D.1 Formal Statement of Lower Bound

In this section of the appendix we formally state and prove our sample complexity lower bound for RichLQR without perfect decodability. The protocol for the lower bound is as follows: The learning algorithm A accesses the system (3) through $n$ trajectories on which it can play any (possibly adaptively chosen) sequence of control inputs $\mathbf{u}_{0:T}$ and observe $\mathbf{y}_{0:T}$. At the end of this process, the algorithm outputs a decoder $\hat{f}_{\mathsf{A}}$, and the prediction performance of the decoder (at time $t = 1$) is measured under an arbitrary roll-in policy (chosen a-priori).

**Theorem 5** (Lower bound for `RichLQR` without perfect decodability.)**.** *Let $\mathbf{w}_t, \varepsilon_t \sim \mathcal{N}(0,1)$, let $n \geq n_0$, where $n_0$ is an absolute constant, and suppose we require that inputs are bounded so that $|\mathbf{u}_t| \leq 64 \ln^{1/2} n$. For every such $n$, there exists a function class $\mathscr{F}$ with $|\mathscr{F}| = 2$ and system with $d_{\mathbf{x}} = d_{\mathbf{u}} = d_{\mathbf{y}} = 1$ and $T = 1$ such that for learning algorithm A using only $n$ trajectories, and any roll-in policy $\pi$, we have*

$$\mathbb{E}_{\mathsf{A}} \mathbb{E}_{\pi} \left[ \left( \hat{f}_{\mathsf{A}}(\mathbf{y}_1) - f_{\star}(\mathbf{y}_1) \right)^2 \right] \geq \Omega(1) \cdot \frac{1}{\ln^{3/2} n}.$$

*Moreover, each $f \in \mathscr{F}$ is $\mathcal{O}(\ln^{1/2} n)$-Lipschitz and invertible, with $f'(y) \geq 1$ for all $y \in \mathbb{R}$.*

Theorem 5 shows that to learn a $\varepsilon$-suboptimal decoder under output noise for a particular function class $\mathscr{F}$ with $|\mathscr{F}| = 2$, any algorithm requires an exponential number of samples. We note however that since the Lipschitz parameter for the functions in the construction grows with $n$ (as $\ln^{1/2} n$), the construction does not rule out a sample complexity guarantee that is polynomial in $1/n$ but exponential in the Lipschitz parameter. Nonetheless, the algorithms we develop in this paper under the perfect decodability assumption enjoy polynomial dependence on both $1/n$ and the Lipschitz parameter, which the lower bound shows is impossible under unit output noise. We remark that the constraint that $|\mathbf{u}_t| \leq 64 \ln^{1/2} n$ can be weakened to $|\mathbf{u}_t| \leq C \ln^{1/2} n$ for any $C \geq 64$ at the cost of weakening the final lower bound to $\frac{1}{C \ln^{3/2} n}$. Finally, we remark that the lower bound only rules out learning a $\varepsilon$-optimal decoder, not an $\varepsilon$-optimal policy; such a lower bound may require a more sophisticated construction.

Beyond Theorem 5, an additional challenge for solving RichLQR without perfect decodability is that the optimal controller is no longer reactive: since the problem is partially observable, the optimal controller will in general depend on the entire history, which makes it difficult to characterize its performance and analyze the suboptimality of data-driven algorithms. We believe that developing more tractable models for RichLQR under weaker decodability assumptions is an important direction for future research.

## D.2 Additional Preliminaries

For an $L_2$-integrable function $f : \mathbb{R} \to \mathbb{R}$, we define the Fourier transform $\widehat{f}$ via

$$\widehat{f}(\omega) = \int e^{-i2\pi\omega x} f(x) dx.$$

For functions $f, g : \mathbb{R} \to \mathbb{R}$, we let $f * g$ denote their convolution, which is given by

$$(f * g)(x) = \int f(x - y) g(y) dy.$$

For a pair of distributions $P \ll Q$ with densities $p$ and $q$, we define

$$D_{\mathrm{KL}}(P \,\|\, Q) = \int p(x) \ln(p(x)/q(x)) dx$$

and

$$\chi^2(P \,\|\, Q) = \int \frac{(p(x) - q(x))^2}{q(x)} dx.$$

### D.3 Proof of Theorem 5

Throughout this proof we use $C$ to denote an absolute numerical constant whose value may change from line to line.

We begin the proof by instantiating the LQR parameters. We set $T = 1$, $d_{\mathbf{u}} = d_{\mathbf{x}} = d_{\mathbf{y}} = 1$, $\mathbf{w}_t \sim \mathcal{N}(0, 1)$ and $\varepsilon_t \sim \mathcal{N}(0, 1)$. We select $a = \frac{1}{2}$ (this choice is arbitrary) and $b = 1$. We assume that $\mathbf{x}_0$ is always initialized to the same value, and this value is known to the learner. The precise value will be specified shortly, but it will be chosen such that $\mathbf{y}_0$ reveals no information about the underlying instance. With the parameters above, the observation $\mathbf{y}_1$ follows the following data-generating process:

$$
\begin{aligned}
\mathbf{y}_1 &= f_\star^{-1}(\mathbf{x}_1) + \varepsilon_1 \\
\mathbf{x}_1 &= \mathbf{u}_0 + \tfrac{1}{2}\mathbf{x}_0 + \mathbf{w}_0.
\end{aligned}
\tag{D.1}
$$

Since $\mathbf{x}_0$ is known to the learner, we reparameterize the control inputs for the sake of notational compactness via $\mathbf{u}_0 := \mathbf{u}_0 - \frac{1}{2}\mathbf{x}_0$, so the data-generating process simplifies to

$$
\begin{aligned}
\mathbf{y}_1 &= f_\star^{-1}(\mathbf{x}_1) + \varepsilon_1 \\
\mathbf{x}_1 &= \mathbf{u}_0 + \mathbf{w}_0.
\end{aligned}
\tag{D.2}
$$

The basic observation underlying our lower bound is that the data-generating process (D.2) is an instance of the classical *error-in-variable regression* problem in the *Berkson* error model [26, 27, 34, 35]. To emphasize the similarity to the setting, we rebind the variables as $Y = \mathbf{y}_1$, $\varepsilon = \varepsilon_1$, $Z = \mathbf{x}_1$, $X = \mathbf{u}_0$, $W = \mathbf{w}_0$, and $m_\star = f_\star^{-1}$, so that Eq. (D.2) becomes

$$
\begin{aligned}
Y &= m_\star(Z) + \varepsilon \\
Z &= X + W.
\end{aligned}
\tag{D.3}
$$

We can interpret $X$ (the control $\mathbf{u}_0$) as a true covariate known to the learner, and $Z$ (the state $\mathbf{x}_1$) as an unobserved noisy version of this covariate obtained by adding the noise $W$. The noisy covariate is passed through the regression function $m_\star$, then the noise $\varepsilon$ is added, leading to the target variable $Y$ (the observation $\mathbf{y}_1$).

Ultra-slow $1/\ln n$-type rates appear in many variants of the error-in-variable regression problem [10, 26, 27], as well as the closely related nonparametric deconvolution problem [9]. Our lower bound is based on Theorem 2 of Meister [27], but with two important changes that add additional complications to the analysis. First, we ensure that the regression functions in our construction are invertible, so that the perfect decodability assumption holds in absence of noise, and second, our lower bound holds even for actively chosen covariates, since these correspond to control inputs chosen by the learner in the RichLQR problem.

Rather than constructing a decoder class $\mathscr{F}$ directly, it will be more convenient to construct a class of encoders $\mathscr{M}$ (so that $m_\star \in \mathscr{M}$), then take $\mathscr{F} = \left\{ m^{-1} \mid m \in \mathscr{M} \right\}$ to be the induced decoder class.

Let $0 < \alpha \le 1$, $\beta \ge 1$, and $\gamma > 0$ be parameters of the construction. We define the following functions:

$$
\begin{aligned}
r(z) &= \gamma z, \\
\phi(z) &= e^{-\frac{z^2}{2\beta^2}}, \\
\psi(z) &= \cos(4\pi\beta z), \\
h(z) &= \alpha\phi(z)\psi(z).
\end{aligned}
\tag{D.4}
$$

We consider two alternate regression functions: $m_0(z) := r(z) + h(z)$ and $m_1(z) := r(z) - h(z)$, and take $\mathscr{M} = \{m_0, m_1\}$. We define $f_i = m_i^{-1}$.

**Lemma D.1.** For $m \in \{m_0, m_1\}$, we have

$$
m'(z) \in [\gamma - 14\alpha\beta, \gamma + 14\alpha\beta].
$$

In light of this lemma, we will leave $\beta \ge 1$ free for the time being, but choose

$$
\alpha = \frac{1}{28\beta^2}, \quad \text{and} \quad \gamma = \frac{1}{\beta},
\tag{D.5}
$$

which ensures that

$$0 < \frac{1}{2\beta} \le m'(z) \le \frac{3}{2\beta}. \tag{D.6}$$

In particular, this implies that $m$ is $\frac{3}{2}$-Lipschitz and invertible (since $\beta \ge 1$).

We now specify the starting state as $\mathbf{x}_0 = \frac{1}{8\beta}$. This ensures that $\psi(\mathbf{x}_0) = \cos(\pi/2) = 0$, so that $m_0(\mathbf{x}_1) = m_1(\mathbf{x}_0)$, and consequently the observation $\mathbf{y}_0$ is statistically independent of the underlying instance.

Let $\mathbf{x}_0^{(i)}, \mathbf{u}_0^{(i)}, \mathbf{x}_1^{(i)}, \mathbf{y}_1^{(i)}$, and so forth denote the realizations of the sytem variables in the $i$th trajectory played by the learner, and let $S = (\mathbf{y}_0^{(1)}, \mathbf{u}_0^{(1)}, \mathbf{y}_1^{(1)}, \mathbf{u}_1^{(1)}), \ldots, (\mathbf{y}_0^{(n)} \mathbf{u}_0^{(n)}, \mathbf{y}_1^{(n)}, \mathbf{u}_1^{(n)})$ denote the observables collected throughout the entire learning process. For $i \in \{0, 1\}$, we let $\mathbb{P}_{S;i}$ denote the law of $S$ when $m_i$ is the true encoding function, and let $\mathbb{E}_i$ denote the expectation under $\mathbb{P}_{S;i}$. We also let $\mathbb{P}_{\mathbf{y}_1|\mathbf{u}_0;i}(\cdot \mid u)$ denote the law of $\mathbf{y}_1$ given $\mathbf{u}_0$ when $m_\star = m_i$ and $p_i(y \mid u)$ be the corresponding density (we suppress dependence on $\mathbf{x}_0$, which takes on the constant value $\frac{1}{8\beta}$ in both instances). Lastly, we let $\mathbb{E}_{\pi;i}$ denote the expectation over $(\mathbf{y}_0, \mathbf{u}_0, \mathbf{y}_1, \mathbf{u}_1)$ when we roll in with $\pi$ and $m_i$ is the underlying encoder.

Let $\hat{f}_{\mathsf{A}}(\cdot)$ be the decoder returned by $\mathsf{A}$, which we assume to be $\sigma(S)$-measurable. We first observe that since the roll-in policy has $|\mathbf{u}_t| \le \beta$ with probability 1, Lemma D.6 implies that

$$\max_{i \in \{0,1\}} \mathbb{E}_i \mathbb{E}_{\pi;i}\big[(\hat{f}_{\mathsf{A}}(\mathbf{y}_1) - f_i(\mathbf{y}_1))^2\big] \ge c \cdot \max_{i \in \{0,1\}} \mathbb{E}_i\Big[\int_{-1}^{1}(\hat{f}_{\mathsf{A}}(y) - f_i(y))^2 dy\Big],$$

meaning that going forward we can dispense with the roll-in policy and lower bound the simpler quantity on the right-hand side above. Now, let $P_i$ denote the density corresponding to the law $\mathbb{P}_{S;i}$. We can further lower bound the worst-case risk of $\mathsf{A}$ as

$$\max_{i \in \{0,1\}} \mathbb{E}_i\Big[\int_{-1}^{1}(\hat{f}_{\mathsf{A}}(y) - f_i(y))^2 dy\Big]$$

$$\ge \frac{1}{2}\Big[\mathbb{E}_0\Big[\int_{-1}^{1}(\hat{f}_{\mathsf{A}}(y) - f_0(y))^2 dy\Big] + \mathbb{E}_1\Big[\int_{-1}^{1}(\hat{f}_{\mathsf{A}}(y) - f_1(y))^2 dy\Big]\Big]$$

$$\ge \frac{1}{2}\int_{-1}^{1}\Big[\int_{\mathbb{R}^{4n}}\big[(\hat{f}_{\mathsf{A}}(y) - f_0(y))^2 + (\hat{f}_{\mathsf{A}}(y) - f_1(y))^2\big]\min\{P_0(S), P_1(S)\}dS\Big]dy$$

$$\ge \frac{1}{4}\int_{-1}^{1}(f_0(y) - f_1(y))^2 dy \cdot \int_{\mathbb{R}^{4n}}\min\{P_0(S), P_1(S)\}dS$$

$$\ge \frac{1}{4}\int_{-1}^{1}(f_0(y) - f_1(y))^2 dy \cdot \Big(1 - \frac{1}{2}\|P_0 - P_1\|_{L_1(\mathbb{R}^{4n})}\Big)$$

$$= \frac{1}{4}\int_{-1}^{1}(f_0(y) - f_1(y))^2 dy \cdot (1 - D_{\mathrm{TV}}(\mathbb{P}_{S;0} \| \mathbb{P}_{S;1})).$$

If we choose $\beta = 64\ln^{1/2} n$ then our key technical lemma, Lemma D.2, implies that $D_{\mathrm{TV}}(\mathbb{P}_{S;0} \| \mathbb{P}_{S;1}) = o(1)$. Lemma D.7 further implies that $\int_{-1}^{1}(f_0(y) - f_1(y))^2 dy \ge \frac{1}{8}\alpha^2\beta$, so that when $n$ is sufficiently large we have

$$\max_{i \in \{0,1\}} \mathbb{E}_i\Big[\int_{-1}^{1}(\hat{f}_{\mathsf{A}}(y) - f_i(y))^2 dx\Big] \ge c \cdot \alpha^2\beta = c\ln^{-3/2} n.$$

$\square$

## D.4 Proofs for Supporting Lemmas

*Proof of Lemma D.1.* We calculate that for $m \in \{m_0, m_1\}$, we have

$$m'(z) = \gamma \pm \alpha\Big(\frac{z}{\beta^2}e^{-\frac{z^2}{2\beta^2}}\cos(2\beta z) + 4\pi\beta e^{-\frac{z^2}{2\beta^2}}\sin(2\beta z)\Big).$$

Observe that $|\cos z|, |\sin z|, e^{-z^2} \le 1$, and

$$\Big|\frac{z}{\beta^2}e^{-\frac{z^2}{2\beta^2}}\Big| \le \frac{1}{\beta}\sup_z\Big|ze^{-\frac{z^2}{2}}\Big| \le \frac{1}{\beta e^{1/2}}.$$

It follows that

$$f'(z) \in \left[\gamma - \alpha\left(\frac{1}{\beta e^{1/2}} + 4\pi\beta\right), \gamma + \alpha\left(\frac{1}{\beta e^{1/2}} + 4\pi\beta\right)\right] \subseteq [\gamma - 14\alpha\beta, \gamma + 14\alpha\beta],$$

where we have used that $\beta \geq 1$. $\qquad\qquad \square$

**Lemma D.2.** If we choose $\beta = 64\ln^{1/2} n$, then for all $n \geq 3$ we have

$$D_{\mathrm{TV}}(\mathbb{P}_{S;0} \,\|\, \mathbb{P}_{S;1}) \leq Cn^{-4}.$$

*Proof of Lemma D.2.* To begin, we apply Pinsker's inequality:

$$D_{\mathrm{TV}}^2(\mathbb{P}_{S;0} \,\|\, \mathbb{P}_{S;1}) \leq \frac{1}{2} D_{\mathrm{KL}}(\mathbb{P}_{S;0} \,\|\, \mathbb{P}_{S;1}).$$

Let $o^{(j)} = (y_0^{(1)}, u_0^{(1)}, y_1^{(2)}, u_1^{(2)})$. We observe that the density $P_i(o^{(1)}, \ldots, o^{(n)})$ factorizes as

$P_i(o^{(1)}, \ldots, o^{(n)}) =$

$\prod_{j=1}^{n} p_{\mathbf{y}_0;i}(y_0^{(j)}) p_{\mathbf{u}_0^{(j)}}(u_0^{(j)} \mid o^{(1)}, \ldots, o^{(t-1)}, y_0^{(j)}) p_{\mathbf{y}_1|\mathbf{u}_0;i}(y_1^{(j)} \mid u_0^{(j)}) p_{\mathbf{u}_1^{(j)}}(u_1^{(j)} \mid o^{(1)}, \ldots, o^{(t-1)}, y_0^{(j)}, u_0^{(j)}, y_1^{(j)}),$

where $p_{\mathbf{y}_0;i}$ is the density for $\mathbf{y}_0$ under instance $i$, $p_{\mathbf{u}_0^{(j)}}$ and $p_{\mathbf{u}_1^{(j)}}$ are the conditional densities for $\mathbf{u}_0^{(j)}$ and $\mathbf{u}_1^{(j)}$ given all preceding observations, and $p_{\mathbf{y}_1|\mathbf{u}_0;i}$ is the conditional density for $\mathbf{y}_1$ given $\mathbf{u}_0$ under instance $i$. The densities $p_{\mathbf{u}_0^{(j)}}$ and $p_{\mathbf{u}_1^{(j)}}$ do not depend on the instance $i$, nor does the density $p_{\mathbf{y}_0;i}$ (recall that the choice of starting state $\mathbf{x}_0 = \frac{1}{8\beta}$ guarantees $m_0(\mathbf{x}_0) = m_1(\mathbf{x}_0)$, so $\mathbf{y}_0 = \varepsilon_0$ in law for both instances). We conclude that the KL divergence telescopes as

$$D_{\mathrm{KL}}(\mathbb{P}_{S;0} \,\|\, \mathbb{P}_{S;1}) = \sum_{j=1}^{n} \mathbb{E}_0\Big[D_{\mathrm{KL}}\Big(\mathbb{P}_{\mathbf{y}_1|\mathbf{u}_0;0}(\cdot \mid \mathbf{u}_0^{(j)}) \,\|\, \mathbb{P}_{\mathbf{y}_1|\mathbf{u}_0;1}(\cdot \mid \mathbf{u}_0^{(j)})\Big)\Big]$$

$$\leq \sum_{j=1}^{n} \mathbb{E}_0\Big[\chi^2\Big(\mathbb{P}_{\mathbf{y}_1|\mathbf{u}_0;0}(\cdot \mid \mathbf{u}_0^{(j)}) \,\|\, \mathbb{P}_{\mathbf{y}_1|\mathbf{u}_0;1}(\cdot \mid \mathbf{u}_0^{(j)})\Big)\Big].$$

Since the algorithm satisfies $|\mathbf{u}_0^{(j)}|, |\mathbf{u}_1^{(j)}| \leq \beta$ almost surely, we can apply Lemma D.3 to each summand, which gives

$$D_{\mathrm{KL}}(\mathbb{P}_{S;0} \,\|\, \mathbb{P}_{S;1}) \leq Cn^{-9}.$$

$\qquad\qquad \square$

**Lemma D.3.** If we choose $\beta = 64\ln n$, then for all $n \geq 3$ and all $|u| \leq \beta$, we have

$$\chi^2\big(\mathbb{P}_{\mathbf{y}_1|\mathbf{u}_0;0}(\cdot \mid u) \,\|\, \mathbb{P}_{\mathbf{y}_1|\mathbf{u}_0;1}(\cdot \mid u)\big) \leq Cn^{-10}. \tag{D.7}$$

*Proof of Lemma D.3.* Recall that we let $p_i$ denote the conditional density for $\mathbb{P}_{\mathbf{y}_1|\mathbf{u}_0;i}(\cdot \mid u)$. Let $p_\varepsilon(\varepsilon) = e^{-\frac{1}{2}\varepsilon^2}$ denote the density of $\varepsilon$ and $p_{\mathbf{w}}(w) = e^{-\frac{1}{2}w^2}$ denote the density of $\mathbf{w}$. Observe that for each $i$, we have

$$p_i(y \mid u) = \frac{1}{\sqrt{2\pi}} \int p_\varepsilon(y - m_i(u + w)) p_{\mathbf{w}}(w) dw.$$

It follows that

$\chi^2\big(\mathbb{P}_{\mathbf{y}_1|\mathbf{u}_0;0}(\cdot \mid u) \,\|\, \mathbb{P}_{\mathbf{y}_1|\mathbf{u}_0;1}(\cdot \mid u)\big)$

$= \frac{1}{\sqrt{2\pi}} \int p_1^{-1}(y \mid u) \cdot \left| \int [p_\varepsilon(y - m_0(u + w)) - p_\varepsilon(y - m_1(u + w))] p_{\mathbf{w}}(w) dw \right|^2 dy.$

By Lemma D.4 (with $\eta = 1/5$), we have

$$p_i^{-1}(y \mid u) \leq 3^{1/2} \exp\left(\frac{(1 + 1/5)(y - \gamma u)^2}{2} + 5\right)$$

$$\leq 3^{1/2} \exp\left(\frac{(1 + 1/5)^2}{2} y^2 + \frac{5(1 + 1/5)}{2} \gamma^2 u^2 + 5\right).$$

Since $|u| \leq \beta$, $\gamma^2 u^2 \leq 1$, so we can further simplify to

$$p_i^{-1}(y \mid u) \leq C \cdot \exp\left(\frac{3}{4}y^2\right).$$

Consequently, we have

$$\chi^2\big(\mathbb{P}_{\mathbf{y}_1|\mathbf{u}_0;0}(\cdot \mid u) \,\|\, \mathbb{P}_{\mathbf{y}_1|\mathbf{u}_0;1}(\cdot \mid u)\big)$$

$$\leq C \int e^{\frac{3}{4}y^2} \left| \int [p_{\varepsilon}(y - m_0(u+w)) - p_{\varepsilon}(y - m_1(u+w))]p_{\mathbf{w}}(w)dw \right|^2 dy.$$

Using the Taylor series representation for $p_{\varepsilon}$, we have

$$p_{\varepsilon}(y - m_i(u+w)) = \sum_{k=0}^{\infty} \frac{1}{k!} p_{\varepsilon}^{(k)}(y)(-m_i(u+w))^k,$$

and so

$$\chi^2\big(\mathbb{P}_{\mathbf{y}_1|\mathbf{u}_0;0}(\cdot \mid u) \,\|\, \mathbb{P}_{\mathbf{y}_1|\mathbf{u}_0;1}(\cdot \mid u)\big)$$

$$\leq C \int e^{\frac{3}{4}y^2} \left| \sum_{k=0}^{\infty} \frac{1}{k!} p_{\varepsilon}^{(k)}(y) \int [(-f_0(u+w))^k - (-f_1(u+w))^k]p_{\mathbf{w}}(w)dw \right|^2 dy.$$

Applying the Cauchy-Schwarz inequality to the series, we can further upper bound by

$$C \int e^{\frac{3}{4}y^2} \left( \sum_{k=0}^{\infty} \frac{2^{-2k}}{k!} (p_{\varepsilon}^{(k)}(y))^2 \right) \left( \sum_{k=0}^{\infty} \frac{2^{2k}}{k!} \left( \int [(m_0(u+w))^k - (m_1(u+w))^k]p_{\mathbf{w}}(w)dw \right)^2 \right) dy$$

$$= C \left( \sum_{k=0}^{\infty} \frac{2^{-2k}}{k!} \int e^{\frac{3}{4}y^2} (p_{\varepsilon}^{(k)}(y))^2 dy \right) \left( \sum_{k=0}^{\infty} \frac{2^{2k}}{k!} \left( \int [(m_0(u+w))^k - (m_1(u+w))^k]p_{\mathbf{w}}(w)dw \right)^2 \right).$$

We first bound the left term involving the density $p_{\varepsilon}$. Let $H_k(y) = (-1)^k e^{\frac{y^2}{2}} \frac{d^k}{dy^k} e^{-\frac{y^2}{2}}$ denote the probabilist's $k$th Hermite polynomial, so that $p_{\varepsilon}^{(k)}(y) = (-1)^k H_k(y) e^{-\frac{1}{2}y^2}$. Then we have

$$\int e^{\frac{3}{4}y^2} \big|p_{\varepsilon}^{(k)}(y)\big|^2 dy = \int e^{\frac{3}{4}y^2} \cdot H_k^2(y) e^{-y^2} dy$$

$$= \int H_k^2(y) e^{-\frac{1}{4}y^2} dy$$

$$\overset{(i)}{=} 2^k \int H_k^2(y/\sqrt{2}) e^{-\frac{1}{2}(y/\sqrt{2})^2} dy$$

$$= \sqrt{2} \cdot 2^k \int H_k^2(y) e^{-\frac{1}{2}y^2} dy$$

$$\leq C \cdot 2^k k!,$$

where $(i)$ uses that $H_k$ is a degree-$k$ polynomial. Applying this inequality for each $k$, we have

$$\sum_{k=0}^{\infty} \frac{2^{-2k}}{k!} \int e^{\frac{3}{4}y^2} (p_{\varepsilon}^{(k)}(y))^2 dy \leq C \cdot \sum_{k=0}^{\infty} 2^{-k} \leq C,$$

and so

$$\chi^2\big(\mathbb{P}_{\mathbf{y}_1|\mathbf{u}_0;0}(\cdot \mid u) \,\|\, \mathbb{P}_{\mathbf{y}_1|\mathbf{u}_0;1}(\cdot \mid u)\big) \leq C \cdot \sum_{k=0}^{\infty} \frac{2^{2k}}{k!} \left( \int [(m_0(u+w))^k - (m_1(u+w))^k]p_{\mathbf{w}}(w)dw \right)^2.$$

Next, using the binomial theorem, for any $x \in \mathbb{R}$ we have

$$(m_0(x))^k - (m_1(x))^k = (r(x) + h(x))^k - (r(x) - h(x))^k$$

$$= \sum_{j=0}^{k} \binom{k}{j} r^{k-j}(x) h^j(x) (1 - (-1)^k)$$

$$= 2 \sum_{j \leq k,\ \text{odd}} \binom{k}{j} r^{k-j}(x) h^j(x),$$

leading to the upper bound

$$\chi^2\big(\mathbb{P}_{\mathbf{y}_1|\mathbf{u}_0;0}(\cdot\mid u)\,\|\,\mathbb{P}_{\mathbf{y}_1|\mathbf{u}_0;1}(\cdot\mid u)\big)$$

$$\leq C\cdot\sum_{k=0}^{\infty}\frac{2^{2k}}{k!}\left(\sum_{j\leq k,\text{ odd}}\binom{k}{j}\int r^{k-j}(u+w)h^j(u+w)p_{\mathbf{w}}(w)dw\right)^2$$

$$\leq C\sum_{k=0}^{\infty}\frac{2^{2k}k}{k!}\sum_{j\leq k,\text{ odd}}\binom{k}{j}\left|\int r^{k-j}(u+w)h^j(u+w)p_{\mathbf{w}}(w)dw\right|^2$$

$$=C\sum_{k=0}^{\infty}\frac{2^{2k}k}{k!}\sum_{j\leq k,\text{ odd}}\binom{k}{j}\left|(r^{k-j}h^j*p_{\mathbf{w}})(u)\right|^2$$

$$\leq C\sum_{k=0}^{\infty}\frac{2^{2k}k}{k!}\sum_{j\leq k,\text{ odd}}\binom{k}{j}\sup_{u\in\mathbb{R}}\left|(r^{k-j}h^j*p_{\mathbf{w}})(u)\right|^2$$

$$\leq C\sum_{k=0}^{\infty}\frac{2^{3k}k}{k!}\max_{j\leq k,\text{ odd}}\sup_{u\in\mathbb{R}}\left|(r^{k-j}h^j*p_{\mathbf{w}})(u)\right|^2,$$

where the equality holds because $p_w$ is symmetric. We now appeal to Lemma D.5 for each term in the sum, which leads to an upper bound of

$$C\sum_{k=0}^{\infty}\frac{2^{3k}k}{k!}\max_{j\leq k,\text{ odd}}\left(\gamma^{k-j}\alpha^j\beta^{(k-j+1)/2}\cdot j\sqrt{(k-j)!}\cdot\exp\left(-2\pi^2\left(\frac{\beta^2}{j}\wedge1\right)\beta^2\right)\right)^2$$

$$\leq C\sum_{k=0}^{\infty}2^{3k}k^3\max_{j\leq k,\text{ odd}}\left(\gamma^{k-j}\alpha^j\beta^{(k-j+1)/2}\cdot\exp\left(-2\pi^2\left(\frac{\beta^2}{j}\wedge1\right)\beta^2\right)\right)^2.$$

Recalling the choice $\alpha=\frac{1}{12\beta^2}$ and $\gamma=1/\beta$, we can upper bound

$$\gamma^{k-j}\alpha^j\beta^{(k-j+1)/2}\leq\beta^{-k/2}$$

for each term above, so we have

$$\leq C\sum_{k=0}^{\infty}2^{3k}k^3\max_{j\leq k,\text{ odd}}\beta^{-k}\cdot\exp\left(-4\pi^2\left(\frac{\beta^2}{j}\wedge1\right)\beta^2\right).$$

Since $\beta\geq64$ for $n\geq3$, we have $\beta^{-k}\leq2^{-6k}$, so we can upper bound the sum above as

$$\leq C\sum_{k=0}^{\infty}2^{-2k}k^3\max_{j\leq k,\text{ odd}}2^{-k}\cdot\exp\left(-4\pi^2\left(\frac{\beta^2}{j}\wedge1\right)\beta^2\right).$$

We now consider two cases for the term in the $\max$ above. First, if $j\leq\beta^2$, then we have $\exp\left(-4\pi^2\left(\frac{\beta^2}{j}\wedge1\right)\beta^2\right)\leq\exp\left(-4\pi^2\beta^2\right)$. Otherwise, we have $k\geq j\geq\beta^2$, so $2^{-k}\leq2^{-\beta^2}$. Putting the two cases together (using that $\exp\left(-4\pi^2\beta^2\right)\leq2^{-\beta^2}$), we get the following coarse upper bound:

$$C2^{-\beta^2}\sum_{k=0}^{\infty}2^{-k}k^3\leq C2^{-\beta^2}.$$

The choice $\beta=64\ln^{1/2}n$ implies that $2^{-\beta^2}\leq n^{-10}$.

$\square$

**Lemma D.4.** Let $\eta\leq1$ be given. Then for each $i\in\{0,1\}$, we have

$$p_i(y\mid u)\geq3^{-1/2}\exp\left(-\left(\frac{(1+\eta)(y-\gamma u)^2}{2}+\frac{1}{\eta}\right)\right).$$

*Proof of Lemma D.4.* We have

$$p_i(y\mid u)=\frac{1}{\sqrt{2\pi}}\int p_{\boldsymbol{\varepsilon}}(y-m_i(u+w))p_{\mathbf{w}}(w)dw$$

$$=\frac{1}{\sqrt{2\pi}}\int\exp\left(-\frac{1}{2}(y-r(u+w)\pm h(u+w))^2\right)p_{\mathbf{w}}(w)dw.$$

Using the AM-GM inequality, we have that for any $\eta > 0$, this is lower bounded by

$$\frac{1}{\sqrt{2\pi}} \int \exp\left(-\frac{1+\eta}{2}(y - r(u+w))^2\right) \exp\left(-\frac{1+1/\eta}{2}h^2(u+w)\right) p_{\mathbf{w}}(w)dw.$$

We will restrict to $\eta < 1$. Since $|h| \le \alpha < 1$ everywhere, we can further lower bound by

$$\frac{\exp\left(-\frac{1+1/\eta}{2}\right)}{\sqrt{2\pi}} \int \exp\left(-\frac{1+\eta}{2}(y - r(u+w))^2\right) p_{\mathbf{w}}(w)dw$$

$$\ge \frac{e^{-\frac{1}{\eta}}}{\sqrt{2\pi}} \int \exp\left(-\frac{1+\eta}{2}(y - r(u+w))^2\right) p_{\mathbf{w}}(w)dw$$

$$= \frac{e^{-\frac{1}{\eta}}}{\sqrt{2\pi}} \int \exp\left(-\frac{1+\eta}{2}(y - r(u+w))^2\right) \exp\left(-\frac{1}{2}w^2\right)dw.$$

Define $\mu = y - \gamma u$, $\sigma^2 = (1 + (1+\eta)\gamma^2)^{-1}$, and $\mu' = (1+\eta)\gamma\sigma^2\mu$. Then by completing the square, we have

$$\exp\left(-\frac{1+\eta}{2}(y - r(u+w))^2\right) \exp\left(-\frac{1}{2}w^2\right) = \exp\left(-\frac{(1+\eta)\mu^2}{2(1+(1+\eta)\gamma^2)}\right) \cdot \exp\left(-\frac{(w-\mu')^2}{2\sigma^2}\right).$$

It follows that

$$\int \exp\left(-\frac{1+\eta}{2}(y - r(u+w))^2\right) \exp\left(-\frac{1}{2}w^2\right)dw = \exp\left(-\frac{(1+\eta)\mu^2}{2(1+(1+\eta)\gamma^2)}\right) \cdot \sqrt{2\pi\sigma^2}$$

$$\ge \exp\left(-\frac{(1+\eta)\mu^2}{2}\right) \cdot \sqrt{\frac{2\pi}{3}}.$$

$$\square$$

**Lemma D.5.** There is a universal constant $C > 0$ such that for all $k$ and $j \le k$ with $j$ odd,

$$\sup_{x\in\mathbb{R}} \left|(r^{k-j}h^j * p_{\mathbf{w}})(x)\right| \le C \cdot \gamma^{k-j}\alpha^j\beta^{(k-j+1)/2} \cdot j\sqrt{(k-j)!} \cdot \exp\left(-2\pi^2\left(\frac{\beta^2}{j} \wedge 1\right)\beta^2\right) \quad \text{(D.8)}$$

*Proof of Lemma D.5.* Let $x \in \mathbb{R}$ be fixed. Then, using the Fourier inversion formula (using that both $r^{k-j}h^j$, $p_{\mathbf{w}}$, and their respective Fourier transforms are $L_2$-integrable), we have

$$\left|(r^{k-j}h^j * p_{\mathbf{w}})(x)\right| = \left|\int e^{i2\pi x\omega}(\widehat{r^{k-j}h^j})(\omega)\widehat{p_{\mathbf{w}}}(\omega)d\omega\right| \le \int \left|(\widehat{r^{k-j}h^j})(\omega)\widehat{p_{\mathbf{w}}}(\omega)\right|d\omega.$$

We proceed to compute the Fourier transform for $r^{k-j}(x)h^j(x) = \gamma^{k-j}\alpha^j x^{k-j}\phi^j(x)\psi^j(x)$. We first observe that $\phi^j(x) = \exp(-\frac{j}{\beta^2} \cdot \frac{z^2}{2})$. Let $b_1 = \frac{\beta^2}{j}$. Then, using that the Fourier transform is self-dual for gaussians (specifically, that the Fourier transform of $e^{-cx^2}$ is $\sqrt{\frac{\pi}{c}}e^{-\frac{\pi^2}{c}\omega^2}$), we have

$$\widehat{\phi^j}(\omega) = \sqrt{2\pi b_1}e^{-2\pi^2 b_1\omega^2}.$$

Next, we recall that for any $f$, the Fourier transform of $x^n f(x)$ is $\left(\frac{i}{2\pi}\right)^n \frac{d^n}{d\omega^n}\widehat{f}(\omega)$, so that

$$\widehat{x^{k-j}\phi^j}(\omega) = \left(\frac{i}{2\pi}\right)^{k-j}\sqrt{2\pi b_1} \cdot \frac{d^{k-j}}{d\omega^{k-j}}e^{-2\pi^2 b\omega^2}$$

$$= \left(\frac{i}{2\pi}\right)^{k-j}\sqrt{2\pi b_1}b_2^{k-j} \cdot H_{k-j}(b_2\omega)e^{-\frac{(b_2\omega)^2}{2}}.$$

where $b_2 := 2\pi\sqrt{b_1}$. Finally, we use that

$$\psi^j(x) = (\cos(4\pi\beta x))^j = \frac{1}{2^j}(e^{i4\pi\beta x} + e^{-i4\pi\beta x})$$

$$= \frac{1}{2^j}\sum_{l=0}^{j}\binom{j}{l}e^{i4\pi\beta x\cdot(j-l)} \cdot e^{-i4\pi\beta x\cdot l}$$

$$= \frac{1}{2^j}\sum_{l=0}^{j}\binom{j}{l}e^{i4\pi\beta x\cdot(j-2l)}$$

We now use that the Fourier transform of $e^{-icx}f(x)$ is $\widehat{f}(\omega - \frac{c}{2\pi})$ to derive

$$\widehat{x^{k-j}\phi^j\psi^j}(\omega) = \frac{1}{2^j}\left(\frac{i}{2\pi}\right)^{k-j}\sqrt{2\pi b_1}b_2^{k-j}\sum_{l=0}^{j}\binom{j}{l}H_{k-j}(b_2(\omega - 2\beta(j-2l)))e^{-\frac{(b_2(\omega-2\beta(j-2l)))^2}{2}}$$

It follows that

$$\int \left|(\widehat{r^{k-j}h^j})(\omega)\widehat{p_{\mathbf{w}}}(\omega)\right|d\omega$$

$$\leq \gamma^{k-j}\alpha^j\frac{1}{2^j}\left(\frac{1}{2\pi}\right)^{k-j}\sqrt{2\pi b_1}b_2^{k-j}\sum_{l=0}^{j}\binom{j}{l}\int\left|H_{k-j}(b_2(\omega - 2\beta(j-2l)))e^{-\frac{(b_2(\omega-2\beta(j-2l)))^2}{2}}\widehat{p_w}(\omega)\right|d\omega$$

$$\leq \gamma^{k-j}\alpha^j\frac{1}{2^j}\left(\frac{1}{2\pi}\right)^{k-j}\sqrt{2\pi b_1}b_2^{k-j}\sum_{l=0}^{j}\binom{j}{l}\int|H_{k-j}(b_2(\omega - 2\beta(j-2l)))|e^{-\frac{(b_2(\omega-2\beta(j-2l)))^2}{2}}e^{-2\pi^2\omega^2}d\omega$$

Now, let $0 \leq l \leq j$ be fixed. We bound

$$\int|H_{k-j}(b_2(\omega - 2\beta(j-2l)))|e^{-\frac{(b_2(\omega-2\beta(j-2l)))^2}{2}}e^{-2\pi^2\omega^2}d\omega$$

$$\leq \underbrace{\int_{(-\beta,\beta)}|H_{k-j}(b_2(\omega - 2\beta(j-2l)))|e^{-\frac{(b_2(\omega-2\beta(j-2l)))^2}{2}}e^{-2\pi^2\omega^2}d\omega}_{(\star)}$$

$$+ \underbrace{\int_{\mathbb{R}\setminus(-\beta,\beta)}|H_{k-j}(b_2(\omega - 2\beta(j-2l)))|e^{-\frac{(b_2(\omega-2\beta(j-2l)))^2}{2}}e^{-2\pi^2\omega^2}d\omega}_{(\star\star)}.$$

For the integral in the term $(\star)$, we drop the $e^{-2\pi^2 w^2}$ term (since it is at most one), and apply Cauchy-Schwarz to bound by

$$\int_{(-\beta,\beta)}|H_{k-j}(b_2(\omega - 2\beta(j-2l)))|e^{-\frac{(b_2(\omega-2\beta(j-2l)))^2}{2}}d\omega$$

$$\leq \sqrt{\int_{(-\beta,\beta)}H_{k-j}^2(b_2(\omega - 2\beta(j-2l)))e^{-\frac{(b_2(\omega-2\beta(j-2l)))^2}{2}}d\omega}\cdot\sqrt{\int_{(-\beta,\beta)}e^{-\frac{(b_2(\omega-2\beta(j-2l)))^2}{2}}d\omega}$$

Observe that since $j$ is odd, $j - 2l$ is also odd, and hence $|j - 2l| \geq 1$. It follows that for $\omega \in (-\beta,\beta)$, $\omega - 2\beta(j-2l) \notin (-\beta,\beta)$, and so

$$\int_{(-\beta,\beta)}e^{-\frac{(b_2(\omega-2\beta(j-2l)))^2}{2}}d\omega \leq \int_{(-\beta,\beta)}e^{-\frac{b_2^2}{2}\beta^2}d\omega \leq 2\beta e^{-\frac{b_2^2}{2}\beta^2}.$$

Leaving the Hermite integral for a moment and moving to the second term $(\star\star)$, we have

$$\int_{\mathbb{R}\setminus(-\beta,\beta)}|H_{k-j}(b_2(\omega - 2\beta(j-2l)))|e^{-\frac{(b_2(\omega-2\beta(j-2l)))^2}{2}}e^{-2\pi^2\omega^2}d\omega$$

$$\leq e^{-2\pi^2\beta^2}\int_{\mathbb{R}\setminus(-\beta,\beta)}|H_{k-j}(b_2(\omega - 2\beta(j-2l)))|e^{-\frac{(b_2(\omega-2\beta(j-2l)))^2}{2}}d\omega$$

$$\leq e^{-2\pi^2\beta^2}\sqrt{\int_{\mathbb{R}\setminus(-\beta,\beta)}H_{k-j}^2(b_2(\omega - 2\beta(j-2l)))e^{-\frac{(b_2(\omega-2\beta(j-2l)))^2}{2}}d\omega}\cdot\sqrt{\int_{\mathbb{R}\setminus(-\beta,\beta)}e^{-\frac{(b_2(\omega-2\beta(j-2l)))^2}{2}}d\omega}$$

$$\leq e^{-2\pi^2\beta^2}\sqrt{\int_{\mathbb{R}\setminus(-\beta,\beta)}H_{k-j}^2(b_2(\omega - 2\beta(j-2l)))e^{-\frac{(b_2(\omega-2\beta(j-2l)))^2}{2}}d\omega}\cdot\sqrt{\frac{2\pi}{b_2}}.$$

Putting both cases together, we have

$$\int|H_{k-j}(b_2(\omega - 2\beta(j-2l)))|e^{-\frac{(b_2(\omega-2\beta(j-2l)))^2}{2}}e^{-2\pi^2\omega^2}d\omega$$

$$\leq C\cdot(\sqrt{\beta}\vee 1/\sqrt{b_2})\exp(-2\pi^2(b_1^2\wedge 1)\beta^2)\cdot\sqrt{\int H_{k-j}^2(b_2(\omega - 2\beta(j-2l)))e^{-\frac{(b_2(\omega-2\beta(j-2l)))^2}{2}}d\omega},$$

where $C$ is a numerical constant. Using a change of variables, we have

$$\sqrt{\int H_{k-j}^2(b_2(\omega - 2\beta(j-2l))e^{-\frac{(b_2(\omega-2\beta(j-2l)))^2}{2}}d\omega} = \frac{1}{\sqrt{b_2}}\sqrt{\int H_{k-j}^2(\omega)e^{-\frac{\omega^2}{2}}d\omega}$$

$$\le \sqrt{\frac{2\pi(k-j)!}{b_2}}.$$

Since this bound holds uniformly for all $l$ and $\sum_{l=0}^{j}\binom{j}{l} = 2^j$, we have

$$\int \left|(\overline{r^{k-j}h^j})(\omega)\widehat{p_{\mathbf{w}}}(\omega)\right|d\omega$$

$$\le C \cdot \gamma^{k-j}\alpha^j\left(\frac{1}{2\pi}\right)^{k-j}\sqrt{2\pi b_1}b_2^{k-j} \cdot \sqrt{\frac{2\pi(k-j)!}{b_2}} \cdot (\sqrt{\beta} \vee 1/\sqrt{b_2})\exp\left(-2\pi^2(b_1^2 \wedge 1)\beta^2\right)$$

$$\le C' \cdot \gamma^{k-j}\alpha^j b_1^{(k-j)/2} \cdot (\sqrt{\beta} \vee 1/\sqrt{b_1}) \cdot \sqrt{(k-j)!} \cdot \exp\left(-2\pi^2(b_1^2 \wedge 1)\beta^2\right)$$

$$\le C'' \cdot \gamma^{k-j}\alpha^j\beta^{(k-j+2)/2}j \cdot \sqrt{(k-j)!} \cdot \exp\left(-2\pi^2(b_1^2 \wedge 1)\beta^2\right).$$

$\square$

**Lemma D.6.** For any non-negative function $g : \mathbb{R} \to \mathbb{R}_+$ and any roll-in policy $\pi$ with $|\mathbf{u}_0| \le \beta$ almost surely,

$$\mathbb{E}_{\pi;i}[g(\mathbf{y}_1)] \ge c \cdot \int_{-1}^{1} g(y)dy \quad \text{for all } i \in \{0, 1\},$$

where $c$ is an absolute numerical constant.

*Proof of Lemma D.6.* Observe that we have

$$\mathbb{E}_{\pi;i}[g(\mathbf{y}_1)] = \mathbb{E}_{\mathbf{u}_0;i}\left[\int_{-\infty}^{\infty} g(y)p_i(y \mid \mathbf{u}_0)dy\right]$$

$$\ge \mathbb{E}_{\mathbf{u}_0;i}\left[\int_{-1}^{1} g(y)p_i(y \mid \mathbf{u}_0)dy\right].$$

Lemma D.4 (with $\eta = 1$) implies that for all $y \in [-1, 1]$ and $|u| \le \beta$,

$$p_i(y \mid u) \ge 3^{-1/2}\exp\left(-\left((y - \gamma u)^2 + 1\right)\right) \ge c.$$

It follows that

$$\mathbb{E}_{\mathbf{u}_0;i}\left[\int_{-1}^{1} g(y)p_i(y \mid \mathbf{u}_0)dy\right] \ge c \cdot \int_{-1}^{1} g(y)dy.$$

$\square$

**Lemma D.7.** If $\beta \ge 1$ and $\alpha$ and $\gamma$ are chosen as in Eq. (D.5), we have

$$\int_{-1}^{1}(f_0(y) - f_1(y))^2 dy \ge \frac{1}{8}\alpha^2\beta.$$

*Proof of Lemma D.7.* Recall that $m_0 = f_0^{-1}$ and $m_1 = f_1^{-1}$. Throughout the proof we will use that

$$\frac{1}{2\beta} \le m_i'(z) \le \frac{3}{2\beta}, \quad \text{and} \quad \frac{2\beta}{3} \le f_i'(y) \le 2\beta.$$

As a first step, we have

$$\int_{-1}^{1}(f_0(y) - f_1(y))^2 dy = \int_{-1}^{1}(f_0(y) - f_0(f_0^{-1}(f_1(y))))^2 dy \ge \frac{4\beta^2}{9}\int_{-1}^{1}(y - f_0^{-1}(f_1(y)))^2 dy,$$

where we have used that $f'(y) \ge \frac{2\beta^2}{3}$ everywhere. Next, using a change of variables, we have

$$\int_{-1}^{1}(y - f_0^{-1}(f_1(y)))^2 dy = \int_{f_1(-1)}^{f_1(1)}\frac{(f_1^{-1}(x) - f_0^{-1}(x))^2}{f_1'(f_1^{-1}(x))}dx$$

$$\ge \frac{1}{2\beta}\int_{f_1(-1)}^{f_1(1)}(f_1^{-1}(x) - f_0^{-1}(x))^2 dx,$$

where the inequality uses that $f_1' \leq 2\beta$ everywhere. Next, we observe that $f_1 = m_1^{-1}$, and that

$$m_1(1) \leq \gamma + \alpha < 1, \quad \text{and} \quad m_1(-1) \geq -\gamma - \alpha > -1.$$

It follows that $f_1(1) \geq 1$ and $f_1(-1) \leq -1$, and consequently

$$\int_{f_1(-1)}^{f_1(1)} (f_1^{-1}(x) - f_0^{-1}(x))^2 dx \geq \int_{-1}^{1} (f_1^{-1}(x) - f_0^{-1}(x))^2 dx$$

$$= \int_{-1}^{1} (m_1(x) - m_0(x))^2 dx$$

$$= 4 \int_{-1}^{1} h^2(x) dx.$$

Finally, we appeal to Lemma D.8, which implies that

$$\int_{-1}^{1} h^2(x) dx \geq \frac{\alpha^2}{2e}.$$

$\square$

**Lemma D.8.** If we choose $\beta \geq 1$, then the function $h$ in (D.4) satisfies

$$\int_{-\beta}^{\beta} h^2(z) dz \geq \frac{\alpha^2 \beta}{2e}, \quad \text{and} \quad \int_{-1}^{1} h^2(z) dz \geq \frac{\alpha^2}{2e}. \tag{D.9}$$

*Proof of Lemma D.8.* First, since we integrate only over the range $(-\beta, \beta)$, $e^{-\frac{z^2}{\beta^2}} \geq e^{-1}$, so we have

$$\int_{-\beta}^{\beta} h^2(z) dz = \alpha^2 \int_{-\beta}^{\beta} e^{-\frac{z^2}{\beta^2}} \cos^2(4\pi\beta z) dz \geq \frac{\alpha^2}{e} \int_{-\beta}^{\beta} \cos^2(4\pi\beta z) dz.$$

Next, we recall that for any $a$, the indefinite integral of $\cos^2(ax)$ satisfies $\int \cos^2(ax) = \frac{x}{2} + \frac{1}{2a} \sin(ax)\cos(ax)$. Applying this above, we have

$$\int_{-\beta}^{\beta} \cos^2(4\pi\beta z) dz = \left. \frac{x}{2} + \frac{1}{8\pi\beta} \sin(4\pi\beta x)\cos(4\pi\beta x) \right|_{-\beta}^{\beta} \geq \beta - \frac{1}{4\pi\beta}.$$

For $\beta > 1$, this is at least $\frac{\beta}{2}$.

Similarly, since $\beta \geq 1$, we have

$$\int_{-1}^{1} h^2(z) dz \geq \frac{\alpha^2}{e} \int_{-1}^{1} \cos^2(4\pi\beta z) dz,$$

and

$$\int_{-1}^{1} \cos^2(4\pi\beta z) dz = \left. \frac{x}{2} + \frac{1}{8\pi\beta} \sin(4\pi\beta x)\cos(4\pi\beta x) \right|_{-1}^{1} \geq 1 - \frac{1}{4\pi\beta} \geq \frac{1}{2}.$$

$\square$

# E   Learning Theory Tools

In this section, we state and prove basic learning-theoretic tools used throughout the proofs for our main results. Appendix E.1 gives the main statements and definitions for these results, and Appendix E.2 proves the results in the order in which they appear. Our results are split into the following categories:

- Appendix E.1.1 introduces a convention for subexponential random variables ("$c$-concentrated") used throughout our proofs and establishes key properties of random variables satisfying this condition (Lemma E.1)

- Appendix E.1.2 gives a concentration properties for Gaussian vectors (Lemma E.2) and establishes a useful change-of-measure lemma (Lemma E.3)

- Appendix E.1.3 gives a generic template (Lemma E.4) for computing conditional expectations for random variables we call *decodable Markov chains* (Definition 4), which arise when analyzing the regression problems used in Algorithm 2.

- Appendix E.1.4 presents Definition 5, which introduces the main notion of covering number used in our analysis, and provides bounds on covering numbers for these function classes used by Algorithm 2.

- Appendix E.1.5 gives prediction error bounds for square loss regression over a general function classes, subject to misspecification error. Proposition E.1 provides guarantees based on a classical notion of misspecification error (which arises in Phase I of Algorithm 2), while Corollary E.2 gives guarantees under a stronger notion of *function-dependent* misspecification error, which is used in the analysis of Phase III.

- Appendix E.1.6 provides guarantees for a principal component analysis (PCA) setup which, in particular, subsumes the dimensionality reduction procedure used in Phase I of Algorithm 2. It provides guarantees for estimating a covariance matrix under persistent error (Proposition E.2), and a corollary regarding overlap between eigenspaces (Corollary E.3).

- Appendix E.1.7 considers linear regression. Proposition E.3 gives bounds for parameter recovery under errors in variables, which is used to recover $A_{\mathrm{id}}$ and $B_{\mathrm{id}}$ in Phase II of Algorithm 2. Proposition E.4 gives a guarantee for covariance estimation, which are used to estimate $\Sigma_{w,\mathrm{id}}$ in Phase II.

- Finally, Appendix E.1.8 gives a parameter recovery bound for regression with measurements which are rank-one outer products of near-Gaussian vectors. This is used to recover the cost matrix $Q_{\mathrm{id}}$ in Phase II.

## E.1   Statement of Guarantees

### E.1.1   Generic Concentration

**Definition 3** ($c$-concentration)**.** *We say that a non-negative random variable $\boldsymbol{z}$ is $c$-concentrated if* $\mathbb{P}[\boldsymbol{z} \ge c\ln(1/\delta)] \le \delta$ *for all $\delta \in (0, 1/e]$. For such random variables, we define $c_{n,\delta} := c\ln(2n/\delta)$.*

This is one of many equivalent (up to numerical constants) definitions for sub-exponential concentration (e.g., [43]). We opt for the term "c-concentrated" to make the dependence on the concentration parameter $c$ precise.

**Lemma E.1** (Truncated concentration)**.** Let $\boldsymbol{z}$ be a non-negative $c$-concentrated random variable. Then, $\boldsymbol{z}$ is $c'$-concentrated for all $c' \ge c$, and $\alpha\boldsymbol{z} + \beta$ is $\alpha c + \beta$-concentrated for all $\alpha, \beta > 0$. Moreover, the the following bounds hold.

1. For any $\delta \in (0, 1/e]$, we have $\mathbb{E}[\boldsymbol{z}\mathbb{I}\{\boldsymbol{z} \ge c\ln(1/\delta)\}] \le c\delta$, and in particular, $\mathbb{E}\max\{c, \boldsymbol{z}\} \le 2c$. For any integer $k \ge 1$, and $\mathbb{E}[\boldsymbol{z}^k] \le 2k!c^k$.

2. Let $\varepsilon^2 \ge \mathbb{E}[\boldsymbol{z}]$, and let $\delta \in (0, 1)$. Suppose $n$ is large enough such that $\psi(n, \delta) \le \frac{\varepsilon^2}{c}$, where we define

$$\psi(n, \delta) := \frac{2\ln(2n/\delta)\ln(2/\delta)}{n}. \tag{E.1}$$

Then with probability at least $1 - \delta$, $\mathbf{z}^{(i)} \sim \mathbf{z}$ satisfy $\frac{1}{n}\sum_i \mathbf{z}^{(i)} \leq 2\varepsilon^2$, where $\mathbf{z}^{(i)} \overset{\text{i.i.d.}}{\sim} \mathbf{z}$ for $1 \leq i \leq n$.

3. Consider the previous claim. Suppose we replace the hypothesis that $\mathbf{z}$ is $c$-concentrated with the assumption that for a given $\delta \in (0,1)$, $\mathbf{z} \leq c \ln(2n/\delta)$ almost surely. Then with failure probability at least $1 - 2\delta$, $\frac{1}{n}\sum_i \mathbf{z}^{(i)} \leq 2\varepsilon^2$.

### E.1.2 Gaussian Concentration and Change of Measure

Our first lemma shows that norms of Gaussian vectors satisfy the $c$-concentration condition.

**Lemma E.2.** Let $\mathbf{x} \sim \mathcal{N}(0, \Sigma)$. Then, $\|\mathbf{x}\|^2$ is $c$-concentrated for $c = 5\mathrm{tr}(\Sigma)$.

Next, we provide a change of measure argument, which is used to establish that Algorithm 1 accurately estimates the system state.

**Lemma E.3** (Gaussian change of measure)**.** Let $\Sigma_1, \Sigma_2 > 0$ be matrices in $\mathbb{R}^{d \times d}$. Let $\mathbf{x}_1 \sim \mathcal{N}(0, \Sigma_1)$, $\mathbf{x}_2 \sim \mathcal{N}(0, \Sigma_2)$, and let $\mathbf{y}_1 \sim q(\cdot \mid \mathbf{x}_1)$, $\mathbf{y}_2 \sim q(\cdot \mid \mathbf{x}_1)$. Let $\hat{h}, h_\star : \mathcal{Y} \to \mathbb{R}^d$ be two functions such that $\max\{\|\hat{h}(y)\|, \|h_\star(y)\|\} \leq L\|f_\star(y)\|$. Suppose that

$$\mathbb{E}_{\mathbf{y}_1}[\|\hat{h}(\mathbf{y}_1) - h_\star(\mathbf{y}_1)\|^2] \leq \varepsilon^2, \quad \text{and} \quad \|I - \Sigma_1^{1/2}\Sigma_2^{-1}\Sigma_1^{1/2}\|_{\mathrm{op}} \leq \frac{1}{14d \ln\left(\frac{80eL^2(1+\|\Sigma_1\|_{\mathrm{op}})}{\varepsilon^2}\right)},$$

for some $\varepsilon > 0$. Then the following error bound holds:

$$\mathbb{E}_{\mathbf{y}_2}[\|\hat{h}(\mathbf{y}_2) - h_\star(\mathbf{y}_2)\|^2] \leq 2\varepsilon^2.$$

### E.1.3 Conditional Expectations for Decodable Markov Chains

**Definition 4** (Decodable Markov chain)**.** *Let $\mathbf{u} \in \mathcal{U}$, $\mathbf{x} \in \mathcal{X}$, $\mathbf{y} \in \mathcal{Y}$ be random variables that form a Markov chain $\mathbf{u} \to \mathbf{x} \to \mathbf{y}$. We say $(\mathbf{u}, \mathbf{x}, \mathbf{y})$ is a* decodable Markov chain *if there exists some function $f_\star : \mathcal{Y} \to \mathcal{X}$ such that $\mathbf{x} = f_\star(\mathbf{y})$ almost surely.*

**Lemma E.4** (Characterization of square loss minimizer)**.** Let $(\mathbf{u}, \mathbf{x}, \mathbf{y})$ be a decodable Markov chain. Then, $\mathbb{E}[\mathbf{u} \mid \mathbf{y} = y] = h_\star(y)$, where

$$h_\star(y) := \mathbb{E}[\mathbf{u} \mid \mathbf{x} = f_\star(y)].$$

Moreover, for any class of functions $\mathscr{H}$ with $h_\star \in \mathscr{H}$, for any $h \in \arg\min_{h' \in \mathscr{H}} \mathbb{E}\|h'(\mathbf{y}) - \mathbf{u}\|^2$, we have

$$h(\mathbf{y}) = h_\star(\mathbf{y}) \quad \text{almost surely in } \mathbf{y}.$$

### E.1.4 Covering Numbers

**Definition 5** (Covering numbers)**.** *Let $(\mathcal{X}, \mathrm{dist})$ be a metric space with pseudometric $\mathrm{dist}$. The covering number $\mathcal{N}(\epsilon, \mathcal{X}, \mathrm{dist})$ is defined as the minimal cardinality of any set $\mathcal{X}' \subseteq \mathcal{X}$ such that*

$$\max_{x \in \mathcal{X}} \min_{x' \in \mathcal{X}'} \mathrm{dist}(x, x') \leq \epsilon.$$

*We say that $\mathcal{X}'$ is a minimal $\epsilon$-cover of $\mathcal{X}$ if it witnesses the condition above and has $|\mathcal{X}'| = \mathcal{N}(\epsilon, \mathcal{X}, \mathrm{dist})$.*

**Lemma E.5.** Let $\mathscr{M} := \{M \in \mathbb{R}^{d \times d_\mathbf{x}} : \|M\|_{\mathrm{op}} \leq b\}$. Then, $\mathcal{N}(b\epsilon, \mathscr{M}, \|\cdot\|_{\mathrm{op}}) \leq (1 + 2/\epsilon)^{dd_\mathbf{x}}$.

### E.1.5 Square Loss Regression

**Proposition E.1** (Square loss regression with misspecification error)**.** Let $(\mathbf{u}, \mathbf{y})$ be a pair of random variables with $\mathbf{u} \in \mathcal{U}$, $\mathbf{y} \in \mathcal{Y}$, and let $\mathbf{e} \in \mathcal{U}$ be an arbitrary "error" random variable. Suppose that $\mathscr{H}$ is a function class that contains the function $h_\star(y) := \mathbb{E}[\mathbf{u} \mid \mathbf{y} = y]$. Consider empirical risk minimizer

$$\hat{h}_n := \arg\min_{h \in \mathscr{H}} \sum_{i=1}^n \|h(\mathbf{y}^{(i)}) + \mathbf{e}^{(i)} - \mathbf{u}^{(i)}\|^2,$$

where $(\mathbf{u}^{(i)}, \mathbf{y}^{(i)}, \mathbf{e}^{(i)})$ are drawn i.i.d. from the law of $(\mathbf{u}, \mathbf{y}, \mathbf{e})$ for $1 \leq i \leq n$. Suppose that there exists a constant $c > 0$ and function $\varphi : \mathcal{Y} \to \mathbb{R}_+$ such that the following properties hold:

- $\|h(y)\|^2 \le \varphi(y)$ for all $h \in \mathscr{H}$.

- The random variables $\varphi(\mathbf{y})$ and $\|\mathbf{e} - \mathbf{u}\|^2$ are $c$-concentrated.

- For all $c' \ge c$ and all $\epsilon \le 1$, the $\sqrt{c'}\epsilon$-covering number of $\mathscr{H}$ in the pseudometric $d_{c',\infty}(h, h') := \sup_{y \in \mathcal{Y}}\{\|h(y) - h'(y)\| : \varphi(y) \le c'\}$ is bounded by a function $\mathcal{N}(\epsilon)$.

Then, with probability at least $1 - \frac{3\delta}{2}$,

$$\mathbb{E}\|\hat{h}_n(\mathbf{y}) - h_\star(\mathbf{y})\|^2 \le \frac{270 c_{n,\delta}}{n} \ln(2\mathcal{N}(1/33n)\delta^{-1}) + 8\mathbb{E}\|\mathbf{e}\|^2,$$

where we recall that $c_{n,\delta} := c\ln(2n/\delta)$.

We now state two corollaries of the above regression. First, a simple corollary for structured function classes of the form $\{M \cdot f\}$, where $M$ are matrices of bounded operator norm, and $f \in \mathscr{F}$ are elements of finite class which satisfy a growth condition like Assumption 5.

**Corollary E.1** (Regression with Structured Function Class)**.** Let $(\mathbf{u}, \mathbf{y})$ be a pair of random variables with $\mathbf{u} \in \mathbb{R}^{d_u}$, $\mathbf{y} \in \mathcal{Y}$, and let $\mathbf{e} \in \mathcal{U}$ be an arbitrary "error" random variable. Suppose that $\mathscr{H}$ is a function class that contains the function $h_\star(y) := \mathbb{E}[\mathbf{u} \mid \mathbf{y} = y]$. Consider empirical risk minimizer

$$\hat{h}_n := \underset{h \in \mathscr{H}}{\arg\min} \sum_{i=1}^{n} \|h(\mathbf{y}^{(i)}) + \mathbf{e}^{(i)} - \mathbf{u}^{(i)}\|^2.$$

where $(\mathbf{u}^{(i)}, \mathbf{y}^{(i)}, \mathbf{e}^{(i)})$ are drawn i.i.d. from the law of $(\mathbf{u}, \mathbf{y}, \mathbf{e})$ for $1 \le i \le n$. Suppose $\mathscr{F} : \mathcal{Y} \to \mathbb{R}^{d_\times}$ is a finite class of functions satisfying $f(y) \le L \max\{1, \|f_\star(y)\|_2\}$ for all $f \in \mathscr{F}$, where $L \ge 1$ without loss of generality. In addition, suppose that $\mathscr{H}$ takes the form

$$\mathscr{H} := \{h(y) = M \cdot f(y) : f \in \mathscr{F} \quad M \in \mathbb{R}^{d_u d_x}, \quad \|M\|_{\mathrm{op}} \le b\}.$$

Lastly, assume that the random variables $\varphi(\mathbf{y})$ and $\|\mathbf{e} - \mathbf{u}\|^2$ are $c$-concentrated, where $\varphi(y)^{1/2} := bL \max\{1, \|f_\star(y)\|_2\}$. Then, with probability at least $1 - \frac{3\delta}{2}$,

$$\mathbb{E}\|\hat{h}_n(\mathbf{y}) - h_\star(\mathbf{y})\|^2 \le \frac{c(d_u d_x + \ln|\mathscr{F}|) \cdot \mathtt{logs}(n, \delta)}{n} + 8\mathbb{E}\|\mathbf{e}\|^2,$$

where define $\mathtt{logs}(n, \delta) := 270 \ln(2n/\delta)\ln(330n/\delta) \lesssim \ln(n/\delta)^2$.

Second, we state a regression bound tailored to the structured regression problems that arise in Phase III of our algorithm.

**Corollary E.2.** Let $\mathcal{Z} = \mathcal{Y} \times \mathcal{Y}$. Let $(\mathbf{v}, \mathbf{z}) \in \mathcal{V} \times \mathcal{Z}$ be a pair of random variables, and let $\mathbf{e} \in \mathcal{V}$ be an arbitrary "error" random variable defined on the same probability space. Let $\mathscr{H}$ be a function class, and let $\phi, \widehat{\phi} : \mathscr{G} \times \mathcal{Z} \to \mathbb{R}^d$ be measurable functions. Suppose that the set $\mathscr{H}$ contains a function $h_\star$ satisfying $\phi(h_\star, z) := \mathbb{E}[\mathbf{v} \mid \mathbf{z} = z]$. Let $\{(\mathbf{z}^{(i)}, \mathbf{v}^{(i)}, \mathbf{e}^{(i)})\}_{i=1}^{n}$ be i.i.d. copies of $(\mathbf{z}, \mathbf{v}, \mathbf{e})$, and define

$$\hat{h} := \underset{h \in \mathscr{H}}{\arg\min} \sum_{i=1}^{n} \|\widehat{\phi}(h, \mathbf{z}^{(i)}) + \mathbf{e}^{(i)} - \mathbf{v}^{(i)}\|^2.$$

Introduce $\delta_\phi(z, h) := \widehat{\phi}(h, \mathbf{z}^{(i)}) - \phi(h, \mathbf{z}^{(i)})$. Suppose that there exists a constant $c > 0$ and a map $\psi : \mathcal{Z} \to \mathbb{R}_+$ such that the following properties hold:

1. $\sup_{h \in \mathscr{H}} \|\phi(h, z)\|^2 + \sup_{h \in \mathscr{H}} \|\delta_\phi(z, h)\|^2 \le \psi(z)^2$.

2. For all $\delta \in (0, 1/e]$, we have $\mathbb{P}[\psi(\mathbf{z})^2 \vee \|\mathbf{e} - \mathbf{v}\|^2 \ge c\ln\delta^{-1}] \le \delta$.

3. $\mathscr{H}$ takes the form $\mathscr{H} = \{h(y) = M \cdot f(y) : f \in \mathscr{F} \quad M \in \mathbb{R}^{d_1 \times d_\times}, \quad \|M\|_{\mathrm{op}} \le b\}$ for some $b > 0$, where $\mathscr{F} : \mathcal{Y} \to \mathbb{R}^{d_\times}$ is a finite class and . Furthermore, there exists $L \ge 1$, matrices $X_1, X_2$ of appropriate dimension, and an arbitrary function $\delta_0 : \mathcal{Z} \to \mathcal{V}$ (which does not depend on $h$) such that

$$\forall f \in \mathscr{F}, \quad \|f(y)\|_2 \le L \max\{1, \|f_\star(y)\|\} \quad \text{for all } y \in \mathcal{Y}$$
$$\forall h \in \mathscr{H}, \quad \widehat{\phi}(h, z) := X_1(h(y_1) - X_2 h(y_2)) + \delta_0(z) \quad \text{for all } z = (y_1, y_2, y_3).$$

4. Finally, $c_\psi \geq 1$ satisifes the following for all all $z = (y_1, y_2, y_3)$

$$bL(\|X_1\|_{\mathrm{op}} + \|X_1 \cdot X_2\|_{\mathrm{op}})(2 + \|f_\star(y_1)\| + \|f_\star(y_2)\|) \leq 2c_\psi\psi(z).$$

Then, with probability at least $1 - \frac{3\delta}{2}$,

$$\mathbb{E}\|\phi(\hat{h}, \mathbf{z}) - \phi(h_\star, \mathbf{z})\|^2 \leq \frac{12c(\ln|\mathscr{F}| + d_1 d_\mathbf{x})\mathtt{logs}(c_\psi n, \delta)}{n} + 16\mathbb{E}\|\mathbf{e}\|^2 + 8\max_{h \in \mathscr{H}}\mathbb{E}\|\delta_\phi(h, \mathbf{z})\|^2,$$

where again we define $\mathtt{logs}(n, \delta) := 270\ln(2n/\delta)\ln(330n/\delta)$, so that $\mathtt{logs}(c_\psi n, \delta) \lesssim \ln^2(c_\psi n/\delta)$.

### E.1.6 Principal Component Analysis

**Proposition E.2** (PCA with errors). Let $\mathscr{H} \subseteq (\mathcal{Y} \to \mathbb{R}^d)$ be a function class, and let $\mathbf{y} \in \mathcal{Y}$ be a random variable. Suppose that there exists a function $\varphi : \mathcal{Y} \to \mathbb{R}_+$ and constants $c$, $L$ such that the following properties hold:

- $\|h(\mathbf{y})\|^2 \leq \max\{c, \varphi(\mathbf{y})\}$ for all $h \in \mathscr{H}$.
- $\varphi(\mathbf{y})$ is $c$-concentrated.

Let $h_\star \in \mathscr{H}$ be given, and let $\Lambda_\star := \mathbb{E}[h_\star(\mathbf{y})h_\star(\mathbf{y})^\top]$. Next, let $\hat{h} \in \mathscr{H}$ be given and define

$$\widehat{\Lambda}_n := \frac{1}{n}\sum_{i=1}^n \hat{h}(\mathbf{y}^{(i)})\hat{h}(\mathbf{y}^{(i)})^\top,$$

where $\mathbf{y}^{(i)} \overset{\text{i.i.d.}}{\sim} \mathbf{y}$. Then with probability $1 - \delta$, we have $\|\widehat{\Lambda}_n - \Lambda_\star\|_{\mathrm{op}} \leq \varepsilon_{\mathrm{pca}, n, \delta}$, where

$$\varepsilon_{\mathrm{pca}, n, \delta} := 3\sqrt{c \cdot \mathbb{E}[\|\hat{h}(\mathbf{y}) - h_\star(\mathbf{y})\|^2]} + 5cn^{-1/2}\ln(2dn/\delta)^{3/2}.$$

**Corollary E.3** (Significant basis overlap). Consider the setting of Proposition E.2, and suppose that $\Lambda_\star := \mathbb{E}[h_\star(\mathbf{y})h_\star^\top(\mathbf{y})] \in \mathbb{R}^{d \times d}$ has $\mathrm{rank}(\Lambda_\star) = d_\mathbf{x}$, so that $\lambda_{d_\mathbf{x}}(\Lambda_\star) > 0$. Let $V_\star \in \mathbb{R}^{d \times d_\mathbf{x}}$ denote be a matrix with orthonormal columns that span the column space the image of $\Lambda_\star$. Likewise, let $\widehat{V} \in \mathbb{R}^{d \times d_\mathbf{x}}$ be a matrix with orthonormal columns span the eigenspace of the top $d_\mathbf{x}$ eigenvectors of $\widehat{\Lambda}_n$. Suppose $\varepsilon_{\mathrm{pca}, n, \delta} \leq \frac{\lambda_{d_\mathbf{x}}(\Lambda_\star)}{4}$. Then on the good event for Proposition E.2, we have $\sigma_{d_\mathbf{x}}(V_\star^\top\widehat{V}_n) \geq 2/3$.

### E.1.7 Linear Regression

**Proposition E.3** (Linear regression with errors in variables). Let $(\mathbf{u}, \mathbf{y}, \mathbf{w}, \mathbf{e}, \boldsymbol{\delta})$ be a collection of random variables defined over a shared probability space, and let $\{(\mathbf{u}^{(i)}, \mathbf{y}^{(i)}\mathbf{w}^{(i)}, \mathbf{e}^{(i)}, \boldsymbol{\delta}^{(i)})\}_{i=1}^n$ be i.i.d. copies. Suppose the following conditions hold:

1. $\mathbf{y} = M_\star\mathbf{u} + \mathbf{w} + \mathbf{e}$ with probability 1, where $M_\star \in \mathbb{R}^{d_\mathbf{y} \times d_\mathbf{u}}$.
2. $\mathbf{w} \mid \mathbf{u}, \boldsymbol{\delta} \sim \mathcal{N}(0, \Sigma_w)$ and $\mathbf{u} \sim \mathcal{N}(0, \Sigma_u)$.
3. We have $\mathbb{E}\|\mathbf{e}\|^2 \leq \varepsilon_\mathbf{e}^2$ and $\mathbb{E}\|\boldsymbol{\delta}\|^2 \leq \varepsilon_{\boldsymbol{\delta}}^2$.
4. $\mathbf{e}$ is $c_\mathbf{e}$-concentrated and $\boldsymbol{\delta}$ is $c_{\boldsymbol{\delta}}$-concentrated for $c_\mathbf{e} \geq \varepsilon_\mathbf{e}^2$ and $c_{\boldsymbol{\delta}} \geq \varepsilon_{\boldsymbol{\delta}}^2$.
5. $\varepsilon_{\boldsymbol{\delta}}^2 \leq \frac{1}{16}\lambda_{\min}(\Sigma_u)$.

Let $\delta \leq 1/e$, and let $n \in \mathbb{N}$ satisfy

1. $\psi(n, \delta) \leq \min\left\{\frac{\varepsilon_\mathbf{e}^2}{c_\mathbf{e}}, \frac{\varepsilon_{\boldsymbol{\delta}}^2}{c_{\boldsymbol{\delta}}}\right\}$, where $\psi(n, \delta) := \frac{2\ln(2n/\delta)\ln(2/\delta)}{n}$.
2. $n \geq c_1(d_\mathbf{u} + \ln(1/\delta))$, for some universal constant $c_1 > 0$.

Then the solution to the least squares problem

$$\widehat{M} = \min_M \sum_{i=1}^n \|M(\mathbf{u}^{(i)} + \boldsymbol{\delta}^{(i)}) - \mathbf{y}^{(i)}\|^2,$$

satisfies the following inequality with probability at least $1 - 4\delta$:

$$\|\widehat{M} - M_\star\|_{\mathrm{op}}^2 \lesssim \lambda_{\min}(\Sigma_u)^{-1}\left(\|M_\star\|_{\mathrm{op}}^2\varepsilon_{\boldsymbol{\delta}}^2 + \varepsilon_{\mathbf{e}}^2 + \frac{\|\Sigma_w\|_{\mathrm{op}}(d_{\mathbf{y}} + d_{\mathbf{u}} + \ln(1/\delta))}{n}\right). \qquad \text{(E.2)}$$

**Proposition E.4.** Consider the setting of [Proposition E.3], and suppose we additionally require that $n \geq c_0(d_{\mathbf{y}} + \ln(1/\delta))$ for some (possibly inflated) universal constant $c_0$. Furthermore, suppose we have $\varepsilon_{\boldsymbol{\delta}}^2\|M_\star\|_{\mathrm{op}}^2 + \varepsilon_{\mathbf{e}}^2 \leq 2\lambda_+$ for some $\lambda_+ \geq \lambda_{\max}(\Sigma_w)$. Then, with probability at least $1 - 7\delta$, [(E.2)] holds, and moreover

$$\left\|\frac{1}{n}\sum_{i=1}^n(\widehat{M}(\mathbf{u}^{(i)} + \boldsymbol{\delta}^{(i)}) - \mathbf{y}^{(i)})^{\otimes 2} - \Sigma_w\right\|_{\mathrm{op}} \lesssim \sqrt{\lambda_+(\varepsilon_{\boldsymbol{\delta}}^2\|M_\star\|_{\mathrm{op}}^2 + \varepsilon_{\mathbf{e}}^2) + \frac{\lambda_+^2(d_{\mathbf{y}} + \ln(1/\delta))}{n}}.$$

### E.1.8 Regression with Matrix Measurements

**Proposition E.5** (Regression with matrix measurements). Let $\mathbf{y} \in \mathcal{Y}$ be a random variable, and let $\mathbf{y}^{(i)} \overset{\text{i.i.d.}}{\sim} \mathbf{y}$ for $1 \leq i \leq n$. Fix two regression functions $\hat{g}, g_\star : \mathcal{Y} \to \mathbb{R}^d$, and suppose that $\mathbf{z} := \max\{\|\hat{g}(\mathbf{y})\|^2, \|g_\star(\mathbf{y})\|^2\}$ is $c$-concentrated, and that $\mathbf{x} := g_\star(\mathbf{y}) \sim \mathcal{N}(0, \Sigma_x)$. Let $Q^\star \geq 0$ be a fixed matrix, and consider the regression.

$$\widetilde{Q} \in \underset{M}{\arg\min} \sum_{i=1}^n \left(g_\star(\mathbf{y}^{(i)})^\top Q^\star g_\star(\mathbf{y}^{(i)}) - \hat{g}(\mathbf{y}^{(i)})^\top M\hat{g}(\mathbf{y}^{(i)})^\top\right)^2.$$

Set $\widehat{Q} := (\frac{1}{2}\widetilde{Q}^\top + \frac{1}{2}\widetilde{Q})_+$, where $(\cdot)_+$ truncates all negative eigenvalues to zero. Then, there is a universal constant $c_0 > 0$ such that if the following conditions hold:

$$\mathbb{E}\|\hat{g}(\mathbf{y}) - g_\star(\mathbf{y})\|^2 \leq \varepsilon^2, \quad \psi(n, \delta/2) \leq \frac{\varepsilon^2}{4c}, \quad n \geq c_0(d^2 + \ln(1/\delta)), \quad \text{and} \quad \varepsilon^2 \leq \frac{\lambda_{\min}(\Sigma_x)^2}{64c\ln(2n/\delta)},$$

then with probability at least $1 - 2\delta$,

$$\|\widehat{Q} - Q^\star\|_{\mathrm{F}}^2 \leq \|\widetilde{Q} - Q^\star\|_{\mathrm{F}}^2 \leq 64c\varepsilon^2\ln(4n/\delta)\cdot\frac{\|Q^\star\|_{\mathrm{op}}^2}{\lambda_{\min}(\Sigma_x)^2}.$$

## E.2 Proofs for Technical Tools

### E.2.1 Proof of Lemma E.1

First observe that if $\mathbf{z}$ is $c$-concentrated, then for $\delta \in (0, 1/e]$, $\ln(1/\delta) \geq 1$, so that $\mathbb{P}[\alpha\mathbf{z} + \beta \geq (\alpha c + \beta)\ln(1/\delta)] \leq \mathbb{P}[\alpha\mathbf{z} \geq \alpha c\ln(1/\delta)] = \mathbb{P}[c\mathbf{z} \geq c\ln(1/\delta)]$. This is at most $\delta$ by the definition of the $c$-concentrated property. We now turn to the enumerated points.

**Point 1.** For $\delta \leq 1/e$, $\mathbb{P}[\mathbf{z} \geq c\ln(1/\delta)] \leq \delta$. Thus, for any $\delta \in (0, 1/e)$, and $u \geq c_\delta \geq c$, we have $\mathbb{P}[\mathbf{z} \geq u] \leq e^{-u/c}$. It follows that for any $\delta \leq 1/e$,

$$\mathbb{E}[\mathbf{z}\mathbb{I}(\mathbf{z} \geq c\ln(1/\delta))] = \int_{u=c\ln(1/\delta)}^\infty \mathbb{P}[\mathbf{z} \geq u]du$$

$$= \int_{u=c\ln(1/\delta)}^\infty e^{-u/c}du = ce^{-c\ln(1/\delta)/c} = c\delta.$$

A similar calculation reveals that

$$\mathbb{E}[\mathbf{z}^k] \leq c^k\int_{u=c^k}^\infty \mathbb{P}[\mathbf{z}^k \geq u]du$$

$$\leq c^k + \int_{u=c^k}^\infty e^{-\frac{u^{1/k}}{c}}du$$

$$= c^k + kc^k\int_{u=1}^\infty e^{-u}u^{k-1}du$$

$$= c^k(1 + k!).$$

**Point 2.** Define the increments $\Delta_i = z^{(i)} - \mathbb{E}[z^{(i)}]$, and $\Delta_{i,\delta} := \Delta_i \mathbb{I}(z^{(i)} \leq c_{n,\delta})$. By a union bound, $\Delta_i = \Delta_{i,\delta}$ for all $i \in [n]$ with probability at least $1 - \delta/2$. Moreover, $-c \leq -\mathbb{E}[z] \leq \Delta_{i,\delta} \leq c_{n,\delta}$, so that by Bennett's inequality ([25], Theorem 3), it holds that with probability at least $1 - \delta/2$,

$$\frac{1}{n}\sum_i \Delta_{i,\delta} \leq \sqrt{\frac{2\mathrm{VAR}[\Delta_{i,\delta}]\ln(2/\delta)}{n}} + \frac{c_{n,\delta}\ln(2/\delta)}{3n}$$

$$\leq \frac{\tau}{2c_{n,\delta}}\mathrm{VAR}[\Delta_{i,\delta}] + \left(\frac{1}{3} + \frac{1}{\tau}\right)\frac{c_{n,\delta}\ln(2/\delta)}{n},$$

for any $\tau > 0$. Moreover, we have $\mathrm{VAR}[\Delta_{i,\delta}] = \mathbb{E}[\Delta_{i,\delta}^2] \leq c_{n,\delta}\mathbb{E}|z^{(i)} - \mathbb{E}[z^{(i)}]| \leq 2c_{n,\delta}\mathbb{E}|z^{(i)}| = 2c_{n,\delta}\mathbb{E}[z^{(i)}]$ by non-negativity of $z^{(i)}$. Hence, with total probability at least $1 - \delta$, we have

$$\frac{1}{n}\sum_i \Delta_i = \frac{1}{n}\sum_i \Delta_i \leq \tau\mathbb{E}[z^{(i)}] + \left(\frac{1}{3} + \frac{1}{\tau}\right)\frac{c_{n,\delta}\ln(2/\delta)}{n}.$$

Recalling that $\Delta_i \leq z^{(i)} - \mathbb{E}[z^{(i)}]$ and taking $\tau = 1/2$, we have that with total probability at least $1 - \delta$,

$$\frac{1}{n}\sum_i z^{(i)} \leq \frac{3}{2}\mathbb{E}[z^{(i)}] + \frac{5}{6}\frac{c_{n,\delta}\ln(1/\delta)}{n}.$$

By assumption, $\mathbb{E}[z^{(i)}] \leq \varepsilon^2$. Hence, for

$$\frac{c_{n,\delta}\ln(2/\delta)}{n} \leq \frac{6}{5}\cdot\frac{1}{2}\varepsilon^2 \leq \varepsilon^2/2,$$

we have that $\frac{1}{n}\sum_i z^{(i)} \leq 2\varepsilon^2$. In particular, since

$$\psi(n,\delta) = \frac{2\ln(2n/\delta)\ln(2/\delta)}{n} = \frac{2c_{n,\delta}\ln(2/\delta)}{cn},$$

we have $\frac{1}{n}\sum_i z^{(i)} \leq 2\varepsilon^2$ for $\psi(n,\delta) \leq \varepsilon^2/c$.

It is simple to verify that all the steps above go through if $z \leq c_{\delta,n} = c\ln(2n/\delta)$ almost surely and $\mathbb{E}[z] \leq c$. Substituting in $c_{n,\delta} := c\ln(2n/\delta)$ concludes. $\qquad\square$

### E.2.2  Proof of Lemma E.2

First, observe that $\mathbb{E}[\|\mathbf{x}\|^2] = \mathrm{tr}(\Sigma)$. Next, from Hsu et al. [15, Proposition 1], we have that

$$\mathbb{P}[\|\mathbf{x}\|^2 \geq \mathrm{tr}(\Sigma) + 2\sqrt{t}\|\Sigma\|_\mathrm{F} + 2t\|\Sigma\|_\mathrm{op}] \leq e^{-t}.$$

Setting $t = \ln(1/\delta) \geq 1$ and bounding $\mathrm{tr}(\Sigma) + 2\sqrt{t}\|\Sigma\|_\mathrm{F} + 2t\|\Sigma\|_\mathrm{op} \leq 5t\cdot\mathrm{tr}(\Sigma) = 5\mathrm{tr}(\Sigma)\ln(1/\delta)$ concludes. $\qquad\square$

### E.2.3  Proof of Lemma E.3

Let $\mathcal{Q}_1$ denote the law of $\mathbf{y}_1$, $\mathcal{Q}_2$ the law of $\mathbf{y}_2$, $\mathcal{P}_1$ the law of $\mathbf{x}_1$, and $\mathcal{P}_2$ the law of $\mathbf{x}_2$. Let $q(y \mid x)$ denote the density of $y$ given $x$. We then have that

$$\mathbb{E}_{\mathbf{y}_2}[\|h_\star(\mathbf{y}_2) - \hat{h}(\mathbf{y}_2)\|^2] = \int_y \|h_\star(y) - \hat{h}(y)\|^2 \mathrm{d}\mathcal{Q}_2(y)$$

$$= \int_{x,y} q(x \mid y)\|h_\star(y) - \hat{h}(y)\|^2 \mathrm{d}\mathcal{P}_2(x)$$

$$= \int_{x,y} q(x \mid y)\frac{\mathrm{d}\mathcal{P}_2(x)}{\mathrm{d}\mathcal{P}_1(x)}\|h_\star(y) - \hat{h}(y)\|^2 \mathrm{d}\mathcal{P}_1(x).$$

Using the standard expression for the density for the multivariate Gaussian distribution, we have the identity

$$\frac{\mathrm{d}\mathcal{P}_2(x)}{\mathrm{d}\mathcal{P}_1(x)} = \det(\Sigma_1\Sigma_2^{-1})^{1/2}\cdot\exp\left(\frac{1}{2}x^\top(\Sigma_1^{-1} - \Sigma_2^{-1})x\right)$$

$$= \det(I + (\Sigma_1^{1/2}\Sigma_2^{-1}\Sigma^{1/2} - I))^{1/2}\cdot\exp\left(\frac{1}{2}x^\top\Sigma_1^{-1/2}(I - \Sigma_1^{1/2}\Sigma_2^{-1}\Sigma^{1/2})\Sigma_1^{-1/2}x\right).$$

Hence, if we set $\eta = \|(\Sigma_1^{1/2}\Sigma_2^{-1}\Sigma_1^{1/2} - I)\|_{\mathrm{op}}$, we have

$$\det(I + (\Sigma_1^{1/2}\Sigma_2^{-1}\Sigma_1^{1/2} - I)) = \prod_{i=1}^{d}\lambda_i(I + (\Sigma_1^{1/2}\Sigma_2^{-1}\Sigma_1^{1/2} - I))$$

$$\leq \prod_{i=1}^{d}(1+\eta)^d \leq \exp(d\eta).$$

Similarly, we may bound

$$\exp\left(\frac{1}{2}x^\top\Sigma_1^{-1/2}(I - \Sigma_1^{1/2}\Sigma_2^{-1}\Sigma_1^{1/2})\Sigma_1^{-1/2}x\right) \leq \exp\left(\frac{\eta}{2}x^\top\Sigma_1^{-1}x\right).$$

Thus,

$$\frac{\mathrm{d}\mathcal{P}_2(x)}{\mathrm{d}\mathcal{P}_1(x)} \leq \exp\left(\frac{\eta}{2}(d + x^\top\Sigma_1^{-1}x)\right).$$

In particular, for any $B > 0$, as long as

$$x^\top\Sigma_1^{-1}x \leq B, \quad \text{and} \quad \eta \leq \frac{2\ln(3/2)}{d+B}, \tag{E.3}$$

we have

$$\frac{\mathrm{d}\mathcal{P}_2(x)}{\mathrm{d}\mathcal{P}_1(x)} \leq 3/2.$$

Henceforth, fix a bound parameter $B$ and assume $\eta \leq \frac{2\ln(3/2)}{d+B} < 1$. We have

$$\mathbb{E}_{\mathbf{y}_2}[\|h_\star(\mathbf{y}_2) - \hat{h}(\mathbf{y}_2)\|^2] = \int_{x,y}q(x\mid y)\frac{\mathrm{d}\mathcal{P}_2(x)}{\mathrm{d}\mathcal{P}_1(x)}\|h_\star(y) - \hat{h}(y)\|^2\mathrm{d}\mathcal{P}_1(x)$$

$$\leq \underbrace{\frac{3}{2}\int_{x,y}q(x\mid y)\|h_\star(y) - \hat{h}(y)\|^2\mathrm{d}\mathcal{P}_1(x)\mathbb{I}(x^\top\Sigma_1^{-1}x \leq B)}_{:=\mathrm{Term}_1}$$

$$+ \underbrace{\int_{x,y}q(x\mid y)\exp\left(\frac{\eta}{2}(d + x^\top\Sigma_1^{-1}x)\right)\|h_\star(y) - \hat{h}(y)\|^2\mathrm{d}\mathcal{P}_1(x)\mathbb{I}(x^\top\Sigma_1^{-1}x > B)}_{:=\mathrm{Term}_2}.$$

To handle the first term, we use the assumed error bound between $h_\star$ and $\hat{h}$:

$$\mathrm{Term}_1 := \frac{3}{2}\int_{x,y}q(x\mid y)\|h_\star(y) - \hat{h}(y)\|^2\mathrm{d}\mathcal{P}_1(x)\mathbb{I}(x^\top\Sigma_1 x \leq B)$$

$$\leq \frac{3}{2}\int_{x,y}q(x\mid y)\|h_\star(y) - \hat{h}(y)\|^2\mathrm{d}\mathcal{P}_1(x) = \frac{3}{2}\mathbb{E}_{\mathbf{y}_1}\|h_\star(\mathbf{y}_1) - \hat{h}(\mathbf{y}_2)\|^2 \leq \frac{3\varepsilon^2}{2}. \tag{E.4}$$

For $\mathrm{Term}_2$, we use the bound $\|h_\star(y) - \hat{h}(y)\|^2 \leq 4L\max\{1, \|f_\star(y)\|^2\} = 4L^2\max\{1, \|x\|^2\}$ to bound

$$\int_{x,y}q(x\mid y)\exp\left(\frac{\eta}{2}(d + x^\top\Sigma_1 x)\right)\|h_\star(y) - \hat{h}(y)\|^2\mathrm{d}\mathcal{P}_1(x)\mathbb{I}(x^\top\Sigma_1^{-1}x > B)$$

$$\leq 4L^2 e^{\frac{d\eta}{2}}\int_x\exp\left(\frac{\eta}{2}x^\top\Sigma_1^{-1}x\right)(1 + \|x\|^2)\mathrm{d}\mathcal{P}_1(x)\mathbb{I}(x^\top\Sigma_1^{-1}x > B).$$

Let us change variables to $u = \Sigma^{-1/2}x$, and let $\mathcal{P}_0$ denote the density of $u$, which is precisely the density of a standard normal $\mathcal{N}(0, I)$ random variable. Then, using the formula the standard normal density,

$$\int_x\exp\left(\frac{\eta}{2}x^\top\Sigma_1^{-1}x\right)(1 + \|x\|^2)\mathbb{I}(x^\top\Sigma_1^{-1}x > B)\mathrm{d}\mathcal{P}_1(x)$$

$$= \int_u\exp\left(\frac{\eta}{2}\|u\|^2\right)\cdot(1 + \|\Sigma^{1/2}u\|^2)\cdot\mathbb{I}(\|u\|^2 > B)\mathrm{d}\mathcal{P}_0(u)$$

$$= \int_u\frac{1}{(2\pi)^{d/2}}\exp\left(-\frac{(1-\eta)}{2}\|u\|^2\right)\cdot(1 + \|\Sigma^{1/2}u\|^2)\cdot\mathbb{I}(\|u\|^2 > B)\mathrm{d}u$$

$$\leq \int_u\frac{1}{(2\pi)^{d/2}}\exp\left(-\frac{(1-\eta)}{2}\|u\|^2\right)(1 + \|\Sigma_1\|_{\mathrm{op}}\|u\|^2)\mathbb{I}(\|u\|^2 > B)\mathrm{d}u.$$

Again, let us rescale via $z \leftarrow (1-\eta)^{-1/2}u$. The determinant of the Jacobian of this transformation is $(1-\eta)^{d/2}$, so that for $B \geq 1$, this is equal to

$$(1-\eta)^{-d/2} \int_z \frac{1}{(2\pi)^{d/2}} \exp\left(-\frac{1}{2}\|u\|^2\right)(1+(1-\eta)\|\Sigma_1\|_{\mathrm{op}}\|z\|^2)\mathbb{I}(\|z\|^2 > B(1-\eta)^{-1})\mathrm{d}z$$

$$= (1-\eta)^{-d/2}\mathbb{E}_{z\sim\mathcal{N}(0,I)}\left[(1+(1-\eta)\|\Sigma_1\|_{\mathrm{op}}\|z\|^2)\mathbb{I}(\|z\|^2 \geq B(1-\eta)^{-1})\right]$$

$$\overset{(i)}{\leq} e^{\eta d}\mathbb{E}_{z\sim\mathcal{N}(0,I)}\left[(1+\|\Sigma_1\|_{\mathrm{op}}\|z\|^2)\mathbb{I}(\|z\|^2 \geq B(1-\eta)^{-1})\right]$$

$$\leq (1+\|\Sigma_1\|_{\mathrm{op}})e^{\eta d}\mathbb{E}_{z\sim\mathcal{N}(0,I)}\left[\|z\|^2\mathbb{I}(\|z\|^2 \geq B(1-\eta)^{-1})\right],$$

where in $(i)$ we observe that $(1-\eta)^{-1/\eta} \leq e^2$ for $\eta \leq 1/2$, and where the last inequality uses that $B \geq 1$. Now, from Lemma E.2, we have that $\|z\|^2$ is $5d$-concentrated. Hence, for $\eta \leq 1/2$, $(1-\eta)^{-1}\|z\|^2$ is $10d$-concentrated. Thus $B = 10d\ln(1/\delta)$ gives $\mathbb{E}_{z\sim\mathcal{N}(0,I)}\left[\|z\|^2\mathbb{I}((1-\eta)^{-1}\|z\|^2 \geq B)\right] \leq 10d\delta$ by Lemma E.1., and therefore

$$\mathrm{Term}_2 = \int_{x,y} q(x \mid y)\exp\left(\frac{\eta}{2}(2d+x^\top\Sigma_1 x)\right)\|h_\star(y) - \hat{h}(y)\|^2\mathrm{d}\mathcal{P}_1(x)\mathbb{I}(x^\top\Sigma_1^{-1}x > B)$$

$$\leq 4(1+\|\Sigma_1\|_{\mathrm{op}})L^2 e^{\eta d}\cdot e^{\eta d/2}\cdot 10d\delta = \delta\cdot(1+\|\Sigma_1\|_{\mathrm{op}})40L^2 e^{3d\eta/2}.$$

In particular, if $\eta \leq 1/2d$ and $\delta = \frac{\varepsilon^2}{80L^2\|\Sigma_1\|_{\mathrm{op}}e}$, we have $\mathrm{Term}_2 \leq \frac{\varepsilon^2}{2}$, and thus $\mathrm{Term}_1 + \mathrm{Term}_2 \leq 2\varepsilon^2$. Gathering our conditions, we require $\eta \leq 1/\max\{2,d\}$, $B = 10d\ln\left(\frac{\varepsilon^2}{80L^2(1+\|\Sigma_1\|_{\mathrm{op}})e}\right)$, and—from Eq. (E.3)—$\eta \leq \frac{2\ln(3/2)}{d+B}$. Altogether, it suffices to select

$$\eta \leq \frac{2\ln(3/2)}{11d\ln\left(\frac{\varepsilon^2}{80L^2(1+\|\Sigma_1\|_{\mathrm{op}})e}\right)} \leq \frac{1}{14d\ln\left(\frac{\varepsilon^2}{80L^2(1+\|\Sigma_1\|_{\mathrm{op}})e}\right)}.$$

$\square$

### E.2.4 Proof of Lemma E.4

By the tower rule and the fact that $\mathbf{u} \to \mathbf{x} \to \mathbf{y}$ is a Markov chain, $\mathbb{E}[\mathbf{u} \mid \mathbf{y} = y] = \mathbb{E}[\mathbb{E}[\mathbf{u} \mid \mathbf{x}, \mathbf{y} = y] \mid \mathbf{y} = y] = \mathbb{E}[\mathbb{E}[\mathbf{u} \mid \mathbf{x}] \mid \mathbf{y} = y]$. Moreover, from decodability, $\mathbb{E}[\mathbb{E}[\mathbf{u} \mid \mathbf{x}] \mid \mathbf{y} = y] = \mathbb{E}[\mathbb{E}[\mathbf{u} \mid \mathbf{x} = f_\star(y)] \mid \mathbf{y} = y] = \mathbb{E}[\mathbf{u} \mid f_\star(y) = \mathbf{x}] = h_\star(y)$.

For the second point, It is well know that any *unrestricted* minimizer of $\|h(\mathbf{y}) - \mathbf{u}\|^2$ over all measurable $h$ satisfies $h = h_0$ almost surely, where $h(y) \coloneqq \mathbb{E}[\mathbf{u} \mid \mathbf{y} = y]$. We verify above that $h_\star(y) = \mathbb{E}[\mathbf{u} \mid \mathbf{y} = y]$, proving the that any unrestricted minimizer $h$ coincideds with $h_\star$. Since $h_\star \in \mathscr{H}$, the same holds for the function class constraint in the lemma statement. $\square$

### E.2.5 Proof of Lemma E.5

Our task is to bound $\mathcal{N}(b\epsilon, \mathscr{M}, \|\cdot\|_{\mathrm{op}})$, where we recall $\mathscr{M} \coloneqq \{M \in \mathbb{R}^{dd_\times} : \|M\|_{\mathrm{op}} \leq b\}$. By rescaling, it suffices to bound $\mathcal{N}(\epsilon, \frac{1}{b}\mathscr{M}, \|\cdot\|_{\mathrm{op}})$. We recognize $\mathscr{M}$ as the operator norm ball in $\mathbb{R}^{d\times d_\times}$ and appeal to the following standard lemma.

**Lemma E.6** ([43], Lemma 5.2). Let $\mathcal{B}$ be the unit ball in $\mathbb{R}^d$ for an arbitrary norm. Then, if dist is the metric induced by the norm, $\mathcal{N}(\epsilon, \mathcal{B}, \mathrm{dist}) \leq (1+\frac{2}{\epsilon})^d$.

$\square$

### E.2.6 Proof of Proposition E.1

Before diving into the meat of the proof, we first establish some basic concentration properties and state a number of definitions. For each realization $(\mathbf{y}, \mathbf{e}, \mathbf{u})$, define

$$\mathcal{E} \coloneqq \{\|\mathbf{e} - \mathbf{u}\|^2 \vee \varphi(\mathbf{y}) \leq c_{n,\delta}\},$$

where we recall that $c_{n,\delta} \coloneqq c\ln(2n/\delta)$. Let $\ell(h) \coloneqq \|h(\mathbf{y}) + \mathbf{e} - \mathbf{u}\|^2$, and let $\mathcal{L}(h) = \mathbb{E}[\ell(h)]$. Furthermore, define $\ell_\delta(h) = \ell(h)\mathbb{I}(\mathcal{E})$ and $\mathcal{L}_\delta(h) = \mathbb{E}[\ell_\delta(h)]$. We first establish the following useful claim.

**Claim E.1.** Then on $\mathcal{E}$, $|\ell(h) - \ell(h')| \le 4\sqrt{c_{n,\delta}}\|h(\mathbf{y}) - h'(\mathbf{y})\|$. Moreover, defining $\mathbf{z} = \|\mathbf{e} - \mathbf{u}\|^2 \vee \varphi(y)$, we have that $\ell(h) \le 4\mathbf{z}$. In particular, on $\mathcal{E}$, $\ell(h) \le 4c_{n,\delta}$.

*Proof of Claim E.1.*
$$
\begin{aligned}
|\ell(h) - \ell(h')| &= \big|\|h'(\mathbf{y}) + \mathbf{e} - \mathbf{u}\|^2 - \|h(\mathbf{y}) + \mathbf{e} - \mathbf{u}\|^2\big| \\
&\le 2|\langle h(\mathbf{y}) - h'(\mathbf{y}), \mathbf{e} - \mathbf{u}\rangle| + (\|h(\mathbf{y})\| + \|h'(\mathbf{y})\|)(\|h(\mathbf{y})\| - \|h'(\mathbf{y})\|) \\
&\le 4\sqrt{c_{n,\delta}}\|h(\mathbf{y}) - h'(\mathbf{y})\|.
\end{aligned}
$$
This proves the first claim. The claim holds because $\ell(h) \le 2\|\mathbf{e} - \mathbf{u}\|^2 + 2\|h(\mathbf{y})^2\| \le 2\|\mathbf{e} - \mathbf{u}\|^2 + 2\max\{\varphi(\mathbf{y}), c\} \le 4\max\{\mathbf{z}, c\}$. $\qquad\square$

Next, Let $\mathcal{E}^{(i)}$ denote the event that $\mathcal{E}$ holds for the $i$th sample, and let $\mathcal{E}_{1:n} = \bigcup_{i\in[n]} \mathcal{E}^{(i)}$. Note that $\mathcal{E}_{1:n}$ occurs with probability at least $1 - 2 \cdot \delta/2 = 1 - \delta$ by the $c$-concentration property and a union bound. On this event, if we define

$$
\mathcal{L}_{n,\delta}(h) := \sum_{i=1}^{n} \ell_{i,\delta}(h), \quad \text{where} \quad \ell_{i,\delta}(h) := \mathbb{I}(\mathcal{E}^{(i)})\|h(\mathbf{y}^{(i)}) + \mathbf{e}^{(i)} - \mathbf{u}^{(i)}\|^2,
$$

we have

$$
\hat{h}_n := \arg\min_{h\in\mathcal{H}} \mathcal{L}_{n,\delta}(h).
$$

Lastly, define the excess risk with respect to the Bayes function $h_\star(y) = \mathbb{E}[\mathbf{u} \mid \mathbf{y} = y]$:

$$
\mathcal{R}_{n,\delta}(h) = \mathcal{L}_{n,\delta}(h) - \mathcal{L}_{n,\delta}(h_\star), \quad \mathcal{R}_\delta(h) = \mathcal{L}_\delta(h) - \mathcal{L}_\delta(h_\star).
$$

Finally, let $\mathcal{H}_0 \subset \mathcal{H}$ denote a finite cover for $\mathcal{H}$ such that, for some $\epsilon > 0$ to be selected at the end of the proof,

$$
\sup_{h\in\mathcal{H}} \inf_{h'\in\mathcal{H}_0} \sup_{y:\varphi(y)\le c_{n,\delta}} \|h(y) - h'(y)\| \le \sqrt{c_{n,\delta}}\epsilon, \tag{E.5}
$$

and let $\hat{h}_0 \in \mathcal{H}_0$ denote the element that witnesses the covering inequality above for $\hat{h}_n$. Note that by Claim E.1 and (E.5), the differences on the truncated losses between $\hat{h}_n$ and $h_0$ satisfy

$$
|\mathcal{L}_{n,\delta}(\hat{h}_n) - \mathcal{L}_{n,\delta}(\hat{h}_0)| \vee |\mathcal{L}_\delta(\hat{h}_n) - \mathcal{L}_\delta(\hat{h}_0)| \le 4\epsilon c_{n,\delta},
$$

whenever $\mathcal{E}_{1:n}$ holds. Thus, on $\mathcal{E}_{1:n}$, when $\hat{h}_n \in \arg\min_{h\in\mathcal{H}} \mathcal{R}_{n,\delta}(h)$, we have

$$
\begin{aligned}
\mathcal{R}_\delta(\hat{h}_n) &= \mathcal{R}_\delta(\hat{h}_n) - \mathcal{R}_{n,\delta}(\hat{h}_n) + \mathcal{R}_{n,\delta}(\hat{h}_n) \\
&\overset{(i)}{\le} \mathcal{R}_\delta(\hat{h}_n) - \mathcal{R}_{n,\delta}(\hat{h}_n) \\
&\le \mathcal{R}_\delta(\hat{h}_0) - \mathcal{R}_{n,\delta}(\hat{h}_0) + 2\max\{|\mathcal{L}_\delta(\hat{h}_0) - \mathcal{L}_\delta(\hat{h}_n)|, |\mathcal{L}_{n,\delta}(\hat{h}_0) - \mathcal{L}_{n,\delta}(\hat{h}_n)|\} \\
&\le \mathcal{R}_\delta(\hat{h}_0) - \mathcal{R}_{n,\delta}(\hat{h}_0) + 8c_{n,\delta}\epsilon., \tag{E.6}
\end{aligned}
$$

where $(i)$ uses that $\mathcal{R}_{n,\delta}(\hat{h}_n)$ is non-positive for the empirical risk minimizer.

**Step 1: Bounding $\mathcal{R}_\delta(n)$.** From the bound $\ell_{i,\delta}(h) \le 4c_{n,\delta}$ (Claim E.1), along with Bennett's inequality (see e.g. Theorem 3 of [25]) and a union bound over $\mathcal{H}_0$, we have, for all $\delta \in (0,1)$, with probability at least $1 - \delta/2$,

$$
\begin{aligned}
\mathcal{R}_\delta(\hat{h}_0) - \mathcal{R}_{n,\delta}(\hat{h}_0) &\le \sqrt{2n^{-1}\mathrm{VAR}[\ell_{i,\delta}(\hat{h}_0) - \ell_{i,\delta}(h_\star)] \cdot \ln(2|\mathcal{H}_0|\delta^{-1})} + \frac{4c_{n,\delta}}{3}\ln(2|\mathcal{H}_0|\delta^{-1}) \\
&\le \frac{\tau}{2c_{n,\delta}}\mathrm{VAR}[\ell_{i,\delta}(\hat{h}_0) - \ell_{i,\delta}(h_\star)] + \frac{c_{n,\delta}}{n}\left(\frac{4}{3} + \frac{1}{\tau}\right)\ln(2|\mathcal{H}_0|\delta^{-1}), \tag{E.7}
\end{aligned}
$$

where the last step uses AM-GM and holds for all $\tau > 0$. Again, by Claim E.1, we have

$$
\begin{aligned}
\mathrm{VAR}[\ell_{i,\delta}(\hat{h}_0) - \ell_{i,\delta}(h_\star)] &\le \mathbb{E}\left[\mathbb{I}(\mathcal{E})\left(\ell(\hat{h}_0) - \ell(h_\star)\right)^2\right] \\
&\le 16c_{n,\delta}\mathbb{E}\left[\mathbb{I}(\mathcal{E})\|\hat{h}_0(\mathbf{y}) - h_\star(\mathbf{y})\|^2\right] \\
&\le 32c_{n,\delta}\mathbb{E}\left[\mathbb{I}(\mathcal{E})\|\hat{h}_0(\mathbf{y}) - \hat{h}_n(\mathbf{y})\|^2\right] + 32c_{n,\delta}\mathbb{E}\left[\mathbb{I}(\mathcal{E})\|\hat{h}_n(\mathbf{y}) - h_\star(\mathbf{y})\|^2\right].
\end{aligned}
$$

From Eq. (E.5), we have $\mathbb{E}\left[\mathbb{I}(\mathcal{E})\|\hat{h}_0(\mathbf{y}) - \hat{h}_n(\mathbf{y})\|^2\right] \leq c_{n,\delta}\epsilon^2$ Moreover, we can always upper bound $\mathbb{E}\left[\mathbb{I}(\mathcal{E})\|\hat{h}_n(\mathbf{y}) - h_\star(\mathbf{y})\|^2\right]$ by removing the indicator. This ultimately yields

$$\mathrm{VAR}[\ell_{i,\delta}(\hat{h}_0) - \ell_{i,\delta}(h_\star)] \leq 32c_{n,\delta}^2\epsilon^2 + 32c_{n,\delta}\mathbb{E}\left[\|\hat{h}_n(\mathbf{y}) - h_\star(\mathbf{y})\|^2\right].$$

Thus, combining the above with Eqs. (E.6) and (E.7), we have

$$\mathcal{R}_\delta(\hat{h}_n) \leq 16\tau\mathbb{E}\left[\|\hat{h}_n(\mathbf{y}) - h_\star(\mathbf{y})\|^2\right] + 16c_{n,\delta}(\epsilon + \tau\epsilon^2) + \frac{c_{n,\delta}}{n}\left(\frac{4}{3} + \frac{1}{\tau}\right)\ln(2|\mathscr{H}_0|\delta^{-1}). \quad \text{(E.8)}$$

**Step 2: Relating $\mathcal{R}_\delta(h)$ to error against $h_\star$.** Recall that $\mathcal{L}_\delta(h) \leq \mathcal{L}(h)$ due to truncation, so that

$$\mathcal{R}_\delta(h) = \mathcal{L}_\delta(h) - \mathcal{L}_\delta(h_\star) \geq \mathcal{L}_\delta(h) - \mathcal{L}(h_\star)$$
$$\geq \mathcal{L}(h) - \mathcal{L}(h_\star) - |\mathcal{L}_\delta(h) - \mathcal{L}(h)|. \quad \text{(E.9)}$$

We further develop

$$\mathcal{L}(h) - \mathcal{L}(h_\star) = \mathbb{E}\left[\|h(\mathbf{y}) + \mathbf{e} - \mathbf{u}\|^2 - \|h_\star(\mathbf{y}) + \mathbf{e} - \mathbf{u}\|^2\right]$$
$$= \mathbb{E}\left[\|h(\mathbf{y}) - \mathbf{u}\|^2 - \|h_\star(\mathbf{y}) - \mathbf{u}\|^2\right] + 2\mathbb{E}\langle\mathbf{e}, h(\mathbf{y}) - h_\star(\mathbf{y})\rangle$$
$$\geq \mathbb{E}\left[\|h(\mathbf{y}) - \mathbf{u}\|^2 - \|h_\star(\mathbf{y}) - \mathbf{u}\|^2\right] - 2\mathbb{E}\|\mathbf{e}\|^2 - \frac{1}{2}\mathbb{E}\|h(\mathbf{y}) - h_\star(\mathbf{y})\|^2,$$

where the last line uses Cauchy-Schwartz and AM-GM Moreover, since $h_\star = \mathbb{E}[\mathbf{y} \mid \mathbf{u}]$, we can see that $\mathbb{E}\left[\|h(\mathbf{y}) - \mathbf{u}\|^2 - \|h_\star(\mathbf{y}) - \mathbf{u}\|^2\right] = \mathbb{E}\|h(\mathbf{y}) - h_\star(\mathbf{y})\|^2$. This yields

$$\mathcal{L}(h) - \mathcal{L}(h_\star) \geq -2\mathbb{E}\|\mathbf{e}\|^2 + \frac{1}{2}\mathbb{E}\|h(\mathbf{y}) - h_\star(\mathbf{y})\|^2.$$

Hence, Eq. (E.9) yields that for all $h$,

$$\mathbb{E}\|h(\mathbf{y}) - h_\star(\mathbf{y})\|^2 \leq 2\mathcal{R}_\delta(h) + 4\mathbb{E}\|\mathbf{e}\|^2 + 2|\mathcal{L}_\delta(h) - \mathcal{L}(h)|.$$

Finally, recalling $\mathbf{z} = \varphi(\mathbf{y})^2 \vee \|\mathbf{e} - \mathbf{u}\|^2$, we have

$$\sup_{h\in\mathscr{H}} 2|\mathcal{L}_\delta(h) - \mathcal{L}(h)| = \sup_{h\in\mathscr{H}} 2\mathbb{E}[\mathbb{I}(\mathcal{E}^c)\ell(h)]$$
$$\leq 8\mathbb{E}[\mathbb{I}(\mathcal{E}^c)\max\{c, \mathbf{z}\}] \qquad \text{(Claim E.1)}$$
$$\leq \frac{8c\delta}{3n}. \qquad \text{(Lemma E.1)}$$

Hence, the previous two displays give

$$\mathbb{E}\|h(\mathbf{y}) - h_\star(\mathbf{y})\|^2 \leq 2\mathcal{R}_\delta(h) + 4\mathbb{E}\|\mathbf{e}\|^2 + \frac{8c\delta}{3n}.$$

Thus, choosing $h = \hat{h}_n$ and combining with Eq. (E.8), we have

$$\mathbb{E}\|\hat{h}_n(\mathbf{y}) - h_\star(\mathbf{y})\|^2 \leq 32\tau\mathbb{E}\left[\|\hat{h}_n(\mathbf{y}) - h_\star(\mathbf{y})\|^2\right] + 32c_{n,\delta}(\epsilon + \tau\epsilon^2)$$
$$+ 2\frac{c_{n,\delta}}{n}\left(\frac{4}{3} + \frac{1}{\tau}\right) + \frac{8c\delta}{3n} + 4\,\mathbb{E}\|\mathbf{e}\|^2.$$

Setting $\tau = \frac{1}{64}$ and using $\epsilon \leq 1$ gives

$$\mathbb{E}\|\hat{h}_n(\mathbf{y}) - h_\star(\mathbf{y})\|^2 \leq \frac{1}{2}\mathbb{E}\left[\|\hat{h}_n(\mathbf{y}) - h_\star(\mathbf{y})\|^2\right]$$
$$+ 33c_{n,\delta}\epsilon + \frac{c_{n,\delta}}{n}\left(\frac{8}{3} + 128\right)\ln(2|\mathscr{H}_0|\delta^{-1}) + 4\mathbb{E}\|\mathbf{e}\|^2 + \frac{8c\delta}{3n}$$
$$\leq \frac{1}{2}\mathbb{E}\left[\|\hat{h}_n(\mathbf{y}) - h_\star(\mathbf{y})\|^2\right] + 33c_{n,\delta}\epsilon + \frac{c_{n,\delta}}{n}134\ln(2|\mathscr{H}_0|\delta^{-1}) + 4\mathbb{E}\|\mathbf{e}\|^2,$$

where in the last line we folded the $8c\delta/3n$ term into the term with the log, bounding $8/3 + 8\delta/3n \leq 16/3 \leq 6$. Rearranging the above yields

$$\mathbb{E}\|\hat{h}_n(\mathbf{y}) - h_\star(\mathbf{y})\|^2 \leq 66c_{n,\delta}\epsilon + \frac{268c_{n,\delta}}{n}\ln(2|\mathscr{H}_0|\delta^{-1}) + 8\mathbb{E}\|\mathbf{e}\|^2.$$

Taking $\epsilon = 1/33n$ concludes the proof.

$\square$

### E.2.7 Proof of Corollary E.1

We verify Conditions 1-3 of Proposition E.1 in succession:

1. Condition 1: By assumption 1 of the corollary, $f(y) \le L \max\{1, \|f_\star(y)\|_2\}$, then $h(y) \le bL \max\{1, \|f_\star(y)\|_2\} := \varphi(y)$ for all $h \in \mathcal{H}$.

2. Condition 2: This is satisfied by assumption 2 of the corollary, $\varphi(\mathbf{y})^{1/2}$ and $\|\mathbf{e} - \mathbf{u}\|^2$ are $c$-concentrated.

3. Condition 3: We bound the covering number. Let $\mathcal{M} := \{M \in \mathbb{R}^{d_u d_x} : \|M\|_{\mathrm{op}} \le b\}$. Then, for an $\epsilon > 0$ to be chosen, let $\mathsf{N}(b\epsilon, \mathcal{M}, \|\cdot\|_{\mathrm{op}}) \le (1 + 2/\epsilon)^{d_u d_x}$ from Lemma E.5, so we may take a $b\epsilon$- cover $\mathcal{M}_\epsilon$ of $\mathcal{M}$ to have cardinality $(1 + 2/\epsilon)^{d_u d_x}$. Define the induced cover $\mathcal{H}_\epsilon := \{Mf : M \in \mathcal{M}_\epsilon, f \in \mathcal{F}\}$, which has $|\mathcal{H}_\epsilon| \le |\mathcal{F}|(1 + 2/\epsilon)^{d_u d_x}$. Given $h = Mf \in \mathcal{H}$, let $h' : M'f$, where $M' \in \mathcal{M}_\epsilon$ satisfies $\|M - M'\|_{\mathrm{op}} \le b\epsilon$. Then,

$$
\begin{aligned}
d_{c', \infty}(h, h') &:= \sup_{y \in \mathcal{Y}} \{\|h(y) - h'(y)\| : \varphi(y)^{1/2} \le \sqrt{c'}\} \\
&:= \sup_{y \in \mathcal{Y}} \{\|(M - M')f(y)\| : \varphi(y)^{1/2} \le \sqrt{c'}\} \\
&\le \sup_{y \in \mathcal{Y}} \{b\epsilon \cdot \|f(y)\| : \varphi(y)^{1/2} \le \sqrt{c'}\} \qquad (\|M - M'\|_{\mathrm{op}} \le b\epsilon) \\
&\le \sup_{y \in \mathcal{Y}} \{b\epsilon \cdot L \max\{1, \|f_\star(y)\|\} : \varphi(y)^{1/2} \le \sqrt{c'}\}
\end{aligned}
$$

(Assumption 1 of Corollary)

$$
\le \sup_{y \in \mathcal{Y}} \{b\epsilon \cdot L \max\{1, \|f_\star(y)\|\} : bL \max\{1, \|f_\star(y)\|\} \le \sqrt{c'}\}\}
$$

(Definition of $\varphi$)

$$
= \epsilon\sqrt{c'}.
$$

Hence, the $\sqrt{c'}\epsilon$ cover of $\mathcal{H}$ in the metric $d_{c', \infty}(h, h')$ is at most the cardinality of $\mathcal{H}_\epsilon$, which is at most $|\mathcal{F}|(1 + 2/\epsilon)^{d_u d_x}$. Thus, we can take $\ln \mathsf{N}(\epsilon) = \ln|\mathcal{F}| + d_u d_x \ln(1 + 2/\epsilon)$ in Condition 3 of Proposition E.1. For $\epsilon \le 1$, this may be upper bounded by $\ln \mathsf{N}(\epsilon) = \ln|\mathcal{F}| + d_u d_x \ln(5/\epsilon)$.

Hence, the conclusion of Proposition E.1 entails that, with probability at least $1 - \frac{3\delta}{2}$,

$$
\begin{aligned}
&\mathbb{E}\|\hat{h}_n(\mathbf{y}) - h_\star(\mathbf{y})\|^2 \\
&\le \frac{270c \ln(2n/\delta)(\ln(2/\delta) + \ln|\mathcal{F}| + d_u d_x \ln(5 \cdot 33n))}{n} + 8\mathbb{E}\|\mathbf{e}\|^2.
\end{aligned}
$$

Recalling that $\mathtt{logs}(n, \delta) := 270 \ln(2n/\delta) \ln(330n/\delta)$, we simplify

$$
\begin{aligned}
270 \cdot \ln(2n/\delta) &\cdot (\ln(2/\delta) + \ln|\mathcal{F}| + d_1 d_\mathbf{x} \ln(3 \cdot 55n)) \\
&\le 270 \ln(2n/\delta)(\ln|\mathcal{F}| + d_1 d_\mathbf{x} \ln(330n/\delta)| \\
&\le (\ln|\mathcal{F}| + d_1 d_\mathbf{x})270 \ln(2n/\delta) \ln(330n/\delta) := (\ln|\mathcal{F}| + d_1 d_\mathbf{x})\mathtt{logs}(n, \delta) \qquad (\text{E.10})
\end{aligned}
$$

which yields our final bound of

$$
\mathbb{E}\|\hat{h}_n(\mathbf{y}) - h_\star(\mathbf{y})\|^2 \le \frac{c(d_u d_x + \ln|\mathcal{F}|) \cdot \mathtt{logs}(n, \delta)}{n} + 8\mathbb{E}\|\mathbf{e}\|^2,
$$

as needed.

$\square$

### E.2.8 Proof of Corollary E.2

We consider

$$
\hat{h} := \arg\min_{h \in \mathcal{H}} \sum_{i=1}^{n} \|\widehat{\phi}(h, \mathbf{z}^{(i)}) + \mathbf{e}^{(i)} - \mathbf{v}^{(i)}\|^2.
$$

Recall the assumption that, for some $h_\star \in \mathcal{H}$, $\phi(h_\star, \mathbf{z}) = \mathbb{E}[\mathbf{u} \mid \mathbf{z}]$, and that $\delta_\phi(h, z) := \phi(h, z) - \phi(h, z)$. Let us set up a correspondence with Proposition E.1.

- $g_h(\mathbf{z}) := \widehat{\phi}(h, \mathbf{z}) - \delta_\phi(h_\star, \mathbf{z})$. Let $\mathcal{G}$ denote the resulting class of functions $\{g_h : h \in \mathcal{H}\}$.
- $\tilde{\mathbf{e}} = \delta_\phi(h_\star, \mathbf{z}) + \mathbf{e}$.

Then, we have

$$\widehat{\phi}(\hat{h}) - \delta_\phi(h_\star, \mathbf{z}) = g_{\hat{h}}(\mathbf{y}) := \arg\min_{g \in \mathcal{G}} \sum_{i=1}^{n} \|g(\mathbf{z}) + \tilde{\mathbf{e}}^{(i)} - \mathbf{v}^{(i)}\|^2.$$

Define $g_\star := g_{h_\star} = \phi(h_\star, \mathbf{z})$ and $\widehat{g} = g_{\hat{h}}$. We then have

$$\begin{aligned}
\mathbb{E}\|\phi(\hat{h}, \mathbf{z}) - \phi(h_\star, \mathbf{z})\|^2 &= \mathbb{E}\|\widehat{g}(\mathbf{z}) - g_\star(\mathbf{z}) + \delta_\phi(\hat{h}, \mathbf{z}) - \delta_\phi(h_\star, \mathbf{z})\|^2 \\
&= 2\mathbb{E}\|\widehat{g}(\mathbf{z}) - g_\star(\mathbf{z})\|^2 + 2\mathbb{E}\|\delta_\phi(\hat{h}, \mathbf{z}) - \delta_\phi(h_\star, \mathbf{z})\|^2 \\
&\leq 2\mathbb{E}\|\widehat{g}(\mathbf{z}) - g_\star(\mathbf{z})\|^2 + 8 \max_{h \in \mathcal{H}} \mathbb{E}\|\delta_\phi(h, \mathbf{z})\|^2 \qquad \text{(E.11)}
\end{aligned}$$

It remains to bound $\mathbb{E}\|\widehat{g}(\mathbf{z}) - g_\star(\mathbf{z})\|^2$.

Note that $g_\star \in \mathcal{G}$, and moreover $g_\star(z) = \phi(h_\star, \mathbf{z})$, which is equal to $\mathbb{E}[\mathbf{u} \mid \mathbf{z}]$ by assumption. Considering the function class $\mathcal{G} = \{g_h : h \in \mathcal{H}\}$ as the function class, $g_\star$ as the Bayes regressor, $\mathbf{v}$ as the target, and $\tilde{\mathbf{e}}$ as the residual noise, let us verify with conditions of Proposition E.1, albeit with slightly inflated constants. We have

1. Define $\tilde{\varphi}(z) = 6\psi^2(z)$. We bound $\|g_h(z)\|^2 \leq \tilde{\varphi}(z)$ via

$$\begin{aligned}
\|g_h(z)\|^2 &\leq (\|\phi(h, z)\| + \|\delta_\phi(h, z)\| + \|\delta_\phi(h_\star, z)\|)^2 \\
&\leq (\sup_{h \in \mathcal{H}} \|\phi(h, z)\| + 2 \sup_{h \in \mathcal{H}} \|\delta_\phi(h, z)\|)^2 \leq 2 \sup_{h \in \mathcal{H}} \|\phi(h, z)\|^2 + 6 \sup_{h \in \mathcal{H}} \|\delta_\phi(h, z)\|^2 \\
&\leq 6(\sup_{h \in \mathcal{H}} \|\phi(h, z)\|^2 + \sup_{h \in \mathcal{H}} \|\delta_\phi(h, z)\|^2) \leq 6\psi(z)^2 := \tilde{\varphi}(z),
\end{aligned}$$

   where the last inequality follows by the first assumption of the lemma.

2. Next, we establish the concentration property for $\tilde{\varphi}(\mathbf{z}) \vee \|\tilde{\mathbf{e}}_t - \mathbf{v}_t\|^2$ that, for $\tilde{c} = 6c$, we have

$$\mathbb{P}[\tilde{\varphi}(\mathbf{z}) \vee \|\tilde{\mathbf{e}}_t - \mathbf{v}_t\|^2 \geq \tilde{c}\ln(1/\delta)] \leq 1/\delta. \qquad \text{(E.12)}$$

   We have that

$$\begin{aligned}
\tilde{\varphi}(z) \vee \|\tilde{\mathbf{e}}_t - \mathbf{v}_t\|^2 &= \tilde{\varphi}(z) \vee \|\delta_\phi(z, h_\star) + \mathbf{e}_t - \mathbf{v}_t\|^2 \\
&\leq \tilde{\varphi}(z) \vee (2\|\delta_\phi(h_\star, z)\|^2 + 2\|\mathbf{e}_t - \mathbf{v}_t\|^2) \\
&\leq (\tilde{\varphi}(z) \vee 2\|\delta_\phi(h_\star, z)\|^2) + 2\|\mathbf{e}_t - \mathbf{v}_t\|^2).
\end{aligned}$$

   Now, by assumption, we have that $2\|\delta_\phi(h_\star, z)\|^2 \leq \tilde{\varphi}(z) = 6\psi(z)^2$, so we may drop the $\delta_\phi$-term. Substituting in the definition of $\tilde{\varphi}(z)$ and bounding $2 \leq 6$ gives

$$\tilde{\varphi}(z) \vee \|\tilde{\mathbf{e}}_t - \mathbf{v}_t\|^2 \leq 6\left(\psi(z)^2 \vee (\|\mathbf{e}_t - \mathbf{v}_t\|^2)\right).$$

   Hence, the desired inequality Eq. (E.12) follows from the second condition of our corollary.

3. Lastly, it remains to verify the covering property from Proposition E.1. Let $\mathcal{M} := \{M \in \mathbb{R}^{d_1 d_{\mathbf{x}}} : \|M_{\text{op}}\| \leq b\}$, let $\mathcal{M}_\epsilon$ denote a $b\epsilon$-cover of $\mathcal{M}$ in $\|\cdot\|_{\text{op}}$, let $\mathcal{H}_\epsilon := \{M \cdot f : M \in \mathcal{M}_\epsilon, f \in \mathcal{F}\}$, and finally set $\mathcal{G}_\epsilon := \{g_h : h \in \mathcal{H}_\epsilon\}$. Our goal will be to show that, for $\epsilon$ adequately chosen, $\mathcal{G}_\epsilon$ is an adequate cover of $\mathcal{G}$.

   Let $g \in \mathcal{G}$. Then, $g = g_h$, where $h = M \cdot f$ for some $f \in \mathcal{F}$ and $M \in \mathcal{M}$. Let $\tilde{h}_\epsilon \in \mathcal{H}_\epsilon$ be selected by selecting $M_\epsilon \in \mathcal{M}_\epsilon$ such that $\|M - M_\epsilon\|_{\text{op}} \leq b\epsilon$, $h_\epsilon := M_\epsilon \cdot f$, and $g_\epsilon := g_{h_\epsilon}$. Then, for any $z$, we have

$$\begin{aligned}
g(z) - g_\epsilon(z) &= \widehat{\phi}(h, z) + \delta_\phi(h_\star, z) - (\widehat{\phi}(h_\epsilon, z) + \delta_\phi(h_\star, z)) \\
&= \widehat{\phi}(h, z) - \widehat{\phi}(h_\epsilon, z) \\
&\stackrel{(i)}{=} X_1(h(y_1) - X_2 h(y_2)) + \delta_0(z) - (X_1(h_\epsilon(y_1) - X_2 h_\epsilon(y_2)) + \delta_0(z)) \\
&= X_1(h(y_1) - h_\epsilon(y_1)) - X_1 X_2(h(y_2) - h_\epsilon(y_2)) \\
&= X_1(M - M_\epsilon)f(y_1) - X_1 X_2(M - M_\epsilon)f(y_2),
\end{aligned}$$

where in $(i)$ we use the functional form of $\widehat{\phi}$ assumed by the lemma. Since $\|M - M_\epsilon\|_{\mathrm{op}} \le b\epsilon$, and $f(y) \le L \max\{1, \|f_\star\|\}$

$$
\begin{aligned}
g(z) - g_\epsilon(z) &\le b\epsilon(\|X_1\|_{\mathrm{op}}\|f(y_1)\| + \|X_1 X_2\|_{\mathrm{op}}\|f(y_2)\|) \\
&\le bL\epsilon(\|X_1\|_{\mathrm{op}} + \|X_1 \cdot X_2\|_{\mathrm{op}})(\max\{1, \|f_\star(y_1)\|\} + \max\{1, \|f_\star(y_2)\|\}) \\
&\le bL\epsilon(\|X_1\|_{\mathrm{op}} + \|X_1 \cdot X_2\|_{\mathrm{op}})(2 \vee \|f_\star(y_1)\| + \|f_\star(y_2)\|).
\end{aligned}
$$

Finally, by assumption, we have that $bL(\|X_1\|_{\mathrm{op}} + \|X_1 \cdot X_2\|_{\mathrm{op}})(2 + \|f_\star(y_1)\| + \|f_\star(y_2)\|) \le 2c_\psi \psi(z)$. Thus, recalling $\tilde{\varphi}(z) = 6\psi(z)^2$, we have

$$
g(z) - g_\epsilon(z) \le c_\psi \epsilon \sqrt{(2\psi(z))^2} \le c_\psi \epsilon \tilde{\varphi}(z)^{1/2}.
$$

It therefore follows that, for all $c'$, and all $\epsilon \le 1$, $\mathscr{G}_{\epsilon/c_\psi}$ is a $\sqrt{c'}\epsilon$-covering number of $\mathscr{G}$ in the pseudometric $d_{c',\infty}(h, h') := \sup_{z\in\mathcal{Z}}\{\|g(z) - g'(z)\| : \varphi(z) \le c'\}$. Hence, we can take $\mathcal{N}(\epsilon) = |\mathscr{G}_\epsilon|$ is applying Proposition E.1.

Hence, Proposition E.1 implies the bound

$$
\mathbb{E}\|\hat{g} - g_\star\|^2 \le \frac{270\tilde{c}_{n,\delta}}{n}\ln(2|\mathscr{G}_{1/33c_\psi n}|\delta^{-1}) + 8\mathbb{E}\|\mathbf{e}\|^2,
$$

where we have $\tilde{c}_{n,\delta} = \tilde{c}\ln(2n/\delta) = 6c\ln(2n\delta)$. Combining the above with Eq. (E.11)

$$
\mathbb{E}\|\phi(\hat{g}, \mathbf{z}) - \phi(g_\star, \mathbf{z})\|^2 \le \frac{12 \cdot 270\ln(2n/\delta)}{n}\ln(2|\mathscr{G}_{1/33c_\psi n}|\delta^{-1}) + 16\mathbb{E}\|\mathbf{e}\|^2 + 8\max_{h\in\mathscr{H}}\mathbb{E}\|\delta_\phi(h, \mathbf{z})\|^2.
$$

Finally, let us bound $|\mathscr{G}_{1/33c_\psi n}|$. From Lemma E.5, we have

$$
\ln|\mathscr{G}_\epsilon| = \ln|\mathscr{H}_\epsilon| = \ln(|\mathscr{F}\|\mathscr{M}_\epsilon|) \le \ln(|\mathscr{F}|) + d_1 d_\mathbf{x}\ln(5/\epsilon).
$$

Thus, repeating the computation Eq. (E.10) in the proof of Corollary E.1,

$$
\begin{aligned}
270\ln(2n/\delta)\ln(2|\mathscr{G}_{1/33c_\psi n}|\delta^{-1}) &\le 270 \cdot \ln(2n/\delta) \cdot (\ln(2/\delta) + \ln|\mathscr{F}| + d_1 d_\mathbf{x}\ln(3 \cdot 55c_\psi n)) \\
&\le (\ln|\mathscr{F}| + d_1 d_\mathbf{x})\mathtt{logs}(c_\psi n, \delta).
\end{aligned}
$$

Thus,

$$
\mathbb{E}\|\phi(\hat{g}, \mathbf{z}) - \phi(g_\star, \mathbf{z})\|^2 \le \frac{12c(\ln|\mathscr{F}| + d_1 d_\mathbf{x})\mathtt{logs}(c_\psi n, \delta)}{n} + 16\mathbb{E}\|\mathbf{e}\|^2 + 8\max_{h\in\mathscr{H}}\mathbb{E}\|\delta_\phi(h, \mathbf{z})\|^2,
$$

concluding the corollary. $\qquad\square$

### E.2.9  Proof of Proposition E.2

Define the matrix $\widehat{\Lambda} = \mathbb{E}[\hat{h}(\mathbf{y})\hat{h}(\mathbf{y})^\top]$. To begin, we have

$$
\|\Lambda_\star - \widehat{\Lambda}_n\|_{\mathrm{op}} \le \|\Lambda_\star - \widehat{\Lambda}\|_{\mathrm{op}} + \|\widehat{\Lambda} - \widehat{\Lambda}_n\|_{\mathrm{op}}.
$$

We now bound the terms on the right-hand side one by one. First,

$$
\begin{aligned}
&\|\Lambda_\star - \widehat{\Lambda}\|_{\mathrm{op}} \\
&= \|\mathbb{E}[\hat{h}(\mathbf{y})\hat{h}(\mathbf{y})^\top - h_\star(\mathbf{y})h_\star(\mathbf{y})^\top\|_{\mathrm{op}} \\
&\le \mathbb{E}[\|\hat{h}(\mathbf{y})\hat{h}(\mathbf{y})^\top - h_\star(\mathbf{y})h_\star(\mathbf{y})^\top\|_{\mathrm{op}}] \\
&\le \mathbb{E}[(\|\hat{h}(\mathbf{y})\| + \|h_\star(\mathbf{y})\|)\|h_\star(\mathbf{y}) - \hat{h}(\mathbf{y})\|] \\
&\le \mathbb{E}[(\|\hat{h}(\mathbf{y})\| + \|h_\star(\mathbf{y})\|)^2]^{1/2}\mathbb{E}[\|h_\star(\mathbf{y}) - \hat{h}(\mathbf{y})\|^2]^{1/2}.
\end{aligned}
$$

Moreover, $(\|\hat{h}(\mathbf{y})\| + \|h_\star(\mathbf{y})\|)^2 \le 4\max\{c, \varphi(\mathbf{y})\}$ by assumption, so this is at most

$$
\|\mathbb{E}[\hat{h}(\mathbf{y})\hat{h}(\mathbf{y})^\top - h_\star(\mathbf{y})h_\star(\mathbf{y})^\top]\|_{\mathrm{op}} \le 2(\mathbb{E}\max\{c, \varphi(\mathbf{y})\})^{1/2})\mathbb{E}[\|h_\star(\mathbf{y}) - \hat{h}(\mathbf{y})\|^2]^{1/2}. \quad \text{(E.13)}
$$

Finally, using Lemma E.1, one can bound $\mathbb{E}\max\{c, \varphi(\mathbf{y})\} \le 2c$, so we can further bound by $3\sqrt{c}\mathbb{E}[\|h_\star(\mathbf{y}) - \hat{h}(\mathbf{y})\|]^{1/2}$.

For the second term, we appeal to truncation. Let $\mathcal{E}$ denote the event $\{\varphi(\mathbf{y}) \le c \ln(2n/\delta)\}$, and let $\mathcal{E}^{(i)}$ denote the analogous event for $\mathbf{y}^{(i)}$. By construction $\mathcal{E}^{(1)}, \dots, \mathcal{E}^{(n)}$ occur simultaneously with probability at least $1 - \delta/2$, so that we may bound

$$\|\widehat{\Lambda} - \widehat{\Lambda}_n\|_{\mathrm{op}} \tag{E.14}$$

$$= \left\| \frac{1}{n} \sum_{i=1}^n \mathbb{E}[\hat{h}(\mathbf{y})\hat{h}(\mathbf{y})^\top] - \mathbb{I}(\mathcal{E}^{(i)})\hat{h}(\mathbf{y}^{(i)})\hat{h}(\mathbf{y}^{(i)})^\top \right\|_{\mathrm{op}}$$

$$\le \left\| \mathbb{E}[(1 - \mathbb{I}(\mathcal{E}))\hat{h}(\mathbf{y})\hat{h}(\mathbf{y})^\top] \right\|_{\mathrm{op}}$$

$$+ \left\| \frac{1}{n} \sum_{i=1}^n \mathbb{E}[\hat{h}(\mathbf{y})\hat{h}(\mathbf{y})^\top \mathbb{I}(\mathcal{E})] - \mathbb{I}(\mathcal{E}^{(i)})\hat{h}(\mathbf{y}^{(i)})\hat{h}(\mathbf{y}^{(i)})^\top \right\|_{\mathrm{op}}. \tag{E.15}$$

We bound the first term above by

$$\left\| \mathbb{E}[(1 - \mathcal{E})\hat{h}(\mathbf{y})\hat{h}(\mathbf{y})^\top \right\|_{\mathrm{op}} \le \mathbb{E}\left[ \left\| (1 - \mathcal{E})\hat{h}(\mathbf{y})\hat{h}(\mathbf{y})^\top \right\|_{\mathrm{op}} \right]$$

$$= \mathbb{E}[(1 - \mathcal{E})\|\hat{h}(\mathbf{y})\|^2]$$

$$\le \mathbb{E}[\mathbb{I}(\varphi(\mathbf{y}) > c \ln(2n/\delta)) \max\{c, \varphi(\mathbf{y})\}]$$

$$\le \mathbb{E}[\mathbb{I}(\varphi(\mathbf{y}) > c \ln(2n/\delta))\varphi(\mathbf{y})] \le \frac{c\delta}{2n}, \tag{E.16}$$

where the last line uses Lemma E.1.

To conclude, let us bound the last term in Eq. (E.15). Define the symmetric matrices $\mathbf{M}^{(i)} := \mathbb{E}[\hat{h}(\mathbf{y})\hat{h}(\mathbf{y})^\top \mathbb{I}(\mathcal{E})] - \hat{h}(\mathbf{y}^{(i)})\hat{h}(\mathbf{y}^{(i)})^\top \mathbb{I}(\mathcal{E}^{(i)})$. Then $\mathbb{E}\mathbf{M}^{(i)} = 0$, we can see that $\|\mathbf{M}^{(i)}\| \le c \ln(2n/\delta)$ almost surely (indeed, if $X, Y \succeq 0$, then $\|X - Y\|_{\mathrm{op}} \le \max\{\|X\|_{\mathrm{op}}, \|Y\|_{\mathrm{op}}\}$), and thus

$$(\mathbf{M}^{(i)})^2 \preceq (c \ln(2n/\delta))^2 I.$$

Hence, by Theorem 1.3 of [40],

$$\mathbb{P}\left[ \left\| \sum_{i=1}^n \mathbf{M}^{(i)} \right\| \ge t \right] \le 2d e^{-t^2/8\sigma^2}, \quad \text{where} \quad \sigma^2 := n(c \ln(2n/\delta))^2.$$

Rearranging, we have that

$$\mathbb{P}\left[ \left\| \frac{1}{n} \sum_{i=1}^n \mathbf{M}^{(i)} \right\| \ge 2c \ln(2n/\delta)\sqrt{2 \ln(2d/\delta)/n} \right] \le \frac{\delta}{2}.$$

Simplifying $2c \ln(2n/\delta)\sqrt{2 \ln(2d/\delta)/n} \le 4cn^{-1/2} \ln(2dn/\delta)^{3/2}$, we have that with probability $1 - \delta/2$, $\frac{1}{n}\sum_{i=1}^n \mathbb{E}[\hat{h}(\mathbf{y})\hat{h}(\mathbf{y})^\top \mathbb{I}(\mathcal{E})] - \mathbb{I}(\mathcal{E})^{(i)}\hat{h}(\mathbf{y}^{(i)})\hat{h}(\mathbf{y}^{(i)})^\top] = \|\sum_{i=1}^n \mathbf{M}^{(i)}\|_{\mathrm{op}} \le 4cn^{-1/2} \ln(2dn/\delta)^{3/2}$. Hence, combining with Eqs. (E.13) and (E.16), we conclude that with probability $1 - \delta$,

$$\|\Lambda_\star - \widehat{\Lambda}_n\|_{\mathrm{op}} \le \|\Lambda_\star - \widehat{\Lambda}\|_{\mathrm{op}} + \|\widehat{\Lambda} - \widehat{\Lambda}_n\|_{\mathrm{op}}$$

$$\le 3\sqrt{c\mathbb{E}[\|\hat{h}(\mathbf{y}) - h_\star(\mathbf{y})\|^2]} + 4cn^{-1/2} \ln(2dn/\delta)^{3/2} + \frac{\delta c}{2n}$$

$$\le 3\sqrt{c\mathbb{E}[\|\hat{h}(\mathbf{y}) - h_\star(\mathbf{y})\|^2]} + 5cn^{-1/2} \ln(2dn/\delta)^{3/2} := \varepsilon_{\mathrm{pca},n}^2.$$

$\square$

### E.2.10 Proof of Corollary E.3

By assumption, $\Lambda_\star$ is rank $d_\mathbf{x}$ and $\lambda_{d_\mathbf{x}}(\Lambda_\star) > 0$. Let $V_\star$ be and eigenbasis for the top $d_\mathbf{x}$ eigenvalues of $\Lambda_\star$, and let $\widehat{V}_n$ be an eigenbasis for the top $d_\mathbf{x}$ eigenvalues of $\widehat{\Lambda}_n$. From the Davis-Kahan sine theorem [5], we have that for any $\alpha \in (0, 1)$,

$$\|(I - V_\star V_\star^\top)\widehat{V}_n\|_{\mathrm{op}} \le (1 - \alpha)^{-1} \lambda_{d_\mathbf{x}}(\Lambda_\star)^{-1} \|\Lambda_\star - \widehat{\Lambda}_n\|_{\mathrm{op}},$$

whenever

$$\|\Lambda_\star - \widehat{\Lambda}_n\|_{\mathrm{op}} \le \alpha \lambda_{d_{\mathbf{x}}}(\Lambda_\star). \tag{E.17}$$

In particular, if $n$ is sufficiently large that for

$$\|\Lambda_\star - \widehat{\Lambda}_n\|_{\mathrm{op}} \le \varepsilon_{\mathrm{pca},n,\delta} \le \frac{1}{4}\lambda_{d_{\mathbf{x}}}(\Lambda_\star),$$

then from Eq. (E.17),

$$\|(I - V_\star V_\star^\top)\widehat{V}_n\|_{\mathrm{op}} \le \frac{4\varepsilon_{\mathrm{pca},n,\delta}}{3\lambda_d(\Lambda_\star)^{-1}} \le \frac{1}{3}.$$

And thus,

$$\sigma_{d_{\mathbf{x}}}(V_\star^\top \widehat{V}_n) \ge \sigma_{d_{\mathbf{x}}}(V_\star V_\star^\top \widehat{V}_n) \ge \sigma_{d_{\mathbf{x}}}(\widehat{V}_n) - \|(I - V_\star V_\star^\top)\widehat{V}_n\|_{\mathrm{op}}$$
$$\ge 1 - 1/3 = 2/3,$$

where the previous display uses that $\sigma_{d_{\mathbf{x}}}(\widehat{V}_n) = 1$ since $\widehat{V}_n$ has orthonormal columns, and that $\|(I - V_\star V_\star^\top)\widehat{V}_n\|_{\mathrm{op}} \le 1/3$. $\qquad\square$

### E.2.11 Proof of Proposition E.3

Let $\mathbf{U}$ be the matrix with $\{\mathbf{u}^{(i)}\}_{i=1}^n$ as rows, and let $\boldsymbol{\Delta}, \mathbf{Y}, \mathbf{W}, \mathbf{E}$ be defined analogously for $\boldsymbol{\delta}, \mathbf{y}, \mathbf{w}$, and $\mathbf{e}$ respectively. Let us assume for now that $(\mathbf{U} + \boldsymbol{\Delta})$ has full row rank; this will be justified momentarily in Claim E.3. Then we have

$$\widehat{M}^\top = (\mathbf{U} + \boldsymbol{\Delta})^\dagger \mathbf{Y}$$
$$= (\mathbf{U} + \boldsymbol{\Delta})^\dagger (\mathbf{U}M_\star^\top + \mathbf{W} + \mathbf{E})$$
$$= (\mathbf{U} + \boldsymbol{\Delta})^\dagger (\mathbf{U}M_\star^\top + \mathbf{W} + \mathbf{E})$$
$$= M_\star^\top + (\mathbf{U} + \boldsymbol{\Delta})^\dagger (-\boldsymbol{\Delta}M_\star^\top + \mathbf{W} + \mathbf{E}). \tag{E.18}$$

Thus,

$$\|\widehat{M} - M_\star\|_{\mathrm{op}} \le \frac{\|\boldsymbol{\Delta}\|_{\mathrm{op}}\|M_\star\|_{\mathrm{op}} + \|\mathbf{E}\|_{\mathrm{op}}}{\sigma_{\min}(\mathbf{U} + \boldsymbol{\Delta})} + \|(\mathbf{U} + \boldsymbol{\Delta})^\dagger \mathbf{W}\|_{\mathrm{op}}. \tag{E.19}$$

**Handling the Gaussian Noise.** We first handle the term $\|(\mathbf{U} + \boldsymbol{\Delta})^\dagger \mathbf{W}\|_{\mathrm{op}}$. Observe that $\mathbf{W}$ is Gaussian conditioned on $\mathbf{U}$ and $\boldsymbol{\Delta}$. Fix a matrix $U$ with $U^\top U > 0$. Fix a vector $v \in \mathbb{R}^{d_{\mathbf{y}}}$ with $\|v\| = 1$, and observe that $\langle v, \mathbf{e}^{(i)} \rangle$ are $\|\Sigma_w\|_{\mathrm{op}}$-subgaussian. Thus, for any matrix $\Lambda > 0$, we have from Abbasi-Yadkori et al. [1, Theorem 3] that conditioned on $\mathbf{U}$ and $\boldsymbol{\Delta}$, with probability at least $1 - \delta$,

$$\|U^\top \mathbf{W}v\|_{(\Lambda + U^\top U)^{-1}} \le \sqrt{2\|\Sigma_e\|_{\mathrm{op}} \ln\left(\frac{\det(\Lambda + U^\top U)^{1/2}\det(\Lambda)^{-1/2}}{\delta}\right)}.$$

Since we are taking $U$ to be fixed (conditioned on $\mathbf{U}$ and $\boldsymbol{\Delta}$), we can take $\Lambda = U^\top U$. This gives, with probability at least $1 - \delta$,

$$\|U^\top \mathbf{W}v\|_{(U^\top U)^{-1}} \le 2\sqrt{2\|\Sigma_w\|_{\mathrm{op}} \ln\left(\frac{2^{d_{\mathbf{u}}/2}}{\delta}\right)} \le 2\sqrt{\|\Sigma_w\|_{\mathrm{op}}(d_{\mathbf{u}} + 2\ln(1/\delta))}.$$

It follows that

$$\|U^\dagger \mathbf{W}v\|_2 \le \sigma_{\min}(U)^{-1}\|U^\top \mathbf{W}v\|_{(U^\top U)^{-1}} \le 2\sigma_{\min}(U)^{-1}\sqrt{\|\Sigma_w\|_{\mathrm{op}}(d_{\mathbf{u}} + 2\ln(1/\delta))}.$$

By a standard covering argument (see, e.g., Vershynin [42, Section 4.2]), we find that with probability at least $1 - \delta$,

$$\|U^\dagger \mathbf{W}\|_{\mathrm{op}} = \sup_{v \in \mathbb{R}^{d_{\mathbf{y}}}:\|v\|=1}\|U^\dagger \mathbf{W}v\| \lesssim \sigma_{\min}(U)^{-1}\sqrt{\|\Sigma_w\|_{\mathrm{op}}(d_{\mathbf{u}} + d_{\mathbf{y}} + \ln(1/\delta))},$$

Taking $U = \mathbf{U} + \boldsymbol{\Delta}$, this implies that with probability at least $1 - \delta$,

$$\|(\mathbf{U} + \boldsymbol{\Delta})^\dagger \mathbf{W}\|_{\mathrm{op}} \lesssim \sigma_{\min}(\mathbf{U} + \boldsymbol{\Delta})^{-1}\sqrt{\|\Sigma_w\|_{\mathrm{op}}(d_{\mathbf{u}} + d_{\mathbf{y}} + \ln(1/\delta))},$$

**Error Terms.**  We have

$$\|\boldsymbol{\Delta}\|_{\mathrm{op}}\|M_\star\|_{\mathrm{op}} + \|\mathbf{E}\|_{\mathrm{op}} \le \|M_\star\|_{\mathrm{op}}\|\boldsymbol{\Delta}\|_F + \|\mathbf{E}\|_F = \left(\|M_\star\|_{\mathrm{op}}\sqrt{\sum_{i=1}^n \|\boldsymbol{\delta}^{(i)}\|^2}\right) + \sqrt{\sum_{i=1}^n \|\mathbf{e}^{(i)}\|^2}.$$

Recall that 1) $\psi(n,\delta) \coloneqq \frac{2\ln(2n/\delta)\ln(2/\delta)}{n}$, 2) $\|\boldsymbol{\delta}^{(i)}\|^2$ is $c_{\boldsymbol{\delta}}$-concentrated and $\|\mathbf{e}^{(i)}\|^2$ are $c_{\mathbf{e}}$ concentrated (Definition 3), and 3) $\mathbb{E}\|\mathbf{e}^{(i)}\|^2 \le \varepsilon_{\mathbf{e}}^2$ and $\mathbb{E}\|\boldsymbol{\delta}^{(i)}\|^2 \le \varepsilon_{\boldsymbol{\delta}}^2$. Lemma E.1 thus implies that for

$$\psi(n,\delta) \le \min\left\{\frac{\varepsilon_{\mathbf{e}}^2}{c_{\mathbf{e}}}, \frac{\varepsilon_{\boldsymbol{\delta}}^2}{c_{\boldsymbol{\delta}}}\right\},$$

the following event holds with probability at least $1 - 2\delta$:

$$\mathcal{E}_{\mathrm{ls},1} \coloneqq \left\{\sum_{i=1}^n \|\boldsymbol{\delta}^{(i)}\|^2 \le 2n\varepsilon_{\boldsymbol{\delta}}^2\right\} \cap \left\{\sum_{i=1}^n \|\mathbf{e}^{(i)}\|^2 \le 2n\varepsilon_{\mathbf{e}}^2\right\}. \qquad (\mathrm{E.20})$$

Clearly, on $\mathcal{E}_{\mathrm{ls},1}$ we have

$$\|\boldsymbol{\Delta}\|_{\mathrm{op}}\|M_\star\|_{\mathrm{op}} + \|\mathbf{E}\|_{\mathrm{op}} \lesssim n^{1/2}(\|M_\star\|_{\mathrm{op}}\varepsilon_{\boldsymbol{\delta}} + \varepsilon_{\mathbf{e}}).$$

**Bounding the least eigenvalue.**  Summarizing the development so far, we have for $\psi(n,\delta) \le \max\{\frac{\varepsilon_{\mathbf{e}}^2}{c_{\mathbf{e}}}, \frac{\varepsilon_{\boldsymbol{\delta}}^2}{c_{\boldsymbol{\delta}}}\}$, with probability at least $1 - 3\delta$,

$$\|\widehat{M} - M_\star\|_{\mathrm{op}} \lesssim \frac{n^{1/2}(\|M_\star\|_{\mathrm{op}}\varepsilon_{\boldsymbol{\delta}} + \varepsilon_{\mathbf{e}}) + \sqrt{\|\Sigma_w\|_{\mathrm{op}}(d_{\mathbf{y}} + d_{\mathbf{u}} + \ln(1/\delta))}}{\sigma_{\min}(\mathbf{U} + \boldsymbol{\Delta})}.$$

Finally, let us lower bound $\sigma_{\min}(\mathbf{U} + \boldsymbol{\Delta})$. We start with the following self-contained result.

**Claim E.2.**  Consider matrices $U, \Delta$, and suppose $\|\Delta\|_{\mathrm{op}}^2 \le \frac{1}{4}\lambda_{\min}(U^\top U)$. Then,

$$(U + \Delta)^\top(U + \Delta) \succeq \frac{1}{4}U^\top U.$$

*Proof.*  By Cauchy-Schwarz and AM-GM, we have the elementary inequality that for two vectors $v, w$ of the same dimension, $\|v + w\|^2 = \|v\|^2 + \|w\|^2 + 2\langle v, w\rangle \ge \frac{1}{2}\|v\|^2 - \|w\|^2$. This entails

$$\begin{aligned}
(U + \Delta)^\top(U + \Delta) &\succeq \frac{1}{2}U^\top U - \Delta^\top\Delta \\
&\succeq \frac{1}{2}U^\top U - I\|\Delta\|_{\mathrm{op}}^2 \\
&= \frac{1}{4}U^\top U + (\frac{1}{4}U^\top U - \|\Delta\|_{\mathrm{op}}^2 I) \\
&\succeq \frac{1}{4}U^\top U,
\end{aligned}$$

where the last line uses the that $\frac{1}{4}U^\top U \succeq \frac{1}{4}\lambda_{\min}(U^\top U)$, and the assumption $\|\Delta\|_{\mathrm{op}}^2 \le \frac{1}{4}\lambda_{\min}(U^\top U)$. $\qquad \square$

**Claim E.3.**  There is a universal constant $c_1 > 0$ such that the following holds. Let $\mathbf{U} \in \mathbb{R}^{n \times d_{\mathbf{u}}}$ be a matrix with rows drawn i.i.d. from $\mathcal{N}(0, \Sigma_u)$ where $\Sigma_u \succeq 0$, and let $\boldsymbol{\Delta}$ be a matrix of the same dimension with $\|\boldsymbol{\Delta}\|_{\mathrm{op}}^2 \le \frac{n}{8}\lambda_{\min}(\Sigma_u)$. Then, for $n \ge c_1(d_{\mathbf{u}} + \ln(1/\delta))$, the following holds with probability $1 - \delta$:

$$(\mathbf{U} + \boldsymbol{\Delta})^\top(\mathbf{U} + \boldsymbol{\Delta}) \succeq \frac{n\Sigma_u}{8} \succeq \frac{n\lambda_{\min}(\Sigma_u)}{8}I. \qquad (\mathrm{E.21})$$

*Proof of Claim E.3.*  From Claim E.2, we have that if $\|\boldsymbol{\Delta}\|_{\mathrm{op}}^2 \le \frac{1}{4}\lambda_{\min}(\mathbf{U}^\top\mathbf{U})$, we have

$$(\mathbf{U} + \boldsymbol{\Delta})^\top(\mathbf{U} + \boldsymbol{\Delta}) \succeq \frac{1}{4}\mathbf{U}^\top\mathbf{U}.$$

If this holds, we have

$$\frac{1}{4}\mathbf{U}^\top\mathbf{U} = \frac{1}{4}\Sigma_u^{1/2}\left(\Sigma^{-1/2}\mathbf{U}^\top\mathbf{U}\Sigma_u^{-1/2}\right)\Sigma_u^{1/2} \geq \lambda_{\min}(\Sigma_u^{-1/2}\mathbf{U}^\top\mathbf{U}\Sigma_u^{-1/2})\Sigma_u.$$

Note that $\mathbf{U}\Sigma_u^{-1/2}$ has standard Gaussian rows, and its number of rows exceeds its number of columns. Thus, from Theorem 5.39 of [41], we have that

$$\mathbb{P}\left[\lambda_{\min}(\Sigma_u^{-1/2}\mathbf{U}^\top\mathbf{U}\Sigma_u^{-1/2})^{1/2} \geq \sqrt{n} - \mathcal{O}(\sqrt{d_{\mathbf{u}}} + \sqrt{\ln(1/\delta)})\right] \geq 1 - \delta. \qquad (\text{E.22})$$

In particular, for $n \geq c_1(d_{\mathbf{u}} + \ln(1/\delta))$ for some universal $c_1$, we have that with probability $1 - \delta$, $\lambda_{\min}(\Sigma_u^{-1/2}\mathbf{U}^\top\mathbf{U}\Sigma_u^{-1/2})^{1/2} \geq \sqrt{n/2}$, and thus when this occurs, and when $\|\boldsymbol{\Delta}\|_{\mathrm{op}}^2 \leq \frac{n\lambda_{\min}(\Sigma_u)}{8} \leq \frac{1}{4}\lambda_{\min}(\mathbf{U}^\top\mathbf{U})$, we have

$$(\mathbf{U} + \boldsymbol{\Delta})^\top(\mathbf{U} + \boldsymbol{\Delta}) \geq \frac{1}{4}\mathbf{U}^\top\mathbf{U} \geq \frac{n}{8}\Sigma_u \geq \frac{n}{8}\lambda_{\min}(\Sigma_u)I.$$

$\square$

Hence, for $n \geq c_1(\ln(1/\delta) + d_{\mathbf{u}})$, $\psi(n,\delta) \leq \max\{\frac{\varepsilon_{\mathbf{e}}^2}{c_{\mathbf{e}}}, \frac{\varepsilon_{\boldsymbol{\delta}}^2}{c_{\boldsymbol{\delta}}}\}$, and $2\varepsilon_{\boldsymbol{\delta}}^2 \leq \frac{n}{8}\lambda_{\min}(\Sigma_u)$ (or equivalently, $\varepsilon_{\boldsymbol{\delta}}^2 \leq \frac{n}{16}\lambda_{\min}(\Sigma_u)$), we find that with total failure probability at least $1 - 4\delta$,

$$\|\widehat{M} - M_\star\|_{\mathrm{op}} \lesssim \frac{n^{1/2}(\|M_\star\|_{\mathrm{op}}\varepsilon_{\boldsymbol{\delta}} + \varepsilon_{\mathbf{e}}) + \sqrt{\|\Sigma_w\|_{\mathrm{op}}(d_{\mathbf{y}} + d_{\mathbf{u}} + \ln(1/\delta))}}{\sigma_{\min}(\mathbf{U} + \boldsymbol{\Delta})}$$

$$\lesssim \frac{(\|M_\star\|_{\mathrm{op}}\varepsilon_{\boldsymbol{\delta}} + \varepsilon_{\mathbf{e}}) + \sqrt{n^{-1/2}\|\Sigma_w\|_{\mathrm{op}}(d_{\mathbf{y}} + d_{\mathbf{u}} + \ln(1/\delta))}}{\sqrt{\lambda_{\min}(\Sigma_u)}}.$$

Hence, under these conditions, with probability $1 - 4\delta$,

$$\|\widehat{M} - M_\star\|_{\mathrm{op}}^2 \lesssim \lambda_{\min}(\Sigma_u)^{-1}\left(\|M_\star\|_{\mathrm{op}}^2\varepsilon_{\boldsymbol{\delta}}^2 + \varepsilon_{\mathbf{e}}^2 + \frac{\|\Sigma_w\|_{\mathrm{op}}(d_{\mathbf{y}} + d_{\mathbf{u}} + \ln(1/\delta))}{n}\right).$$

$\square$

### E.2.12 Proof of Proposition E.4

Assume that the events of the proof of Proposition E.2 above; this contributes a failure probability of $4\delta$. To begin, we have that

$$\sum_{i=1}^n(\widehat{M}(\mathbf{u}^{(i)} + \boldsymbol{\delta}^{(i)}) - \mathbf{y}^{(i)})^{\otimes 2}$$

$$= \sum_{i=1}^n((\widehat{M} - M_\star)(\mathbf{u}^{(i)} + \boldsymbol{\delta}^{(i)}) - \mathbf{w}^{(i)} - \mathbf{e}^{(i)})^{\otimes 2}$$

$$= \left((\mathbf{U} + \boldsymbol{\Delta})(\widehat{M} - M_\star)^\top + \boldsymbol{\Delta}M_\star^\top - \mathbf{W} - \mathbf{E}\right)^\top\left((\mathbf{U} + \boldsymbol{\Delta})(\widehat{M} - M_\star)^\top + \boldsymbol{\Delta}M_\star^\top - \mathbf{W} - \mathbf{E}\right).$$

From Eq. (E.18), and the fact that $\mathbf{U} + \boldsymbol{\Delta}$ has full rank under the high probability events of Proposition E.2, we have

$$\widehat{M}^\top = M_\star^\top + (\mathbf{U} + \boldsymbol{\Delta})^\dagger(-\boldsymbol{\Delta}M_\star^\top + \mathbf{W} + \mathbf{E}).$$

This yields

$$(\mathbf{U} + \boldsymbol{\Delta})(\widehat{M}^\top - M_\star^\top) = P_{\mathbf{U}+\boldsymbol{\Delta}}(-\boldsymbol{\Delta}M_\star^\top + \mathbf{W} + \mathbf{E}),$$

where $P_{\mathbf{U}+\boldsymbol{\Delta}} := (\mathbf{U} + \boldsymbol{\Delta})(\mathbf{U} + \boldsymbol{\Delta})^\dagger \in \mathbb{R}^{n \times n}$ is the projection onto the row space of $\mathbf{U} + \boldsymbol{\Delta}$, which has dimension $d_{\mathbf{u}}$. Thus, we find that

$$\sum_{i=1}^n(\widehat{M}(\mathbf{u}^{(i)} + \boldsymbol{\delta}^{(i)}) - \mathbf{y}^{(i)})^{\otimes 2}$$

$$= \left((I - P_{\mathbf{U}+\boldsymbol{\Delta}})(-\boldsymbol{\Delta}M_\star^\top + \mathbf{W} + \mathbf{E})\right)^\top\left((I - P_{\mathbf{U}+\boldsymbol{\Delta}})(-\boldsymbol{\Delta}M_\star^\top + \mathbf{W} + \mathbf{E})\right).$$

Rearranging, and using that $I - P_{\mathbf{U}+\mathbf{\Delta}}$ is a projection operator, we have that

$$\left\| \frac{1}{n} \sum_{i=1}^{n} (\widehat{M}(\mathbf{u}^{(i)} + \boldsymbol{\delta}^{(i)}) - \mathbf{y}^{(i)})^{\otimes 2} - \frac{1}{n} \mathbf{W}^\top (I - P_{\mathbf{U}+\mathbf{\Delta}}) \mathbf{W} \right\|_{\mathrm{op}} \leq 2 \|\mathbf{W}\|_{\mathrm{op}} \|\mathbf{E} - \mathbf{\Delta} M_\star^\top\|_{\mathrm{op}} + \|\mathbf{E} - \mathbf{\Delta} M_\star^\top\|_{\mathrm{op}}^2.$$

We can now bound this quantity using the following claim.

**Claim E.4.** Suppose that $\lambda_+ \geq \lambda_{\max}(\Sigma_w)$, and $\varepsilon_{\boldsymbol{\delta}}^2 \|M_\star\|_{\mathrm{op}}^2 + \varepsilon_{\mathbf{e}}^2 \leq 2\lambda_+$. Suppose the event $\mathcal{E}_{\mathrm{ls},1}$ of Eq. (E.20) holds, and $n \geq c' \sqrt{d_{\mathbf{y}} + \ln(1/\delta)}$ for $c'$ sufficiently large. Then,

$$\left\| \frac{1}{n} \sum_{i=1}^{n} (\widehat{M}(\mathbf{u}^{(i)} + \boldsymbol{\delta}^{(i)}) - \mathbf{y}^{(i)})^{\otimes 2} - \frac{1}{n} \mathbf{W}^\top (I - P_{\mathbf{U}+\mathbf{\Delta}}) \mathbf{W} \right\|_{\mathrm{op}} \lesssim \sqrt{\frac{\lambda_+ (\varepsilon_{\boldsymbol{\delta}}^2 \|M_\star\|_{\mathrm{op}}^2 + \varepsilon_{\mathbf{e}}^2)}{n}}.$$

*Proof.* On the event $\mathcal{E}_{\mathrm{ls},1}$ of Eq. (E.20), recall that

$$\|\mathbf{E} - \mathbf{\Delta} M_\star^\top\|_{\mathrm{op}} \leq \varepsilon_{\boldsymbol{\delta}} \|M_\star\|_{\mathrm{op}} + \varepsilon_{\mathbf{e}}.$$

In addition, for $n \geq c'(d_{\mathbf{y}} + \ln(1/\delta))$ for some sufficiently large numerical constant $c'$, a suitable analogue of Eq. (E.22) implies that with an additional probability $1 - \delta$,

$$\|\mathbf{W}\|_{\mathrm{op}} \leq \lambda_{\max}(\Sigma_w)^{1/2} \|\Sigma_w^{-1/2} \mathbf{W}\|_{\mathrm{op}} \leq 2\sqrt{n} \lambda_+^{1/2}.$$

Hence, for $\varepsilon_{\boldsymbol{\delta}}^2 \|M_\star\|_{\mathrm{op}}^2 + \varepsilon_{\mathbf{e}}^2 \leq \lambda_+$, we have that with total probability at least $1 - 5\delta$ (including events from the previous proposition),

$$\left\| \frac{1}{n} \sum_{i=1}^{n} (\widehat{M}(\mathbf{u}^{(i)} + \boldsymbol{\delta}^{(i)}) - \mathbf{y}^{(i)})^{\otimes 2} - \frac{1}{n} \mathbf{W}^\top (I - P_{\mathbf{U}+\mathbf{\Delta}}) \mathbf{W} \right\|_{\mathrm{op}} \lesssim \sqrt{\lambda_+(\Sigma_w)(\varepsilon_{\boldsymbol{\delta}}^2 \|M_\star\|_{\mathrm{op}} + \varepsilon_{\mathbf{e}}^2)}.$$

$\square$

To conclude the proof, we bound

$$\left\| \frac{1}{n} \sum_{i=1}^{n} (\widehat{M}(\mathbf{u}^{(i)} + \boldsymbol{\delta}^{(i)}) - \mathbf{y}^{(i)})^{\otimes 2} - \Sigma_w \right\|_{\mathrm{op}}$$

$$\leq \left\| \frac{1}{n} \sum_{i=1}^{n} (\widehat{M}(\mathbf{u}^{(i)} + \boldsymbol{\delta}^{(i)}) - \mathbf{y}^{(i)})^{\otimes 2} - \frac{1}{n} \mathbf{W}^\top (I - P_{\mathbf{U}+\mathbf{\Delta}}) \mathbf{W} \right\|_{\mathrm{op}}$$

$$+ \left\| \Sigma_w - \frac{1}{n} \mathbf{W}^\top (I - P_{\mathbf{U}+\mathbf{\Delta}}) \mathbf{W} \right\|_{\mathrm{op}}. \tag{E.23}$$

The following claim bounds the second term.

**Claim E.5.** Suppose $n \geq c''(d_{\mathbf{u}} + \ln(1/\delta))$, where $c'' > 0$ is a suitably large numerical constant. For any upper bound $\lambda_+ \geq \lambda_{\max}(\Sigma_w)$, with probability $1 - 2\delta$,

$$\left\| \Sigma_w - \frac{1}{n} \mathbf{W}^\top (I - P_{\mathbf{U}+\mathbf{\Delta}}) \mathbf{W} \right\|_{\mathrm{op}} \leq \lambda_+ \sqrt{\frac{d_{\mathbf{y}} + \ln(1/\delta)}{n}}.$$

*Proof.* Define $P_{\mathbf{U}+\mathbf{\Delta}}^c := I - P_{\mathbf{U}+\mathbf{\Delta}}$. Then we have

$$\left\| \Sigma_w - \frac{1}{n} \mathbf{W}^\top P_{\mathbf{U}+\mathbf{\Delta}}^c \mathbf{W} \right\|_{\mathrm{op}} \leq \frac{1}{n} \lambda_{\max}(\Sigma_w) \| n I - \underbrace{(\mathbf{W}\Sigma_w^{-1/2})^\top P_{\mathbf{U}+\mathbf{\Delta}}^c (\mathbf{W}\Sigma_w^{-1/2})}_{:= \mathbf{M}} \|_{\mathrm{op}}.$$

Now, observe that since $P_{\mathbf{U}+\mathbf{\Delta}}^c \in \mathbb{R}^{n \times n}$ is a projection matrix with rank $n - d_{\mathbf{u}}$, and $\mathbf{W}\Sigma_w^{-1/2} \in \mathbb{R}^{n \times d_{\mathbf{y}}}$, the matrix $\mathbf{M} = (\mathbf{W}\Sigma_w^{-1/2})^\top P_{\mathbf{U}+\mathbf{\Delta}}^c (\mathbf{W}\Sigma_w^{-1/2})$ is identical in distribution to $\mathbf{G}^\top \mathbf{G}$, where $\mathbf{G} \in \mathbb{R}^{(n-d_{\mathbf{u}}) \times d_{\mathbf{y}}}$ has i.i.d. unit Gaussian entries. Theorem 5.39 of [41] guarantees that with probability at least $1 - 2\delta$,

$$\sqrt{n - d_{\mathbf{u}}} - \mathcal{O}\left(\sqrt{d_{\mathbf{y}} + \ln(1/\delta)}\right) \leq \sigma_{\min}(\mathbf{G}) \leq \sigma_{\max}(\mathbf{G}) \leq \sqrt{n - d_{\mathbf{u}}} + \mathcal{O}\left(\sqrt{d_{\mathbf{y}} + \ln(1/\delta)}\right).$$

This implies that for $n \ge c''(d_{\mathbf{u}} + d_{\mathbf{y}} + \ln(1/\delta))$ for some universal constant $c''$, we have that

$$(n - d_{\mathbf{u}}) - \mathcal{O}\Big(\sqrt{d_{\mathbf{y}} + \ln(1/\delta)}\Big)\sqrt{n - d_{\mathbf{u}}} \le \lambda_{\min}(\mathbf{M}) \le \lambda_{\max}(\mathbf{M}) \le (n - d_{\mathbf{u}}) + \mathcal{O}\Big(\sqrt{d_{\mathbf{y}} + \ln(1/\delta)}\Big)\sqrt{n - d_{\mathbf{u}}}.$$

Hence, on this event (and again for $n \ge c''(d_{\mathbf{u}} + \ln(1/\delta))$ for $c''$ suitably large),

$$\|nI - \mathbf{M}\|_{\mathrm{op}} \le d_{\mathbf{u}} + \mathcal{O}\Big(\sqrt{d_{\mathbf{y}} + \ln(1/\delta)}\Big)\sqrt{n - d_{\mathbf{u}}} \lesssim \sqrt{n(d_{\mathbf{y}} + \ln(1/\delta))}.$$

as needed. $\qquad\square$

In total, combining Claims E.4 and E.5 and Eq. (E.23), we conclude that on the events of the previous proposition, and with an additional $3\delta$ failure probability,

$$\left\|\frac{1}{n}\sum_{i=1}^{n}(\widehat{M}(\mathbf{u}^{(i)} + \boldsymbol{\delta}^{(i)}) - \mathbf{y}^{(i)})^{\otimes 2} - \Sigma_w\right\|_{\mathrm{op}}$$

$$\lesssim \sqrt{\lambda_{\max}(\Sigma_w)(\varepsilon_{\boldsymbol{\delta}}^2\|M_\star\|_{\mathrm{op}}^2 + \varepsilon_{\mathbf{e}}^2) + \frac{\lambda_{\max}(\Sigma_w)^2(d + \ln(1/\delta))}{n}}, \qquad (E.24)$$

provided that $\varepsilon_{\boldsymbol{\delta}}^2\|M_\star\|_{\mathrm{op}}^2 + \varepsilon_{\mathbf{e}}^2 \le \lambda_{\max}(\Sigma_w)$, and $n \ge c(d_{\mathbf{y}} + \ln(1/\delta))$ for some universal constant $c$.

### E.2.13 Proof of Proposition E.5

We observe that since $Q^\star$ lies in the convex PSD cone, $\|\widehat{Q} - Q^\star\|_{\mathrm{F}} \le \|(\frac{1}{2}\widetilde{Q}^\top + \frac{1}{2}\widetilde{Q}) - Q^\star\|_{\mathrm{F}}$ by the Pythagorean theorem. In more detail, we have the following result.

**Claim E.6.** Let $A, B \in \mathbb{R}^{d \times d}$, and let $B \ge 0$. Then $\|A_+ - B\|_F \le \|A - B\|_F$.

*Proof.* Let $A = A_+ + A_-$, so that $A_+ \ge 0$ and $A_- \le 0$. Then we have

$$\|A - B\|_F^2 - \|A_+ - B\|_F^2 = \|A_+ + A_- - B\|_F^2 - \|A_+ - B\|_F^2 = \|A_-\|_F^2 + \langle A_-, A_+ - B\rangle.$$

Now, note that $\langle A_-, A_+ - B\rangle = \langle -A_-, B\rangle \ge 0$, since $\langle X, Y\rangle \ge 0$ whenever $X, Y \ge 0$. $\qquad\square$

Moreover, $Q^\star = (Q^\star)^\top$, so $\|(\frac{1}{2}\widetilde{Q}^\top + \frac{1}{2}\widetilde{Q}) - Q^\star\|_{\mathrm{F}} \le \|\widetilde{Q} - Q^\star\|_{\mathrm{F}}$ by the triangle inequality. Thus, we conclude

$$\|\widehat{Q} - Q^\star\|_{\mathrm{F}} \le \|\widetilde{Q} - Q^\star\|_{\mathrm{F}}.$$

Next, let us introduce $\mathbf{v}_i := \mathrm{vec}(g_\star(\mathbf{y}^{(i)})g_\star(\mathbf{y}^{(i)})^\top) \in \mathbb{R}^{d^2}$ and $\hat{\mathbf{v}}_i := \mathrm{vec}(\hat{g}(\mathbf{y}^{(i)})\hat{g}(\mathbf{y}^{(i)})^\top)$. Let $\mathbf{V} \in \mathbb{R}^{n \times d^2}$ denote the matrix whose rows are $\mathbf{v}_i$ and $\widehat{\mathbf{V}}$ analogouly for $\hat{\mathbf{v}}$. Then, we have that

$$\mathrm{vec}(\widehat{Q}) = \widehat{\mathbf{V}}^\dagger \mathbf{V} \mathrm{vec}(Q^\star)$$
$$= \widehat{\mathbf{V}}^\dagger \widehat{\mathbf{V}} \mathrm{vec}(Q^\star) + \widehat{\mathbf{V}}^\dagger(\mathbf{V} - \widehat{\mathbf{V}})\mathrm{vec}(Q^\star)$$
$$= \mathrm{vec}(Q^\star) + \widehat{\mathbf{V}}^\dagger(\mathbf{V} - \widehat{\mathbf{V}})\mathrm{vec}(Q^\star),$$

provided that $\widehat{\mathbf{V}}$ is full rank (which we ultimately verify), where we recall that $\widehat{\mathbf{V}}^\dagger = (\widehat{\mathbf{V}}^\top\widehat{\mathbf{V}})^{-1}\widehat{\mathbf{V}}^\top$ in this case. Next, we bound

$$\|\mathrm{vec}(\widehat{Q}) - \mathrm{vec}(Q^\star)\|^2 \le \frac{1}{\sigma_{\min}(\widehat{\mathbf{V}})^2}\|(\mathbf{V} - \widehat{\mathbf{V}})\mathrm{vec}(Q^\star)\|_2^2$$

$$= \frac{1}{\lambda_{\min}(\widehat{\mathbf{V}}^\top\widehat{\mathbf{V}})}\sum_{i=1}^{n}\big\langle\mathrm{vec}(g_\star(\mathbf{y}^{(i)})g_\star(\mathbf{y}^{(i)})^\top) - \mathrm{vec}(\hat{g}(\mathbf{y}^{(i)})\hat{g}(\mathbf{y}^{(i)})^\top, \mathrm{vec}(Q^\star)\big\rangle^2$$

$$= \frac{1}{\lambda_{\min}(\widehat{\mathbf{V}}^\top\widehat{\mathbf{V}})}\sum_{i=1}^{n}\big\langle g_\star(\mathbf{y}^{(i)})g_\star(\mathbf{y}^{(i)})^\top - \hat{g}(\mathbf{y}^{(i)})\hat{g}(\mathbf{y}^{(i)})^\top, Q^\star\big\rangle^2$$

$$= \frac{1}{\lambda_{\min}(\widehat{\mathbf{V}}^\top\widehat{\mathbf{V}})}\sum_{i=1}^{n}\big\langle g_\star(\mathbf{y}^{(i)})g_\star(\mathbf{y}^{(i)})^\top - \hat{g}(\mathbf{y}^{(i)})\hat{g}(\mathbf{y}^{(i)})^\top, Q^\star\big\rangle^2$$

$$\overset{(a)}{\le} \frac{\|Q^\star\|_{\mathrm{op}}^2}{\lambda_{\min}(\widehat{\mathbf{V}}^\top\widehat{\mathbf{V}})}\sum_{i=1}^{n}\|g_\star(\mathbf{y}^{(i)})g_\star(\mathbf{y}^{(i)})^\top - \hat{g}(\mathbf{y}^{(i)})\hat{g}(\mathbf{y}^{(i)})^\top\|_{\mathrm{nuc}}^2$$

$$\overset{(b)}{\le} \frac{2\|Q^\star\|_{\mathrm{op}}^2}{\lambda_{\min}(\widehat{\mathbf{V}}^\top\widehat{\mathbf{V}})}\sum_{i=1}^{n}\|g_\star(\mathbf{y}^{(i)})g_\star(\mathbf{y}^{(i)})^\top - \hat{g}(\mathbf{y}^{(i)})\hat{g}(\mathbf{y}^{(i)})^\top\|_{\mathrm{F}}^2,$$

where $(a)$ uses Hölder's inequality ($|\langle A, B \rangle| \le \|A\|_{\text{op}} \|B\|_{\text{nuc}}$), and $(b)$ uses that $g_\star(\mathbf{y}^{(i)}) g_\star(\mathbf{y}^{(i)})^\top - \hat{g}(\mathbf{y}^{(i)}) \hat{g}(\mathbf{y}^{(i)})^\top$ has rank 2, so its nuclear norm is at most $\sqrt{2}$ times its Frobenius norm. Recognizing $\sum_{i=1}^{n} \|g_\star(\mathbf{y}^{(i)}) g_\star(\mathbf{y}^{(i)})^\top - \hat{g}(\mathbf{y}^{(i)}) \hat{g}(\mathbf{y}^{(i)})^\top\|_{\text{F}}^2 = \|\widehat{\mathbf{V}} - \mathbf{V}\|_{\text{F}}^2$, we obtain

$$\|\text{vec}(\widetilde{Q}) - \text{vec}(Q^\star)\|^2 \le \frac{4\|\mathbf{V} - \widehat{\mathbf{V}}\|_{\text{F}}^2 \|Q^\star\|_{\text{op}}^2}{\lambda_{\min}(\widehat{\mathbf{V}}^\top \widehat{\mathbf{V}})}. \tag{E.25}$$

Next, we give the following bound.

**Claim E.7.** Suppose that $\psi(n, \delta) \le \frac{\varepsilon^2}{4c}$. Then, with probability $1 - \delta$, we have the bound $\|\mathbf{V} - \widehat{\mathbf{V}}\|_{\text{op}}^2 \le \|\mathbf{V} - \widehat{\mathbf{V}}\|_{\text{F}}^2 \le 8 c_{n,\delta} \varepsilon^2$, where $c_{n,\delta} = c \ln(2n/\delta)$.

*Proof.* To begin, observe that

$$\begin{aligned}
\|\mathbf{V} - \widehat{\mathbf{V}}\|_{\text{op}}^2 &\le \|\mathbf{V} - \widehat{\mathbf{V}}\|_{\text{F}}^2 \\
&= \sum_{i=1}^{n} \|\text{vec}(g_\star(\mathbf{y}^{(i)}) g_\star(\mathbf{y}^{(i)})^\top) - \text{vec}(\hat{g}(\mathbf{y}^{(i)}) \hat{g}(\mathbf{y}^{(i)})^\top)\|_2^2 \\
&\le \sum_{i=1}^{n} \|g_\star(\mathbf{y}^{(i)}) g_\star(\mathbf{y}^{(i)})^\top - \hat{g}(\mathbf{y}^{(i)}) \hat{g}(\mathbf{y}^{(i)})^\top\|_{\text{F}}^2 \\
&\le \sum_{i=1}^{n} \left( (\|g_\star(\mathbf{y}^{(i)})\| + \|g_\star(\mathbf{y}^{(i)})\|)^2 \|g_\star(\mathbf{y}^{(i)}) - \hat{g}(\mathbf{y}^{(i)})\|^2 \right).
\end{aligned}$$

Introduce the event $\mathcal{E} := \max\{\|g_\star(\mathbf{y})\|^2, \|\hat{g}(\mathbf{y})\|^2\} \le c_{n,\delta} := c \ln(2n/\delta)$, and let $\mathcal{E}^{(i)}$ denote the analogous event for $\mathbf{y}^{(i)}$. Let $\mathcal{E}^{(1:n)} = \bigcap_{i=1}^{n} \mathcal{E}^{(i)}$. Then $\mathcal{E}^{(1:n)}$ holds with probability at least $1 - \delta/2$, and on this event the above display is at most

$$\|\mathbf{V} - \widehat{\mathbf{V}}\|_{\text{op}}^2 \le 4 c_{n,\delta} \sum_{i=1}^{n} \mathbb{I}(\mathcal{E}^{(i)}) \|g_\star(\mathbf{y}^{(i)}) - \hat{g}(\mathbf{y}^{(i)})\|^2.$$

Next, define the random variable $\boldsymbol{\delta}_i := \mathbb{I}(\mathcal{E}^{(i)}) \|g_\star(\mathbf{y}^{(i)}) - \hat{g}(\mathbf{y}^{(i)})\|^2$, and we observe that $\boldsymbol{\delta}_i \le 4c \ln(2n/\delta)$ with probability 1. Thus, by applying Lemma E.1 with $c \leftarrow 4c$, we have that for any $\varepsilon^2 \ge \mathbb{E}[\mathbb{I}(\mathcal{E}^{(i)}) \|g_\star(\mathbf{y}^{(i)}) - \hat{g}(\mathbf{y}^{(i)})\|^2]$, with probability at least $1 - \delta/2$,

$$\sum_{i=1}^{n} \mathbb{I}(\mathcal{E}^{(i)}) \|g_\star(\mathbf{y}^{(i)}) - \hat{g}(\mathbf{y}^{(i)})\|^2 \le 2\varepsilon^2,$$

as soon as $\psi(n, \delta) \le \frac{\varepsilon^2}{4c}$. In paricular, since $\|g_\star(\mathbf{y}^{(i)}) - \hat{g}(\mathbf{y}^{(i)})\|^2 \ge 0$, it is valid to select $\varepsilon^2 \ge \mathbb{E}[\|g_\star(\mathbf{y}^{(i)}) - \hat{g}(\mathbf{y}^{(i)})\|^2]$. Hence, for such $\varepsilon^2$, we conclude that with total probability at least $1 - \delta$,

$$\|\mathbf{V} - \widehat{\mathbf{V}}\|_{\text{op}}^2 \le 8 c_{n,\delta} \varepsilon^2.$$

$\square$

Denote the event of Claim E.7 by $\mathcal{E}_1$. Then on $\mathcal{E}_1$, Eq. (E.25) implies

$$\|\widetilde{Q} - Q^\star\|_{\text{F}}^2 \le \|Q^\star\|_{\text{F}}^2 \frac{8 c_{n,\delta} \varepsilon^2}{\lambda_{\min}(\widehat{\mathbf{V}}^\top \widehat{\mathbf{V}})}. \tag{E.26}$$

Next, from Claim E.2, we have that

$$\|\widehat{\mathbf{V}} - \mathbf{V}\|_{\text{op}}^2 \le \frac{1}{4} \lambda_{\min}(\mathbf{V}^\top \mathbf{V}) \quad \text{implies} \quad \lambda_{\min}(\widehat{\mathbf{V}}^\top \widehat{\mathbf{V}}) \ge \frac{1}{4} \lambda_{\min}(\mathbf{V}^\top \mathbf{V}).$$

And thus, on $\mathcal{E}_1$, we have that

$$\varepsilon^2 \le \frac{\lambda_0}{32 c_{n,\delta}} \le \frac{1}{4} \lambda_{\min}(\mathbf{V}^\top \mathbf{V}) \quad \text{implies} \quad \lambda_{\min}(\widehat{\mathbf{V}}^\top \widehat{\mathbf{V}}) \ge \frac{1}{4} \lambda_{\min}(\mathbf{V}^\top \mathbf{V}). \tag{E.27}$$

Let us now lower bound $\lambda_{\min}(\mathbf{V}^\top \mathbf{V})$ with high probability. We observe that $\lambda_{\min}(\mathbf{V}^\top \mathbf{V}) \geq \lambda_0$ for some $\lambda_0 > 0$ if and only if

$$\forall M \in \mathbb{R}^{d \times d}: \quad \sum_{i=1}^{n} \langle g_\star(\mathbf{y}^{(i)}) g_\star(\mathbf{y}^{(i)})^\top, M \rangle^2 \geq \lambda_0 \|M\|_{\mathrm{F}}^2,$$

if and only if

$$\forall M \in \mathbb{R}^{d \times d}: \quad \sum_{i=1}^{n} \langle \mathbf{x}^{(i)} \mathbf{x}^{(i)\top}, M \rangle^2 \geq \lambda_0 \|M\|_{\mathrm{F}}^2, \tag{E.28}$$

where $\mathbf{x}^{(i)} := g_\star(\mathbf{y}^{(i)}) \overset{\text{i.i.d.}}{\sim} \mathcal{N}(0, \Sigma_x)$ by assumption.

**Claim E.8.** Let $\mathbf{x}^{(i)} \overset{\text{i.i.d.}}{\sim} \mathcal{N}(0, \Sigma_x)$. Then, for $n \geq c_1 d$, the following holds with probability $1 - e^{-c_2 n}$:

$$\forall M \in \mathbb{R}^{d \times d}: \quad \sum_{i=1}^{n} \langle \mathbf{x}^{(i)} \mathbf{x}^{(i)\top}, M \rangle^2 \geq \frac{1}{2} \lambda_{\min}(\Sigma_x)^2 \|M\|_{\mathrm{F}}^2,$$

where $c_1$ is a numerical constant

*Proof.* Define $\tilde{\mathbf{x}}^{(i)} := \Sigma_x^{-1/2} \mathbf{v}^{(i)}$. Note that $\tilde{\mathbf{x}}^{(i)} \sim \mathcal{N}(0, I)$. From Wainwright [43, Theorem 10.12], we find that for any $\alpha > 0$, for all $n \geq c_1 \alpha d$, with probability $1 - e^{-c_2 n}$, the following holds simultaneously for all matrices $M$ satisfies $\|M\|_{\mathrm{nu}}^2 \leq \alpha \|M\|_{\mathrm{F}}^2$

$$\sum_{i=1}^{n} \langle \tilde{\mathbf{x}}^{(i)} \tilde{\mathbf{x}}^{(i)\top}, M \rangle^2 \geq \frac{1}{2} \|M\|_{\mathrm{F}}^2, \tag{E.29}$$

where $\|M\|_{\mathrm{nuc}} = \sum_{i=1}^{n} \sigma_i(M)$ denotes the matrix nuclear norm. By Cauchy-Schwartz, $\|M\|_{\mathrm{nu}}^2 \leq d\|M\|_{\mathrm{F}}^2$ for all matrices $M \in \mathbb{R}^{d \times d}$. This means that we capture *all* matrices $M$ by setting $\alpha = d$, and thus, for $n \geq c_1 d^2$, then with probability $1 - e^{-c_2 n}$, Eq. (E.29) holds for all $M \in \mathbb{R}^{d \times d}$ simultaneously. When this holds, we have that for all such $M$,

$$\sum_{i=1}^{n} \langle \mathbf{x}^{(i)} \mathbf{x}^{(i)\top}, M \rangle^2 = \sum_{i=1}^{n} \langle \tilde{\mathbf{x}}^{(i)} \tilde{\mathbf{x}}^{(i)\top}, \Sigma_x^{1/2} M \Sigma_x^{1/2} \rangle^2 \geq \frac{1}{2} \|\Sigma_x^{1/2} M \Sigma_x^{1/2}\|_{\mathrm{F}}^2.$$

Moreover, we have that

$$
\begin{aligned}
\|\Sigma_x^{1/2} M \Sigma_x^{1/2}\|_{\mathrm{F}}^2 &= \mathrm{tr}(\Sigma_x^{1/2} M \Sigma_x M^\top \Sigma_x^{1/2}) \\
&\geq \lambda_{\min}(\Sigma_x) \mathrm{tr}(\Sigma_x^{1/2} M M^\top \Sigma_x^{1/2}) \\
&= \lambda_{\min}(\Sigma_x) \mathrm{tr}(M^\top \Sigma_x M) \geq \lambda_{\min}(\Sigma_x)^2 \mathrm{tr}(M^\top M) = \lambda_{\min}(\Sigma_x)^2 \|M\|_{\mathrm{F}}^2.
\end{aligned}
$$

$\square$

Denote the event of Claim E.8 by $\mathcal{E}_2$. Then, on $\mathcal{E}_2$, we can take $\lambda_0 = \frac{1}{2}\lambda_{\min}(\Sigma_x)^2$ in Eq. (E.28), and thus on $\mathcal{E}_1$, Eq. (E.27) yields that

$$\varepsilon^2 \leq \frac{\lambda_{\min}(\Sigma_x)^2}{64 c_{n,\delta}} \leq \frac{1}{4}\lambda_0 \quad \text{implies} \quad \lambda_{\min}(\widehat{\mathbf{V}}^\top \widehat{\mathbf{V}}) \geq \frac{\lambda_{\min}(\Sigma_x)^2}{8}.$$

Thus, by Eq. (E.26), we have that on $\mathcal{E}_1 \cap \mathcal{E}_2$,

$$\|\widetilde{Q} - Q^\star\|_{\mathrm{F}}^2 = \|\mathrm{vec}(\widetilde{Q}) - q_\star\|^2 \leq 4 \cdot 16 c_{n,\delta} \varepsilon^2 \cdot \frac{\|Q^\star\|_{\mathrm{op}}^2}{\lambda_{\min}(\Sigma_x)^2},$$

giving us the desired inequality. Since $\mathbb{P}(\mathcal{E}_1 \cap \mathcal{E}_2) \geq 1 - \delta - e^{-c_2 n}$ for $n \geq c_1 d^2$, we have that if $n \geq c(d^2 + \ln(1/\delta))$ for some universal constant $c$, $\mathbb{P}(\mathcal{E}_1 \cap \mathcal{E}_2) \geq 1 - 2\delta$, yielding our desired failure probability. Recalling that $c_{n,\delta} := c \ln(2/\delta)$ concludes. $\square$

## F   Linear Control Theory

In this section we recall some basic results for the classical LQR problem in the fully observed setting with known dynamics. The main result for this section is Theorem 7, which bounds the regret of any policy for the RichLQR in terms of decoding errors. Proofs are deferred to the end of the section.

### F.1 Basic Technical Results

**Lemma F.1.** Let $X$ be any matrix with $\rho(X) < 1$. Then for any $Y > 0$, there exists a unique solution $P > 0$ to the Lyapunov equation

$$P = X^\top P X + Y. \tag{F.1}$$

Moreover, $X$ is $(\alpha, \gamma)$-strongly stable for $\alpha = \|P^{1/2}\|_{\mathrm{op}} \|P^{-1/2}\|_{\mathrm{op}}$ and $\gamma = \|I - P^{-1/2} Y P^{-1/2}\|_{\mathrm{op}}^{1/2}$.

This lemma immediately implies the following strong stability guarantees for the closed-loop and open-loop dynamics for LQR.

**Proposition F.1.** $A_{\mathrm{cl},\infty} := A + BK_\infty$ is $(\alpha_\infty, \gamma_\infty)$-strongly stable, where $\alpha_\infty := \|P_\infty^{1/2}\|_{\mathrm{op}} \|P_\infty^{-1/2}\|_{\mathrm{op}}$ and $\gamma_\infty := \|I - P_\infty^{-1/2} Q P_\infty^{-1/2}\|_{\mathrm{op}}^{1/2} < 1$.[9]

**Proposition F.2.** If we define $\alpha_A := \|\Sigma_A^{1/2}\|_{\mathrm{op}} \|\Sigma_A^{-1/2}\|_{\mathrm{op}}$ and $\gamma_A := \|I_{d_{\mathbf{x}}} - \Sigma_A^{-1}\|^{1/2} < 1$, then $A$ is $(\alpha_A, \gamma_A)$-strongly stable, where $\Sigma_A$ is the unique solution to the Lyapunov equation

$$\Sigma = A\Sigma A^\top + I_{d_{\mathbf{x}}}. \tag{F.2}$$

We also make use of the following bound on the operator norm for the infinite-horizon covariance matrix.

**Proposition F.3.** We have $\|\Sigma_\infty\|_{\mathrm{op}} \le \|R\|_{\mathrm{op}} + \|B\|_{\mathrm{op}}^2 \|P_\infty\|_{\mathrm{op}} \le 2\Psi_\star^3$.

### F.2 Value Functions

Toward proving our main regret decomposition, in this section we establish some basic technical results regarding the value functions and Q-functions for the fully observed LQR problem. Our first result concerns finite-horizon value functions for linear controller.

**Lemma F.2.** Consider the RichLQR setting (1) under Assumption 1, and consider a state feedback controller $\pi_K(y) = Kf_\star(y)$, where $f_\star$ is the true decoder. Define

$$
\begin{aligned}
\mathbf{V}_{t:T}^K(x) &= \mathbb{E}_{\pi_K}\left[ \sum_{s=t}^T \mathbf{x}_s^\top Q \mathbf{x}_s + \mathbf{u}_s^\top R \mathbf{u}_s \mid \mathbf{x}_t = x \right], \\
\mathbf{Q}_{t:T}^K(x,u) &= \mathbb{E}_{\pi_K}\left[ \sum_{s=t}^T \mathbf{x}_s^\top Q \mathbf{x}_s + \mathbf{u}_s^\top R \mathbf{u}_s \mid \mathbf{x}_t = x, \mathbf{u}_t = u \right].
\end{aligned} \tag{F.3}
$$

Then we have

$$
\begin{aligned}
\mathbf{V}_{t:T}^K(x) &= \sum_{s=t}^T \left\| (A+BK)^{s-t} x \right\|_{Q+K^\top RK}^2 + F_{t:T}(A,B,Q,R,K,\Sigma_w), \\
\mathbf{Q}_{t:T}^K(x,u) &= \|x\|_Q^2 + \|u\|_R^2 + \mathbf{V}_{t+1:T}^K(Ax+Bu) + G_{t:T}(A,B,Q,R,K,\Sigma_w),
\end{aligned} \tag{F.4}
$$

where $F_{t:T}$ and $G_{t:T}$ are functions that depend on the system parameters and time horizon, but not the state or control inputs.

In light of Lemma F.2, it will be convenient to define

$$\overline{\mathbf{V}}_{t:T}^K(x) = \sum_{s=t}^T \left\| (A+BK)^{s-t} x \right\|_{Q+K^\top RK}^2, \tag{F.5}$$

which is simply the value function in Eq. (F.4) in the absence of noise. Our next result concerns the infinite-horizon value functions that arise in the noiseless setting.

**Lemma F.3** ([3]). Consider the optimal infinite horizon controller $\pi_\infty(x) = K_\infty x$, and define $\mathbf{V}_\infty(x) = \|x\|_{P_\infty}^2$. Then $\mathbf{V}_\infty$ is the infinite-horizon cost for playing $\pi_\infty$ starting from $\mathbf{x}_1 = x$ under the noiseless dynamics

$$\mathbf{x}_{t+1} = A\mathbf{x}_t + B\mathbf{u}_t.$$

Moreover, if we define $\mathbf{Q}_\infty(x, u) = \|x\|_Q^2 + \|u\|_R^2 + \|Ax + Bu\|_{P_\infty}^2$, we have

$$\pi_\infty(x) = \underset{u \in \mathbb{R}^{d_\mathbf{u}}}{\arg\min}\, \mathbf{Q}_\infty(x, u).$$

Finally, we have

$$P_\infty = \sum_{k=0}^\infty ((A + BK_\infty)^\top)^k (Q + K_\infty^\top R K_\infty)(A + BK_\infty)^k$$

The following lemma shows that the infinite-horizon value functions are well-approximated by their finite-horizon counterparts.

**Lemma F.4.** For all $x \in \mathbb{R}^{d_\times}$ and all $t \le T$, we have

$$\left| \overline{\mathbf{V}}_{t:T}^{K_\infty}(x) - \mathbf{V}_\infty(x) \right| \le \mathcal{O}(\alpha_\infty^2 (1 - \gamma_\infty^2)^{-1} \Psi_\star^3) \cdot \gamma_\infty^{2(T-t+1)} \|x\|_2^2$$

and

$$\left| \overline{\mathbf{Q}}_{t:T}^{K_\infty}(x, u) - \mathbf{Q}_\infty(x, u) \right| \le \mathcal{O}(\alpha_\infty^2 (1 - \gamma_\infty^2)^{-1} \Psi_\star^5) \cdot \gamma_\infty^{2(T-t)} (\|x\|_2^2 + \|u\|_2^2).$$

Lastly, we establish a Lipschitz property for the finite-horizon $Q$-functions.

**Lemma F.5.** For all $x \in \mathbb{R}^{d_\times}$ and $u, u' \in \mathbb{R}^{d_\mathbf{u}}$,

$$\left| \overline{\mathbf{Q}}_{t:T}^{K_\infty}(x, u) - \overline{\mathbf{Q}}_{t:T}^{K_\infty}(x, u') \right| \le \mathcal{O}(\Psi_\star^3) \cdot (\|x\|_2 \vee \|u\|_2 \vee \|u'\|_2) \|u - u'\|_2.$$

## F.3 Perturbation Bound for the Optimal Controller

To analyze the quality of the certainty-equivalent controller used in RichID-CE, we use the following perturbation bound.

**Theorem 6** (Mania et al. [24]). *Suppose we have matrices $(\widehat{A}, \widehat{B}, \widehat{Q})$ for which there exists an invertible transformation $G$ such that*

$$\left\| \widehat{A} - GAG^{-1} \right\|_{\mathrm{op}} \vee \left\| \widehat{B} - GB \right\|_{\mathrm{op}} \vee \left\| \widehat{Q} - G^{-\top} Q G^{-1} \right\|_{\mathrm{op}} \le \varepsilon.$$

*Suppose that $\|G\|_{\mathrm{op}} \vee \|G^{-1}\|_{\mathrm{op}} \le C_{\mathrm{sim}}$. Let $\widehat{K}$ be the optimal infinite-horizon controller for $(\widehat{A}, \widehat{B}, \widehat{Q}, R)$. Then once $\varepsilon \le c_{\mathrm{stable}} \cdot \gamma_\infty \cdot C_{\mathrm{sim}}^{-15} \alpha_\infty^{-4} (1 - \gamma_\infty^2)^{-2} \Psi_\star^{-11}$, where $c_{\mathrm{stable}}$ is a sufficiently small numerical constant,*

$$\left\| \widehat{K} - K_\infty G^{-1} \right\|_{\mathrm{op}} \le \mathcal{O}(C_{\mathrm{sim}}^{11} \alpha_\infty^2 (1 - \gamma_\infty^2)^{-1} \Psi_\star^9) \cdot \varepsilon, \tag{F.6}$$

*and we are guaranteed that $A + B\widehat{K}$ is $(\alpha_\infty, \bar{\gamma}_\infty)$-strongly stable, where $\bar{\gamma}_\infty = (1 + \gamma_\infty)/2$.*

## F.4 Regret Decomposition

The following theorem is the main result from this section, and shows that any policy of the form $\widehat{\pi}_t(\mathbf{y}_{1:t}) = \widehat{K} \hat{f}_t(\mathbf{y}_{1:t})$ (in particular, the policy returned by Phase III of RichID-CE), has low regret whenever $\widehat{K}$ accurately approximates $K_\infty$ and $\hat{f}_t$ has low prediction error on the state distribution induced by $\widehat{\pi}_{1:t-1}$.

**Theorem 7.** *Consider a randomized policy of the form $\widehat{\pi}_t(\mathbf{y}_{1:t}) = \widehat{K} \hat{f}_t(\mathbf{y}_{1:t}) + \mathbf{v}_t$, where $\mathbb{E}[\mathbf{v}_t \mid \mathbf{y}_{1:t}] = 0$. Suppose we are guaranteed that*

$$\left\| \widehat{K} - K_\infty \right\|_{\mathrm{op}} \le \varepsilon_K \le \|K_\infty\|, \quad \text{and} \quad \mathbb{E}_{\widehat{\pi}} \left\| \hat{f}_t(\mathbf{y}_{1:t}) - f_\star(\mathbf{y}_t) \right\|_2^2 \le \varepsilon_f^2 \quad \text{for all } t.$$

*Suppose that $\|\hat{f}_t\|_2 \le \bar{b}$ almost surely, that $\mathbb{E}\|\mathbf{v}_t\|_2^2 \le \sigma_\mathbf{v}^2$, and that $\mathbb{E}_{\widehat{\pi}}\|\mathbf{x}_t\|_2^2 \le c_\mathbf{x}^2$, where $\bar{b}, c_\mathbf{x} \ge 1$. Then for any $0 \le \tau \le T$, we have*

$$J_T(\widehat{\pi}) - J_T(\pi_\infty) \tag{F.7}$$

$$\le C_1 \cdot (\varepsilon_f^2 + \varepsilon_K^2 + \sigma_\mathbf{v}^2)(T - \tau)/T + C_2(\varepsilon_f + c_\mathbf{x} \cdot \varepsilon_K + \sigma_\mathbf{v})\tau/T + C_3 \exp(-2\ln(1/\gamma_\infty)\tau)(T - \tau)/T,$$

*where $C_1 \le \mathcal{O}(\Psi_\star^5 c_\mathbf{x}^2)$, $C_2 \le \mathcal{O}(\bar{b}\Psi_\star^5 c_\mathbf{x}(1 \vee \sigma_\mathbf{v}))$, and $C_3 \le \mathcal{O}(\alpha_\infty^2 \Psi_\star^7 (c_\mathbf{x}^2 \vee \sigma_\mathbf{v}^2 \vee \bar{b}^2))$.*

### F.5 Proofs for Linear Control Theory Results

*Proof of Lemma F.1.* Existence of a unique solution to the Lyapunov equation is a standard result [3]. Now, define $L = P^{1/2} X P^{-1/2}$. Then the Lyapunov equation (F.1) is equivalent to

$$L^\top L + P^{-1/2} Y P^{-1/2} = I.$$

This implies that

$$\|L\|_{\mathrm{op}}^2 = \left\|L^\top L\right\|_{\mathrm{op}} \le \underbrace{\left\|I - P^{-1/2} Y P^{-1/2}\right\|_{\mathrm{op}}}_{=\gamma^2} < 1.$$

Moreover, since $L = P^{1/2} Y P^{-1/2}$, we may take $\alpha = \|P^{1/2}\|_{\mathrm{op}} \|P^{-1/2}\|_{\mathrm{op}}$. $\qquad\square$

*Proof of Lemma F.2.* Since we have perfect decodability, $\pi_K$ operates directly on the true state, and so we may overload $\pi_K(x) = Kx$. To begin, we observe that if we begin at $\mathbf{x}_t = x$ and follow $\pi_K$, we have

$$\mathbf{x}_s = (A + BK)^{s-t} x + \sum_{i=t}^{s-1} (A + BK)^{s-i-1} \mathbf{w}_i.$$

It follows that

$$\mathbf{V}_{t:T}^K(x) = \mathbb{E}\left[ \sum_{s=t}^T \left\| (A + BK)^{s-t} x + \sum_{i=t}^{s-1} (A + BK)^{s-i-1} \mathbf{w}_i \right\|_{Q+K^\top RK}^2 \right].$$

However, since $\mathbf{w}_t$ are zero-mean and independent, we can expand the norm and cancel the cross terms, which allows us to write this as

$$\mathbf{V}_{t:T}^K(x) = \sum_{s=t}^T \left\| (A + BK)^{s-t} x \right\|_{Q+K^\top RK}^2 + F_{t:T}(A, B, Q, R, K, \Sigma_w). \tag{F.8}$$

The expression for $\mathbf{Q}_{t:T}^K$ immediately follows, since we have

$$\mathbf{Q}_{t:T}^K(x, u) = \mathbb{E}_{\pi_K}\left[ \|x\|_Q^2 + \|u\|_R^2 + \mathbf{V}_{t+1:T}(Ax + Bu + \mathbf{w}_t) \right].$$

The fact that $\mathbf{V}_{t+1:T}^K$ is quadratic and $\mathbf{w}_t$ is zero-mean again allows us to factor out the noise. $\qquad\square$

*Proof of Lemma F.4.* Since $\overline{\mathbf{V}}_{t:T}^K = \overline{\mathbf{V}}_{1:T-t+1}^K$, we focus on the case $t = 1$ without loss of generality. Observe that we have

$$\overline{\mathbf{V}}_{1:T}^{K_\infty}(x) = \sum_{k=0}^{T-1} \left\| (A + BK_\infty)^k x \right\|_{Q+K_\infty^\top RK_\infty}^2 = \left\langle x, \sum_{k=0}^{T-1} ((A + BK_\infty)^\top)^k (Q + K_\infty^\top RK_\infty)(A + BK_\infty)^k x \right\rangle.$$

Using the expression for $P_\infty$ from Lemma F.3, it follows that

$$\left| \overline{\mathbf{V}}_{1:T}^{K_\infty}(x) - \mathbf{V}_\infty(x) \right| \le \|x\|_2^2 \cdot \left\| \sum_{k=T}^\infty ((A + BK_\infty)^\top)^k (Q + K_\infty^\top RK_\infty)(A + BK_\infty)^k \right\|_{\mathrm{op}}$$

$$\le \|x\|_2^2 \cdot 2\Psi_\star^3 \sum_{k=T}^\infty \left\| (A + BK_\infty)^k \right\|_{\mathrm{op}}^2.$$

Now, using Proposition F.1, we are guaranteed that $\left\| (A + BK_\infty)^k \right\|_{\mathrm{op}} \le \alpha_\infty \gamma_\infty^k$, so we have

$$\sum_{k=T}^\infty \left\| (A + BK_\infty)^k \right\|_{\mathrm{op}}^2 \le \alpha_\infty^2 \sum_{k=T}^\infty \gamma_\infty^{2k} = \alpha_\infty^2 \gamma_\infty^{2T} (1 - \gamma_\infty^2)^{-1}.$$

This is establishes the bound on the error to $\mathbf{V}_\infty$. The error bound for the $Q$-functions follows immediately, since

$$\left| \overline{\mathbf{Q}}_{t:T}^{K_\infty}(x, u) - \mathbf{Q}_\infty(x, u) \right| = \left| \overline{\mathbf{V}}_{t+1:T}^{K_\infty}(Ax + Bu) - \mathbf{V}_\infty(Ax + Bu) \right|.$$

$\qquad\square$

*Proof of Lemma F.5.* We first compute that for any $x, x'$,

$$\left|\overline{\mathbf{V}}_{t+1:T}^{K_\infty}(x) - \overline{\mathbf{V}}_{t+1:T}^{K_\infty}(x')\right| \le 2(\|x\|_2 \vee \|x'\|_2)\|x - x'\|_2 \left\|\sum_{s=t}^{T}((A + BK_\infty)^\top)^{s-t}(Q + K_\infty^\top R K_\infty)(A + BK_\infty)^{s-t}\right\|_{\mathrm{op}}$$

$$\le 2(\|x\|_2 \vee \|x'\|_2)\|x - x'\|_2 \left\|\sum_{s=0}^{\infty}((A + BK_\infty)^\top)^{s-t}(Q + K_\infty^\top R K_\infty)(A + BK_\infty)^{s-t}\right\|_{\mathrm{op}}$$

$$= 2(\|x\|_2 \vee \|x'\|_2)\|x - x'\|_2 \|P_\infty\|_{\mathrm{op}}.$$

As a consequence, for all $x$ and $u, u'$, we have

$$\left|\overline{\mathbf{Q}}_{t:T}^{K_\infty}(x, u) - \overline{\mathbf{Q}}_{t:T}^{K_\infty}(x, u')\right| \le \left|\|u\|_R^2 - \|u'\|_R^2\right| + \left|\overline{\mathbf{V}}_{t+1:T}^{K_\infty}(Ax + Bu) - \overline{\mathbf{V}}_{t+1:T}^{K_\infty}(Ax + Bu')\right|$$

$$\le 2\Psi_\star(\|u\|_2 \vee \|u'\|_2)\|u - u'\|_2 + \left|\overline{\mathbf{V}}_{t+1:T}^{K_\infty}(Ax + Bu) - \overline{\mathbf{V}}_{t+1:T}^{K_\infty}(Ax + Bu')\right|$$

$$\le 2\Psi_\star(\|u\|_2 \vee \|u'\|_2)\|u - u'\|_2 + 2\|P_\infty\|(\|Ax + Bu\|_2 \vee \|Ax + Bu'\|_2)\|B(u - u')\|_2$$

$$\le \mathcal{O}(\Psi_\star^3) \cdot (\|x\|_2 \vee \|u\|_2 \vee \|u'\|_2)\|u - u'\|_2.$$

$\square$

*Proof of Theorem 6.* We first consider the case where $G$ is the identity matrix. We apply Proposition 2 of [24], which implies that[10]

$$\left\|\widehat{P} - P_\infty\right\|_{\mathrm{op}} \le \mathcal{O}(\alpha_\infty^2(1 - \gamma_\infty^2)^{-1}\Psi_\star^6) \cdot \varepsilon,$$

as long as $\varepsilon \le c \cdot (1 - \gamma_\infty^2)^2 \alpha_\infty^{-4}\Psi_\star^{-11}$, where $c$ is a sufficiently small numerical constant. Proposition 1 of [24] now implies that

$$\left\|\widehat{K} - K_\infty\right\|_{\mathrm{op}} \le \mathcal{O}(\alpha_\infty^2(1 - \gamma_\infty^2)^{-1}\Psi_\star^9) \cdot \varepsilon.$$

The strong stability result follows by observing that

$$\left\|P_\infty^{1/2}(A + B\widehat{K})P_\infty^{-1/2}\right\|_{\mathrm{op}} \le \left\|P_\infty^{1/2}(A + BK_\infty)P_\infty^{-1/2}\right\|_{\mathrm{op}} + \left\|P_\infty^{1/2}B(\widehat{K} - K_\infty)P_\infty^{-1/2}\right\|_{\mathrm{op}}$$

$$\le \gamma_\infty + \alpha_\infty\Psi_\star\left\|\widehat{K} - K_\infty\right\|_{\mathrm{op}}.$$

In the general case, we apply the reasoning above with $A' = BAG^{-1}$, $B' = GB$, and $Q' = G^{-T}QG^{-1}$, and $R$, and observe that the optimal controller for this system is $K_\infty G^{-1}$. The same perturbation bound holds, but with $\Psi_\star$ scaled up by at most $C_{\mathrm{sim}}^2$ and $\alpha_\infty$ scaled up by at most $C_{\mathrm{sim}}$.

$\square$

*Proof of Theorem 7.* Before beginning the proof, we collect some helpful norm bounds. We have:

$$\mathbb{E}_{\widehat{\pi}}\|\mathbf{x}_t\|_2^2 \le c_{\mathbf{x}}^2, \tag{F.9}$$

$$\mathbb{E}_{\widehat{\pi}}\|\widehat{\pi}_t(\mathbf{y}_{1:t})\|_2^2 \le \mathbb{E}_{\widehat{\pi}}\left\|\widehat{K}\widehat{f}_t(\mathbf{y}_{1:t})\right\|_2^2 + \sigma_{\mathbf{v}}^2 \le 42\Psi_\star^2\overline{b}^2 + \sigma_{\mathbf{v}}^2, \tag{F.10}$$

$$\mathbb{E}_{\widehat{\pi}}\|\pi_t^\star(\mathbf{y}_t)\|_2^2 \le \|K_\infty\|_{\mathrm{op}}^2 \mathbb{E}_{\widehat{\pi}}\|\mathbf{x}_t\|_2^2 \le \Psi_\star^2 c_{\mathbf{x}}^2, \tag{F.11}$$

where (F.10) follows because $\|\widehat{K}\|_{\mathrm{op}} \le 2\|K_\infty\|_{\mathrm{op}}$, so that $\|\widehat{K}\widehat{f}(\mathbf{y}_{1:t})\| \le 2\|K_\infty\|_{\mathrm{op}}\overline{b} \le 2\Psi_\star\overline{b}$ almost surely.

As a first-step, using the standard performance difference lemma [17], we have

$$J_T(\widehat{\pi}) - J_T(\pi_\infty) = \mathbb{E}_{\widehat{\pi}}\left[\frac{1}{T}\sum_{t=1}^{T}\mathbf{Q}_{t:T}^{K_\infty}(\mathbf{x}_t, \widehat{\pi}(\mathbf{y}_{1:t})) - \mathbf{Q}_{t:T}^{K_\infty}(\mathbf{x}_t, \pi_\infty(\mathbf{y}_t))\right],$$

where we have used Assumption 1, which implies that the Q-functions for $\pi_\infty$ have the form in Eq. (F.4).

Let $T_0 = T - \tau$. We handle the timesteps before and after $T_0$ separately. For the first case, where $t \leq \tau$, we apply Lemma F.4, which implies that

$$\mathbb{E}_{\widehat{\pi}}\left[\sum_{t=1}^{T_0} \mathbf{Q}_{t:T}^{K_\infty}(\mathbf{x}_t, \widehat{\pi}(\mathbf{y}_{1:t})) - \mathbf{Q}_{t:T}^{K_\infty}(\mathbf{x}_t, \pi_\infty(\mathbf{y}_t))\right]$$

$$\leq \mathbb{E}_{\widehat{\pi}}\left[\sum_{t=1}^{T_0} \mathbf{Q}_\infty(\mathbf{x}_t, \widehat{\pi}(\mathbf{y}_{1:t})) - \mathbf{Q}_\infty(\mathbf{x}_t, \pi_\infty(\mathbf{y}_t))\right]$$

$$+ \mathcal{O}(\alpha_\infty^2 (1 - \gamma_\infty^2)^{-1} \Psi_\star^5) \cdot \sum_{t=1}^{T_0} \gamma_\infty^{2(T-t)} (\mathbb{E}_{\widehat{\pi}} \|\mathbf{x}_t\|_2^2 + \mathbb{E}\|\widehat{\pi}(\mathbf{y}_{1:t})\|_2^2 + \mathbb{E}\|\pi^\star(\mathbf{y}_{1:t})\|_2^2)$$

We simplify the error term above to

$$\gamma_\infty^{2\tau}(T - \tau) \cdot \mathcal{O}\left(\alpha_\infty^2 \Psi_\star^7 (c_\mathbf{x}^2 \vee \sigma_\mathbf{v}^2 \vee \bar{b}^2)\right).$$

To handle the summands, we observe that since $\pi_\infty(\mathbf{y}_t) = \arg\min_{u \in \mathbb{R}^{d_\mathbf{u}}} \mathbf{Q}_\infty(\mathbf{x}, u)$, and since $\mathbf{Q}_\infty$ is a strongly convex quadratic with Hessian $P_\infty + B^\top P_\infty B = \Sigma_\infty$, the first-order conditions for optimality imply that

$$\mathbf{Q}_\infty(\mathbf{x}_t, \widehat{\pi}(\mathbf{y}_{1:t})) - \mathbf{Q}_\infty(\mathbf{x}_t, \pi_\infty(\mathbf{y}_t)) = \|\widehat{\pi}(\mathbf{y}_{1:t}) - \pi_\infty(\mathbf{y}_t)\|_{\Sigma_\infty}^2.$$

Thus, since $\|\Sigma_\infty\|_{\mathrm{op}} \leq 2\Psi_\star^3$ (Proposition F.3), we have

$$\mathbb{E}_{\widehat{\pi}}\left[\sum_{t=1}^{T_0} \mathbf{Q}_\infty(\mathbf{x}_t, \widehat{\pi}(\mathbf{y}_{1:t})) - \mathbf{Q}_\infty(\mathbf{x}_t, \pi_\infty(\mathbf{y}_t))\right] \leq 2\Psi_\star^3 \sum_{t=1}^{T_0} \mathbb{E}_{\widehat{\pi}} \|\widehat{\pi}(\mathbf{y}_{1:t}) - \pi^\star(\mathbf{y}_t)\|_2^2.$$

Now, for each $t$, we have

$$\mathbb{E}_{\widehat{\pi}} \|\widehat{\pi}(\mathbf{y}_{1:t}) - \pi^\star(\mathbf{y}_t)\|_2^2 = \mathbb{E}_{\widehat{\pi}} \|\widehat{K}\hat{f}_t(\mathbf{y}_{1:t}) + \mathbf{v}_t - K_\infty f_\star(\mathbf{y}_t)\|_2^2$$

$$\leq \mathbb{E}_{\widehat{\pi}} \|\widehat{K}\hat{f}_t(\mathbf{y}_{1:t}) - K_\infty f_\star(\mathbf{y}_t)\|_2^2 + \sigma_\mathbf{v}^2$$

$$\leq 2\mathbb{E}_{\widehat{\pi}} \|\widehat{K}\hat{f}_t(\mathbf{y}_{1:t}) - \widehat{K} f_\star(\mathbf{y}_t)\|_2^2 + 2\mathbb{E}_{\widehat{\pi}} \|(\widehat{K} - K_\infty)f_\star(\mathbf{y}_t)\|_2^2 + 2\sigma_\mathbf{v}^2$$

$$\leq 8\Psi_\star^2 \mathbb{E}_{\widehat{\pi}} \|\hat{f}_t(\mathbf{y}_{1:t}) - f_\star(\mathbf{y}_t)\|_2^2 + 24c_\mathbf{x}^2 \|\widehat{K} - K_\infty\|_{\mathrm{op}}^2 + 2\sigma_\mathbf{v}^2$$

$$\leq 8\Psi_\star^2 \varepsilon_f^2 + 24c_\mathbf{x}^2 \varepsilon_K^2 + 2\sigma_\mathbf{v}^2. \tag{F.11}$$

Collecting terms, this gives a coarse bound of

$$\mathbb{E}_{\widehat{\pi}}\left[\sum_{t=1}^{T_0} \mathbf{Q}_\infty(\mathbf{x}_t, \widehat{\pi}(\mathbf{y}_{1:t})) - \mathbf{Q}_\infty(\mathbf{x}_t, \pi_\infty(\mathbf{y}_t))\right] \leq \mathcal{O}\left(\Psi_\star^5 c_\mathbf{x}^2 (T - \tau)(\varepsilon_f^2 + \varepsilon_K^2 + \sigma_\mathbf{v}^2)\right).$$

We now bound the terms after time $T_0$. Using Lemma F.5, we have

$$\mathbb{E}_{\widehat{\pi}}\left[\sum_{t=T_0}^{T} \mathbf{Q}_{t:T}^{K_\infty}(\mathbf{x}_t, \widehat{\pi}(\mathbf{y}_{1:t})) - \mathbf{Q}_{t:T}^{K_\infty}(\mathbf{x}_t, \pi_\infty(\mathbf{y}_t))\right]$$

$$\leq \mathcal{O}(\Psi_\star^3) \mathbb{E}_{\widehat{\pi}}\left[\sum_{t=T_0}^{T} (\|\mathbf{x}_t\|_2 + \|\widehat{\pi}_t(\mathbf{y}_{1:t})\|_2 + \|\pi_\infty(\mathbf{y}_t)\|_2)\|\widehat{\pi}(\mathbf{y}_{1:t}) - \pi_\infty(\mathbf{y}_t)\|_2\right]$$

$$\leq \mathcal{O}(\bar{b}\Psi_\star^4) \mathbb{E}_{\widehat{\pi}}\left[\sum_{t=T_0}^{T} (\|\mathbf{x}_t\|_2 + \|\mathbf{v}_t\|_2)\|\widehat{\pi}(\mathbf{y}_{1:t}) - \pi_\infty(\mathbf{y}_t)\|_2\right]$$

$$\leq \mathcal{O}(\bar{b}\Psi_\star^4) \sum_{t=T_0}^{T} \left(\sqrt{\mathbb{E}_{\widehat{\pi}} \|\mathbf{x}_t\|_2^2} + \sqrt{\mathbb{E}\|\mathbf{v}_t\|_2^2}\right)\sqrt{\mathbb{E}_{\widehat{\pi}} \|\widehat{\pi}(\mathbf{y}_{1:t}) - \pi_\infty(\mathbf{y}_t)\|_2^2}$$

$$\leq \mathcal{O}(\bar{b}\Psi_\star^4 (c_\mathbf{x} + \sigma_\mathbf{v})) \sum_{t=T_0}^{T} \sqrt{\mathbb{E}_{\widehat{\pi}} \|\widehat{\pi}(\mathbf{y}_{1:t}) - \pi_\infty(\mathbf{y}_t)\|_2^2}$$

$$\leq \mathcal{O}(\bar{b}\Psi_\star^4 (c_\mathbf{x} + \sigma_\mathbf{v})(\Psi_\star \varepsilon_f + c_\mathbf{x}\varepsilon_K + \sigma_\mathbf{v}) \cdot \tau)$$

$$\leq \mathcal{O}(\bar{b}\Psi_\star^5 c_\mathbf{x}(1 \vee \sigma_\mathbf{v})(\varepsilon_f + c_\mathbf{x}\varepsilon_K + \sigma_\mathbf{v}) \cdot \tau),$$

where the second-to-last inequality uses Eq. (F.11).

$\square$

# G  Proofs for RichID Phase I and II

The section is organized as follows.

- Appendix G.1 contains preliminaries. Appendix G.1.1 establishes the relevant Gaussian marginals and conditionals, Appendix G.1.2 specifies the burn-in parameter $\kappa_0$, and Appendix G.1.3 addresses relevant properties of the function class $\mathscr{H}_{\mathrm{id}}^{\circ}$.
- Appendix G.2 provides proofs for Phase I, in particular Theorem 2 and its more granular statement, Theorem 2a.
- Appendix G.3 provides proofs for Phase II, including Theorem 3/Theorem 3a.

## G.1  Preliminaries

Recall that in the identification phase, for each $t \geq 0$, we take $\mathbf{u}_t \sim \mathcal{N}(0, I_{d_{\mathbf{u}}})$. We recall that the controllability matrices are given by $\mathcal{C}_k = [A^{k-1}B \mid \ldots \mid B]$, and define the following matrices:

$$\Sigma_{k,\mathrm{id}} := A^k \Sigma_0 (A^k)^\top + \sum_{s=0}^{k-1} (A^s)(\Sigma_w + BB^\top)(A^s)^\top. \tag{G.1}$$

$$\Sigma_{\infty,\mathrm{id}} := \sum_{s=0}^{\infty} (A^s)(\Sigma_w + BB^\top)(A^s)^\top. \tag{G.2}$$

We also recall the definition of $\kappa_0$ and $\kappa_1$:

$$\kappa_0 := \left\lceil \frac{1}{1-\gamma_\star} \ln\left( \frac{84 \Psi_\star^5 \alpha_\star^4 d_{\mathbf{x}} \ln(1000 n_{\mathrm{id}})}{(1-\gamma_\star)^2} \right) \right\rceil. \tag{G.3}$$

$$\kappa_1 := \kappa_0 + \kappa. \tag{G.4}$$

Finally, we define

$$\mathbf{v} := (\mathbf{u}_{\kappa_0}^\top, \ldots, \mathbf{u}_{\kappa_1-1}^\top)^\top \in \mathbb{R}^{\kappa d_{\mathbf{u}}},$$

and we recall the definition of the function class used in the regression problem for Phase I:

$$\mathscr{H}_{\mathrm{id}}^{\circ} := \left\{ Mf(\cdot) \mid f \in \mathscr{F}, \ M \in \mathbb{R}^{\kappa d_{\mathbf{u}} \times d_{\mathbf{x}}}, \ \|M\|_{\mathrm{op}} \leq \sqrt{\Psi_\star} \right\}, \tag{G.5}$$

which corresponds to choosing $r_{\mathrm{id}} = \sqrt{\Psi_\star}$.

### G.1.1  Marginals and Conditions

To compute the Bayes regression function for Phase I we use the following results, which are readily verified.

**Fact G.1** (Marginals for Phase I)**.** Fix $\kappa, \kappa_0$ and define. $\kappa_1 := \kappa_0 + \kappa$. Then $\mathbf{v}, \mathbf{x}_{\kappa_1}$ are jointly Gaussian are jointly gaussian and mean zero. Moreover, $\mathbf{x}_{\kappa_1} \sim \mathcal{N}(0, \Sigma_{\kappa_1,\mathrm{id}})$, $\mathbf{v} \sim \mathcal{N}(0, I_{kd_{\mathbf{u}}})$, and $\mathbb{E}[\mathbf{v}\mathbf{x}_k^\top] = \mathcal{C}_\kappa^\top$.

**Fact G.2** (Gaussian Expectation)**.** Let $(U, X)$ be jointly Gaussian random variables with distribution

$$(U, X) \sim \mathcal{N}\left(0, \begin{bmatrix} \Sigma_{UU} & \Sigma_{UX} \\ \Sigma_{XU} & \Sigma_{XX} \end{bmatrix}\right).$$

Then we have $\mathbb{E}[U \mid X = x] = \Sigma_{UX}\Sigma_{XX}^{-1}x$.

### G.1.2  Selecting the Burn-In Time

**Lemma G.1.** Fix an integer $n_{\mathrm{id}} \in \mathbb{N}$. Then as long as $\kappa_0$ satisfies Eq. (G.3), we have that for any $k, k' \geq \kappa_0$ (including $k = \infty$), the following properties hold.

1. The following bounds hold with respect to the PSD ordering:

$$\frac{9}{10}\Sigma_\star \preceq \Sigma_{k,\mathrm{id}} \preceq \frac{11}{10}\Sigma_\star \preceq \frac{11}{5}\Psi_\star^2 \alpha_\star^2 (1-\gamma_\star)^{-1} \cdot I.$$

2. Fix $\varepsilon > 0$. For any $h_1, h_2 \in \mathscr{H}_{\mathrm{id}}$ with $\mathbb{E}\|h_1(\mathbf{y}_k) - h_2(\mathbf{y}_k)\|^2 \le \varepsilon^2$, we have
$$\mathbb{E}\|h_1(\mathbf{y}_{k'}) - h_2(\mathbf{y}_{k'})\|^2 \le 2\max\{\varepsilon^2, \Psi_\star L^2/n_{\mathrm{id}}\}.$$

3. The controllability matrices satisfy the following bounds:
$$1 \wedge \sigma_{\min}(\mathcal{C}_k^\top \Sigma_{k',\mathrm{id}}^{-1/2}) \ge \sigma_{d_\mathbf{x}}(\mathcal{C}_k)\sqrt{\frac{5(1-\gamma_\star)}{11\Psi_\star^2\alpha_\star^2}}, \tag{G.6}$$

and
$$1 \wedge \sigma_{\min}(\mathcal{C}_k^\top \Sigma_{k',\mathrm{id}}^{-1/2}) \cdot \sigma_{\min}(\Sigma_{k',\mathrm{id}}^{-1/2}) \ge \frac{5(1-\gamma_\star)\sigma_{d_\mathbf{x}}(\mathcal{C}_k)}{11\Psi_\star^2\alpha_\star^2}. \tag{G.7}$$

4. $\left\|\mathcal{C}_k^\top \Sigma_{k',\mathrm{id}}^{-1}\right\|_{\mathrm{op}} \le \sqrt{\Psi_\star}$, provided $k \ge k'$ (but in fact, not requiring $k \ge \kappa_0$).

*Proof.* Our proof starts with the following claim, which shows that the covariance matrices for $k$ and $k'$ are very close under the conditions of the lemma.

**Claim G.1.** Fix $\epsilon \in (0, 1/2)$. For all $k, k' \ge \frac{1}{1-\gamma_\star} \ln \frac{6\epsilon^{-1}\Psi_\star^2\alpha_\star^2}{1-\gamma_\star}$, we have that
$$\max\{\|I - \Sigma_{k,\mathrm{id}}^{-1/2}\Sigma_{k',\mathrm{id}}\Sigma_{k,\mathrm{id}}^{-1/2}\|_{\mathrm{op}}, \|I - \Sigma_{k,\mathrm{id}}^{1/2}\Sigma_{k',\mathrm{id}}^{-1}\Sigma_{k,\mathrm{id}}^{1/2}\|_{\mathrm{op}}\} \le \epsilon.$$

*Proof of Claim G.1.* Since $\Sigma_{k,\mathrm{id}} \ge \Sigma_w \ge \Psi_\star^{-1}I$ by definition, we have that
$$\begin{aligned}
\|I - \Sigma_{k,\mathrm{id}}^{-1/2}\Sigma_{k',\mathrm{id}}\Sigma_{k,\mathrm{id}}^{-1/2}\|_{\mathrm{op}} &= \|\Sigma_{k,\mathrm{id}}^{-1/2}(\Sigma_{k,\mathrm{id}} - \Sigma_{k',\mathrm{id}})\Sigma_{k,\mathrm{id}}^{-1/2}\| \\
&\le \|\Sigma_{k,\mathrm{id}}^{-1}\|\|\Sigma_{k',\mathrm{id}} - \Sigma_{k,\mathrm{id}}\| \le \Psi_\star\|\Sigma_{k',\mathrm{id}} - \Sigma_{k,\mathrm{id}}\|.
\end{aligned}$$
By the same token,
$$\begin{aligned}
\|I - \Sigma_{k,\mathrm{id}}^{1/2}\Sigma_{k',\mathrm{id}}^{-1}\Sigma_{k,\mathrm{id}}^{1/2}\|_{\mathrm{op}} &\le \epsilon \\
\iff (1-\epsilon)I &\le \Sigma_{k,\mathrm{id}}^{1/2}\Sigma_{k,\mathrm{id}}^{-1}\Sigma_{k,\mathrm{id}}^{1/2} \le (1+\epsilon)I \\
\iff (1-\epsilon)\Sigma_{k,\mathrm{id}}^{-1} &\le \Sigma_{k',\mathrm{id}}^{-1} \le (1+\epsilon)\Sigma_{k,\mathrm{id}}^{-1} &\text{(conjugation)} \\
\iff (1+\epsilon)^{-1}\Sigma_{k,\mathrm{id}} &\le \Sigma_{k',\mathrm{id}} \le (1-\epsilon)^{-1}\Sigma_{k,\mathrm{id}} &\text{(inversion)} \\
\iff (1+\epsilon)^{-1}I &\le \Sigma_{k,\mathrm{id}}^{-1/2}\Sigma_{k',\mathrm{id}}\Sigma_{k,\mathrm{id}}^{-1/2} \le (1+\epsilon)^{-1}\Sigma_{k,\mathrm{id}}^{-1/2} \\
\iff \|I - \Sigma_{k,\mathrm{id}}^{-1/2}\Sigma_{k',\mathrm{id}}\Sigma_{k,\mathrm{id}}^{-1/2}\|_{\mathrm{op}} &\le \max\{1 - (1+\epsilon)^{-1}, (1-\epsilon)^{-1} - 1\}.
\end{aligned}$$
In particular, for $\epsilon \le 1/2$, $\|I - \Sigma_{k,\mathrm{id}}^{1/2}\Sigma_{k',\mathrm{id}}^{-1}\Sigma_{k,\mathrm{id}}^{1/2}\|_{\mathrm{op}} \le \epsilon$ as long as $\|I - \Sigma_{k,\mathrm{id}}^{-1/2}\Sigma_{k',\mathrm{id}}\Sigma_{k,\mathrm{id}}^{-1/2}\|_{\mathrm{op}} \le 2\epsilon$. Combining with the above,
$$\max\{\|I - \Sigma_{k,\mathrm{id}}^{-1/2}\Sigma_{k',\mathrm{id}}\Sigma_{k,\mathrm{id}}^{-1/2}\|_{\mathrm{op}}, \|I - \Sigma_{k,\mathrm{id}}^{-1/2}\Sigma_{k',\mathrm{id}}\Sigma_{k,\mathrm{id}}^{-1/2}\|_{\mathrm{op}}\} \le \epsilon \tag{G.8}$$
$$\text{if } \|\Sigma_{k,\mathrm{id}} - \Sigma_{k',\mathrm{id}}\| \le \frac{\epsilon}{2\Psi_\star} \le \epsilon \le 1/2.$$

Next, for any $k, k'$, using strong stability implies
$$\begin{aligned}
\|\Sigma_{k,\mathrm{id}} - \Sigma_{k',\mathrm{id}}\|_{\mathrm{op}} &\le \|A^k(\Sigma_0)(A^k)^\top - A^{k'}(\Sigma_0)(A^{k'})^\top\|_{\mathrm{op}} + \left\|\sum_{i=\min\{k,k'\}+1}^{\max\{k,k'\}} (A^i)\Sigma_w(A^i)^\top\right\|_{\mathrm{op}} \\
&\le \Psi_\star\alpha_\star^2\left(2\gamma_\star^{2\min\{k,k'\}} + \sum_{i=\min\{k,k'\}+1}^{\max\{k,k'\}} \gamma_\star^{2s}\right) \\
&\le \frac{3\Psi_\star\alpha_\star^2\gamma_\star^{2\min\{k,k'\}}}{1-\gamma_\star}.
\end{aligned}$$
Hence, for a given $\epsilon > 0$, we have
$$\|\Sigma_{k,\mathrm{id}} - \Sigma_{k',\mathrm{id}}\|_{\mathrm{op}} \le \epsilon \text{ for } \min\{k, k'\} \ge \frac{1}{1-\gamma_\star} \ln \frac{3\epsilon^{-1}\Psi_\star\alpha_\star^2}{1-\gamma_\star}.$$
The bound now follows by combining with Eq. (G.8), and shrinking $\epsilon$ by a factor of 2. $\qquad\square$

Next, we require a basic operator norm bound for $\Sigma_{\infty,\mathrm{id}}$.

**Claim G.2.** $\|\Sigma_{\infty,\mathrm{id}}\|_{\mathrm{op}} \le 2\Psi_\star^2 \alpha_\star^2 (1-\gamma_\star)^{-1}$.

*Proof of Claim G.2.* $\|\Sigma_{\infty,\mathrm{id}}\|_{\mathrm{op}} = \|\sum_{i=0}^\infty (A^i)(\Sigma_w + BB^\top)(A^i)^\top\|_{\mathrm{op}} \le (\|\Sigma_w\|_{\mathrm{op}} + \|B\|_{\mathrm{op}}^2)\sum_{i=0}^\infty \|A^i\|_{\mathrm{op}}^2$. We can bound $(\|\Sigma_w\|_{\mathrm{op}} + \|B\|_{\mathrm{op}}^2) \le 2\Psi_\star^2$ and $\|A^i\|_{\mathrm{op}}^2 \le \alpha_\star^2 \gamma_\star^{2i}$, so that $\|\Sigma_{\infty,\mathrm{id}}\|_{\mathrm{op}} \le 2\Psi_\star^2\alpha_\star^2 \sum_{i\ge 0}\gamma_\star^{2i} \le \Psi_\star^2\alpha_\star^2(1-\gamma_\star)^{-1}$. $\qquad\square$

We now proceed with the proof of the lemma. We prove points 1 through 4 in order.

1. We have that $\frac{9}{10}\Sigma_{\infty,\mathrm{id}} \preceq \Sigma_{k,\mathrm{id}} \preceq \frac{11}{10}\Sigma_{\infty,\mathrm{id}}$ if and only $\|I - \Sigma_{\infty,\mathrm{id}}^{-1/2}\Sigma_{k,\mathrm{id}}\Sigma_{\infty,\mathrm{id}}^{-1/2}\| \le 1/10$. Hence, the bounds hold by selecting $k \leftarrow \infty$, $k' \leftarrow k$, and invoking Claim G.1 for our choice of $\kappa_0$. Moreover, by Claim G.2, $\Sigma_{\infty,\mathrm{id}} \preceq 2I\Psi_\star^2\alpha_\star^2(1-\gamma_\star)^{-1}$, yielding the last inequality.

2. For point 2, every $h \in \mathscr{H}_{\mathrm{id}}$ satisfies $\|h(y)\| \le L\sqrt{\Psi_\star}\max\{1, \|f_\star(y)\|\}$; see the definition of the class $\mathscr{H}_{\mathrm{id}}$ in Eq. (G.5). Hence, given two elements $h, h' \in \mathscr{H}_{\mathrm{id}}$ with $\mathbb{E}_{\mathbf{y}_k \sim \mathcal{N}(0,\Sigma_{k,\mathrm{id}})}\|h(\mathbf{y}_k) - h'(\mathbf{y}_k)\| \le \varepsilon^2$, Lemma E.3 ensures that

$$\mathbb{E}_{\mathbf{y}_{k'} \sim \mathcal{N}(0,\Sigma_{k',\mathrm{id}})}\|h(\mathbf{y}_{k'}) - h'(\mathbf{y}_{k'})\| \le 2\max\{\varepsilon^2, \Psi_\star L^2/n_{\mathrm{id}}\},$$

provided that $\|I - \Sigma_{k,\mathrm{id}}^{1/2}\Sigma_{k',\mathrm{id}}^{-1}\Sigma_{k,\mathrm{id}}^{1/2}\|_{\mathrm{op}} \le \dfrac{1}{14d_{\mathbf{x}}\ln(80en_{\mathrm{id}}(1 + \|\Sigma_{k,\mathrm{id}}\|_{\mathrm{op}}))}$.

Using that $\Psi_\star, \alpha_\star \ge 1$ and the previous bound, $(1 + \|\Sigma_{k,\mathrm{id}}\|_{\mathrm{op}}) \le \frac{22}{5}\Psi_\star^2\alpha_\star^2(1-\gamma_\star)^{-1}$, we have that as long as

$$\|I - \Sigma_{k,\mathrm{id}}^{1/2}\Sigma_{k',\mathrm{id}}^{-1}\Sigma_{k,\mathrm{id}}^{1/2}\|_{\mathrm{op}} \le \frac{1}{14d_{\mathbf{x}}\ln(16\cdot 22en_{\mathrm{id}}\Psi_\star^2\alpha_\star^2(1-\gamma_\star)^{-1})}, \tag{G.9}$$

we obtain the desired inequality: $\mathbb{E}_{\mathbf{y}_{k'} \sim \mathcal{N}(0,\Sigma_{k',\mathrm{id}})}\|h(\mathbf{y}_{k'}) - h'(\mathbf{y}_{k'})\| \le 2\max\{\varepsilon^2, \Psi_\star L^2/n_{\mathrm{id}}\}$. Finally to obtain the guarantee in Eq. (G.9), we require

$$\min\{k, k'\} \ge \frac{1}{1-\gamma_\star}\ln\left(\frac{6\cdot 14\Psi_\star^3\alpha_\star d_{\mathbf{x}}\ln(16\cdot 22en_{\mathrm{id}}\Psi_\star^2\alpha_\star^2(1-\gamma_\star)^{-1})}{1-\gamma_\star}\right).$$

Simplifying constants, a sufficient condition is that

$$\min\{k, k'\} \ge \frac{1}{1-\gamma_\star}\ln\left(\frac{84\Psi_\star^3\alpha_\star^2 d_{\mathbf{x}}\ln(1000n_{\mathrm{id}}\Psi_\star^2\alpha_\star^2(1-\gamma_\star)^{-1})}{1-\gamma_\star}\right).$$

Finally, since $\ln(xy) = \ln(x) + \ln(y) \le y\ln(x)$ for $x \ge e$ and $y \ge 1$, we can further simplify to the sufficient condition

$$\min\{k, k'\} \ge \frac{1}{1-\gamma_\star}\ln\left(\frac{84\Psi_\star^5\alpha_\star^4 d_{\mathbf{x}}\ln(1000n_{\mathrm{id}})}{(1-\gamma_\star)^2}\right) := \kappa_0.$$

which is precisely the condition in Eq. (G.3).

3. For the third point, we start with 1

$$\lambda_{\min}(\Sigma_{k',\mathrm{id}}^{-1/2}) = \sqrt{\frac{1}{\lambda_{\max}(\Sigma_{k',\mathrm{id}})}} \ge \sigma_{d_{\mathbf{x}}}(\mathcal{C}_k)\sqrt{\frac{5(1-\gamma)}{11\Psi_\star^2\alpha_\star^2}} \tag{G.10}$$

To prove the first point of Eq. (G.7), we bound

$$\sigma_{d_{\mathbf{x}}}(\mathcal{C}_k^\top\Sigma_{k',\mathrm{id}}^{-1/2}) \ge \sigma_{d_{\mathbf{x}}}(\mathcal{C}_k)\lambda_{\min}(\Sigma_{k',\mathrm{id}}^{-1/2}) \ge \sigma_{d_{\mathbf{x}}}(\mathcal{C}_k)\sqrt{\frac{5(1-\gamma)}{11\Psi_\star^2\alpha_\star^2}}, \tag{G.11}$$

where we use the first point of the lemma in the last step (Eq. (G.10)). To see that this lower bound (the RHS of Eq. (G.11)) is less than 1 (acounting for the $\wedge 1$ in the LHS of Eq. (G.7)), we observe $\Sigma_{k',\mathrm{id}} \succeq \mathcal{C}_k\mathcal{C}_k^\top$ for $k' \ge k$, and thus $\sigma_{d_{\mathbf{x}}}(\mathcal{C}_k^\top\Sigma_{k',\mathrm{id}}^{-1/2}) \le 1$.

Proving the second part of Eq. (G.7) follows by combining Eqs. (G.10) and (G.11). The resultant lower bound is also less than 1, since the (RHS of Eq. (G.10)) $\le 1$, and $\sigma_{d_{\mathbf{x}}}(\mathcal{C}_k^\top\Sigma_{k',\mathrm{id}}^{-1/2}) \le 1$ as well.

4. Finally, $\|\mathcal{C}_k^\top \Sigma_{k',\mathrm{id}}^{-1}\|_{\mathrm{op}} \le \|\mathcal{C}_k^\top (\Sigma_{k',\mathrm{id}}^{-1/2})\|_{\mathrm{op}} \|\Sigma_{k',\mathrm{id}}^{-1/2}\|_{\mathrm{op}}$. Since $\Sigma_{k',\mathrm{id}} \succeq \mathcal{C}_k \mathcal{C}_k^\top$ for $k' \ge k$, $\|\mathcal{C}_k^\top (\Sigma_{k',\mathrm{id}}^{-1/2})\|_{\mathrm{op}} \le 1$. Moreover, since $\Sigma_{k',\mathrm{id}} \succeq \Sigma_w \succeq \Psi_\star^{-1} I$, $\|\Sigma_{k',\mathrm{id}}^{-1/2}\|_{\mathrm{op}} \le \sqrt{\Psi_\star}$, as needed.

$\square$

### G.1.3 Properties of the Class $\mathscr{H}_{\mathrm{id}}$

**Lemma G.2.** Let $\kappa_0$ satisfy Eq. (G.3), and define

$$c_{\mathrm{conc,id}} := \frac{12 L^2 \Psi_\star^3 \alpha_\star^2}{1 - \gamma_\star}. \tag{G.12}$$

Then, for all $k \ge \kappa_0$:

1. $\max_{h \in \mathscr{H}_{\mathrm{id}}} \|h(\mathbf{y}_k)\|^2 \le L^2 \Psi_\star \max\{1, \|f_\star(\mathbf{y}_k)\|^2\}$, and both are $(d_\mathbf{x} \cdot c_{\mathrm{conc,id}})$-concentrated.

2. For any matrix $V$ with $\|V\|_{\mathrm{op}} \le 1$ (e.g., any $V$ with orthonormal columns) and any $h, h' \in \mathscr{H}_{\mathrm{id}}$, the random variable $\|V^\top h(\mathbf{y}_k)\|^2$ is $(d_\mathbf{x} c_{\mathrm{conc,id}})$ concentrated, and $\|V^\top h(\mathbf{y}_k) - V^\top h'(\mathbf{y}_k)\|^2$ is $(4 d_\mathbf{x} c_{\mathrm{conc,id}})$-concentrated.

*Proof.* Let us first reason about the concentration of $\|f_\star(\mathbf{y}_k)\|^2$. Under perfect decodability, $\|f_\star(\mathbf{y}_k)\|^2 = \|\mathbf{x}_k\|^2$, which is $5 \mathrm{tr}(\Sigma_{k,\mathrm{id}}) \le 5 d_\mathbf{x} \|\Sigma_{k,\mathrm{id}}\|_{\mathrm{op}}$-concentrated by Lemma E.2. Moreover, from Lemma G.1, we have that $5 d_\mathbf{x} \|\Sigma_{k,\mathrm{id}}\|_{\mathrm{op}} \le 11 \Psi_\star^2 \alpha_\star^2 (1 - \gamma_\star)^{-1}$.

To finish proving the first point, observe that that $\max_{h \in \mathscr{H}_{\mathrm{id}}} \|h(y)\|^2 \le L^2 \Psi_\star \max\{1, \|f_\star(y)\|^2\}$ (Eq. (G.5)). From Lemma E.1, we recall that if a random variable $z$ is $c$-concentrated, then $\alpha(z + \beta)$ is $\alpha(c + \beta)$ concentrated for $\beta, \alpha > 0$. Hence, $\max_{h \in \mathscr{H}_{\mathrm{id}}} \|h(y)\|^2$ is $L^2 \Psi_\star (1 + 11 d_\mathbf{x} \Psi_\star^2 \alpha_\star^2 (1 - \gamma_\star)^{-1}) \le d_\mathbf{x} \cdot 12 L^2 \Psi_\star^3 \alpha_\star^2 (1 - \gamma_\star)^{-1} = d_\mathbf{x} c_{\mathrm{conc,id}}$-concentrated, as needed.

The proof of the second point is analogous. First, we note that for $\|V\|_{\mathrm{op}} \le 1$, $\|V^\top h(\mathbf{y}_k)\|^2 \le \|h(\mathbf{y}_k)\|^2$ and $\|V^\top (h(\mathbf{y}_k) - h'(\mathbf{y}_k))\|^2 \le 4 L^2 \Psi_\star \max\{1, \|f_\star(y)\|^2\}$. Combined with the concentration result for $\|f_\star(\mathbf{y}_k)\|^2$ above, this yields the result. $\square$

**Lemma G.3.** Let $\kappa_0$ satisfy Eq. (G.3), let $\kappa \in \mathbb{N}$, and define $\kappa_1 := \kappa_0 + \kappa$. Then for all $x \in \mathbb{R}^{d_\mathbf{x}}$ and $y \in \mathrm{supp}\, q(\cdot \mid x)$ we have:

$$\mathbb{E}[\mathbf{v} \mid \mathbf{y}_{\kappa_1} = y] = \mathcal{C}_\kappa^\top \Sigma_{\kappa_1,\mathrm{id}}^{-1} x =: h_{\star,\mathrm{id}}(y),$$

and $h_{\star,\mathrm{id}} \in \mathscr{H}_{\mathrm{id}}$.

*Proof.* Since $\mathbf{v} \to \mathbf{x}_{\kappa_1} \to \mathbf{y}_{\kappa_1}$ forms a Markov chain, and $\mathbf{x}_{\kappa_1} = f_\star(\mathbf{y}_{\kappa_1})$ almost surely, we have that $(\mathbf{v}, \mathbf{x}_{\kappa_1}, \mathbf{y}_{\kappa_1})$ is *decodable* in the sense of Definition 4. Thus, by Lemma E.4, $\mathbb{E}[\mathbf{v} \mid \mathbf{y}_{\kappa_1} = y] = \mathbb{E}[\mathbf{v} \mid \mathbf{x}_{\kappa_1} = f_\star(y)]$. By Fact G.1, $(\mathbf{v}, \mathbf{x}_{\kappa_1})$ are jointly Gaussian and mean zero, and $\mathbb{E}[\mathbf{x}_{\kappa_1} \mathbf{x}_{\kappa_1}^\top] = \Sigma_{\kappa_1,\mathrm{id}}$ and $\mathbb{E}[\mathbf{v} \mathbf{x}_{\kappa_1}^\top] = \mathcal{C}_\kappa^\top$. Thus, $\mathbb{E}[\mathbf{v} \mid \mathbf{x}_{\kappa_1} = x] = \mathcal{C}_\kappa^\top \Sigma_{\kappa_1,\mathrm{id}}^{-1} x$ (Fact G.2), giving that $\mathbb{E}[\mathbf{v} \mid \mathbf{y}_{\kappa_1} = y] = \mathcal{C}_\kappa^\top \Sigma_{\kappa_1,\mathrm{id}}^{-1} f_\star(y) = h_{\star,\mathrm{id}}(y)$, as needed.

To see that $h_{\star,\mathrm{id}} \in \mathscr{H}_{\mathrm{id}}$, we observe that $h_{\star,\mathrm{id}} = M f_\star$ for $M = \mathcal{C}_\kappa^\top \Sigma_{\kappa_1,\mathrm{id}}^{-1}$. By Lemma G.1 part 4, we have $\|M\|_{\mathrm{op}} \le \Psi_\star^{1/2}$. Thus, from the definition of $\mathscr{H}_{\mathrm{id}}$ in Eq. (G.5) and the fact that $f_\star \in \mathscr{F}$ by the realizability assumption, we conclude that $h_{\star,\mathrm{id}} \in \mathscr{H}_{\mathrm{id}}$. $\square$

### G.2 Proof of Decoder Recovery (Theorem 2)

We first state the full version of Theorem 2, which asserts that Phase I recovers a decoder that accurately predicts the state under Gaussian roll-in, up to a well-conditioned similarity transformation.

**Theorem 2a.** *For a universal constant $\bar{c}_{\mathrm{id},1} \ge 8$, define*

$$\varepsilon_{\mathrm{id},h}^2 = \bar{c}_{\mathrm{id},1} \frac{\ln^2\left(\frac{n_{\mathrm{id}}}{\delta}\right)(d_\mathbf{u} \kappa + d_\mathbf{x} c_{\mathrm{conc,id}})(\ln|\mathscr{F}| + d_\mathbf{u} d_\mathbf{x} \kappa)}{n_{\mathrm{id}}}, \tag{G.13}$$

*and assume that $n_{\mathrm{id}}$ is sufficiently large such that*

$$\varepsilon_{\mathrm{id},h}\sqrt{d_{\mathbf{x}}c_{\mathrm{conc,id}}} \le \frac{(1-\gamma_\star)\sigma_{d_{\mathbf{x}}}(\mathcal{C}_\kappa)^2}{71\alpha_\star^2\Psi_\star^2}.$$

*Then, with probability at least $1 - 3\delta$, there exists an invertible matrix $S_{\mathrm{id}} \in \mathbb{R}^{d_{\mathbf{x}}^2}$ satisfying*

$$1 \wedge \sigma_{\min}(S_{\mathrm{id}}) \ge \sigma_{\min,\mathrm{id}} := \frac{\sigma_{\min}(\mathcal{C}_\kappa)(1-\gamma_\star)}{4\Psi_\star^2\alpha_\star^2}, \quad \text{and} \quad 1 \vee \|S_{\mathrm{id}}\|_{\mathrm{op}} \le \sigma_{\max,\mathrm{id}} := \sqrt{\Psi_\star},$$

*such that the function $f_{\star,\mathrm{id}}(y) := S_{\mathrm{id}}f_\star(y)$ and the learned decoder $\hat{f}_{\mathrm{id}}$ satisfy*

$$\mathbb{E}\|f_{\star,\mathrm{id}}(\mathbf{y}_{\kappa_1}) - \hat{f}_{\mathrm{id}}(\mathbf{y}_{\kappa_1})\|^2 \le \varepsilon_{\mathrm{id},h}^2.$$

*In particular, for*

$$n_{\mathrm{id}} = \Omega_\star(d_{\mathbf{x}}d_{\mathbf{u}}\kappa(\ln|\mathscr{F}| + d_{\mathbf{u}}d_{\mathbf{x}}\kappa)),$$

*we have that*

$$\mathbb{E}\|f_{\star,\mathrm{id}}(\mathbf{y}_{\kappa_1}) - \hat{f}_{\mathrm{id}}(\mathbf{y}_{\kappa_1})\|^2 \le \mathcal{O}_\star\left(\frac{d_{\mathbf{u}}\kappa(\ln|\mathscr{F}| + d_{\mathbf{u}}d_{\mathbf{x}}\kappa)\ln^2(\frac{n_{\mathrm{id}}}{\delta})}{n_{\mathrm{id}}}\right).$$

*Proof.* The proof of this theorem follows from two propositions which we establish in the sequel. The first, Proposition G.1, demonstrates that the learned function $\hat{h}_{\mathrm{id}}$ satisfies the following bound with probability $1 - 3\delta/2$:

$$\mathbb{E}_{\mathbf{y}_{\kappa_1}}[\|\hat{h}_{\mathrm{id}}(\mathbf{y}_{\kappa_1}) - h_{\star,\mathrm{id}}(\mathbf{y}_{\kappa_1})\|^2] \le \varepsilon_{\mathrm{id},h}^2,$$

where $h_{\star,\mathrm{id}}(y) := \mathcal{C}_\kappa^\top\Sigma_{\kappa_1,\mathrm{id}}^{-1}f_\star(y)$. Now recall that the function $\hat{f}_{\mathrm{id}}(y)$ is constructed as $\widehat{V}_{\mathrm{id}}^\top h_{\star,\mathrm{id}}(y)$, where $\widehat{V}_{\mathrm{id}}$ has orthonormal columns. Defining $S_{\mathrm{id}} = \widehat{V}_{\mathrm{id}}^\top\mathcal{C}_\kappa^\top\Sigma_{\kappa_1,\mathrm{id}}^{-1}$, we see that $\widehat{V}_{\mathrm{id}}^\top h_{\star,\mathrm{id}}(y) = S_{\mathrm{id}}f_\star(y) = f_{\star,\mathrm{id}}(y)$. Thus, since $\widehat{V}_{\mathrm{id}}$ has operator norm 1,

$$\mathbb{E}_{\mathbf{y}_{\kappa_1}}[\|\hat{f}_{\mathrm{id}}(\mathbf{y}_{\kappa_1}) - f_{\star,\mathrm{id}}(\mathbf{y}_{\kappa_1})\|^2] = \mathbb{E}_{\mathbf{y}_{\kappa_1}}[\|\widehat{V}_{\mathrm{id}}^\top(\hat{h}_{\mathrm{id}}(\mathbf{y}_{\kappa_1}) - h_{\star,\mathrm{id}}(\mathbf{y}_{\kappa_1}))\|^2]$$
$$\le \mathbb{E}_{\mathbf{y}_{\kappa_1}}[\|\hat{h}_{\mathrm{id}}(\mathbf{y}_{\kappa_1}) - h_{\star,\mathrm{id}}(\mathbf{y}_{\kappa_1})\|^2] \le \varepsilon_{\mathrm{id},h}^2.$$

To conclude, the norm bounds for the matrix $S_{\mathrm{id}}$ are provided by Proposition G.2, which hold with probability at least $1 - \delta$. $\square$

### G.2.1  Prediction Error Guarantee for $\hat{h}_{\mathrm{id}}$

**Proposition G.1.** Let $(\mathbf{y}_{\kappa_1}^{(i)}, \mathbf{v}^{(i)})_{i=1}^{n_{\mathrm{id}}}$ be as described in Algorithm 3 and Section 2.1, and let

$$\hat{h}_{\mathrm{id}} \in \underset{h \in \mathscr{H}_{\mathrm{id}}}{\arg\min} \sum_{i=1}^{n_{\mathrm{id}}} \|h(\mathbf{y}_{\kappa_1}^{(i)}) - \mathbf{v}^{(i)}\|^2.$$

Then there is a universal constant $\bar{c}_{\mathrm{id},1} \ge 8$ such that with probability at least $1 - \frac{3}{2}\delta$, we have

$$\mathbb{E}_{\mathbf{y}_{\kappa_1}}[\|\hat{h}_{\mathrm{id}}(\mathbf{y}_{\kappa_1}) - h_{\star,\mathrm{id}}(\mathbf{y}_{\kappa_1})\|^2] \le \varepsilon_{\mathrm{id},h}^2 \le \mathcal{O}_\star\left(\frac{\ln^2(\frac{n_{\mathrm{id}}}{\delta})d_{\mathbf{u}}\kappa(\ln|\mathscr{F}| + d_{\mathbf{u}}d_{\mathbf{x}}\kappa)}{n_{\mathrm{id}}}\right).$$

We let $\mathcal{E}_{\mathrm{id},h}$ denote the event that this inequality holds.

*Proof.* From Lemma G.3, we have $\mathbb{E}[\mathbf{v} \mid \mathbf{y}_{\kappa_1} = y] = h_{\star,\mathrm{id}}(y)$, so $h_{\star,\mathrm{id}} \in \mathscr{H}_{\mathrm{id}}$. To prove the result, we simply apply our general-purpose error bound for least-square regression, Corollary E.1, with $\mathbf{y} = \mathbf{y}_{\kappa_1}$, $\mathbf{u} = \mathbf{v}$, and $\mathbf{e} = 0$. We verify that each precondition for the proposition holds.

- **Structure of function class.** $\mathscr{F}$ is finite, and by Assumption 5, $\|f(y)\| \le L\max\{1, \|f_\star(y)\|\}$. Moreover, $\mathscr{H}_{\mathrm{id}} := \{M \cdot f : \|M\| \le \sqrt{\Psi_\star}, f \in \mathscr{F}\}$.

- **Concentration Property.** By Lemma G.2, defining $\varphi(y) := L^2 \Psi_\star \max\{1, \|f_\star(y)\|^2\}$ we see that $\varphi(\mathbf{y}_{\kappa_1})$ is $d_\mathbf{x} c_{\text{conc,id}}$-concentrated. Moreover, since $\mathbf{v} \sim \mathcal{N}(0, I_{d_\mathbf{u}\kappa})$, we have that $\|\mathbf{v}\|^2$ is $5d_\mathbf{u}\kappa$-concentrated by Lemma E.2. Hence, $c = 5d_\mathbf{u}\kappa + d_\mathbf{x} c_{\text{conc,id}}$ is a valid choice for the concentration constant $c$ in Proposition E.1.

Thus, Corollary E.1 with $c \lesssim d_\mathbf{u}\kappa + d_\mathbf{x} c_{\text{conc,id}}$ implies that

$$
\mathbb{E}_{\mathbf{y}_{\kappa_1}}[\|\hat{h}_{\text{id}}(\mathbf{y}_{\kappa_1}) - h_{\star,\text{id}}(\mathbf{y}_{\kappa_1})\|^2] \lesssim \frac{\ln^2(\frac{n_{\text{id}}}{\delta})(d_\mathbf{u}\kappa + d_\mathbf{x} c_{\text{conc,id}})(\ln|\mathscr{F}| + d_\mathbf{u} d_\mathbf{x}\kappa)}{n_{\text{id}}}
$$

$$
= \mathcal{O}_\star\left(\frac{\ln^2(\frac{n_{\text{id}}}{\delta})(d_\mathbf{u}\kappa + d_\mathbf{x})(\ln|\mathscr{F}| + d_\mathbf{u} d_\mathbf{x}\kappa)}{n_{\text{id}}}\right)
$$

$$
= \mathcal{O}_\star\left(\frac{\ln^2(\frac{n_{\text{id}}}{\delta})d_\mathbf{u}\kappa(\ln|\mathscr{F}| + d_\mathbf{u} d_\mathbf{x}\kappa)}{n_{\text{id}}}\right),
$$

where the last simplification uses that controllability requires $d_\mathbf{u}\kappa \geq d_\mathbf{x}$. $\qquad\square$

### G.2.2 Dimension Reduction

**Proposition G.2.** Suppose that $\kappa_0$ satisfies Eq. (G.3). Let $\widehat{V}_{\text{id}} \in \mathbb{R}^{\kappa d_\mathbf{u} \times d_\mathbf{x}}$ be an eigenbasis for the top $d_\mathbf{x}$ eigenvalues of $\widehat{\Lambda}_n$, where we define

$$
\widehat{\Lambda}_n := \frac{1}{n}\sum_{i=1}^n \hat{h}_{\text{id}}(\mathbf{y}_{\kappa_1}^{(i)})\hat{h}_{\text{id}}(\mathbf{y}_{\kappa_1}^{(i)})^\top.
$$

Further, let $S_{\text{id}} := \widehat{V}_{\text{id}}^\top \mathcal{C}_\kappa^\top \Sigma_{\kappa_1,\text{id}}^{-1} \in \mathbb{R}^{d_\mathbf{x} \times d_\mathbf{x}}$. Then if

$$
\sqrt{d_\mathbf{x} c_{\text{conc,id}}}\varepsilon_{\text{id},h} \leq \frac{(1-\gamma_\star)\sigma_{d_\mathbf{x}}(\mathcal{C}_\kappa)^2}{71\alpha_\star^2 \Psi_\star^2}, \tag{G.14}
$$

we have that with probability at least $1 - \delta$, an event $\mathcal{E}_{\text{id,pca}}$ occurs such that on $\mathcal{E}_{\text{id},h} \cap \mathcal{E}_{\text{id,pca}}$,

$$
1 \wedge \sigma_{\min}(S_{\text{id}}) \geq \sigma_{\min,\text{id}} := \frac{\sigma_{\min}(\mathcal{C}_\kappa)(1-\gamma_\star)}{4\Psi_\star^2\alpha_\star^2}, \quad \text{and} \quad 1 \vee \|S_{\text{id}}\|_{\text{op}} \leq \sigma_{\max,\text{id}} := \sqrt{\Psi_\star}.
$$

*Proof.* Introduce $\Lambda_\star := \mathcal{C}_\kappa \Sigma_{\kappa_1,\text{id}}^{-1} \mathcal{C}_\kappa^\top$, and let $V_{\text{id}} \in \mathbb{R}^{\kappa d_\mathbf{u} \times d_\mathbf{x}}$ be an eigenbasis for its $d_\mathbf{x}$ non-zero eigenvectors. From Lemma G.3 and Fact G.1, we have

$$
\mathbb{E}[h_{\star,\text{id}}(\mathbf{y}_{\kappa_1})h_{\star,\text{id}}(\mathbf{y}_{\kappa_1})^\top] = \mathcal{C}_\kappa \Sigma_{\kappa_1,\text{id}}^{-1}\mathbb{E}[f_\star(\mathbf{y}_{\kappa_1})f_\star(\mathbf{y}_{\kappa_1})^\top]\Sigma_{\kappa_1,\text{id}}^{-1}\mathcal{C}_\kappa^\top
$$

$$
= \mathcal{C}_\kappa \Sigma_{\kappa_1,\text{id}}^{-1}\mathbb{E}[\mathbf{x}_{\kappa_1}\mathbf{x}_{\kappa_1}^\top]\Sigma_{\kappa_1,\text{id}}^{-1}\mathcal{C}_\kappa^\top
$$

$$
= \mathcal{C}_\kappa \Sigma_{\kappa_1,\text{id}}^{-1}\Sigma_{\kappa_1,\text{id}}\Sigma_{\kappa_1,\text{id}}^{-1}\mathcal{C}_\kappa^\top = \mathcal{C}_\kappa \Sigma_{\kappa_1,\text{id}}^{-1}\mathcal{C}_\kappa^\top := \Lambda_\star.
$$

We apply Proposition E.2 and Corollary E.3, with $\widehat{\Lambda}_n$ and $\Lambda_\star$ as above. To apply this proposition, first observe that $\|\hat{h}_{\text{id}}(\mathbf{y}_{\kappa_1})\|^2$ is $c = d_\mathbf{x} c_{\text{conc,id}}$-concentrated by Lemma G.2. Morever, on $\mathcal{E}_{\text{id},h}$, we have $\mathbb{E}[\|\hat{h}_{\text{id}}(\mathbf{y}_{\kappa_1}) - h_{\star,\text{id}}(\mathbf{y}_{\kappa_1})\|^2 \leq \varepsilon_{\text{id},h}^2$. Thus, the term $\varepsilon_{\text{pca},n,\delta}$ in Proposition E.2, specializes to

$$
\varepsilon_{\text{id,pca}} := 3\sqrt{d_\mathbf{x} c_{\text{conc,id}}}\varepsilon_{\text{id},h} + 5d_\mathbf{x} c_{\text{conc,id}} n_{\text{id}}^{-1/2}\ln(2\kappa d_\mathbf{u} n_{\text{id}}/\delta)^{3/2}.
$$

By the fact that $n_{\text{id}} \geq \kappa d_\mathbf{u}$ (it can be verified that this is required to ensure the upper bound on $\varepsilon_{\text{id},h}$), we can bound

$$
d_\mathbf{x} c_{\text{conc,id}} n_{\text{id}}^{-1/2}\ln(2\kappa d_\mathbf{u} n_{\text{id}}/\delta)^{3/2} \leq \sqrt{d_\mathbf{x} c_{\text{conc,id}}}\varepsilon_{\text{id},h},
$$

where $\varepsilon_{\text{id},h}$ is as in Proposition G.1. Hence, we can bound

$$
\varepsilon_{\text{id,pca}} = 3\sqrt{d_\mathbf{x} c_{\text{conc,id}}}\varepsilon_{\text{id},h} + 5d_\mathbf{x} c_{\text{conc,id}} n_{\text{id}}^{-1/2}\ln(2\kappa d_\mathbf{u} n_{\text{id}}/\delta)^{3/2} \leq 8\sqrt{d_\mathbf{x} c_{\text{conc,id}}}\varepsilon_{\text{id},h}.
$$

Thus, if we denote the event above by $\mathcal{E}_{\text{id,pca}}$, we that have on $\mathcal{E}_{\text{id},h} \cap \mathcal{E}_{\text{id,pca}}$,

$$
\|\widehat{V}_{\text{id}}^\top(\widehat{\Lambda}_n - \Lambda_\star)\widehat{V}_{\text{id}}\|_{\text{op}} = \|\widehat{\Lambda}_n - \Lambda_\star\|_{\text{op}} \leq 8\sqrt{d_\mathbf{x} c_{\text{conc,id}}}\varepsilon_{\text{id},h}.
$$

From the fourth point of Lemma G.1, we have that $\lambda_{\min}(\Lambda_\star) \geq \frac{5(1-\gamma_\star)\sigma_{d_\mathbf{x}}(\mathcal{C}_\kappa)^2}{11\alpha_\star^2\Psi_\star^2}$. Hence, Corollary E.3 ensures that under the $\mathcal{E}_{\mathrm{id},h} \cap \mathcal{E}_{\mathrm{id,pca}}$, we have

$$\sigma_{\min}(\widehat{V}_{\mathrm{id}}^\top V_{\mathrm{id}}) \geq \frac{2}{3}, \tag{G.15}$$

as long as

$$\sqrt{d_\mathbf{x}c_{\mathrm{conc,id}}}\varepsilon_{\mathrm{id},h} \leq \frac{(1-\gamma_\star)\sigma_{d_\mathbf{x}}(\mathcal{C}_\kappa)^2}{71\alpha_\star^2\Psi_\star^2} \leq \frac{1}{4\cdot 8}\cdot\frac{5(1-\gamma_\star)\sigma_{d_\mathbf{x}}(\mathcal{C}_\kappa)^2}{11\alpha_\star^2\Psi_\star^2},$$

which is precisely the condition Eq. (G.14) required by the theorem. To conclude, let us bound the singular values of $S_{\mathrm{id}} := \widehat{V}_{\mathrm{id}}^\top\mathcal{C}_\kappa^\top\Sigma_{\kappa_1,\mathrm{id}}^{-1}$ under the assumption that the bound above holds. Since $\widehat{V}_{\mathrm{id}}$ has orthonormal columns, have that

$$\|S_{\mathrm{id}}\| \leq \|\mathcal{C}_\kappa^\top\Sigma_{\kappa_1,\mathrm{id}}^{-1}\| \leq \sqrt{\Psi_\star}. \tag{Lemma G.1}$$

On the other hand, we can lower bound

$$\sigma_{\min}(S_{\mathrm{id}}) = \sigma_{\min}(\widehat{V}_{\mathrm{id}}^\top\mathcal{C}_\kappa^\top\Sigma_{\kappa_1,\mathrm{id}}^{-1}) \geq \sigma_{\min}(\widehat{V}_{\mathrm{id}}^\top\mathcal{C}_\kappa^\top\Sigma_{\kappa_1,\mathrm{id}}^{-1/2})\sigma_{\min}(\Sigma_{\kappa_1,\mathrm{id}}^{-1/2}), \tag{G.16}$$

where we have used that $\Sigma_{\kappa_1,\mathrm{id}}^{-1/2}$ and $\widehat{V}_{\mathrm{id}}^\top\mathcal{C}_\kappa^\top\Sigma_{\kappa_1,\mathrm{id}}^{-1/2}$ are square. We now prove the following claim, which is also reused in a number of subsequent proofs.

**Claim G.3.** On $\mathcal{E}_{\mathrm{id},h} \cap \mathcal{E}_{\mathrm{id,pca}}$, we have

$$\sigma_{\min}(\widehat{V}_{\mathrm{id}}^\top\mathcal{C}_\kappa^\top\Sigma_{\kappa_1,\mathrm{id}}^{-1/2}) \geq \frac{2}{3}\sigma_{d_\mathbf{x}}(\mathcal{C}_\kappa^\top\Sigma_{\kappa_1,\mathrm{id}}^{-1/2}).$$

*Proof of Claim G.3.* Since $V_{\mathrm{id}}$ is an eigenbasis for $\mathcal{C}_\kappa^\top\Sigma_{\kappa_1,\mathrm{id}}^{-1}\mathcal{C}_\kappa$, we have that $\mathcal{C}_\kappa^\top\Sigma_{\kappa_1,\mathrm{id}}^{-1/2} = V_{\mathrm{id}}V_{\mathrm{id}}^\top\mathcal{C}_\kappa^\top\Sigma_{\kappa_1,\mathrm{id}}^{-1/2}$, and $\sigma_{d_\mathbf{x}}(V_{\mathrm{id}}^\top\mathcal{C}_\kappa^\top\Sigma_{\kappa_1,\mathrm{id}}^{-1/2}) = \sigma_{d_\mathbf{x}}(\mathcal{C}_\kappa^\top\Sigma_{\kappa_1,\mathrm{id}}^{-1/2})$. Thus,

$$\begin{aligned}
\sigma_{\min}(\widehat{V}_{\mathrm{id}}^\top\mathcal{C}_\kappa^\top\Sigma_{\kappa_1,\mathrm{id}}^{-1/2}) &= \sigma_{\min}(\widehat{V}_{\mathrm{id}}^\top V_{\mathrm{id}}V_{\mathrm{id}}^\top\mathcal{C}_\kappa^\top\Sigma_{\kappa_1,\mathrm{id}}^{-1/2}) \\
&\geq \sigma_{\min}(\widehat{V}_{\mathrm{id}}^\top V_{\mathrm{id}})\sigma_{\min}(V_{\mathrm{id}}^\top\mathcal{C}_\kappa^\top\Sigma_{\kappa_1,\mathrm{id}}^{-1/2}) \\
&\geq \frac{2}{3}\sigma_{\min}(V_{\mathrm{id}}^\top\mathcal{C}_\kappa^\top\Sigma_{\kappa_1,\mathrm{id}}^{-1/2}), \tag{by Eq. (G.15)}
\end{aligned}$$

where the first inequality uses that $\sigma_{\min}(XY) \geq \sigma_{\min}(X)\sigma_{\min}(Y)$ for $X, Y \in \mathbb{R}^{d_\mathbf{x}\times d_\mathbf{x}}$. Again, since $V_{\mathrm{id}}$ is an eigenbasis for the non-zero eigenvalues of $\mathcal{C}_\kappa^\top\Sigma_{\kappa_1,\mathrm{id}}^{-1/2}$, we have

$$\sigma_{\min}(V_{\mathrm{id}}^\top\mathcal{C}_\kappa^\top\Sigma_{\kappa_1,\mathrm{id}}^{-1/2}) = \sigma_{d_\mathbf{x}}(\mathcal{C}_\kappa^\top\Sigma_{\kappa_1,\mathrm{id}}^{-1/2}).$$

$\square$

Combining the above claim with Eq. (G.16), we have

$$\begin{aligned}
\sigma_{\min}(S_{\mathrm{id}}) &\geq \frac{2}{3}\sigma_{\min}(\mathcal{C}_\kappa^\top\Sigma_{\kappa_1,\mathrm{id}}^{-1/2})\sigma_{\min}(\Sigma_{\kappa_1,\mathrm{id}}^{-1/2}) \\
&\geq \frac{5\cdot 2\sigma_{\min}(\mathcal{C}_\kappa)(1-\gamma_\star)}{11\cdot 3\Psi_\star^2\alpha_\star^2} \tag{Eq. (G.7) in Lemma G.1} \\
&\geq \frac{\sigma_{\min}(\mathcal{C}_\kappa)(1-\gamma_\star)}{4\Psi_\star^2\alpha_\star^2},
\end{aligned}$$

as needed. $\square$

## G.3 Estimating Costs and Dynamics (Theorem 3)

To begin this section, we state the full version of Theorem 3, which shows that Phase II (Algorithm 4) accurately recovers the system matrices, noise covariance, and state cost up to a similarity transformation.

**Theorem 3a.** *For a possibly inflated numerical constant $\bar{c}_{\mathrm{id},1}$ in Eq. (G.13), suppose $\varepsilon_{\mathrm{id},h}$ satisfies*

$$\varepsilon_{\mathrm{id},h}\sqrt{\ln(2n_{\mathrm{id}}/\delta)} \le \frac{\sigma_{d_{\mathbf{x}}}(\mathcal{C}_\kappa)^2(1-\gamma_\star)^2}{80 \cdot 12 L^2 \Psi_\star^5 \alpha_\star^4 d_{\mathbf{x}}}.$$

*Then, with probability at least $1 - 11\delta$ over both Phase I and Phase II, the following bounds hold:*

$$\|\widehat{Q}_{\mathrm{id}} - Q_{\mathrm{id}}\|_{\mathrm{op}} \lesssim \varepsilon_{\mathrm{id},h} \cdot \frac{\alpha_\star^6 \Psi_\star^7 \sqrt{c_{\mathrm{conc,id}} d_{\mathbf{x}} \ln(n_{\mathrm{id}}/\delta)}}{(1-\gamma_\star)^3 \sigma_{\min}(\mathcal{C}_\kappa)^4},$$

$$\|[\widehat{A}_{\mathrm{id}}; \widehat{B}_{\mathrm{id}}] - [A_{\mathrm{id}}; B_{\mathrm{id}}]\|_{\mathrm{op}} \lesssim \varepsilon_{\mathrm{id},h} \cdot \frac{\Psi_\star^{7/2} \alpha_\star^3}{\sigma_{\min}(\mathcal{C}_\kappa)^2 (1-\gamma_\star)^{3/2}},$$

$$\|\widehat{\Sigma}_{w,\mathrm{id}} - \Sigma_{w,\mathrm{id}}\|_{\mathrm{op}} \lesssim \varepsilon_{\mathrm{id},h} \cdot \frac{\Psi_\star^{5/2} \alpha_\star^4}{\sigma_{\min}(\mathcal{C}_\kappa)(1-\gamma_\star)}.$$

*In particular, for*

$$n_{\mathrm{id}} = \Omega_\star\big(d_{\mathbf{x}}^2 d_{\mathbf{u}} \kappa(\ln|\mathscr{F}| + d_{\mathbf{u}} d_{\mathbf{x}} \kappa) \max\{1, \sigma_{\min}(\mathcal{C}_\kappa)^{-4}\}\big),$$

*we have that*

$$\|\widehat{Q}_{\mathrm{id}} - Q_{\mathrm{id}}\|_{\mathrm{op}} \vee \|[\widehat{A}_{\mathrm{id}}; \widehat{B}_{\mathrm{id}}] - [A_{\mathrm{id}}; B_{\mathrm{id}}]\|_{\mathrm{op}} \vee \|\widehat{\Sigma}_{w,\mathrm{id}} - \Sigma_{w,\mathrm{id}}\|_{\mathrm{op}}$$
$$\le \mathcal{O}_\star\Big(n_{\mathrm{id}}^{-1/2} \cdot \sqrt{d_{\mathbf{x}} d_{\mathbf{u}} \kappa(\ln|\mathscr{F}| + d_{\mathbf{u}} d_{\mathbf{x}} \kappa) \ln(n_{\mathrm{id}}/\delta)^4}\Big)$$

### G.3.1 Preliminaries for Theorem 3

Before proceeding with the proof of Theorem 3, we recall some notation. First, following Appendix G.3, we let

$$\mathcal{E}_{\mathrm{id},h} \quad \text{and} \quad \mathcal{E}_{\mathrm{id,pca}}$$

denote the events from Proposition G.1 and Proposition G.2, respectively. Next, we introduce some functions used throughout the proof and prove some basic facts about them.

**Definition 6.** *Recall that $S_{\mathrm{id}} := \widehat{V}_{\mathrm{id}}^\top \mathcal{C}_\kappa^\top \Sigma_{\kappa_1,\mathrm{id}}^{-1}$ (Proposition G.2). Define functions $f_{\star,\mathrm{id}}, \hat{f}_{\mathrm{id}}, \mathrm{err}_{f,\mathrm{id}} : \mathcal{Y} \to \mathbb{R}^{d_{\mathbf{x}}}$ via:*

$$\hat{f}_{\mathrm{id}} := \widehat{V}_{\mathrm{id}}^\top \hat{h}_{\mathrm{id}}, \quad f_{\star,\mathrm{id}} := S_{\mathrm{id}} f_\star, \quad \mathrm{err}_{f,\mathrm{id}} := \hat{f}_{\mathrm{id}} - f_{\star,\mathrm{id}}.$$

**Lemma G.4.** *Let $\kappa$ satisfy the conditions of Lemma G.1. Then, under the good event $\mathcal{E}_{\mathrm{id},h} \cap \mathcal{E}_{\mathrm{id,pca}}$,*

1. $\mathbb{E}\|\mathrm{err}_{f,\mathrm{id}}(\mathbf{y}_{\kappa_1})\|^2 \le \varepsilon_{\mathrm{id},h}^2$.

2. *For any $k \ge \kappa_0$ (in particular, for $k = \kappa_1 + 1$), $\mathbb{E}\|\mathrm{err}_{f,\mathrm{id}}(\mathbf{y}_k)\|^2 \le 2\varepsilon_{\mathrm{id},h}^2$*

3. *For any $k \ge \kappa_1$ (in particular, for $k \in \{\kappa_1, \kappa_1 + 1\}$), $\max\{\|\hat{f}_{\mathrm{id}}(\mathbf{y}_k)\|, \|f_{\star,\mathrm{id}}(\mathbf{y}_k)\|^2\}$ is $d_{\mathbf{x}} c_{\mathrm{conc,id}}$-concentrated, and $\|\mathrm{err}_{f,\mathrm{id}}(\mathbf{y}_k)\|^2$ is $4 d_{\mathbf{x}} c_{\mathrm{conc,id}}$-concentrated.*

4. *We have that $f_{\star,\mathrm{id}}(\mathbf{y}_{\kappa_1}) = S_{\mathrm{id}} \mathbf{x}_{\kappa_1}$ is zero-mean Gaussian, with*

$$\mathbb{E}\big[f_{\star,\mathrm{id}}(\mathbf{y}_{\kappa_1}) f_{\star,\mathrm{id}}(\mathbf{y}_{\kappa_1})^\top\big] = S_{\mathrm{id}}\big(\mathbb{E}\big[\mathbf{x}_{\kappa_1} \mathbf{x}_{\kappa_1}^\top\big]\big) S_{\mathrm{id}}^\top \succeq I \cdot \frac{\sigma_{d_{\mathbf{x}}}(\mathcal{C}_\kappa)^2(1-\gamma_\star)}{10 \Psi_\star^2 \alpha_\star^2}.$$

*Proof.* For point 1, we have that $f_{\star,\mathrm{id}} := S_{\mathrm{id}} f_\star = \widehat{V}_{\mathrm{id}}^\top \mathcal{C}_\kappa^\top \Sigma_{\kappa_1,\mathrm{id}}^{-1} f_\star = \widehat{V}_{\mathrm{id}}^\top h_{\star,\mathrm{id}}$. Hence,

$$\mathbb{E}\|\mathrm{err}_{f,\mathrm{id}}(\mathbf{y}_{\kappa_1})\|^2 = \mathbb{E}\|\widehat{V}_{\mathrm{id}}^\top(\hat{h}_{\mathrm{id}} - h_{\star,\mathrm{id}})\|^2 \overset{(i)}{\le} \mathbb{E}\|\hat{h}_{\mathrm{id}} - h_{\star,\mathrm{id}}\|^2 \overset{(ii)}{\le} \varepsilon_{\mathrm{id},h}^2,$$

where inequality $(i)$ uses that $\widehat{V}_{\mathrm{id}}$ has orthonormal columns, and inequality $(ii)$ uses the definition of $\mathcal{E}_{\mathrm{id},h}$ (Proposition G.1). Points 2 and 3 follow from Lemma G.1 and Lemma G.2, respectively.

Finally, for point 4, we use that $S_{\mathrm{id}} = \widehat{V}_{\mathrm{id}}^{\mathsf{T}} \mathcal{C}_\kappa^{\mathsf{T}} \Sigma_{\kappa_1,\mathrm{id}}^{-1}$ to write

$$
\begin{aligned}
\mathbb{E}\big[ f_{\star,\mathrm{id}}(\mathbf{x}_{\kappa_1}) f_{\star,\mathrm{id}}(\mathbf{x}_{\kappa_1})^{\mathsf{T}} \big] &= S_{\mathrm{id}} \mathbb{E}\big[ \mathbf{x}_{\kappa_1} \mathbf{x}_{\kappa_1}^{\mathsf{T}} \big] S_{\mathrm{id}}^{\mathsf{T}} \\
&= S_{\mathrm{id}} (\Sigma_{\kappa_1,\mathrm{id}}) S_{\mathrm{id}}^{\mathsf{T}} \\
&= \widehat{V}_{\mathrm{id}}^{\mathsf{T}} \mathcal{C}_\kappa^{\mathsf{T}} \Sigma_{\kappa_1,\mathrm{id}}^{-1} \Sigma_{\kappa_1,\mathrm{id}} \Sigma_{\kappa_1,\mathrm{id}}^{-1} \mathcal{C}_\kappa \widehat{V}_{\mathrm{id}} \\
&= \widehat{V}_{\mathrm{id}}^{\mathsf{T}} \mathcal{C}_\kappa^{\mathsf{T}} \Sigma_{\kappa_1,\mathrm{id}}^{-1} \mathcal{C}_\kappa \widehat{V}_{\mathrm{id}}.
\end{aligned}
$$

Hence,

$$
\begin{aligned}
\lambda_{\min}\big( S_{\mathrm{id}} \mathbb{E}\big[ \mathbf{x}_{\kappa_1} \mathbf{x}_{\kappa_1}^{\mathsf{T}} \big] S_{\mathrm{id}}^{\mathsf{T}} \big) &\geq \sigma_{\min}\big( \widehat{V}_{\mathrm{id}}^{\mathsf{T}} \mathcal{C}_\kappa^{\mathsf{T}} \Sigma_{\kappa_1,\mathrm{id}}^{-1/2} \big)^2 \\
&\geq \frac{4}{9} \sigma_{\min}\big( \mathcal{C}_\kappa^{\mathsf{T}} \Sigma_{\kappa_1,\mathrm{id}}^{-1/2} \big)^2 && \text{(Claim G.3)} \\
&\geq \frac{4}{9} \cdot \frac{5 \sigma_{d_{\mathbf{x}}}(\mathcal{C}_\kappa)^2 (1-\gamma_\star)}{11 \Psi_\star^2 \alpha_\star^2} && \text{(Lemma G.1)} \\
&\geq \frac{\sigma_{d_{\mathbf{x}}}(\mathcal{C}_\kappa)^2 (1-\gamma_\star)}{10 \Psi_\star^2 \alpha_\star^2},
\end{aligned}
$$

as needed. $\qquad\square$

### G.3.2 Estimation of $A_{\mathrm{id}}$, $B_{\mathrm{id}}$, and $\Sigma_{w,\mathrm{id}}$

We first show that Phase II recovers the system matrices and system noise covariance.

**Proposition G.3.** Define

$$
(\widehat{A}_{\mathrm{id}}, \widehat{B}_{\mathrm{id}}) \in \underset{(A,B)}{\arg\min} \; \frac{1}{n_{\mathrm{id}}} \sum_{i=2n_{\mathrm{id}}+1}^{n_{\mathrm{id}}} \| \hat{f}_{\mathrm{id}}(\mathbf{y}_{\kappa_1+1}^{(i)}) - A \hat{f}_{\mathrm{id}}(\mathbf{y}_{\kappa_1}^{(i)}) - B \mathbf{u}_{\kappa_1}^{(i)} \|^2,
$$

$$
\widehat{\Sigma}_{w,\mathrm{id}} = \frac{1}{n_{\mathrm{id}}} \sum_{i=2n_{\mathrm{id}}+1}^{n_{\mathrm{id}}} (\hat{f}_{\mathrm{id}}(\mathbf{y}_{\kappa_1+1}^{(i)}) - \widehat{A}_{\mathrm{id}} \hat{f}_{\mathrm{id}}(\mathbf{y}_{\kappa_1}^{(i)}) - \widehat{B}_{\mathrm{id}} \mathbf{u}_{\kappa_1}^{(i)})^{\otimes 2}.
$$

Further, define the matrices $A_{\mathrm{id}} := S_{\mathrm{id}} A S_{\mathrm{id}}^{-1}$, $B_{\mathrm{id}} := S_{\mathrm{id}} B$, and $\Sigma_{w,\mathrm{id}} := S_{\mathrm{id}} \Sigma_w S_{\mathrm{id}}^{\mathsf{T}}$. Suppose $\varepsilon_{\mathrm{id},h}$ satisfies Eq. (G.14) (which is the preqrequisite of $\mathcal{E}_{\mathrm{id,pca}}$ of Proposition G.2), and $n \geq c_0(d_{\mathbf{x}} + \ln(1/\delta))$ for some universal constant $c_0$. Then on $\mathcal{E}_{\mathrm{id,pca}} \cap \mathcal{E}_{\mathrm{id},h}$, the following event, designated $\mathcal{E}_{\mathrm{id,ls}}$, holds with probability $1 - 7\delta$:

$$
\| [\widehat{A}_{\mathrm{id}}; \widehat{B}_{\mathrm{id}}] - [A_{\mathrm{id}}; B_{\mathrm{id}}] \|_{\mathrm{op}} \lesssim \frac{\Psi_\star^{7/2} \alpha_\star^3}{\sigma_{\min}(\mathcal{C}_\kappa)^2 (1-\gamma_\star)^{3/2}} \varepsilon_{\mathrm{id},h},
$$

$$
\| \widehat{\Sigma}_{w,\mathrm{id}} - \Sigma_{w,\mathrm{id}} \|_{\mathrm{op}} \lesssim \frac{\Psi_\star^{5/2} \alpha_\star^4}{\sigma_{\min}(\mathcal{C}_\kappa)(1-\gamma_\star)} \varepsilon_{\mathrm{id},h}.
$$

*Proof.* We cast the regressions above as an instance of error-in-variable regression, then apply our general guarantees for this problem (Propositions E.3 and E.4). We restate the guarantees here:

**Proposition E.3** (Linear regression with errors in variables). Let $(\mathbf{u}, \mathbf{y}, \mathbf{w}, \mathbf{e}, \boldsymbol{\delta})$ be a collection of random variables defined over a shared probability space, and let $\big\{ (\mathbf{u}^{(i)}, \mathbf{y}^{(i)} \mathbf{w}^{(i)}, \mathbf{e}^{(i)}, \boldsymbol{\delta}^{(i)}) \big\}_{i=1}^{n}$ be i.i.d. copies. Suppose the following conditions hold:

1. $\mathbf{y} = M_\star \mathbf{u} + \mathbf{w} + \mathbf{e}$ with probability 1, where $M_\star \in \mathbb{R}^{d_{\mathbf{y}} \times d_{\mathbf{u}}}$.

2. $\mathbf{w} \mid \mathbf{u}, \boldsymbol{\delta} \sim \mathcal{N}(0, \Sigma_w)$ and $\mathbf{u} \sim \mathcal{N}(0, \Sigma_u)$.

3. We have $\mathbb{E}\|\mathbf{e}\|^2 \leq \varepsilon_{\mathbf{e}}^2$ and $\mathbb{E}\|\boldsymbol{\delta}\|^2 \leq \varepsilon_{\boldsymbol{\delta}}^2$.

4. $\mathbf{e}$ is $c_{\mathbf{e}}$-concentrated and $\boldsymbol{\delta}$ is $c_{\boldsymbol{\delta}}$-concentrated for $c_{\mathbf{e}} \geq \varepsilon_{\mathbf{e}}^2$ and $c_{\boldsymbol{\delta}} \geq \varepsilon_{\boldsymbol{\delta}}^2$.

5. $\varepsilon_{\boldsymbol{\delta}}^2 \leq \frac{1}{16}\lambda_{\min}(\Sigma_u)$.

Let $\delta \leq 1/e$, and let $n \in \mathbb{N}$ satisfy

1. $\psi(n,\delta) \leq \min\left\{\frac{\varepsilon_{\mathbf{e}}^2}{c_{\mathbf{e}}}, \frac{\varepsilon_{\boldsymbol{\delta}}^2}{c_{\boldsymbol{\delta}}}\right\}$, where $\psi(n,\delta) := \frac{2\ln(2n/\delta)\ln(2/\delta)}{n}$.

2. $n \geq c_1(d_{\mathbf{u}} + \ln(1/\delta))$, for some universal constant $c_1 > 0$.

Then the solution to the least squares problem

$$\widehat{M} = \min_M \sum_{i=1}^n \|M(\mathbf{u}^{(i)} + \boldsymbol{\delta}^{(i)}) - \mathbf{y}^{(i)}\|^2,$$

satisfies the following inequality with probability at least $1 - 4\delta$:

$$\|\widehat{M} - M_\star\|_{\mathrm{op}}^2 \lesssim \lambda_{\min}(\Sigma_u)^{-1}\left(\|M_\star\|_{\mathrm{op}}^2 \varepsilon_{\boldsymbol{\delta}}^2 + \varepsilon_{\mathbf{e}}^2 + \frac{\|\Sigma_w\|_{\mathrm{op}}(d_{\mathbf{y}} + d_{\mathbf{u}} + \ln(1/\delta))}{n}\right). \tag{E.2}$$

**Proposition E.4.** Consider the setting of Proposition E.3, and suppose we additionally require that $n \geq c_0(d_{\mathbf{y}} + \ln(1/\delta))$ for some (possibly inflated) universal constant $c_0$. Furthermore, suppose we have $\varepsilon_{\boldsymbol{\delta}}^2\|M_\star\|_{\mathrm{op}}^2 + \varepsilon_{\mathbf{e}}^2 \leq 2\lambda_+$ for some $\lambda_+ \geq \lambda_{\max}(\Sigma_w)$. Then, with probability at least $1 - 7\delta$, (E.2) holds, and moreover

$$\left\|\frac{1}{n}\sum_{i=1}^n (\widehat{M}(\mathbf{u}^{(i)} + \boldsymbol{\delta}^{(i)}) - \mathbf{y}^{(i)})^{\otimes 2} - \Sigma_w\right\|_{\mathrm{op}} \lesssim \sqrt{\lambda_+(\varepsilon_{\boldsymbol{\delta}}^2\|M_\star\|_{\mathrm{op}}^2 + \varepsilon_{\mathbf{e}}^2) + \frac{\lambda_+^2(d_{\mathbf{y}} + \ln(1/\delta))}{n}}.$$

To distinguish between our present notation and the notation of these propositions, we mark the terms to which we apply the proposition with a tilde. Define

$$\begin{aligned}
\tilde{\mathbf{y}} &:= \hat{f}_{\mathrm{id}}(\mathbf{y}_{\kappa_1+1}^{(i)}), \\
\tilde{\mathbf{e}} &:= \mathrm{err}_{f,\mathrm{id}}(\mathbf{y}_{\kappa_1+1}^{(i)}) = (f_{\star,\mathrm{id}} - \hat{f}_{\mathrm{id}})(\mathbf{y}_{\kappa_1+1}^{(i)}), \\
\tilde{\boldsymbol{\delta}} &:= [-\mathrm{err}_{f,\mathrm{id}}(\mathbf{y}_{\kappa_1})^\top; 0_{d_{\mathbf{u}}}^\top]^\top, \\
\tilde{\mathbf{u}} &:= [f_{\star,\mathrm{id}}(\mathbf{y}_{\kappa_1})^\top; \mathbf{u}_{\kappa_1}^\top]^\top, \\
\widetilde{\mathbf{w}} &:= -S_{\mathrm{id}}\mathbf{w}_{\kappa_1}, \\
d_{\tilde{y}} &:= d_{\mathbf{x}}, \\
d_{\tilde{u}} &:= d_{\mathbf{x}} + d_{\mathbf{u}}, \\
\tilde{M} &:= [A_{\mathrm{id}}; B_{\mathrm{id}}].
\end{aligned}$$

To proceed, we verify that this correspondence satisfies the conditions of the propositions above.

**Claim G.4.** It holds that $\tilde{M}\tilde{\mathbf{u}} = \tilde{\mathbf{y}} + \tilde{\mathbf{e}} + \widetilde{\mathbf{w}}$ and

$$[\widehat{A}_{\mathrm{id}}; \widehat{B}_{\mathrm{id}}] \in \arg\min_M \sum_{i=1}^n \|\tilde{\mathbf{y}}^{(i)} - M(\tilde{\mathbf{u}}^{(i)} + \tilde{\boldsymbol{\delta}}^{(i)})\|^2.$$

*Proof.* Observe that we have the dynamics

$$A\mathbf{x}_{\kappa_1} + B\mathbf{u}_{\kappa_1} = \mathbf{x}_{\kappa_1+1} - \mathbf{w}_{\kappa_1}$$

and

$$Af_\star(\mathbf{y}_{\kappa_1}) + B\mathbf{u}_{\kappa_1} = f_\star(\mathbf{y}_{\kappa_1+1}) - \mathbf{w}_{\kappa_1}.$$

Thus, recalling that $f_{\star,\mathrm{id}}(y) = S_{\mathrm{id}}f_\star(y)$, we have

$$S_{\mathrm{id}}AS_{\mathrm{id}}^{-1}f_{\star,\mathrm{id}}(\mathbf{y}_{\kappa_1}) + S_{\mathrm{id}}B\mathbf{u}_{\kappa_1} = f_{\star,\mathrm{id}}(\mathbf{y}_{\kappa_1+1}) - S_{\mathrm{id}}\mathbf{w}_{\kappa_1}.$$

In our new notation, this implies that

$$[A_{\mathrm{id}}; B_{\mathrm{id}}]\tilde{\mathbf{u}} = f_{\star,\mathrm{id}}(\mathbf{y}_{\kappa_1+1}) + \widetilde{\mathbf{w}}.$$

Finally, writing $f_{\star,\mathrm{id}}(\mathbf{y}_{\kappa_1}) = \tilde{\mathbf{y}} + \tilde{\mathbf{e}}$ yields the first part of the claim. The second part follows from similar manipulations. $\square$

Next, we check the Gaussianity and covariance properties of $\tilde{\mathbf{u}}, \widetilde{\mathbf{w}}$.

**Claim G.5.** The following properties hold:

- $\widetilde{\mathbf{w}}, \tilde{\mathbf{u}}$, and $\tilde{\boldsymbol{\delta}}$ are mutually independent.

- $\widetilde{\mathbf{w}} \sim \mathcal{N}(0, \Sigma_{w,\mathrm{id}})$, where $\lambda_{\max}(\Sigma_{w,\mathrm{id}}) \leq 1$.

- $\tilde{\mathbf{u}} \sim \mathcal{N}(0, \Sigma_{\tilde{u}})$, where $\lambda_{\min}(\Sigma_{\tilde{u}}) \geq \frac{\sigma_{d_{\mathbf{x}}}(\mathcal{C}_\kappa)^2(1-\gamma_\star)}{10\Psi_\star^2\alpha_\star^2}$.

*Proof of Claim G.5.* The first point of the claim follows because $\widetilde{\mathbf{w}}$ is determined by $\mathbf{w}_{\kappa_1}$ and $\tilde{\mathbf{u}}$ and $\tilde{\boldsymbol{\delta}}$ are determined by $\mathbf{y}_{\kappa_1}$ and $\mathbf{u}_{\kappa_1}$, respectively.

For the second claim, we have that $\widetilde{\mathbf{w}} = -S_{\mathrm{id}}\mathbf{w}$. Since $\mathbb{E}[\mathbf{w}\mathbf{w}^\top] = \Sigma_w$, $\mathbb{E}[\widetilde{\mathbf{w}}\widetilde{\mathbf{w}}^\top] = S_{\mathrm{id}}\Sigma_w S_{\mathrm{id}}^\top$. Recalling that $S_{\mathrm{id}} = \widehat{V}_{\mathrm{id}}^\top \mathcal{C}_\kappa^\top \Sigma_{\kappa_1,\mathrm{id}}^{-1}$ for some orthonormal $\widehat{V}_{\mathrm{id}}$, we find that

$$\lambda_{\max}(S_{\mathrm{id}}\Sigma_w S_{\mathrm{id}}^\top) \leq \lambda_{\max}(\mathcal{C}_\kappa^\top \Sigma_{\kappa_1,\mathrm{id}}^{-1}\Sigma_w \Sigma_{\kappa_1,\mathrm{id}}^{-1}\mathcal{C}_\kappa)$$
$$= \sigma_{\max}^2(\mathcal{C}_\kappa^\top \Sigma_{\kappa_1,\mathrm{id}}^{-1}\Sigma_w^{1/2}) \leq \left(\sigma_{\max}(\mathcal{C}_\kappa^\top \Sigma_{\kappa_1,\mathrm{id}}^{-1/2})\sigma_{\max}(\Sigma_{\kappa_1,\mathrm{id}}^{-1/2}\Sigma_w^{1/2})\right)^2.$$

Since $\Sigma_{\kappa_1,\mathrm{id}} \succeq \Sigma_w$ and since $\Sigma_{\kappa_1,\mathrm{id}} \succeq \mathcal{C}_\kappa\mathcal{C}_\kappa^\top$, we have that $\sigma_{\max}(\mathcal{C}_\kappa^\top \Sigma_{\kappa_1,\mathrm{id}}^{-1/2}), \sigma_{\max}(\Sigma_{\kappa_1,\mathrm{id}}^{-1/2}\Sigma_w^{1/2}) \leq 1$, and we conclude that $\lambda_{\max}(S_{\mathrm{id}}\Sigma_w S_{\mathrm{id}}^\top) \leq 1$ as needed.

For the second-to-last claim, we have

$$\tilde{\mathbf{u}} = \begin{bmatrix} f_{\star,\mathrm{id}}(\mathbf{y}_{\kappa_1}) \\ \mathbf{u}_{\kappa_1} \end{bmatrix} = \begin{bmatrix} S_{\mathrm{id}}\mathbf{x}_{\kappa_1} \\ \mathbf{u}_{\kappa_1} \end{bmatrix}.$$

Since $\mathbf{x}_{\kappa_1} \perp \mathbf{u}_{\kappa_1}$, we have that

$$\mathbb{E}[\tilde{\mathbf{u}}\tilde{\mathbf{u}}^\top] = \begin{bmatrix} S_{\mathrm{id}}\mathbb{E}[\mathbf{x}_{\kappa_1}\mathbf{x}_{\kappa_1}^\top]S_{\mathrm{id}}^\top & 0 \\ 0 & I \end{bmatrix} \succeq \begin{bmatrix} I \cdot \frac{\sigma_{d_{\mathbf{x}}}(\mathcal{C}_\kappa)^2(1-\gamma_\star)}{10\Psi_\star^2\alpha_\star^2} & 0 \\ 0 & I \end{bmatrix},$$

where the last inequality uses part 4 of Lemma G.4. One can verify that the lower bound on the upper left block is less than 1. Thus,

$$\lambda_{\min}(\mathbb{E}[\tilde{\mathbf{u}}\tilde{\mathbf{u}}^\top]) \geq \frac{\sigma_{d_{\mathbf{x}}}(\mathcal{C}_\kappa)^2(1-\gamma_\star)}{10\Psi_\star^2\alpha_\star^2}.$$

$\square$

Lastly, we check the relevant concentration properties for the errors $\tilde{\mathbf{e}}$ and $\tilde{\boldsymbol{\delta}}$.

**Claim G.6.** The following bounds hold:

- $\tilde{\boldsymbol{\delta}}$ and $\tilde{\mathbf{e}}$ are both $c_{\mathrm{ls}} := 4c_{\mathrm{conc,id}}$-concentrated, and satisfy $\mathbb{E}\|\tilde{\boldsymbol{\delta}}\|^2 \vee \mathbb{E}\|\tilde{\mathbf{e}}\|^2 \leq 2\varepsilon_{\mathrm{id},h}^2 =: \varepsilon_{\mathrm{ls}}^2$.

- For $n \geq n_{\mathrm{id}}$, we have that $\varepsilon_{\mathrm{ls}}/c_{\mathrm{ls}} \geq \psi(n,\delta) = \frac{2\ln(2n/\delta)\ln(2/\delta)}{n}$.

- $\|\tilde{M}\|_{\mathrm{op}} \leq \frac{4\Psi_\star^{5/2}\alpha_\star^2}{\sigma_{\min}(\mathcal{C}_\kappa)(1-\gamma_\star)}$, and thus

$$(1 + \|\tilde{M}\|_{\mathrm{op}}^2)\varepsilon_{\mathrm{ls}}^2 \leq \frac{65\Psi_\star^5\alpha_\star^4}{\sigma_{\min}(\mathcal{C}_\kappa)^2(1-\gamma_\star)^2}\varepsilon_{\mathrm{ls}}^2 \leq \frac{130\Psi_\star^5\alpha_\star^4}{\sigma_{\min}(\mathcal{C}_\kappa)^2(1-\gamma_\star)^2}\varepsilon_{\mathrm{id},h}^2.$$

- For $\varepsilon_{\mathrm{id},h}$ satisfying Eq. (G.14), we have $\varepsilon_{\mathrm{ls}}^2 \leq \frac{1}{16}\lambda_{\min}(\Sigma_{\tilde{u}})$ and thus $(1 + \|\tilde{M}\|_{\mathrm{op}}^2)\varepsilon_{\mathrm{ls}}^2 \leq \lambda_+ =: 1$

*Proof.* The first claims follows from Lemma G.4.

The second claim uses that, examining the definition of $\varepsilon_{\mathrm{id},h}$ in Proposition G.1, we have $\varepsilon_{\mathrm{id},h} \leq 4\psi(n_{\mathrm{id}},\delta)/c_{\mathrm{conc},\mathrm{id}}$, implying $\varepsilon_{\mathrm{ls}}/c_{\mathrm{ls}} \geq \psi(n_{\mathrm{id}},\delta)$. Lastly use that $n \geq n_{\mathrm{id}}$ and that $n \mapsto \psi(n,\delta)$ is decreasing.

For the third point,

$$\|\tilde{M}\|_{\mathrm{op}} = \|[A_{\mathrm{id}}; B_{\mathrm{id}}]\|_{\mathrm{op}} = \|[S_{\mathrm{id}}AS_{\mathrm{id}}^{-1}; S_{\mathrm{id}}B]\|_{\mathrm{op}} \leq 2(1 \vee \sigma_{\min}^{-1}(S_{\mathrm{id}}))(\|S_{\mathrm{id}}A\|_{\mathrm{op}} \vee \|S_{\mathrm{id}}B\|_{\mathrm{op}}).$$

Recalling $S_{\mathrm{id}} = \widehat{V}_{\mathrm{id}}^{\top} \mathcal{C}_{\kappa}^{\top} \Sigma_{\kappa_1,\mathrm{id}}^{-1}$, we use that $\|\widehat{V}_{\mathrm{id}}\|_{\mathrm{op}} \leq 1$ and $\Sigma_{\kappa_1,\mathrm{id}} \geq \mathcal{C}_{\kappa}^{\top}$ to bound

$$\|S_{\mathrm{id}}A\|_{\mathrm{op}} \vee \|S_{\mathrm{id}}B\|_{\mathrm{op}} \leq \|\Sigma_{\kappa_1,\mathrm{id}}^{-1/2}A\|_{\mathrm{op}} \vee \|\Sigma_{\kappa_1,\mathrm{id}}^{-1/2}B\|_{\mathrm{op}}.$$

Since $\Sigma_{\kappa_1,\mathrm{id}} \geq A\Sigma_w A^{\top} \geq AA^{\top}/\lambda_{\min}(\Sigma_w) \geq AA^{\top}\Psi_{\star}^{-1}$ and $\Sigma_{\kappa_1,\mathrm{id}} \geq BB^{\top}$, we conclude that $\|S_{\mathrm{id}}A\|_{\mathrm{op}} \vee \|S_{\mathrm{id}}B\|_{\mathrm{op}} \leq \Psi_{\star}^{1/2}$. Lastly, using $\sigma_{\min}(S_{\mathrm{id}}) \geq \sigma_{\min,\mathrm{id}} = \frac{\sigma_{\min}(\mathcal{C}_{\kappa})(1-\gamma_{\star})}{4\Psi_{\star}^2\alpha_{\star}^2}$ from Proposition G.2, we conclude that

$$\|\tilde{M}\|_{\mathrm{op}} \leq \sqrt{\Psi_{\star}} \frac{8\Psi_{\star}^2\alpha_{\star}^2}{\sigma_{\min}(\mathcal{C}_{\kappa})(1-\gamma_{\star})} = \frac{8\Psi_{\star}^{5/2}\alpha_{\star}^2}{\sigma_{\min}(\mathcal{C}_{\kappa})(1-\gamma_{\star})},$$

as needed. The following inequality follows directly:

$$(\|\tilde{M}\|_{\mathrm{op}}^2 + 1)\varepsilon_{\mathrm{ls}}^2 \leq \frac{65\Psi_{\star}^5\alpha_{\star}^4}{\sigma_{\min}(\mathcal{C}_{\kappa})^2(1-\gamma_{\star})^2}\varepsilon_{\mathrm{ls}}^2 \leq \frac{130\Psi_{\star}^5\alpha_{\star}^4}{\sigma_{\min}(\mathcal{C}_{\kappa})^2(1-\gamma_{\star})^2}\varepsilon_{\mathrm{id},h}^2. \qquad (\text{G.17})$$

For the fourth point, we examine the condition in Eq. (G.14), $\sqrt{d_{\mathbf{x}}c_{\mathrm{conc},\mathrm{id}}}\varepsilon_{\mathrm{id},h} \leq \frac{(1-\gamma_{\star})\sigma_{d_{\mathbf{x}}}(\mathcal{C}_{\kappa})^2}{71\alpha_{\star}^2\Psi_{\star}^2}$, which is equivalent to

$$\varepsilon_{\mathrm{id},h}^2 \leq \frac{(1-\gamma_{\star})^2\sigma_{d_{\mathbf{x}}}(\mathcal{C}_{\kappa})^2}{d_{\mathbf{x}}c_{\mathrm{conc},\mathrm{id}}71^2\alpha_{\star}^2\Psi_{\star}^2} \cdot (\sigma_{d_{\mathbf{x}}}(\mathcal{C}_{\kappa})^2/\alpha_{\star}^2\Psi_{\star}^2).$$

Using the definition of $c_{\mathrm{conc},\mathrm{id}} = 12\Psi_{\star}^3\alpha_{\star}^2 L^2/(1-\gamma_{\star})$ and that $d_{\mathbf{x}}, L \geq 1$, this further implies that

$$\varepsilon_{\mathrm{id},h}^2 \leq \frac{(1-\gamma_{\star})^2\sigma_{d_{\mathbf{x}}}(\mathcal{C}_{\kappa})^2}{12 \cdot 71^2\alpha_{\star}^2\Psi_{\star}^2} \cdot (\sigma_{d_{\mathbf{x}}}(\mathcal{C}_{\kappa})^2/\alpha_{\star}^2\Psi_{\star}^2) \cdot \frac{(1-\gamma_{\star})}{\Psi_{\star}^3\alpha_{\star}^2}.$$

One can verify that $(\sigma_{d_{\mathbf{x}}}(\mathcal{C}_{\kappa})^2/\alpha_{\star}^2\Psi_{\star}^2) \leq 1$ using the same arguments as in Lemma G.1. Thus, under the above condition,

$$\varepsilon_{\mathrm{id},h}^2 \leq \frac{(1-\gamma_{\star})^2\sigma_{d_{\mathbf{x}}}(\mathcal{C}_{\kappa})^2}{12 \cdot 71^2\alpha_{\star}^2\Psi_{\star}^2} \cdot \frac{(1-\gamma_{\star})}{\Psi_{\star}^3\alpha_{\star}^2}.$$

Recalling $\lambda_{\min}(\Sigma_{\tilde{u}}) \geq \frac{\sigma_{d_{\mathbf{x}}}(\mathcal{C}_{\kappa})^2(1-\gamma_{\star})}{10\Psi_{\star}^2\alpha_{\star}^2}$ from Claim G.5, we can directly verify that this implies $\varepsilon_{\mathrm{ls}}^2 = 2\varepsilon_{\mathrm{id},h}^2 \leq \frac{1}{16}\lambda_{\min}(\Sigma_{\tilde{u}})$. Similarly, using the bound in Eq. (G.17) we can check that $(\|\tilde{M}\|_{\mathrm{op}}^2 + 1)\varepsilon_{\mathrm{ls}}^2 \leq \lambda_+ := 1$. $\qquad \square$

To summarize, the claims above verify that, for $\varepsilon_{\mathrm{id},h}$ satisfying Eq. (G.14), and $n_{\mathrm{id}} \geq c_0(d_{\tilde{u}} + d_{\tilde{y}} + \ln(1/\delta))$, the conditions for Propositions E.3 and E.4 hold. It follows that with probability at least $1 - 7\delta$,

$$\|[\widehat{A}_{\mathrm{id}}; \widehat{B}_{\mathrm{id}}] - [A_{\mathrm{id}}; B_{\mathrm{id}}]\|_{\mathrm{op}}^2 \lesssim \lambda_{\min}(\Sigma_{\tilde{u}})^{-1}\left((1 + \|\tilde{M}\|_{\mathrm{op}}^2)\varepsilon_{\mathrm{ls}}^2 + \frac{\|\Sigma_{w,\mathrm{id}}\|_{\mathrm{op}}(d_{\tilde{y}} + d_{\tilde{u}} + \ln(1/\delta))}{n_{\mathrm{id}}}\right),$$

and

$$\|\widehat{\Sigma}_{w,\mathrm{id}} - \Sigma_{w,\mathrm{id}}\|_{\mathrm{op}} \lesssim \sqrt{\lambda_+(1 + \|\tilde{M}\|_{\mathrm{op}}^2)\varepsilon_{\mathrm{ls}}^2 + \frac{\lambda_+^2(d_{\mathbf{y}} + \ln(1/\delta))}{n_{\mathrm{id}}}}.$$

Finally, to conclude, we note that $d_{\tilde{y}} + d_{\tilde{u}} = 2d_{\mathbf{x}} + d_{\mathbf{u}}$, so that (also recalling $d_{\mathbf{u}} \leq d_{\mathbf{x}}$) it suffices to ensure $\varepsilon_{\mathrm{id},h}$ satisfying Eq. (G.14), and $n_{\mathrm{id}} \geq c_0(d_{\mathbf{x}} + \ln(1/\delta))$ for a larger universal constant $c_0$.

Moreover, we can check that $(d_{\mathbf{x}} + \ln(1/\delta))/n_{\mathrm{id}} \lesssim \varepsilon_{\mathrm{id},h}^2$, which together with the bound $\|\Sigma_{w,\mathrm{id}}\|_{\mathrm{op}} \leq \lambda_+ = 1$ means that the dominant terms above are the terms $(1 + \|\tilde{M}\|_{\mathrm{op}}^2)\varepsilon_{\mathrm{ls}}^2 \lesssim \frac{\Psi_\star^5 \alpha_\star^4}{\sigma_{\min}(\mathcal{C}_\kappa)^2(1-\gamma_\star)^2}\varepsilon_{\mathrm{id},h}^2$.

Thus, recalling $\lambda_{\min}(\Sigma_{\tilde{u}}) \gtrsim \frac{\sigma_{d_{\mathbf{x}}}(\mathcal{C}_\kappa)^2(1-\gamma_\star)}{\Psi_\star^2 \alpha_\star^2}$ from Claim G.5, we find

$$\|[\widehat{A}_{\mathrm{id}}; \widehat{B}_{\mathrm{id}}] - \tilde{M}\|_{\mathrm{op}} \lesssim \sqrt{\frac{\Psi_\star^2 \alpha_\star^2}{\sigma_{d_{\mathbf{x}}}(\mathcal{C}_\kappa)^2(1-\gamma_\star)} \cdot \frac{\Psi_\star^5 \alpha_\star^4}{\sigma_{\min}(\mathcal{C}_\kappa)^2(1-\gamma_\star)^2}\varepsilon_{\mathrm{id},h}^2}$$

$$\leq \frac{\Psi_\star^{7/2} \alpha_\star^3}{\sigma_{\min}(\mathcal{C}_\kappa)^2(1-\gamma_\star)^{3/2}}\varepsilon_{\mathrm{id},h}.$$

and

$$\|\widehat{\Sigma}_{w,\mathrm{id}} - \Sigma_{w,\mathrm{id}}\|_{\mathrm{op}} \lesssim \frac{\Psi_\star^{5/2} \alpha_\star^4}{\sigma_{\min}(\mathcal{C}_\kappa)(1-\gamma_\star)}\varepsilon_{\mathrm{id},h}.$$

$\square$

### G.3.3 Recovering the Cost Matrix $Q_{\mathrm{id}}$

We now show that Phase II successfully recovers the system cost matrix $Q$ up to a similarity transform.

**Proposition G.4** (Guarantee for Recovery of $Q_{\mathrm{id}}$). Recall the estimator

$$\widehat{Q}_{\mathrm{id}} = \left(\frac{1}{2}\widetilde{Q}_{\mathrm{id}} + \frac{1}{2}\widetilde{Q}_{\mathrm{id}}^\top\right)_+,$$

$$\widetilde{Q}_{\mathrm{id}} = \min_Q \sum_{i=2n_{\mathrm{id}}+1}^{3n_{\mathrm{id}}} \left(\mathbf{c}_{\kappa_1}^{(i)} - (\mathbf{u}_{\kappa_1}^{(i)})^\top R \mathbf{u}_{\kappa_1}^{(i)} - \hat{f}_{\mathrm{id}}(\mathbf{y}_{\kappa_1}^{(i)})^\top Q \hat{f}_{\mathrm{id}}(\mathbf{y}_{\kappa_1}^{(i)})\right)^2.$$

Suppose that the following conditions hold for a sufficiently large numerical constant $c_0$:

$$n_{\mathrm{id}} \geq c_0(d_{\mathbf{x}}^2 + \ln(1/\delta)), \quad \text{and} \quad \varepsilon_{\mathrm{id},h}\sqrt{\ln(2n_{\mathrm{id}}/\delta)} \leq \frac{\sigma_{d_{\mathbf{x}}}(\mathcal{C}_\kappa)^2(1-\gamma_\star)}{8 \cdot 10 c_{\mathrm{conc,id}}\Psi_\star^2 \alpha_\star^2 d_{\mathbf{x}}}.$$

Then with probability at least $1 - 2\delta$, we have

$$\|\widehat{Q}_{\mathrm{id}} - Q_{\mathrm{id}}\|_{\mathrm{op}} \lesssim \frac{\alpha_\star^6 \Psi_\star^7 \sqrt{c_{\mathrm{conc,id}}d_{\mathbf{x}}\ln(n_{\mathrm{id}}/\delta)}}{(1-\gamma_\star)^3 \sigma_{\min}(\mathcal{C}_\kappa)^4} \cdot \varepsilon_{\mathrm{id},h}.$$

*Proof.* We first rewrite the regression as a special case of Proposition E.5, which offers a generic guarantee for matrix regression with rank-one measurements. Observe that we have

$$\mathbf{c}_{\kappa_1} - \mathbf{u}_{\kappa_1}^\top R \mathbf{u}_{\kappa_1} = \left(f_\star(\mathbf{y}_{\kappa_1})^\top Q f_\star(\mathbf{y}_{\kappa_1}) + \mathbf{u}_{\kappa_1}^\top R \mathbf{u}_{\kappa_1}\right) - \mathbf{u}_{\kappa_1}^\top R \mathbf{u}_{\kappa_1}$$

$$= f_\star(\mathbf{y}_{\kappa_1})^\top Q f_\star(\mathbf{y}_{\kappa_1})$$

$$= f_{\star,\mathrm{id}}(\mathbf{y}_{\kappa_1})^\top S_{\mathrm{id}}^{-\top} Q S_{\mathrm{id}}^{-1} f_{\star,\mathrm{id}}(\mathbf{y}_{\kappa_1})$$

$$= f_{\star,\mathrm{id}}(\mathbf{y}_{\kappa_1})^\top Q_{\mathrm{id}} f_{\star,\mathrm{id}}(\mathbf{y}_{\kappa_1}).$$

Thus, the above regression is equivalent to solving

$$\widetilde{Q}_{\mathrm{id}} = \min_Q \sum_{i=2n_{\mathrm{id}}+1}^{3n_{\mathrm{id}}} \left(f_{\star,\mathrm{id}}(\mathbf{y}_{\kappa_1}^{(i)})^\top Q_{\mathrm{id}} f_{\star,\mathrm{id}}(\mathbf{y}_{\kappa_1}^{(i)}) - \hat{f}_{\mathrm{id}}(\mathbf{y}_{\kappa_1}^{(i)})^\top Q \hat{f}_{\mathrm{id}}(\mathbf{y}_{\kappa_1}^{(i)})\right)^2.$$

We now recall the statement of Proposition E.5.

**Proposition E.5** (Regression with matrix measurements). Let $\mathbf{y} \in \mathcal{Y}$ be a random variable, and let $\mathbf{y}^{(i)} \overset{\text{i.i.d.}}{\sim} \mathbf{y}$ for $1 \leq i \leq n$. Fix two regression functions $\hat{g}, g_\star : \mathcal{Y} \to \mathbb{R}^d$, and suppose that

$\mathbf{z} \coloneqq \max\{\|\hat{g}(\mathbf{y})\|^2, \|g_\star(\mathbf{y})\|^2\}$ is $c$-concentrated, and that $\mathbf{x} \coloneqq g_\star(\mathbf{y}) \sim \mathcal{N}(0, \Sigma_x)$. Let $Q^\star \geq 0$ be a fixed matrix, and consider the regression.

$$\widetilde{Q} \in \operatorname*{arg\,min}_{M} \sum_{i=1}^{n} \left( g_\star(\mathbf{y}^{(i)})^\top Q^\star g_\star(\mathbf{y}^{(i)}) - \hat{g}(\mathbf{y}^{(i)})^\top M \hat{g}(\mathbf{y}^{(i)})^\top \right)^2.$$

Set $\widehat{Q} \coloneqq (\frac{1}{2}\widetilde{Q}^\top + \frac{1}{2}\widetilde{Q})_+$, where $(\cdot)_+$ truncates all negative eigenvalues to zero. Then, there is a universal constant $c_0 > 0$ such that if the following conditions hold:

$$\mathbb{E}\|\hat{g}(\mathbf{y}) - g_\star(\mathbf{y})\|^2 \leq \varepsilon^2, \quad \psi(n, \delta/2) \leq \frac{\varepsilon^2}{4c}, \quad n \geq c_0(d^2 + \ln(1/\delta)), \quad \text{and} \quad \varepsilon^2 \leq \frac{\lambda_{\min}(\Sigma_x)^2}{64c\ln(2n/\delta)},$$

then with probability at least $1 - 2\delta$,

$$\|\widehat{Q} - Q^\star\|_F^2 \leq \|\widetilde{Q} - Q^\star\|_F^2 \leq 64c\varepsilon^2 \ln(4n/\delta) \cdot \frac{\|Q^\star\|_{\mathrm{op}}^2}{\lambda_{\min}(\Sigma_x)^2}.$$

To apply the proposition, we make the following substitutions:

$$Q^\star \leftarrow Q_{\mathrm{id}}, \quad \widehat{Q} \leftarrow \widehat{Q}_{\mathrm{id}}, \quad g_\star \leftarrow f_{\star,\mathrm{id}}, \quad \hat{g} \leftarrow \hat{f}_{\mathrm{id}}, \quad \Sigma_x \leftarrow \mathbb{E}\big[f_{\star,\mathrm{id}}(\mathbf{y}_{\kappa_1})f_{\star,\mathrm{id}}(\mathbf{y}_{\kappa_1})^\top\big], \quad n \leftarrow n_{\mathrm{id}}.$$

We now verify that the conditions for the proposition are satisfied.

1. From Lemma G.4, $\max\{\|\hat{f}_{\mathrm{id}}(\mathbf{y}_{\kappa_1})\|^2, \|f_{\star,\mathrm{id}}(\mathbf{y}_{\kappa_1})\|^2\}$ is $d_{\mathbf{x}}c_{\mathrm{conc,id}}$-concentrated, and $\mathbb{E}\|\hat{f}_{\mathrm{id}}(\mathbf{y}_{\kappa_1}) - f_{\star,\mathrm{id}}(\mathbf{y}_{\kappa_1})\|^2 \leq \varepsilon_{\mathrm{id},h}^2$.

2. We have that $\psi(n_{\mathrm{id}}, \delta/2) \leq \frac{\varepsilon_{\mathrm{id},h}^2}{d_{\mathbf{x}}c_{\mathrm{conc,id}}}$ by examining the definition of $\varepsilon_{\mathrm{id},h}$ in Proposition G.1.

3. We have that $f_{\star,\mathrm{id}}(\mathbf{y}_{\kappa_1}) = S_{\mathrm{id}}\mathbf{x}_{\kappa_1}$ is zero-mean Gaussian, with

$$\mathbb{E}\big[f_{\star,\mathrm{id}}(\mathbf{y}_{\kappa_1})f_{\star,\mathrm{id}}(\mathbf{y}_{\kappa_1})^\top\big] \geq I \cdot \frac{\sigma_{d_{\mathbf{x}}}(\mathcal{C}_\kappa)^2(1 - \gamma_\star)}{10\Psi_\star^2\alpha_\star^2}$$

by Lemma G.4. Hence, the conditions

$$\varepsilon_{\mathrm{id},h}\sqrt{\ln(2n_{\mathrm{id}}/\delta)} \leq \frac{\sigma_{d_{\mathbf{x}}}(\mathcal{C}_\kappa)^2(1 - \gamma_\star)}{8 \cdot 10 c_{\mathrm{conc,id}}\Psi_\star^2\alpha_\star^2 d_{\mathbf{x}}}, \quad \text{and} \quad n_{\mathrm{id}} \geq c_0(d_{\mathbf{x}}^2 + \ln(1/\delta)),$$

suffice to satisfy the third condition of the proposition.

We conclude that when the conditions above hold, with probability at least $1 - 2\delta$,

$$\|\widehat{Q}_{\mathrm{id}} - Q_{\mathrm{id}}\|_{\mathrm{op}} \leq \|\widehat{Q}_{\mathrm{id}} - Q_{\mathrm{id}}\|_F$$
$$\lesssim \frac{\alpha_\star^2\Psi_\star^2\sqrt{c_{\mathrm{conc,id}}d_{\mathbf{x}}\ln(n_{\mathrm{id}}/\delta)}\varepsilon_{\mathrm{id},h}}{(1 - \gamma_\star)\sigma_{\min}(\mathcal{C}_\kappa)^2}\|Q_{\mathrm{id}}\|_{\mathrm{op}}.$$

Finally, we bound

$$\|Q_{\mathrm{id}}\|_{\mathrm{op}} = \|S_{\mathrm{id}}^{-1}Q_{\mathrm{id}}S_{\mathrm{id}}^{-1}\|_{\mathrm{op}} \leq \|Q_{\mathrm{id}}\|_{\mathrm{op}}\sigma_{\min}^{-2}(S_{\mathrm{id}})$$
$$= \Psi_\star\sigma_{\min}^{-2}(S_{\mathrm{id}})$$
$$\leq \Psi_\star \left(\frac{\sigma_{\min}(\mathcal{C}_\kappa)(1 - \gamma_\star)}{4\Psi_\star^2\alpha_\star^2}\right)^{-2}$$
$$\lesssim \frac{\Psi_\star^5\alpha_\star^4}{\sigma_{\min}(\mathcal{C}_\kappa)^2(1 - \gamma_\star)^2}.$$

Thus, altogether,

$$\|\widehat{Q}_{\mathrm{id}} - Q_{\mathrm{id}}\|_{\mathrm{op}} \leq \|\widehat{Q}_{\mathrm{id}} - Q_{\mathrm{id}}\|_F$$
$$\lesssim \frac{\alpha_\star^6\Psi_\star^7\sqrt{c_{\mathrm{conc,id}}d_{\mathbf{x}}\ln(n_{\mathrm{id}}/\delta)}}{(1 - \gamma_\star)^3\sigma_{\min}(\mathcal{C}_\kappa)^4}\varepsilon_{\mathrm{id},h}.$$

$\square$

### G.3.4 Concluding the Proof of Theorem 3a

In total, by combining Propositions G.3 and G.4 and conditioning on the probability $1 - 4\delta$ event from Propositions G.1 and G.2, we have that as long as (for some universal $c_0$),

$$\varepsilon_{\mathrm{id},h}\sqrt{\ln(2n_{\mathrm{id}}/\delta)} \leq \frac{\sigma_{d_{\mathbf{x}}}(\mathcal{C}_\kappa)^2(1-\gamma_\star)}{8 \cdot 10 c_{\mathrm{conc,id}}\Psi_\star^2\alpha_\star^2 d_{\mathbf{x}}},$$

$$\varepsilon_{\mathrm{id},h} \text{ satisifies Eq. (G.14)},$$

$$\text{and} \quad n_{\mathrm{id}} \geq c_0(d_{\mathbf{x}}^2 + \ln(1/\delta)),$$

then with total failure probability at most $1 - 9\delta - 4\delta$,

$$\|\widehat{Q}_{\mathrm{id}} - Q_{\mathrm{id}}\|_{\mathrm{op}} \lesssim \frac{\alpha_\star^6\Psi_\star^7\sqrt{c_{\mathrm{conc,id}}d_{\mathbf{x}}\ln(n_{\mathrm{id}}/\delta)}}{(1-\gamma_\star)^3\sigma_{\min}(\mathcal{C}_\kappa)^4}\varepsilon_{\mathrm{id},h},$$

$$\|[\widehat{A}_{\mathrm{id}};\widehat{B}_{\mathrm{id}}] - [A_{\mathrm{id}};B_{\mathrm{id}}]\|_{\mathrm{op}} \lesssim \frac{\Psi_\star^{7/2}\alpha_\star^3}{\sigma_{\min}(\mathcal{C}_\kappa)^2(1-\gamma_\star)^{3/2}}\varepsilon_{\mathrm{id},h},$$

$$\|\widehat{\Sigma}_{w,\mathrm{id}} - \Sigma_{w,\mathrm{id}}\|_{\mathrm{op}} \lesssim \frac{\Psi_\star^{5/2}\alpha_\star^4}{\sigma_{\min}(\mathcal{C}_\kappa)(1-\gamma_\star)}\varepsilon_{\mathrm{id},h}.$$

To simplify the conditions slightly, we observe that since $\varepsilon_{\mathrm{id},h} \geq \bar{c}_{\mathrm{id},1}\frac{d_{\mathbf{x}}^2}{n_{\mathrm{id}}}$ for some universal constant $\bar{c}_{\mathrm{id},1} \geq 4$, by inflating this constant, we can ensure that $n_{\mathrm{id}} \geq c_0(d_{\mathbf{x}}^2 + \ln(1/\delta))$. Next let us consolidate the conditions

$$\varepsilon_{\mathrm{id},h}\sqrt{\ln(2n_{\mathrm{id}}/\delta)} \leq \frac{\sigma_{d_{\mathbf{x}}}(\mathcal{C}_\kappa)^2(1-\gamma_\star)}{8 \cdot 10 c_{\mathrm{conc,id}}\Psi_\star^2\alpha_\star^2 d_{\mathbf{x}}}, \quad \text{and} \quad \varepsilon_{\mathrm{id},h} \text{ satisifies Eq. (G.14)}.$$

Restating Eq. (G.14), we require that

$$\varepsilon_{\mathrm{id},h}\sqrt{d_{\mathbf{x}}c_{\mathrm{conc,id}}} \leq \frac{(1-\gamma_\star)\sigma_{d_{\mathbf{x}}}(\mathcal{C}_\kappa)^2}{71\alpha_\star^2\Psi_\star^2}.$$

Since $c_{\mathrm{conc,id}}d_{\mathbf{x}} \geq 1$, it suffices to take

$$\varepsilon_{\mathrm{id},h}\sqrt{\ln(2n_{\mathrm{id}}/\delta)} \leq \frac{\sigma_{d_{\mathbf{x}}}(\mathcal{C}_\kappa)^2(1-\gamma_\star)}{80 c_{\mathrm{conc,id}}\Psi_\star^2\alpha_\star^2 d_{\mathbf{x}}}.$$

Recalling that $c_{\mathrm{conc,id}} = \frac{12L^2\Psi_\star^3\alpha_\star^2}{1-\gamma_\star}$, the final condition,

$$\varepsilon_{\mathrm{id},h}\sqrt{\ln(2n_{\mathrm{id}}/\delta)} \leq \frac{\sigma_{d_{\mathbf{x}}}(\mathcal{C}_\kappa)^2(1-\gamma_\star)^2}{80 \cdot 12L^2\Psi_\star^5\alpha_\star^4 d_{\mathbf{x}}}.$$

$\square$

## H  Proofs for RichID Phase III

**Section organization.**  This section is dedicated to the proof of Theorem 4, which is the main result concerning Phase III of RichID-CE (cf. Section 2.3). We state a number of intermediate results, leading up to the proof of theorem. In Appendix H.2, we present a performance bound for the state decoders $(\hat{f}_t)_{t\geq 1}$ as a function of the decoding error of the initial state decoder $\hat{f}_1$. Appendix H.3 is dedicated to the perfomance of $\hat{f}_1$, as this requires extra steps to the decode the initial state $\mathbf{x}_0$. In Appendix H.4, we combine these results to prove Theorem 4. Finally, Appendix I contains the proofs of all the intermediate results.

We recall that the definition of the decoders $(\hat{f}_\tau)$ requires a clipping step (see (14)). Performing this step allows us to use standard concentration tools to bound the decoding error for the predictors that come out of the regression problems solved in Phase III (see Lemma H.3). The impact of clipping on the prediction error is low: In Theorem 9 we show that the probability of ever clipping is very small, so long as the clipping parameter $\bar{b}$ is chosen appropriately.

## H.1 Preliminaries

Before proceeding to the main results, we first provide additional notation and definitions, as well as some basic lemmas which will be used in subsequent proofs.

**Additional notation.** For $t \geq 0$ and a policy $\pi : \bigcup_{\tau=1}^{\infty} \mathcal{Y}^{\tau} \to \mathbb{R}^{d_{\mathbf{u}}}$, where $\pi(\mathbf{y}_{0:t})$ maps past and current observations $\mathbf{y}_{0:t}$ to the current action $\mathbf{u}_t$, we let $\mathbb{P}_{\pi}$ and $\mathbb{E}_{\pi}$ be the probability and expectation with respect to the system's dynamics and policy $\pi$. We will use $\mathcal{E}$ to denote events which hold over the randomness in the learning procedure, and $\mathscr{E}$ to denote events which hold under a given rollout from, say, $\mathbb{E}_{\pi}$.

Throughout this section we let $\widehat{\pi}$ denote the policy returned by Algorithm 5.

**Basic definitions for Phase III.** To simplify presentation, we assume going forward that $S_{\mathrm{id}} = I_{d_{\mathbf{x}}}$ at the cost of increasing problem-dependent parameters such as $\Psi_{\star}$ and $\alpha_{\star}$ by a factor of $\|S_{\mathrm{id}}\|_{\mathrm{op}} \vee \|S_{\mathrm{id}}^{-1}\|_{\mathrm{op}}$—we make this reasoning precise in the proof of Theorem 1a. We therefore drop the subscript id, so that the system parameters we take as a given are $(\widehat{A}, \widehat{B}, \widehat{Q}, \widehat{\Sigma}_w)$. We will consider the following function class

$$\mathcal{H}_{\mathrm{op}} := \left\{ M \cdot f(\cdot) \mid f \in \mathcal{F}, \ M \in \mathbb{R}^{d_{\mathbf{x}} \times d_{\mathbf{x}}}, \ \|M\|_{\mathrm{op}} \leq \Psi_{\star}^3 \right\}, \tag{H.1}$$

that is, we take $r_{\mathrm{op}} = \Psi_{\star}^3$ (note that the final value for $r_{\mathrm{op}}$ when Algorithm 5 is invoked within Algorithm 1 will be inflated to account for the similarity transformation above).

In what follows, we will construct a sequence of functions $(\hat{f}_t : \mathcal{Y}^{t+1} \to \mathbb{R}^{d_{\mathbf{x}}})$ which map observations $(\mathbf{y}_{0:t})$ to estimates of the true states $(\mathbf{x}_t = f_{\star}(\mathbf{y}_t))$. We will denote by $\widehat{\pi}$ the randomized policy defined by $\widehat{\pi}(\mathbf{y}_{0:t}) := \widehat{K}\hat{f}_t(\mathbf{y}_{0:t}) + \mathbf{v}_t$, for all $t \geq 0$, where $\mathbf{v}_t \sim \mathcal{N}(0, \sigma^2 I_{d_{\mathbf{u}}})$ for some $\sigma \in (0, 1]$ to be determined later. Furthermore, for $t \geq 0$, we define the policy $\widetilde{\pi}_t$ which satisfies

$$\widetilde{\pi}_t(\mathbf{y}_{0:\tau}) = \begin{cases} \widehat{\pi}(\mathbf{y}_{0:\tau}), & \text{if } \tau \leq t; \\ \mathbf{v}_{\tau} \sim \mathcal{N}(0, \sigma^2 I_{d_{\mathbf{u}}}), & \text{otherwise.} \end{cases} \tag{H.2}$$

**Additional problem parameters.** Our final results for this section are stated in $\mathcal{O}_{\star}(\cdot)$, but we state many of our intermediate results with precise dependence on the problem parameters. To simplify these statements, we use the following definitions.

**Definition 7** (Aggregated problem parameters)**.**

$$\Psi_{\Sigma} := \alpha_A^2 \|\Sigma_0\|_{\mathrm{op}} + \|\Sigma_{\infty,\mathrm{id}}\|_{\mathrm{op}} + 1, \tag{H.3}$$

$$\mathsf{dev}_x := \frac{3\alpha_A^2}{1 - \gamma_A} \cdot \Psi_{\star}^2 \|\widehat{K}\|_{\mathrm{op}}^2, \tag{H.4}$$

$$\Psi_M := \max \left\{ 1, \varepsilon_{\mathrm{sys}}, \|\mathcal{M}\|_{\mathrm{op}}, \max_{k \in \kappa} \|M_k\|_{\mathrm{op}} \right\},$$

$$L_{\mathrm{op}} := \Psi_{\star}^3 L. \tag{H.5}$$

We simplify our intermediate results to get the final $\mathcal{O}_{\star}(\cdot)$-based bound for Theorem 4 in Appendix H.4.

### H.1.1 Approximation Error for Plug-In Estimators.

Recall that Phase III uses the estimates for $\widehat{A}$, $\widehat{B}$, and so forth from Phase II to form plug-in estimates for a number of important system parameters. Before proceeding, we give some guarantees on the error of these estimates as a function of the error from Phase II.

For $k \in [\kappa]$, recall that we define the matrices $M_k \in \mathbb{R}^{k d_{\mathbf{u}} \times d_{\mathbf{x}}}$ and $\mathcal{M} \in \mathbb{R}^{(\kappa+1)\kappa/2 \times \kappa}$ by

$$M_k := \mathcal{C}_k^{\top} (\mathcal{C}_k \mathcal{C}_k^{\top} + \sigma^{-2}\Sigma_w + \cdots + \sigma^{-2}A^{k-1}\Sigma_w (A^{k-1})^{\top})^{-1},$$

$$\mathcal{M} := [M_1^{\top}, (M_2 A)^{\top}, \ldots, (M_{\kappa} A^{k-1})^{\top}]^{\top} \quad \text{where} \quad \mathcal{C}_k := [A^{k-1}B \mid \ldots \mid B] \in \mathbb{R}^{d_{\mathbf{x}} \times k d_{\mathbf{u}}}. \tag{H.6}$$

We also let $\widehat{M}_k$ and $\widehat{\mathcal{M}}$ be the *plug-in estimators* of $M_k$, and $\mathcal{M}$ respectively, obtained by replacing $A$, $B$, and $\Sigma_w$ in the definitions of $M_k$ and $\mathcal{M}$ by the previously derived estimators $\widehat{A}$, $\widehat{B}$, and $\widehat{\Sigma}_w$, respectively (see Section 2.2).

For $0 < \varepsilon_{\text{sys}} \le 1 \wedge \|\Sigma_w\|_{\text{op}} \wedge \|\Sigma_w\|_{\text{op}}^{-1}$, let $\mathcal{E}_{\text{sys}}$ be the event

$$\mathcal{E}_{\text{sys}} \coloneqq \left\{ \max_{k \in [\kappa]} \left\{ \begin{array}{c} \|\widehat{K} - K_\infty\|_{\text{op}}, \ \|\widehat{M_k}\widehat{A}^k - M_k A^k\|_{\text{op}}, \\ \|\widehat{M_k}\widehat{A}^k\widehat{B} - M_k A^k B\|_{\text{op}}, \ \|\widehat{B}\widehat{K} - B\widehat{K}\|_{\text{op}}, \\ \|I_{d_{\text{x}}} - \widehat{\Sigma}_w \Sigma_w^{-1}\|_{\text{op}}, \ \|\widehat{\Sigma}_w^{-1} - \Sigma_w^{-1}\|_{\text{op}}, \\ \|\widehat{A} - A\|_{\text{op}}, \ \|\widehat{B} - B\|_{\text{op}}, \ \|\widehat{\mathcal{M}} - \mathcal{M}\|_{\text{op}} \end{array} \right\} \le \varepsilon_{\text{sys}} \right\} \bigcap \mathcal{E}_{\text{stab}}, \tag{H.7}$$

where

$$\mathcal{E}_{\text{stab}} \coloneqq \left\{ \begin{array}{c} (A + B\widehat{K}) \text{ is } (\alpha_\infty, \bar{\gamma}_\infty)\text{-strongly stable with } \bar{\gamma}_\infty \coloneqq (1 + \gamma_\infty)/2, \\ \text{and } \widehat{A} \text{ is } (\alpha_A, \bar{\gamma}_A)\text{-strongly stable with } \bar{\gamma}_A \coloneqq (1 + \gamma_A)/2. \end{array} \right\}. \tag{H.8}$$

**Lemma H.1.** Suppose that $\sigma^2 = \mathcal{O}_\star(1)$ and

$$\left\|\widehat{A} - A\right\|_{\text{op}} \vee \left\|\widehat{B} - B\right\|_{\text{op}} \vee \left\|\widehat{Q} - Q\right\|_{\text{op}} \vee \left\|\widehat{\Sigma}_w - \Sigma_w\right\|_{\text{op}} \le \varepsilon_{\text{id}}.$$

Then once $\varepsilon_{\text{id}} \le c_{\text{sys}} \coloneqq \text{poly}(\gamma_\star(1 - \gamma_\star), \alpha_\star^{-1}, \Psi_\star^{-1}) = \check{\mathcal{O}}(1)$, we have that $\mathcal{E}_{\text{sys}}$ holds for

$$\varepsilon_{\text{sys}} \le \mathcal{O}_\star(\varepsilon_{\text{id}}).$$

### H.1.2 Conditioning for the $\mathcal{M}$ Matrix

Assumption 8 is central to the results in this section. In particular, we will use the following implication of this assumption.

**Lemma H.2.** Let $\mathcal{M}_{\sigma^2}$ denote the value of the matrix $\mathcal{M}$ in (H.6) for noise parameter $\sigma^2$. Then $\|\mathcal{M}_{\sigma^2}\|_{\text{op}} = \mathcal{O}_\star(1)$ whenever $\sigma^2 = \mathcal{O}_\star(1)$. Moreover, suppose Assumption 8 holds. Then there exists $\bar{\sigma} = \check{\mathcal{O}}(\lambda_{\mathcal{M}})$, such that for all $\sigma^2 \le \bar{\sigma}^2$, we have

$$\lambda_{\min}^{1/2}(\mathcal{M}_{\sigma^2}^\top \mathcal{M}_{\sigma^2}) \ge \lambda_{\mathcal{M}} \cdot \sigma^2/2 > 0, \tag{H.9}$$

where $\lambda_{\mathcal{M}}$ is as in Assumption 8.

Throughout this section, we make the following assumption, which will eventually be justified by the choice of $\sigma$ in RichID-CE.

**Assumption 10.** $\sigma \le 1$ is sufficiently small such that Eq. (H.9) holds.

In particular, this assumption implies that the matrix $\mathcal{M}$ in (H.6) has full row rank.

## H.2 Learning State Decoders for Rounds $t \ge 1$

We now prove that Phase III successfully learns decoders for $t \ge 1$ with high probability, up to an error term determined by the auxiliary predictor $\hat{f}_{A,0}$ produced during the separate initial state learning phase; the error of this predictor is handled in the next subsection. For the rest of this subsection, we assume the iteration $t \ge 0$ of Algorithm 5 is fixed, meaning we already have $\hat{f}_t$ and our goal is to compute $\hat{f}_{t+1}$. We introduce the following quantities.

- Let $(\mathbf{y}_\tau)_{\tau \ge 0}$ be the observations induced by following the policy $\widetilde{\pi}_t$ defined in (H.2).
- Let $\{(\mathbf{y}_\tau^{(i)}, \mathbf{x}_\tau^{(i)}, \mathbf{v}_\tau^{(i)})\}_{i \in [2n]}$ be i.i.d. copies of $(\mathbf{y}_\tau, \mathbf{x}_\tau, \mathbf{v}_\tau)$, where $(\mathbf{v}_\tau)$ are the random Gaussian vectors used by the policy $\widetilde{\pi}_t$. This is simply the data collected by the $t$th iteration of the loop in Algorithm 5.

We also adopt the shorthand $n = n_{\text{op}}$.

**Learning the decoder at a single step.** Let us recall some notation. For round $t$, we already have a state decoder $\hat{f}_t \colon \mathcal{Y}^{t+1} \to \mathbb{R}^{d_{\text{x}}}$ produced by the previous iteration. As the first step, for each $k \in [\kappa]$, with $\mathscr{H}_{\text{op}}$ as in (H.1), Algorithm 5 solves

$$\hat{h}_{t,k} \in \operatorname*{arg\,min}_{h \in \mathscr{H}_{\text{op}}} \sum_{i=1}^n \left\| \widehat{M_k} \left( h(\mathbf{y}_{t+k}^{(i)}) - \widehat{A}^k h(\mathbf{y}_t^{(i)}) - \widehat{A}^{k-1}\widehat{B}\widehat{K}\hat{f}_t(\mathbf{y}_{0:t}^{(i)}) \right) - \mathbf{v}_{t:t+k-1}^{(i)} \right\|^2. \tag{H.10}$$

Using the solutions of the above regressions for $k \in [\kappa]$, the algorithm constructs the stacked vector

$$\widehat{\phi}_t(\mathbf{y}_{0:t+\kappa}) := [\widehat{\phi}_{t,1}(\hat{h}_{t,1}, \mathbf{y}_{0:t}, \mathbf{y}_{t+1})^\top, \ldots, \widehat{\phi}_{t,\kappa}(\hat{h}_{t,\kappa}, \mathbf{y}_{0:t}, \mathbf{y}_{t+\kappa})^\top]^\top \in \mathbb{R}^{(1+\kappa)\kappa/2},$$

where

$$\widehat{\phi}_{t,k}(h, \mathbf{y}_{0:t}, \mathbf{y}_{t+k}) := \widehat{M}_k \left( h(\mathbf{y}_{t+k}) - \widehat{A}^k h(\mathbf{y}_t) - \widehat{A}^{k-1} \widehat{B}\widehat{K} \hat{f}_t(\mathbf{y}_{0:t}) \right), \ k \in [\kappa]. \tag{H.11}$$

Finally, the algorithm computes the intermediate estimator $\hat{h}_t$:

$$\hat{h}_t \in \underset{h \in \mathscr{H}_{\mathrm{op}}}{\arg\min} \sum_{i=n+1}^{2n} \left\| \widehat{\mathcal{M}} \left( h(\mathbf{y}_{t+1}^{(i)}) - \widehat{A}h(\mathbf{y}_t^{(i)}) - \widehat{B}\widehat{K}\hat{f}_t(\mathbf{y}_{0:t}^{(i)}) \right) - \widehat{\phi}_t(\mathbf{y}_{0:t+\kappa}^{(i)}) \right\|^2. \tag{H.12}$$

Our first guarantee for this section shows that the function $\hat{h}_t$ estimates the system's noise $\mathbf{w}_t$ up to a linear transformation given by the matrix $\mathcal{M}$.

**Theorem 8.** *Let $t \geq 0$ and $\bar{b} > 0$ be given. For $h \in \mathscr{H}_{\mathrm{op}}$ and $\hat{f}_t : \mathcal{Y}^{t+1} \to \mathcal{X}$, let*

$$\phi_t(h, \mathbf{y}_{0:t+1}) := \mathcal{M}(h(\mathbf{y}_{t+1}) - Ah(\mathbf{y}_t) - B\widehat{K}\hat{f}_t(\mathbf{y}_{0:t})).$$

*If the event $\mathcal{E}_{\mathrm{sys}}$ holds and $\|\hat{f}_t(\mathbf{y}_{0:t})\| \leq \bar{b}$ a.s., then for $\hat{h}_t$ as in (H.12), with probability at least $1 - \delta$,*

$$\mathbb{E}_{\widehat{\pi}} \left[ \left\| \phi_t(\hat{h}_t, \mathbf{y}_{0:t+1}) - \mathcal{M}(\mathbf{w}_t + B\mathbf{v}_t) \right\|^2 \right] \leq \varepsilon_{\mathrm{noise}}^2(\delta), \tag{H.13}$$

*where*

$$\varepsilon_{\mathrm{noise}}^2(\delta) \lesssim \kappa(1 + \ln(\kappa)) \left( \frac{c_{w,\phi}(\ln|\mathscr{F}| + d_{\mathbf{x}}^2)\ln^2(n\kappa/\delta)}{n} + L_{\mathrm{op}}^2 \varepsilon_{\mathrm{sys}}^2 \left( d_{\mathbf{x}}\Psi_\Sigma + \mathsf{dev}_x \bar{b}^2 \right) \right) \tag{H.14}$$

*with*

$$c_{w,\phi} := 30\kappa d_{\mathbf{u}}\sigma^2 + 18\alpha_A^2 L_{\mathrm{op}}^2 \Psi_M^2 \left( 32 d_{\mathbf{x}}\Psi_\Sigma + 3\bar{b}^2 \cdot \mathsf{dev}_x \right). \tag{H.15}$$

For the remainder of the subsection, we let $\delta \in (0, e^{-1}]$ be fixed and define

$$\mathcal{E}_{\mathrm{noise}} := \left\{ \mathbb{E}_{\widehat{\pi}} \left[ \left\| \phi_\tau(\hat{h}_\tau, \mathbf{y}_{0:\tau+1}) - \mathcal{M}(\mathbf{w}_\tau + B\mathbf{v}_\tau) \right\|^2 \right] \leq \varepsilon_{\mathrm{noise}}^2(\delta), \text{ for all } 0 \leq \tau \leq t \right\}. \tag{H.16}$$

**From noise estimate to state estimate.** Now that we can estimate the noise at round $t$ using $\hat{h}_t$, we build a state decoder $\hat{f}_{t+1}$ for round $t+1$ by combining $\hat{h}_t$ with the decoder $\hat{f}_t$. Recall that $\tilde{f}_0 \equiv \hat{f}_0 \equiv 0$, and that Algorithm 5 forms $\hat{f}_t$ for all $t \geq 1$ via

$$\hat{f}_{t+1}(\cdot) := \tilde{f}_{t+1}(\cdot)\mathbb{I}\{\|\tilde{f}_{t+1}(\cdot)\| \leq \bar{b}\}, \text{ where } \tilde{f}_{t+1}(\mathbf{y}_{0:t+1}) := \hat{h}_t(\mathbf{y}_{t+1}) + \widehat{A}\hat{f}_t(\mathbf{y}_{0:t}) - \widehat{A}\hat{h}_t(\mathbf{y}_t), \tag{H.17}$$

where $\bar{b} > 0$ is the clipping parameter. Note that we treat $\bar{b}$ as a free parameter throughout this section unless explicitly specified. The case $t = 0$ needs special care as it requires decoding the initial state; we set

$$\hat{f}_1(\cdot) := \tilde{f}_1(\cdot)\mathbb{I}\{\|\tilde{f}_1(\cdot)\| \leq \bar{b}\}, \text{ where } \tilde{f}_1(\mathbf{y}_{0:1}) := \hat{h}_1(\mathbf{y}_1) + \hat{f}_{A,0}(\mathbf{y}_0) - \widehat{A}\hat{h}_0(\mathbf{y}_0), \tag{H.18}$$

and $\hat{f}_{A,0}(\mathbf{y}_0)$ is the estimator for $Af_\star(\mathbf{y}_0)$ which we will construct in the next subsection.

Our goal now is to prove that the $\hat{f}_{t+1}$ is good whenever $\hat{h}_0, \ldots, \hat{h}_t$ are good. To this end, we first give a guarantee on the unprojected decoder $\tilde{f}_{t+1}$, which shows that it has low prediction error for trajectories in which the event

$$\mathcal{E}_{0:t} := \left\{ \tilde{f}_\tau(\mathbf{y}_{0:\tau}) = \hat{f}_\tau(\mathbf{y}_{0:\tau}), \text{ for all } 0 \leq \tau \leq t \right\} \tag{H.19}$$

occurs.

**Lemma H.3.** *Let $t \geq 0$ be given. Let $(\tilde{f}_\tau)_{\tau \in [t+1]}$ and $\mathcal{E}_{0:t}$ be defined as in (H.17), and (H.19), respectively. If the events $\mathcal{E}_{\mathrm{sys}}$ and $\mathcal{E}_{\mathrm{noise}}$ hold, then for $\varepsilon_{\mathrm{noise}}$ as in (H.14), we have*

$$\mathbb{E}_{\widehat{\pi}} \left[ \max_{0 \leq \tau \leq t} \|\tilde{f}_{\tau+1}(\mathbf{y}_{0:\tau+1}) - f_\star(\mathbf{y}_{\tau+1})\|^2 \cdot \mathbb{I}\{\mathcal{E}_{0:t}\} \right] \leq \varepsilon_{\mathrm{dec},t}^2, \tag{H.20}$$

*where*

$$\varepsilon_{\mathrm{dec},t}^2 := 3\alpha_A^2(1 - \gamma_A)^{-2} \left( \varepsilon_{\mathrm{sys}}^2 \bar{b}^2 + \varepsilon_{\mathrm{init}}^2 + \sigma_{\min}(\mathcal{M})^{-2}\varepsilon_{\mathrm{noise}}^2 t \right), \text{ and } \varepsilon_{\mathrm{init}}^2 := \mathbb{E}_{\widehat{\pi}}[\|\hat{f}_{A,0}(\mathbf{y}_0) - Af_\star(\mathbf{y}_0)\|^2]. \tag{H.21}$$

For the next theorem, we show that the even $\mathscr{E}_{0:t}$ occurs with overwhelming probability whenever the clipping parameter $\bar{b}$ is selected appropriately. We need the following definitions. For $t \geq 0$ and $\eta > 0$, let

$$z_t \coloneqq \sum_{\tau=0}^{t} (A + B\widehat{K})^{t-\tau} (B\mathbf{v}_\tau + \mathbf{w}_\tau)$$

denote the contribution of the process noise and Gaussian inputs to the state $\mathbf{x}_{t+1}$. The associated covariance of this random variable when $t \to \infty$ is given by

$$\Sigma_{z,\infty} \coloneqq \sum_{\tau=0}^{\infty} (A + B\widehat{K})^\tau (\sigma^2 BB^\top + \Sigma_w)((A + B\widehat{K})^\tau)^\top. \tag{H.22}$$

The sum in (H.22) converges under the event $\mathcal{E}_{\text{sys}}$, since in this case $\|(A + B\widehat{K})^t\|_{\text{op}} \leq \alpha_\infty \bar{\gamma}_\infty^t$, for all $t \geq 0$, and $\bar{\gamma}_\infty < 1$; see Eq. (H.8). Finally, we consider the following useful event:

$$\mathscr{E}_{0:t}' \coloneqq \left\{ \alpha_\infty^2 \|\mathbf{x}_0\|_2^2 + \|z_\tau\|^2 \leq (d_{\mathbf{u}} \, \alpha_\infty^2 \|\Sigma_0\|_{\text{op}} + d_{\mathbf{x}} \|\Sigma_{z,\infty}\|_{\text{op}}) \ln(2\eta), \ \text{for all } 0 \leq \tau \leq t \right\}.$$

Lastly, we define the following term which guides how we select the clipping in the definition of $(\hat{f}_t)$ in (H.17):

$$\bar{b}_\infty \coloneqq \frac{6(1 - \gamma_\infty)^{-1} \alpha_\infty \Psi_\star \varepsilon_{\text{dec},t} \sqrt{\eta} + \sqrt{2(d_{\mathbf{u}} \alpha_\infty^2 \|\Sigma_0\|_{\text{op}} + d_{\mathbf{x}} \|\Sigma_{z,\infty}\|_{\text{op}}) \ln(2\eta)}}{1 - 2\alpha_\infty \varepsilon_{\text{sys}} (1 - \gamma_\infty)^{-1}}, \tag{H.23}$$

where $\eta > e$ is a free parameter.

We now show that if the clipping parameter $\bar{b}$ in (H.17) is chosen sufficiently large, then under a given execution of $\widehat{\pi}$, the clipping operator is never actived (i.e. $\mathscr{E}_{0:t}$ holds) with high enough probability, provided that the clipping operator is not activated at $t = 1$.

**Theorem 9.** *Let $t \geq 0$, $\eta > 0$, and $\bar{b} > 0$ be given. Let $\varepsilon_{\text{dec},t}$, $\Sigma_{z,\infty}$, and $\bar{b}_\infty$ be defined as in (H.21), (H.22), and (H.23), respectively. If* (**I**) *the events $\mathcal{E}_{\text{sys}}$ and $\mathcal{E}_{\text{noise}}$ hold;* (**II**) *$\varepsilon_{\text{sys}} < (1 - \gamma_\infty)(2\alpha_\infty)^{-1}$; and* (**III**) *$\bar{b} \geq \bar{b}_\infty$, then*

$$\mathbb{P}_{\widehat{\pi}}[\mathscr{E}_{0:t} \wedge \mathscr{E}_{0:t}'] \geq \mathbb{P}_{\widehat{\pi}}[\tilde{f}_1(\mathbf{y}_{0:1}) = \hat{f}_1(\mathbf{y}_{0:1})] - 2(t+1)/\eta.$$

**Concluding the guarantee for the state decoders.** We now put together the preceding results to give the main guarantee for our state decoders $(\hat{f}_t)$ for $t \geq 1$.

**Theorem 10.** *Let $T \geq 0$, $\eta > 0$, and $\bar{b} > 0$ be given. Under the conditions* (**I**), (**II**), *and* (**III**) *of Theorem 9, we have*

$$\mathbb{E}_{\widehat{\pi}}\left[ \max_{0 \leq t \leq T} \|\hat{f}_t(\mathbf{y}_{0:t}) - f_\star(\mathbf{y}_t)\|^2 \right] \leq \varepsilon_{\text{dec},t}^2 + (4T^{1/2}c_{\mathbf{x}} + 2\bar{b}^2)\left( \frac{4T}{\eta} + 1 - \mathbb{P}_{\widehat{\pi}}[\{\tilde{f}_0(\mathbf{y}_0) = \hat{f}_0(\mathbf{y}_0)\} \wedge \mathscr{E}_0'] \right),$$

*where $c_{\mathbf{x}} \coloneqq 30 d_{\mathbf{x}} \Psi_\Sigma + 2\bar{b}^2 \cdot \mathsf{dev}_x$.*

## H.3 Learning the Initial State

Theorem 10 ensures that the decoders $\hat{f}_0, \ldots, \hat{f}_T$ have low error only if the initial error $\varepsilon_{\text{init}}^2 \coloneqq \mathbb{E}_{\widehat{\pi}}[\|\hat{f}_{A,0}(\mathbf{y}_0) - A f_\star(\mathbf{y}_0)\|^2]$ is small. In this subsection, we show that the extra initial state learning procedure in Algorithm 5 ensures that this happens with high probability.

Recall that $n_{\text{init}} \in \mathbb{N}$ denotes the sample size used by Algorithm 5 for learning the initial state. During the initial state learning phase (Line 19 through Line 25 of Algorithm 5), the algorithm gathers data by following a policy we denote $\pi_{\text{ol}}$ which plays random noise $(\mathbf{v}_\tau)$, where $\mathbf{v}_\tau \sim \mathcal{N}(0, \sigma^2 I_{d_{\mathbf{u}}})$, for $\tau \geq 0$ and $\sigma \in (0, 1]$.

Let $\hat{h}_{\text{ol},0} \coloneqq \hat{h}_0$ (recall that the "ol" subscript refers to *open loop*), where we recall that $\hat{h}_0$ is computed on Line 17 of Algorithm 5 prior to the initial state learning phase, using the procedure analyzed Appendix H.2. In particular, by instantiating the result of Theorem 8 with $\hat{f}_0 \equiv 0$, we get that under the event $\mathcal{E}_{\text{sys}}$, for any $\delta \in (0, 1/e]$, with probability at least $1 - \delta$,

$$\mathbb{E}_{\pi_{\text{ol}}}\left[ \left\| \hat{h}_{\text{ol},0}(\mathbf{y}_1) - A\hat{h}_{\text{ol},0}(\mathbf{y}_0) - B\mathbf{v}_0 - \mathbf{w}_0 \right\|^2 \right] \leq \sigma_{\min}(\mathcal{M})^{-2} \cdot \varepsilon_{\text{noise}}^2(\delta). \tag{H.24}$$

We recall that the minimum singular value of $\mathcal{M}$ is bounded away from zero for all sufficiently small $\sigma > 0$ (see Lemma H.2). It follows from (H.24) that $\hat{h}_{\mathrm{ol},0}(\mathbf{y}_1) - A\hat{h}_{\mathrm{ol},0}(\mathbf{y}_0) - B\mathbf{v}_0$ can be used as an estimator for the noise vector $\mathbf{w}_0$. Using this estimator, we solve the following regression problem in Line 21:

$$\hat{h}_{\mathrm{ol},1} \in \underset{h \in \mathscr{H}_{\mathrm{op}}}{\arg\min} \sum_{i=1}^{n_{\mathrm{init}}} \left\| h(\mathbf{y}_1^{(i)}) - \left( \hat{h}_{\mathrm{ol},0}(\mathbf{y}_1^{(i)}) - \widehat{A}\hat{h}_{\mathrm{ol},0}(\mathbf{y}_0^{(i)}) - \widehat{B}\mathbf{v}_0^{(i)} \right) \right\|^2, \qquad (\text{H.25})$$

where $\{(\mathbf{y}_\tau^{(i)}, \mathbf{x}_\tau^{(i)}, \mathbf{v}_\tau^{(i)})\}_{1 \le i \le n_{\mathrm{init}}}$, are the fresh i.i.d. trajectories generated by the policy $\pi_{\mathrm{ol}}$ on Line 19.

We first show that up to a linear transformation, this regression recovers the vector $A\mathbf{x}_0$ (our target), plus a linear combination $B\mathbf{v}_0 + \mathbf{w}_0$ of the system noise and injected noise for $t = 0$. This guarantee is quite useful: Since we can already predict $B\mathbf{v}_0 + \mathbf{w}_0$ well via Eq. (H.24), we will be able to extract $A\mathbf{x}_0$ from this representation.

**Lemma H.4.** Let $\hat{h}_{\mathrm{ol},1}$ be defined as in (H.25), and let $\Sigma_1 \coloneqq \sigma^2 BB^\top + A\Sigma_0 A^\top + \Sigma_w$. If $\mathcal{E}_{\mathrm{sys}}$ holds, then for any $\delta \in (0, 1/e]$, with probability at least $1 - 5\delta/2$, we have

$$\mathbb{E}_{\pi_{\mathrm{ol}}} \left[ \| \hat{h}_{\mathrm{ol},1}(\mathbf{y}_1) - \Sigma_w \Sigma_1^{-1}(\mathbf{w}_0 + B\mathbf{v}_0 + A\mathbf{x}_0) \|^2 \right] \le \varepsilon_{\mathrm{ol},1}^2, \qquad (\text{H.26})$$

where we have

$$\varepsilon_{\mathrm{ol},1}^2 \lesssim \frac{c_1(d_{\mathbf{x}}^2 + \ln|\mathscr{F}|) \ln^2 \frac{n_{\mathrm{init}}}{\delta}}{n_{\mathrm{init}}} + \sigma_{\min}(\mathcal{M})^{-2} \varepsilon_{\mathrm{noise}}^2(\delta) + \varepsilon_{\mathrm{sys}}^2 L_{\mathrm{op}}^2 (1 + d_{\mathbf{x}} \|\Sigma_0\|_{\mathrm{op}} + \sigma^2 d_{\mathbf{u}}),$$

and

$$c_1 \coloneqq L_{\mathrm{op}}^2 \Psi_\star^2 (1 + d_{\mathbf{u}}\sigma^2 + d_{\mathbf{x}}(\|\Sigma_1\|_{\mathrm{op}} + \|\Sigma_0\|_{\mathrm{op}})). \qquad (\text{H.27})$$

To make use of this lemma, we must invert the linear transformation $\Sigma_w \Sigma_1^{-1}$. In fact, the prediction error guarantee from Lemma H.4 implies that we can estimate $\Sigma_{\mathrm{cov}} \coloneqq \Sigma_w \Sigma_1^{-1} \Sigma_w$ (where $\Sigma_1$ is as in Lemma H.4) by computing

$$\widehat{\Sigma}_{\mathrm{cov}} \coloneqq \frac{1}{n} \sum_{i=n_{\mathrm{init}}+1}^{2n_{\mathrm{init}}} \hat{h}_{\mathrm{ol},1}(\mathbf{y}_1^{(i)}) \hat{h}_{\mathrm{ol},1}(\mathbf{y}_1^{(i)})^\top, \qquad (\text{H.28})$$

where $\{(\mathbf{y}_\tau^{(i)}, \mathbf{x}_\tau^{(i)}, \mathbf{v}_\tau^{(i)})\}_{n_{\mathrm{init}} < i \le 2n_{\mathrm{init}}}$, are fresh i.i.d. trajectories generated by the policy $\pi_{\mathrm{ol}}$. To see this, observe that by (H.26) implies that up to the error $\varepsilon_{\mathrm{ol},1}$, (H.28) is an estimator for the covariance matrix of the Gaussian vector $\Sigma_w \Sigma_1^{-1}(\mathbf{w}_0 + B\mathbf{v}_0 + A\mathbf{x}_0)$ which is just $\Sigma_w \Sigma_1^{-1} \Sigma_w$. The following lemma gives a guarantee for the estimated covariance $\widehat{\Sigma}_{\mathrm{cov}}$.

**Lemma H.5.** Let $c_{\mathrm{cov}} \coloneqq L_{\mathrm{op}}^2 (1 + (3d_{\mathbf{x}} + 2)\|\sigma^2 BB^\top + A\Sigma_0 A^\top + \Sigma_w\|_{\mathrm{op}})$ and

$$\varepsilon_{\mathrm{cov}}' \coloneqq 3\varepsilon_{\mathrm{ol},1}\sqrt{c_{\mathrm{cov}}} + 5c_{\mathrm{cov}}\ln(2d_{\mathbf{x}} n_{\mathrm{init}}/\delta)^{3/2} n_{\mathrm{init}}^{-1/2}. \qquad (\text{H.29})$$

Suppose that $n_{\mathrm{init}}$ is large enough such that

$$\varepsilon_{\mathrm{cov}}' < \sigma_{\min}(\Sigma_{\mathrm{cov}})/2, \quad \text{where} \quad \Sigma_{\mathrm{cov}} \coloneqq \Sigma_w \Sigma_1^{-1} \Sigma_w \preceq \Sigma_w, \qquad (\text{H.30})$$

and $\Sigma_1$ is as in Lemma H.4. Then under the event $\mathcal{E}_{\mathrm{sys}}$, with probability at least $1 - (3\kappa + 4)\delta$,

$$\|I_{d_{\mathbf{x}}} - \widehat{\Sigma}_w \widehat{\Sigma}_{\mathrm{cov}}^{-1} \Sigma_w \Sigma_1^{-1}\|_{\mathrm{op}} \le \varepsilon_{\mathrm{cov}}, \quad \|\widehat{\Sigma}_{\mathrm{cov}}\|_{\mathrm{op}} \le 2\|\Sigma_{\mathrm{cov}}\|_{\mathrm{op}}, \quad \text{and} \quad \sigma_{\min}(\widehat{\Sigma}_{\mathrm{cov}}) \ge \sigma_{\min}(\Sigma_{\mathrm{cov}})/2 \qquad (\text{H.31})$$

where

$$\varepsilon_{\mathrm{cov}} \coloneqq 2\Psi_\star \left( \varepsilon_{\mathrm{sys}} + \frac{2\|\Sigma_w^{-1}\|_{\mathrm{op}} \varepsilon_{\mathrm{cov}}'}{\sigma_{\min}(\Sigma_w \Sigma_1^{-1} \Sigma_w)} \right). \qquad (\text{H.32})$$

Lemma H.5 shows that $\widehat{\Sigma}_w \widehat{\Sigma}_{\mathrm{cov}}^{-1} \approx (\Sigma_w \Sigma_1^{-1})^{-1}$, which is exactly what we require to invert the linear transformation in Eq. (H.26). To finish up, we solve the regression problem (Line 23)

$$\tilde{h}_{\mathrm{ol},0} \in \underset{h \in \mathscr{H}_{\mathrm{op}}}{\arg\min} \sum_{i=n_{\mathrm{init}}+1}^{2n_{\mathrm{init}}} \left\| h(\mathbf{y}_0^{(i)}) - \hat{h}_{\mathrm{ol},1}(\mathbf{y}_1^{(i)}) \right\|^2. \qquad (\text{H.33})$$

Note that the argument to $h$ in Eq. (H.33) is $\mathbf{y}_0$, while the argument to $\hat{h}_{\mathrm{ol},1}$ is $\mathbf{y}_1$, so that the Bayes predictor, by Eq. (B.6), is equal to $h(\mathbf{y}_0) = \Sigma_w \Sigma_1^{-1} A \mathbf{x}_0$. Motivated by this observation, the final step is to set

$$\hat{f}_{A,0}(\mathbf{y}_0) = \widehat{\Sigma}_w \widehat{\Sigma}_{\mathrm{cov}}^{-1} \tilde{h}_{\mathrm{ol},0}(\mathbf{y}_0).$$

Our main theorem for this subsection gives the desired prediction error guarantee for this predictor.

**Theorem 11.** *Let $\tilde{h}_{\mathrm{ol},0}$ be as in Eq. (H.33), and set $\hat{f}_{A,0}(\mathbf{y}_0) \coloneqq \widehat{\Sigma}_w \widehat{\Sigma}_{\mathrm{cov}}^{-1} \tilde{h}_{\mathrm{ol},0}(\mathbf{y}_0)$. If $\mathcal{E}_{\mathrm{sys}}$ holds and Eq. (H.30) is satisfied, then for any $\delta \in (0, 1/e]$, with probability at least $1 - (3\kappa + 9)\delta$, the following properties hold:*

1. *The estimator $\hat{f}_{A,0}$ satisfies*

$$\mathbb{E}_{\widehat{\pi}}\left[\left\|\hat{f}_{A,0}(\mathbf{y}_0) - A f_\star(\mathbf{y}_0)\right\|^2\right] \lesssim \varepsilon_{\mathrm{init}}^2, \tag{H.34}$$

   *where*

$$\varepsilon_{\mathrm{init}}^2 \coloneqq \|\Sigma_w^{-1}\|_{\mathrm{op}}^2 \|\Sigma_{\mathrm{cov}}\|_{\mathrm{op}}^2 \left(\frac{c_0(d_\mathbf{x}^2 + \ln|\mathscr{F}|)\ln(\frac{n_{\mathrm{init}}}{\delta})^2}{n_{\mathrm{init}}} + \varepsilon_{\mathrm{ol},1}^2\right) + d_\mathbf{x}\varepsilon_{\mathrm{cov}}^2 \|A\|_{\mathrm{op}}^2 \|\Sigma_0\|,$$

   *with $\varepsilon_{\mathrm{ol},1}$ as in Lemma H.4, $\varepsilon_{\mathrm{cov}}$ as in Eq. (H.32), and $c_0 \coloneqq 32 L_{\mathrm{op}}^2 \Psi_\star^3 d_\mathbf{x}$.*

2. *Let $\eta > e$ be given, and let $\tilde{f}_1$ and $\hat{f}_1$ be defined as in Line 28 of Algorithm 5. If*

$$\bar{b}^2 \geq \bar{b}_0^2 \ln(2\eta), \quad \text{where} \quad \bar{b}_0^2 \coloneqq 10^4 d_\mathbf{x} L_{\mathrm{op}}^2 \Psi_\star^{12}(1 + \|\Sigma_0\|_{\mathrm{op}} + \|\Sigma_1\|_{\mathrm{op}}). \tag{H.35}$$

   *then we have*

$$\mathbb{P}_{\widehat{\pi}}[\tilde{f}_1(\mathbf{y}_{0:1}) = \hat{f}_1(\mathbf{y}_{0:1})] \geq 1 - \eta^{-1}. \tag{H.36}$$

## H.4 Master Theorem for Phase III

By combining Theorems 10 and 11, we derive the proof of Theorem 4.

*Proof of Theorem 4.* Let $\eta > e$ and $\sigma^2 \leq 1$ be fixed. Introduce the shorthand $\lambda = \lambda_{\min}(\mathcal{M}^\top \mathcal{M})$, and recall from Lemma H.2 that $\lambda = \mathcal{O}_\star(1)$ whenever $\sigma^2 =\leq 1$. Lastly, let us set $n_{\mathrm{init}} = n_{\mathrm{op}}$.

Let us begin with some initial parameter choices. First, we set $n_{\mathrm{init}} = n_{\mathrm{op}}$. Following Lemma H.1, we assume that $\varepsilon_{\mathrm{id}} = \check{\mathcal{O}}(1)$ is sufficiently small such that $\mathcal{E}_{\mathrm{sys}}$ holds and $\varepsilon_{\mathrm{sys}} \leq \frac{1-\gamma_\infty}{8\alpha_\infty} \leq \mathcal{O}_\star(\varepsilon_{\mathrm{id}}) = \mathcal{O}_\star(1)$. Next, following Lemma H.2, we assume that $\sigma \leq 1$ is chosen such that $\sigma = \check{\mathcal{O}}(\lambda_\mathcal{M})$ and $\lambda = \lambda_{\min}(\mathcal{M}^\top \mathcal{M}) \geq \lambda_\mathcal{M}^2 \sigma^4 / 4$.

Since $\sigma^2 = \mathcal{O}_\star(1)$, we observe from (H.23) that

$$\bar{b}_\infty^2 \leq \mathcal{O}_\star(\varepsilon_{\mathrm{dec},T}^2 \eta + (d_\mathbf{u} + d_\mathbf{x})\ln(\eta)),$$

and from (H.35) we have

$$\bar{b}_0^2 \ln(2\eta) \leq \mathcal{O}_\star(d_\mathbf{x}\ln(\eta)).$$

Let us assume for now that $\bar{b}^2 = \Omega_\star(d_\mathbf{x} + d_\mathbf{u})$; we will specify a precise choice at the end. Note that choosing $\bar{b} \geq \bar{b}_\infty \vee \bar{b}_0 \ln(2\eta)$ is non-trivial, since our bound on $\bar{b}_\infty$ depends on $\varepsilon_{\mathrm{dec},T}$, which itself depends on $\bar{b}$. Nonetheless, we will show that an appropriate choice of $\bar{b}$ solves this recurrence.

Let $\delta \leq 1/e$ be given. Define $\mathsf{logs} = (d_\mathbf{x}^2 + \ln|\mathscr{F}|)\ln^2(n_{\mathrm{op}}/\delta) \vee \ln^3(n_{\mathrm{op}}/\delta)$. As a first step, we simplify the various parameters defined in this section using the $\mathcal{O}_\star(\cdot)$ notation. In particular, we

have

$$c_{w,\phi} = \mathcal{O}_\star(\kappa d_{\mathbf{u}} + \bar{b}^2) = \mathcal{O}_\star(\kappa \bar{b}^2),$$

$$\varepsilon_{\text{noise}}^2(\delta) = \mathcal{O}_\star\left(\kappa\left(\frac{c_{w,\phi}\mathsf{logs}}{n_{\text{op}}} + \varepsilon_{\text{sys}}(d_{\mathbf{x}} + \bar{b}^2)\right)\right) = \mathcal{O}_\star\left(\frac{\kappa^2\bar{b}^2\mathsf{logs}}{n_{\text{op}}} + \kappa\bar{b}^2\varepsilon_{\text{sys}}^2\right),$$

$$c_{\mathbf{x}} = \mathcal{O}_\star(d_{\mathbf{x}} + \bar{b}^2) = \mathcal{O}_\star(\bar{b}^2),$$

$$c_1 = \mathcal{O}_\star(d_{\mathbf{x}} + d_{\mathbf{u}}),$$

$$\varepsilon_{\text{ol},1}^2 = \mathcal{O}_\star\left(\frac{c_1\mathsf{logs}}{n_{\text{op}}} + \lambda^{-1}\varepsilon_{\text{noise}}^2(\delta) + \varepsilon_{\text{sys}}^2(d_{\mathbf{x}} + d_{\mathbf{u}})\right) = \mathcal{O}_\star\left(\lambda^{-1}\cdot\left(\frac{\kappa^2\bar{b}^2\mathsf{logs}}{n_{\text{op}}} + \kappa\bar{b}^2\varepsilon_{\text{sys}}^2\right)\right),$$

$$c_{\text{cov}} = \mathcal{O}_\star(d_{\mathbf{x}}),$$

$$(\varepsilon_{\text{cov}}')^2 = \mathcal{O}_\star\left(\varepsilon_{\text{ol},1}^2 d_{\mathbf{x}} + \frac{d_{\mathbf{x}}^2\ln(n_{\text{op}}/\delta)^3}{n_{\text{op}}}\right) = \mathcal{O}_\star\left(\lambda^{-1}\cdot\left(\frac{\kappa^2 d_{\mathbf{x}}\bar{b}^2\mathsf{logs}}{n_{\text{op}}} + \kappa d_{\mathbf{x}}\varepsilon_{\text{sys}}^2\bar{b}^2\right)\right),$$

$$\varepsilon_{\text{cov}}^2 = \mathcal{O}_\star(\varepsilon_{\text{sys}}^2 + (\varepsilon_{\text{cov}}')^2).$$

We now appeal to Theorem 11. Simplifying the upper bounds, we are guaranteed that with probability at least $1 - \mathcal{O}(\kappa\delta)$, we have

$$\mathbb{P}_{\widehat{\pi}}[\tilde{f}_1(\mathbf{y}_{0:1}) = \hat{f}_1(\mathbf{y}_{0:1})] \geq 1 - \eta^{-1}$$

and

$$\mathbb{E}_{\widehat{\pi}}\left[\left\|\hat{f}_{A,0}(\mathbf{y}_0) - A f_\star(\mathbf{y}_0)\right\|^2\right] \lesssim \varepsilon_{\text{init}}^2$$

where

$$\varepsilon_{\text{init}}^2 = \mathcal{O}_\star\left(\frac{d_{\mathbf{x}}\mathsf{logs}}{n_{\text{op}}} + \varepsilon_{\text{ol},1}^2 + d_{\mathbf{x}}\varepsilon_{\text{cov}}^2\right) = \mathcal{O}_\star\left(\lambda^{-1}\cdot\left(\frac{\kappa^2 d_{\mathbf{x}}^2\bar{b}^2\mathsf{logs}}{n_{\text{op}}} + \kappa d_{\mathbf{x}}^2\varepsilon_{\text{sys}}^2\bar{b}^2\right)\right).$$

We now appeal to Theorem 8 and Theorem 10. By the union bound, and in light of Eq. (H.36), we have that with probability at least $1 - \mathcal{O}(\kappa T\delta)$,

$$\mathbb{E}_{\widehat{\pi}}\left[\max_{1\leq t\leq T}\|\hat{f}_t(\mathbf{y}_{0:t}) - f_\star(\mathbf{y}_t)\|^2\right] \leq \mathcal{O}_\star\left(\varepsilon_{\text{dec},T}^2 + (T^{1/2}c_{\mathbf{x}} + \bar{b}^2)T/\eta\right) \tag{H.37}$$

where

$$\varepsilon_{\text{dec},T}^2 = \mathcal{O}_\star\left(\varepsilon_{\text{init}}^2 + \lambda^{-1}T\varepsilon_{\text{noise}}^2 + \varepsilon_{\text{sys}}^2\bar{b}^2\right) = \mathcal{O}_\star\left(\lambda^{-1}T\cdot\left(\frac{\kappa^2 d_{\mathbf{x}}^2\bar{b}^2\mathsf{logs}}{n_{\text{op}}} + \kappa d_{\mathbf{x}}^2\varepsilon_{\text{sys}}^2\bar{b}^2\right)\right).$$

Hence, we can simplify to

$$\mathbb{E}_{\widehat{\pi}}\left[\max_{1\leq t\leq T}\|\hat{f}_t(\mathbf{y}_{0:t}) - f_\star(\mathbf{y}_t)\|^2\right]$$

$$\leq \mathcal{O}_\star\left(T^{3/2}\bar{b}^2\eta^{-1} + \lambda^{-1}T\cdot\left(\frac{\kappa^2 d_{\mathbf{x}}^2\bar{b}^2\mathsf{logs}}{n_{\text{op}}} + \kappa d_{\mathbf{x}}^2\varepsilon_{\text{sys}}^2\bar{b}^2\right)\right)$$

$$\leq \mathcal{O}_\star\left(T^{3/2}\bar{b}^2\eta^{-1} + \varepsilon_0^2 + \lambda^{-1}T\kappa d_{\mathbf{x}}^2\varepsilon_{\text{sys}}^2\bar{b}^2\right), \tag{H.38}$$

where $\varepsilon_0^2 := \lambda^{-1}T\cdot\frac{\kappa^2 d_{\mathbf{x}}^2\bar{b}^2\mathsf{logs}}{n_{\text{op}}}$.

It remains to choose $\eta$ and ensure that the condition on $\bar{b}$ is satisfied. We choose $\eta = \varepsilon_0^{-2}$. Since $\varepsilon_{\text{dec},T}^2 \leq \mathcal{O}_\star(\varepsilon_0^2 + \lambda^{-1}T\kappa d_{\mathbf{x}}^2\varepsilon_{\text{sys}}^2\bar{b}^2)$, this implies

$$\bar{b}_0^2\ln(2\eta) \vee \bar{b}_\infty^2 = \Theta_\star(\varepsilon_{\text{dec},T}^2\eta + (d_{\mathbf{u}} + d_{\mathbf{x}})\ln(\eta))$$

$$= \mathcal{O}_\star\left(1 + \varepsilon_0^{-2}\cdot\lambda^{-1}T\kappa d_{\mathbf{x}}^2\varepsilon_{\text{sys}}^2\bar{b}^2 + (d_{\mathbf{u}} + d_{\mathbf{x}})\ln(\varepsilon_0^{-2})\right).$$

It follows that if $\varepsilon_{\text{sys}}^2 \leq \check{\mathcal{O}}\left(\frac{\varepsilon_0^2\lambda}{\kappa d_{\mathbf{x}}^2\bar{b}^2 T}\right)$, we have

$$\bar{b}_0^2\ln(2\eta) \vee \bar{b}_\infty^2 \leq \mathcal{O}_\star\left((d_{\mathbf{u}} + d_{\mathbf{x}})\ln(\varepsilon_0^{-2})\right) = \mathcal{O}_\star\left((d_{\mathbf{u}} + d_{\mathbf{x}})\ln(n_{\text{op}})\right).$$

Hence, we can satisfy the constraint that $\bar{b}_0 \vee \bar{b}_\infty \leq \bar{b}$ by choosing $\bar{b} = \Theta_\star((d_{\mathbf{u}} + d_{\mathbf{x}}) \ln(n_{\mathrm{op}}))$. Returning to the final error bound, we have

$$
\begin{aligned}
&\mathbb{E}_{\widehat{\pi}} \left[ \max_{1 \leq t \leq T} \| \hat{f}_t(\mathbf{y}_{0:t}) - f_\star(\mathbf{y}_t) \|^2 \right] \\
&\leq \mathcal{O}_\star \left( \bar{b}^2 T^{3/2} \eta^{-1} + \varepsilon_0^2 + \lambda^{-1} T \kappa d_{\mathbf{x}}^2 \varepsilon_{\mathrm{sys}}^2 \bar{b}^2 \right) \\
&= \mathcal{O}_\star \left( \bar{b}^2 T^{3/2} \varepsilon_0^2 + \lambda^{-1} T \kappa d_{\mathbf{x}}^2 \varepsilon_{\mathrm{sys}}^2 \bar{b}^2 \right) \\
&= \mathcal{O}_\star \left( \lambda^{-1} T^3 \kappa^2 (d_{\mathbf{x}} + d_{\mathbf{u}})^4 \ln^2(n_{\mathrm{op}}) \cdot \left( \frac{\mathsf{logs}}{n_{\mathrm{op}}} + \varepsilon_{\mathrm{id}}^2 \right) \right).
\end{aligned}
$$

To simplify, we recall that (1) $\lambda^{-1} \leq 4\lambda_{\mathcal{M}}^{-2} \sigma^{-4}$, and (2) $\mathsf{logs} = \mathcal{O}_\star((d_{\mathbf{x}}^2 + \ln|\mathscr{F}|) \ln^3(n_{\mathrm{op}}/\delta))$. Moreover, our condition on $\varepsilon_{\mathrm{sys}}$ above implies that

$$
\varepsilon_{\mathrm{sys}}^2 \leq \check{\mathcal{O}}(\mathsf{logs}/n_{\mathrm{op}}),
$$

which means it suffices to take $\varepsilon_{\mathrm{id}}^2 = \check{\mathcal{O}}(\varepsilon_{\mathrm{sys}}^2) = \check{\mathcal{O}}(\mathsf{logs}/n_{\mathrm{op}})$ as well. Hence, for the final bound, we can simplify to

$$
\mathcal{O}_\star \left( \frac{\lambda_{\mathcal{M}}^{-2}}{\sigma^4} T^3 \kappa^2 (d_{\mathbf{x}} + d_{\mathbf{u}})^4 \cdot \frac{(d_{\mathbf{x}}^2 + \ln|\mathscr{F}|) \ln^5(n_{\mathrm{op}}/\delta)}{n_{\mathrm{op}}} \right).
$$

$\square$

# I  Supporting Proofs for Appendix H

## I.1  A Truncation Bound for the Iterates

Before proceeding with the main proofs in this section, we state a lemma which bounds the magnitudes of the states under the event that $\hat{f}_\tau$ returns state estimates bounded by $\bar{b}$. This bound is used by a number of subsequent proofs.

**Lemma I.1.** Let $\bar{b} > 0$ and $t \geq 0$. If $\| \hat{f}_\tau(\mathbf{y}_{0:\tau}) \| \leq \bar{b}$ a.s. for all $\tau \geq 0$, then for all $\delta \in (0, 1/e]$ and all $\tau \geq 0$, we have that

$$
\mathbb{P}_{\widetilde{\pi}_t} \left[ \| \mathbf{x}_\tau \|^2 \geq 30 d_{\mathbf{x}} \Psi_\Sigma + 2\bar{b}^2 \cdot \mathsf{dev}_x \ln \frac{2}{\delta} \right] \leq \delta, \quad \text{and} \tag{I.1}
$$

$$
\mathbb{E}_{\widetilde{\pi}_t} \left[ \| \mathbf{x}_\tau \|^2 \right] \leq 3 d_{\mathbf{x}} \Psi_\Sigma + \mathsf{dev}_x \bar{b}^2. \tag{I.2}
$$

Moreover, both displays above also hold with $\widetilde{\pi}_t$ replaced by $\widehat{\pi}_t$.

*Proof.* Let $\tau \geq 0$ be fixed. By the system's dynamics and the definition of $\widetilde{\pi}_t$ (cf. Eq. (H.2)), we have

$$
\mathbf{x}_\tau = A^\tau \mathbf{x}_0 + \sum_{s=0}^{\tau-1} A^{\tau-s-1} (B\widehat{K} \hat{f}_s(\mathbf{y}_{0:s}) \mathbb{I}_{s \leq t} + B\mathbf{v}_s + \mathbf{w}_s). \tag{I.3}
$$

Thus, by Jensen's inequality, and using strong stability of $A$,

$$
\| \mathbf{x}_\tau \|^2 \leq 3\|A^\tau \mathbf{x}_0\|^2 + 3 \left\| \sum_{s=0}^{\tau-1} A^{\tau-s-1} (B\mathbf{v}_s + \mathbf{w}_s) \right\|^2 + 3 \left\| \sum_{s=0}^{t \wedge (\tau-1)} A^{\tau-s-1} B\widehat{K} \hat{f}_s(\mathbf{y}_{0:s}) \right\|^2,
$$

$$
\leq 3\alpha_A^2 \|\mathbf{x}_0\|^2 + 3\|\boldsymbol{\xi}_\tau\|^2 + \frac{3\alpha_A^2}{1 - \gamma_A^2} \cdot \|B\widehat{K}\|_{\mathrm{op}}^2 \cdot \bar{b}^2, \tag{I.4}
$$

where $\boldsymbol{\xi}_\tau := \sum_{s=0}^{\tau-1} A^{\tau-s-1} (B\mathbf{v}_s + \mathbf{w}_s) \sim \mathcal{N}(0, \Sigma_\xi)$, with $\Sigma_\xi \preceq \Sigma_{\infty,\mathrm{id}}$ since $\sigma \leq 1$ under Assumption 10 (cf. Eq. (G.2)). By Lemma I.11, the expression above implies that

$$
\mathbb{P}_{\widetilde{\pi}_t} \left[ \| \mathbf{x}_\tau \|^2 \geq \left( (\alpha_A^2 \|\Sigma_0\|_{\mathrm{op}} + \|\Sigma_{\infty,\mathrm{id}}\|_{\mathrm{op}}) (9d_{\mathbf{x}} + 6) + \frac{3\alpha_A^2}{1 - \gamma_A^2} \cdot \|B\widehat{K}\|_{\mathrm{op}}^2 \bar{b}^2 \right) \ln(2/\delta) \right] \leq \delta. \tag{I.5}
$$

which we simplify to

$$\mathbb{P}_{\widetilde{\pi}_t}\left[\|\mathbf{x}_\tau\|^2 \ge \left(2(\alpha_A^2\|\Sigma_0\|_{\mathrm{op}} + \|\Sigma_{\infty,\mathrm{id}}\|_{\mathrm{op}})(9d_{\mathbf{x}} + 6) + \frac{3\alpha_A^2}{1-\gamma_A^2}\cdot\|B\widehat{K}\|_{\mathrm{op}}^2\bar{b}^2\right)\ln(1/\delta)\right] \le \delta.$$

Substituting in the definition of $\mathrm{dev}_x$ and $\Psi_\Sigma$ (Eqs. (H.3) and (H.4)), with $\gamma_A^2 \le 1$ establishes (I.1). We now show Eq. (I.2). by (I.4) and Lemma I.9, we have

$$\mathbb{E}_{\widetilde{\pi}_t}\left[\|\mathbf{x}_\tau\|^2\right] \le 2\alpha_A^2\mathbb{E}_{\widetilde{\pi}_t}\left[\|\mathbf{x}_0\|^2\right] + 3\mathbb{E}_{\widetilde{\pi}_t}\left[\|\boldsymbol{\xi}_\tau\|^2\right] + \frac{3\alpha_A^2}{1-\gamma_A^2}\cdot\|B\widehat{K}\|_{\mathrm{op}}^2\bar{b}^2,$$

$$\le 3d_{\mathbf{x}}(\|\Sigma_0\|_{\mathrm{op}} + \|\Sigma_{\infty,\mathrm{id}}\|_{\mathrm{op}}) + \frac{3\alpha_A^2}{1-\gamma_A^2}\cdot\|B\widehat{K}\|_{\mathrm{op}}^2\bar{b}^2. \tag{I.6}$$

The second part of the lemma follows from (I.5) and (I.6) by the fact that $\widetilde{\pi}_t$ and $\widehat{\pi}$ coincide up to round $t$ (inclusive). $\qquad\square$

## I.2    Proof of Lemma H.1

*Proof of Lemma H.1.* The bounds on $\|\widehat{A} - A\|_{\mathrm{op}}$ and $\|\widehat{B} - B\|_{\mathrm{op}}$ immediately follow from the conditions of the lemma. To show that $\widehat{A}$ is $(\alpha_A, \bar{\gamma}_A)$-strongly stable, we observe that if $S$ is the matrix that witnesses strong stability for $A$, we have

$$\|S^{-1}\widehat{A}S\|_{\mathrm{op}} \le \|S^{-1}(\widehat{A} - A)S\|_{\mathrm{op}} + \|S^{-1}AS\|_{\mathrm{op}} \le \alpha_A\varepsilon_{\mathrm{id}} + \gamma_A.$$

Hence, once $\varepsilon_{\mathrm{id}} \le \frac{1-\gamma_A}{2\alpha_A}$, we have $\|S^{-1}\widehat{A}S\|_{\mathrm{op}} \le \bar{\gamma}_A$.

Next, we appeal to Theorem 6, which implies that once $\varepsilon_{\mathrm{id}} \le c\cdot\alpha_\infty^{-4}(1-\gamma_\infty^2)^2\Psi_\star^{-11}$ for a sufficiently small numerical constant $c$, we have

$$\|\widehat{K} - K_\infty\|_{\mathrm{op}} \le \mathcal{O}_\star(\varepsilon_{\mathrm{id}}),$$

and $A + B\widehat{K}$ is $(\alpha_\infty, \bar{\gamma}_\infty)$-strongly stable for $\bar{\gamma}_\infty = (1 + \gamma_\infty)/2$. In particular, for $\varepsilon_{\mathrm{id}}$ sufficiently small we have $\|\widehat{K}\|_{\mathrm{op}} \le 2\|K_\infty\|_{\mathrm{op}}$, so that

$$\|\widehat{B}\widehat{K} - B\widehat{K}\|_{\mathrm{op}} \le \mathcal{O}_\star(\varepsilon_{\mathrm{id}}).$$

Next, we observe that once $\varepsilon_{\mathrm{id}} \le \Psi_\star^{-1}/2 \le \lambda_{\min}(\Sigma_w)/2$, we have $\lambda_{\min}(\widehat{\Sigma}_w) \ge \lambda_{\min}(\Sigma_w)/2$, and so we can apply Proposition I.1 to deduce that

$$\|I_{d_{\mathbf{x}}} - \widehat{\Sigma}_w\Sigma_w^{-1}\|_{\mathrm{op}} \vee \|\widehat{\Sigma}_w^{-1} - \Sigma_w^{-1}\|_{\mathrm{op}} \le \mathcal{O}_\star(\varepsilon_{\mathrm{id}}).$$

Finally, we bound the errors for the terms involving $\widehat{M}_k$ and $\widehat{\mathcal{M}}$. We first show that to do this, it suffices to bound $\max_{1\le k\le\kappa}\|\widehat{M}_k - M_k\|_{\mathrm{op}}$. First, as long as $\varepsilon_{\mathrm{id}} = \mathcal{O}_\star(1)$, we have

$$\|\widehat{M}_k\widehat{A}^k\widehat{B} - M_kA^kB\|_{\mathrm{op}} \le \mathcal{O}_\star(\|\widehat{M}_k\widehat{A}^k - M_kA^k\|_{\mathrm{op}} + \varepsilon_{\mathrm{id}}),$$

by triangle inequality. Next, we have

$$\|\widehat{M}_k\widehat{A}^k - M_kA^k\|_{\mathrm{op}} \le \|\widehat{A}^k\|_{\mathrm{op}}\|\widehat{M}_k - M_k\|_{\mathrm{op}} + \|M_k\|_{\mathrm{op}}\|\widehat{A}^k - A^k\|_{\mathrm{op}}$$

$$\le \|\widehat{A}^k\|_{\mathrm{op}}\|\widehat{M}_k - M_k\|_{\mathrm{op}} + \mathcal{O}_\star(\|\widehat{A}^k - A^k\|_{\mathrm{op}}).$$

By Lemma I.2, once $\varepsilon_{\mathrm{id}} \le \frac{(1-\gamma_A)}{2\alpha_A}$, this is upper bounded by

$$\mathcal{O}_\star(\bar{\gamma}_A^{k-1}k\cdot(\|\widehat{M}_k - M_k\|_{\mathrm{op}} + \varepsilon_{\mathrm{id}})).$$

Note that $\max_{k\ge1}\bar{\gamma}_A^{k-1}k \le \sum_{k=1}^\infty\bar{\gamma}_A^{k-1}k \le 1/(1-\bar{\gamma}_A)^2 = \mathcal{O}_\star(1)$, so the bound bove further simplifies to

$$\mathcal{O}_\star(\|\widehat{M}_k - M_k\|_{\mathrm{op}} + \varepsilon_{\mathrm{id}}).$$

Finally, by similar reasoning, we have

$$
\begin{aligned}
\|\widehat{\mathcal{M}} - \mathcal{M}\|_{\mathrm{op}} &\le \sum_{k=1}^{\kappa} \|\widehat{M}_k \widehat{A}^k - M_k A^k\|_{\mathrm{op}} \\
&\le \sum_{k=1}^{\kappa} \mathcal{O}\big(\bar{\gamma}_A^{k-1} k \big(\|\widehat{M}_k - M_k\|_{\mathrm{op}} + \varepsilon_{\mathrm{id}}\big)\big) \\
&\le \mathcal{O}\Big(\max_{1 \le k \le \kappa} \|\widehat{M}_k - M_k\|_{\mathrm{op}} + \varepsilon_{\mathrm{id}}\Big) \cdot \sum_{k=1}^{\infty} \bar{\gamma}_A^{k-1} k \\
&\le \mathcal{O}_\star\Big(\max_{1 \le k \le \kappa} \|\widehat{M}_k - M_k\|_{\mathrm{op}} + \varepsilon_{\mathrm{id}}\Big).
\end{aligned}
$$

Finally, we appeal to Lemma I.3, which implies that $\max_{1 \le k \le \kappa} \|\widehat{M}_k - M_k\|_{\mathrm{op}} = \mathcal{O}_\star(\varepsilon_{\mathrm{id}})$. $\qquad\square$

### I.2.1 Supporting Results

**Proposition I.1.** Let $X, Y \in \mathbb{R}^{d \times d}$ be positive definite matrices with $\|X - Y\|_{\mathrm{op}} \le \varepsilon$. Then we have $\|I - XY^{-1}\|_{\mathrm{op}} \le \|Y^{-1}\| \cdot \varepsilon$ and $\|X^{-1} - Y^{-1}\|_{\mathrm{op}} \le \|X^{-1}\|_{\mathrm{op}} \|Y^{-1}\|_{\mathrm{op}} \cdot \varepsilon$.

*Proof of Proposition I.1.* The result follows by the inequalities

$$
\|X^{-1} - Y^{-1}\|_{\mathrm{op}} \le \|X^{-1}\|_{\mathrm{op}} \cdot \|I - XY^{-1}\|_{\mathrm{op}},
$$

and

$$
\|I - XY^{-1}\| \le \|Y^{-1}\|_{\mathrm{op}} \cdot \|Y - X\|_{\mathrm{op}}.
$$

$\qquad\square$

**Lemma I.2.** Suppose $\|\widehat{A} - A\|_{\mathrm{op}} \le \frac{(1-\gamma_A)}{2\alpha_A}$. Then for all $k \ge 1$,

$$
\|\widehat{A}^k - A^k\|_{\mathrm{op}} \le \alpha_A^2 \bar{\gamma}_A^{k-1} k \|\widehat{A} - A\|_{\mathrm{op}},
$$

where $\bar{\gamma}_A = (1 + \gamma_A)/2$. Furthermore, we have $\|\widehat{A}^k\|_{\mathrm{op}} \le 2\alpha_A \bar{\gamma}_A^{k-1} k$.

*Proof of Lemma I.2.* Using Lemma 5 of Mania et al. [24], we are guaranteed that[11]

$$
\|\widehat{A}^k - A^k\|_{\mathrm{op}} \le \alpha_A^2 \big(\alpha_A \|\widehat{A} - A\|_{\mathrm{op}} + \gamma_A\big)^{k-1} k \|\widehat{A} - A\|_{\mathrm{op}}.
$$

The condition in the lemma statement ensures that $\alpha_A \|\widehat{A} - A\|_{\mathrm{op}} + \gamma_A \le \bar{\gamma}_A$, leading to the first result. As a consequence, we also have

$$
\begin{aligned}
\|\widehat{A}^k\|_{\mathrm{op}} &\le \|A^k\|_{\mathrm{op}} + \|\widehat{A}^k - A^k\|_{\mathrm{op}} \\
&\le \alpha_A \gamma_A^k + \alpha_A^2 \bar{\gamma}_A^{k-1} k \|\widehat{A} - A\|_{\mathrm{op}} \\
&\le \alpha_A \gamma_A^k + \alpha_A (1 - \bar{\gamma}_A) \bar{\gamma}_A^{k-1} k \\
&\le 2\alpha_A \bar{\gamma}_A^{k-1} k.
\end{aligned}
$$

$\qquad\square$

**Lemma I.3.** If $\varepsilon_{\mathrm{id}} \le \frac{(1-\gamma_A)}{2\alpha_A} \wedge \frac{\Psi_\star^{-1}}{2}$ and $\sigma^2 = \mathcal{O}_\star(1)$, then for all $1 \le k \le \kappa$, $\|\widehat{M}_k - M_k\|_{\mathrm{op}} \le \mathcal{O}_\star(\varepsilon_{\mathrm{id}})$.

*Proof of Lemma I.3.* Let $k$ be fixed. Define

$$
\Sigma_k = \sum_{i=1}^{k} A^{i-1} \Sigma_w (A^\top)^{i-1}, \quad \text{and} \quad \widehat{\Sigma}_k = \sum_{i=1}^{k} \widehat{A}^{i-1} \widehat{\Sigma}_w (\widehat{A}^\top)^{i-1},
$$

so that we have

$$
M_k = \mathcal{C}_k^\top (\mathcal{C}_k \mathcal{C}_k^\top + \sigma^{-2} \Sigma_k)^{-1}, \quad \text{and} \quad \widehat{M}_k = \widehat{\mathcal{C}}_k^\top (\widehat{\mathcal{C}}_k \widehat{\mathcal{C}}_k^\top + \sigma^{-2} \widehat{\Sigma}_k)^{-1},
$$

where $\widehat{\mathcal{C}}_k := [\widehat{A}^{k-1}\widehat{B} \mid \cdots \mid \widehat{B}]$.

As a starting point, we have by Lemma I.4 that $\|\mathcal{C}_k - \widehat{\mathcal{C}}_k\|_{\mathrm{op}} \le \mathcal{O}_\star(\varepsilon_{\mathrm{id}})$ once $\varepsilon_{\mathrm{id}} \le \frac{(1-\gamma_A)}{2\alpha_A}$. As such our task will mainly boil down to relating the error of $\widehat{M}_k$ to that of $\widehat{\mathcal{C}}_k$. We will use going forward that $\|\mathcal{C}_k\|_{\mathrm{op}} \vee \|\widehat{\mathcal{C}}_k\|_{\mathrm{op}} = \mathcal{O}_\star(1)$.

For the first step, by the triangle inequality we have

$$\|\widehat{M}_k - M_k\|_{\mathrm{op}}$$
$$\le \|\mathcal{C}_k - \widehat{\mathcal{C}}_k\|_{\mathrm{op}}\left\|(\mathcal{C}_k\mathcal{C}_k^\top + \sigma^{-2}\Sigma_k)^{-1}\right\|_{\mathrm{op}} + \|\widehat{\mathcal{C}}_k\|_{\mathrm{op}}\left\|(\mathcal{C}_k\mathcal{C}_k^\top + \sigma^{-2}\Sigma_k)^{-1} - (\widehat{\mathcal{C}}_k\widehat{\mathcal{C}}_k^\top + \sigma^{-2}\widehat{\Sigma}_k)^{-1}\right\|_{\mathrm{op}}.$$

Now, note that $\mathcal{C}_k\mathcal{C}_k^\top + \sigma^{-2}\Sigma_k \succeq \sigma^{-2}\Sigma_w$, so $\|(\mathcal{C}_k\mathcal{C}_k^\top + \sigma^{-2}\Sigma_k)^{-1}\|_{\mathrm{op}} = \mathcal{O}_\star(\sigma^2)$. Similarly, as long as $\varepsilon_{\mathrm{id}} \le \Psi_\star^{-1}/2 \le \lambda_{\min}(\Sigma_w)/2$, we have $\lambda_{\min}(\widehat{\Sigma}_w) \ge \lambda_{\min}(\Sigma_w)/2 > 0$, so we have $\|(\widehat{\mathcal{C}}_k\widehat{\mathcal{C}}_k^\top + \sigma^{-2}\widehat{\Sigma}_k)^{-1}\|_{\mathrm{op}} = \mathcal{O}_\star(\sigma^2)$. This leads allows us to simplify the bound above to

$$\|\widehat{M}_k - M_k\|_{\mathrm{op}} \le \mathcal{O}_\star(\sigma^2\varepsilon_{\mathrm{id}}) + \mathcal{O}_\star\left(\left\|(\mathcal{C}_k\mathcal{C}_k^\top + \sigma^{-2}\Sigma_k)^{-1} - (\widehat{\mathcal{C}}_k\widehat{\mathcal{C}}_k^\top + \sigma^{-2}\widehat{\Sigma}_k)^{-1}\right\|_{\mathrm{op}}\right),$$

and moreover, by invoking Proposition I.1 with the aforementioned operator norm bounds for the inverse matrices, we can further upper bound by

$$\le \mathcal{O}_\star(\sigma^2\varepsilon_{\mathrm{id}}) + \mathcal{O}_\star\left(\sigma^4\left\|(\mathcal{C}_k\mathcal{C}_k^\top + \sigma^{-2}\Sigma_k) - (\widehat{\mathcal{C}}_k\widehat{\mathcal{C}}_k^\top + \sigma^{-2}\widehat{\Sigma}_k)\right\|_{\mathrm{op}}\right)$$
$$\le \mathcal{O}_\star(\sigma^2\varepsilon_{\mathrm{id}}) + \mathcal{O}_\star\left(\sigma^4\|\mathcal{C}_k\mathcal{C}_k^\top - \widehat{\mathcal{C}}_k\widehat{\mathcal{C}}_k^\top\|_{\mathrm{op}} + \sigma^2\|\Sigma_k - \widehat{\Sigma}_k\|_{\mathrm{op}}\right),$$
$$\le \mathcal{O}_\star(\varepsilon_{\mathrm{id}} + \|\Sigma_k - \widehat{\Sigma}_k\|_{\mathrm{op}}),$$

where the final step uses that $\sigma^2 = \mathcal{O}_\star(1)$ to simplify. Finally, we bound

$$\|\Sigma_k - \widehat{\Sigma}_k\|_{\mathrm{op}} \le \sum_{i=1}^{k}\|A^{i-1} - \widehat{A}^{i-1}\|_{\mathrm{op}}\|\Sigma_w(A^\top)^{i-1}\|_{\mathrm{op}} + \|\widehat{A}^{i-1}\|_{\mathrm{op}}\|\Sigma_w - \widehat{\Sigma}_w\|_{\mathrm{op}}\|(A^\top)^{i-1}\|_{\mathrm{op}}$$
$$+ \|\widehat{A}^{i-1}\widehat{\Sigma}_w\|_{\mathrm{op}}\|A^{i-1} - \widehat{A}^{i-1}\|_{\mathrm{op}}.$$

By Lemma I.2, once $\varepsilon_{\mathrm{id}} \le \frac{(1-\gamma_A)}{2\alpha_A}$, we have $\|\widehat{A}^i\|_{\mathrm{op}} \le \mathcal{O}_\star(\bar{\gamma}_A^{i-1}i)$ and $\|\widehat{A}^i - A^i\|_{\mathrm{op}} = \mathcal{O}_\star(\bar{\gamma}_A^{i-1}i\varepsilon_{\mathrm{id}})$. We also have $\|A^i\|_{\mathrm{op}} = \mathcal{O}_\star(\bar{\gamma}_\infty^i)$ and $\|\Sigma_w\|_{\mathrm{op}} \vee \|\widehat{\Sigma}_w\|_{\mathrm{op}} = \mathcal{O}_\star(1)$, so we can bound the sum above as

$$\|\Sigma_k - \widehat{\Sigma}_k\|_{\mathrm{op}} \le \mathcal{O}_\star\left(\varepsilon_{\mathrm{id}} \cdot \left(1 + \sum_{i=2}^{k}\bar{\gamma}_A^{2(i-2)}i^2\right)\right) \le \mathcal{O}_\star\left(\varepsilon_{\mathrm{id}} \cdot \left(1 + \sum_{i=2}^{\infty}\bar{\gamma}_A^{2(i-2)}i^2\right)\right) = \mathcal{O}_\star(\varepsilon_{\mathrm{id}}).$$

$\square$

**Lemma I.4.** If $\varepsilon_{\mathrm{id}} \le \frac{(1-\gamma_A)}{2\alpha_A}$ then for all $1 \le k \le \kappa$, $\|\widehat{\mathcal{C}}_k - \mathcal{C}_k\|_{\mathrm{op}} \le \mathcal{O}_\star(\varepsilon_{\mathrm{id}})$.

*Proof of Lemma I.4.* Let $k$ be fixed. As a first step, we use the block structure to bound

$$\|\widehat{\mathcal{C}}_k - \mathcal{C}_k\|_{\mathrm{op}} \le \sum_{i=1}^{k}\|\widehat{A}^{i-1}\widehat{B} - A^{i-1}B\|_{\mathrm{op}}$$
$$\le \sum_{i=1}^{k}\|\widehat{B}\|_{\mathrm{op}}\|\widehat{A}^{i-1} - A^{i-1}\|_{\mathrm{op}} + \|A^{i-1}\|_{\mathrm{op}}\|\widehat{B} - B\|_{\mathrm{op}}.$$

By Lemma I.2, once $\varepsilon_{\mathrm{id}} \le \frac{(1-\gamma_A)}{2\alpha_A}$, we have $\|\widehat{A}^i\|_{\mathrm{op}} \le \mathcal{O}_\star(\bar{\gamma}_A^{i-1}i)$ and $\|\widehat{A}^i - A^i\|_{\mathrm{op}} = \mathcal{O}_\star(\bar{\gamma}_A^{i-1}i \cdot \varepsilon_{\mathrm{id}})$. We further have $\|A^i\|_{\mathrm{op}} = \mathcal{O}_\star(\bar{\gamma}_\infty^i)$ and $\|B\|_{\mathrm{op}} \vee \|\widehat{B}\|_{\mathrm{op}} = \mathcal{O}_\star(1)$, since $\varepsilon_{\mathrm{id}} = \mathcal{O}_\star(1)$. Plugging in these bounds above and simplifying, we have

$$\|\widehat{\mathcal{C}}_k - \mathcal{C}_k\|_{\mathrm{op}} \le \mathcal{O}_\star\left(\varepsilon_{\mathrm{id}}\left(1 + \sum_{i=2}^{k}\bar{\gamma}_A^{i-2}i\right)\right) \le \mathcal{O}_\star\left(\varepsilon_{\mathrm{id}}\left(1 + \sum_{i=2}^{\infty}\bar{\gamma}_A^{i-2}i\right)\right) = \mathcal{O}_\star(\varepsilon_{\mathrm{id}}).$$

$\square$

## I.3 Proof of Lemma H.2

Let $\overline{\mathcal{M}}$ be as in Eq. (20). For the first point, it is easy to see that $\|M_k\| = \mathcal{O}_\star(1)$ whenever $\sigma^2 = \mathcal{O}_\star(1)$ using strong stability. It follows that

$$\|\mathcal{M}_{\sigma^2}\|_{\mathrm{op}} \le \sum_{k=1}^{\kappa} \|A^{k-1}\| \|M_k\| \le \mathcal{O}_\star\left(\sum_{k=1}^{\kappa} \gamma_A^{k-1}\right) = \mathcal{O}_\star(1).$$

To prove the second point, we first recall the following result.

**Lemma I.5** ([31], Theorem 2.2). Let $X, Y \in \mathbb{R}^{d \times d}$. If $X$ is non-singular and $r := \|X^{-1}Y\|_{\mathrm{op}} < 1$, then $X + Y$ is non-singular and $\|(X + Y)^{-1} - X^{-1}\|_{\mathrm{op}} \le \|Y\|_{\mathrm{op}} \|X^{-1}\|_{\mathrm{op}}^2 / (1 - r)$.

Let $k$ be fixed. We set $X = \sum_{i=1}^{k} A^{i-1} \Sigma_w (A^{i-1})^\top$ and $Y = \sigma^2 \mathcal{C}_k \mathcal{C}_k^\top$. Since $X \ge \Sigma_w > 0$, we have that $\|X^{-1}\|_{\mathrm{op}} = \mathcal{O}_\star(1)$ and $\|Y\|_{\mathrm{op}} = \mathcal{O}_\star(\sigma^2)$. Moreover, $\|\mathcal{C}_k\|_{\mathrm{op}} = \mathcal{O}_\star(1)$. This implies that for any fixed $\varepsilon > 0$, there exists $\bar{\sigma} = \check{\mathcal{O}}(\varepsilon)$ such that for all $\sigma^2 \le \bar{\sigma}^2$,

$$\varepsilon \ge \left\| \mathcal{C}_k^\top \left( \sigma^2 \mathcal{C}_k \mathcal{C}_k^\top + \sum_{i=1}^{k} A^{i-1} \Sigma_w (A^{i-1})^\top \right)^{-1} - \mathcal{C}_k^\top \left( \sum_{i=1}^{k} A^{i-1} \Sigma_w (A^{i-1})^\top \right)^{-1} \right\|_{\mathrm{op}},$$

$$= \left\| M_k/\sigma^2 - \mathcal{C}_k^\top \left( \sum_{i=1}^{k} A^{i-1} \Sigma_w (A^{i-1})^\top \right)^{-1} \right\|_{\mathrm{op}}. \tag{I.7}$$

Define $\overline{M}_k = \mathcal{C}_k^\top \left( \sum_{i=1}^{k} A^{i-1} \Sigma_w (A^{i-1})^\top \right)^{-1}$. Then using the definitions of $\mathcal{M}$ and $\overline{\mathcal{M}}$, we have that

$$\|\mathcal{M}_{\sigma^2}^\top \mathcal{M}_{\sigma^2}/\sigma^4 - \overline{\mathcal{M}}^\top \overline{\mathcal{M}}\|_{\mathrm{op}} \le 2 \left( \|\mathcal{M}_{\sigma^2}/\sigma^2\|_{\mathrm{op}} \vee \|\overline{\mathcal{M}}\|_{\mathrm{op}} \right) \cdot \|\mathcal{M}_{\sigma^2}/\sigma^2 - \overline{\mathcal{M}}\|_{\mathrm{op}}$$

$$= \mathcal{O}_\star\left( \|\mathcal{M}_{\sigma^2}/\sigma^2 - \overline{\mathcal{M}}\|_{\mathrm{op}} \right)$$

and

$$\|\mathcal{M}_{\sigma^2}/\sigma^2 - \overline{\mathcal{M}}\|_{\mathrm{op}} \le \sum_{k=1}^{\kappa} \|A^{k-1}\|_{\mathrm{op}} \|M_k/\sigma^2 - \overline{M}_k\|_{\mathrm{op}}$$

$$\le \mathcal{O}_\star\left( \max_{1 \le k \le \kappa} \|M_k/\sigma^2 - \overline{M}_k\|_{\mathrm{op}} \right).$$

Together with (I.7), this implies that

$$\|\mathcal{M}_{\sigma^2}^\top \mathcal{M}_{\sigma^2}/\sigma^4 - \overline{\mathcal{M}}^\top \overline{\mathcal{M}}\|_{\mathrm{op}} \le \mathcal{O}_\star(\varepsilon), \quad \forall \sigma^2 \le \bar{\sigma}^2. \tag{I.8}$$

Now, note that

$$|\lambda_{\min}(\mathcal{M}_{\sigma^2}^\top \mathcal{M}_{\sigma^2}/\sigma^4) - \lambda_{\min}(\overline{\mathcal{M}}^\top \overline{\mathcal{M}})| \le \|\mathcal{M}_{\sigma^2}^\top \mathcal{M}_{\sigma^2}/\sigma^4 - \overline{\mathcal{M}}^\top \overline{\mathcal{M}}\|_{\mathrm{op}}.$$

Combining this with (I.8) implies that, for all $\sigma^2 \le \bar{\sigma}^2$,

$$\left| \lambda_{\min}^{1/2}(\mathcal{M}_{\sigma^2}^\top \mathcal{M}_{\sigma^2})/\sigma^2 - \lambda_{\mathcal{M}} \right| \le \mathcal{O}_\star(\varepsilon). \tag{I.9}$$

Finally, by definition of the $\mathcal{O}_\star(\cdot)$ notation, there exists $\varepsilon = \check{\mathcal{O}}(\lambda_{\mathcal{M}})$ such that the right-hand side of Eq. (I.9) is at most $\lambda_{\mathcal{M}}/2$, which yields the desired result.

$\square$

## I.4 Proof of Theorem 8

### I.4.1 Regression Bound for $\phi_{t,k}$

Let $t \ge 0$ and $k \in [\kappa]$ be fixed and introduce the shorthand

$$\mathbf{z}_{t,k} := (\mathbf{y}_{0:t}, \mathbf{y}_{t+k}). \tag{I.10}$$

Define the "true" $\phi$-function

$$\phi_{t,k}(h, \mathbf{z}_{t,k}) \coloneqq M_k \left( h(\mathbf{y}_{t+k}) - A^k h(\mathbf{y}_t) - A^{k-1} B \widehat{K} \hat{f}_t(\mathbf{y}_{0:t}) \right), \tag{I.11}$$

for $h \in \mathcal{H}_{\mathrm{op}}$, and its plug-in estimate analogue:

$$\widehat{\phi}_{t,k}(h, \mathbf{z}_{t,k}) \coloneqq \widehat{M}_k \left( h(\mathbf{y}_{t+k}) - \widehat{A}^k h(\mathbf{y}_t) - \widehat{A}^{k-1} \widehat{B} \widehat{K} \hat{f}_t(\mathbf{y}_{0:t}) \right).$$

Finally, define their difference by

$$\delta_{t,k}(h, \mathbf{z}_{t,k}) \coloneqq \widehat{\phi}_{t,k}(h, \mathbf{z}_{t,k}) - \phi_{t,k}(h, \mathbf{z}_{t,k}).$$

**Lemma I.6.** For $k \in [\kappa]$ define the error bound

$$\psi_{t,k}(\mathbf{z}_{t,k})^2 \coloneqq 6\alpha_A^2 L_{\mathrm{op}}^2 \Psi_M^2 \left( 2 + \|\mathbf{x}_t\|_2^2 + \|\mathbf{x}_{t+k}\|_2^2 + \Psi_\star^2 \|\widehat{K}\|_{\mathrm{op}}^2 \bar{b}^2 \right), \tag{I.12}$$

which is well defined since $\mathbf{x}_\tau = f_\star(\mathbf{y}_\tau)$ due to the decodability assumption. Further, introduce the error constant

$$c_{w,\phi} \coloneqq 50 k d_{\mathbf{u}} \sigma^2 + 30 \alpha_A^2 L_{\mathrm{op}}^2 \Psi_M^2 \left( 62 d_{\mathbf{x}} \Psi_\Sigma + 5\bar{b}^2 \cdot \mathsf{dev}_x \right). \tag{I.13}$$

Then, recalling $\mathbf{v} \coloneqq [\mathbf{v}_t^\top, \ldots, \mathbf{v}_{t+k}^\top]^\top$, for all $\delta \in (0, 1/e]$ and $h \in \mathcal{H}_{\mathrm{op}}$, the following results hold.

1. We have the bound

$$\sup_{h \in \mathcal{H}_{\mathrm{op}}} \|\phi_{t,k}(h, \mathbf{z}_{t,k})\|^2 + \sup_{h \in \mathcal{H}_{\mathrm{op}}} \|\delta_{t,k}(h, \mathbf{z}_{t,k})\|^2 \leq \psi_{t,k}(\mathbf{z}_{t,k})^2. \tag{I.14}$$

2. We have the bound

$$L_{\mathrm{op}}(\|\widehat{M}_k\|_{\mathrm{op}} + \|\widehat{M}_k \widehat{A}^k\|_{\mathrm{op}})(2 + \|f_\star(\mathbf{y}_t)\|_2 + \|f_\star(\mathbf{y}_{t+k})\|_2) \lesssim \psi_{t,k}(\mathbf{z}_{t,k}). \tag{I.15}$$

3. For all $\delta \in (0, 1/e]$,

$$\mathbb{P}_{\widetilde{\pi}_t} \left[ \psi_{t,k}(\mathbf{z}_{t,k})^2 + \|\delta_{t,k}(h, \mathbf{z}_{t,k}) - \mathbf{v}\|^2 \geq \frac{5}{3} c_{w,\phi} \ln(1/\delta) \right] \leq \delta. \tag{I.16}$$

4. We have

$$\mathbb{E}_{\widetilde{\pi}_t} \left[ \sup_{h \in \mathcal{H}_{\mathrm{op}}} \|\delta_{t,k}(h, \mathbf{z}_{t,k})\|^2 \right] \leq 24 L_{\mathrm{op}}^2 \varepsilon_{\mathrm{sys}}^2 \left( d_{\mathbf{x}} \Psi_\Sigma + \mathsf{dev}_x \bar{b}^2 \right). \tag{I.17}$$

*Proof.* For notational convenience, we will drop the subscripts $t, k$ in the expressions of $\phi_{t,k}, \widehat{\phi}_{t,k}, \psi_{t,k}, \delta_{t,k}$, and $\mathbf{z}_{t,k}$. Let $h \in \mathcal{H}_{\mathrm{op}}$ be fixed throughout.

1. **Proof of Eq. (I.14)** By Jensen's inequality and Cauchy-Schwarz, we have

$$\|\phi(h, \mathbf{z})\|^2 \leq 3\|M_k\|_{\mathrm{op}}^2 \left( \|h(\mathbf{y}_{t+k})\|^2 + \alpha_A^2 \gamma_A^{2k} \|h(\mathbf{y}_t)\|^2 + \alpha_A^2 \gamma_A^{2k-2} \|B\widehat{K}\|_{\mathrm{op}}^2 \bar{b}^2 \right),$$

$$\overset{(i)}{\leq} 3\|M_k\|_{\mathrm{op}}^2 \left( L_{\mathrm{op}}^2 \cdot (1 + \alpha_A^2 + \|\mathbf{x}_{t+k}\|^2 + \alpha_A^2 \|\mathbf{x}_t\|^2) + \alpha_A^2 \|B\widehat{K}\|_{\mathrm{op}}^2 \bar{b}^2 \right),$$

$$\overset{(ii)}{\leq} 3\Psi_M^2 \alpha_A^2 L_{\mathrm{op}}^2 \left( 2 + \|\mathbf{x}_{t+k}\|^2 + \|\mathbf{x}_t\|^2 + \Psi_\star^2 \|\widehat{K}\|_{\mathrm{op}}^2 \bar{b}^2 \right), \tag{I.18}$$

where inequality $(i)$ follows by the definition of the function class $\mathcal{H}_{\mathrm{op}}$, and $(ii)$ uses that $\alpha_A, L_{\mathrm{op}} \geq 1$, $\|M_k\|_{\mathrm{op}} \leq \Psi_M$, and $\|B\| \leq \Psi_\star$. Similarily, we also have

$$\|\delta(h, \mathbf{z})\|^2 \leq 3L_{\mathrm{op}}^2 \|\widehat{M}_k - M_k\|_{\mathrm{op}}^2 (1 + \|\mathbf{x}_{t+k}\|^2) + 3L_{\mathrm{op}}^2 \|\widehat{M}_k \widehat{A}^k - M_k A^k\|_{\mathrm{op}}^2 (1 + \|\mathbf{x}_t\|^2)$$

$$+ 3\|\widehat{M}_k \widehat{A}^{k-1} \widehat{B} - M_k A^{k-1} B\|_{\mathrm{op}}^2 \|\widehat{K}\|_{\mathrm{op}}^2 \bar{b}^2,$$

$$\leq 3L_{\mathrm{op}}^2 \varepsilon_{\mathrm{sys}}^2 (1 + \|\mathbf{x}_{t+k}\|^2) + 3L_{\mathrm{op}}^2 \varepsilon_{\mathrm{sys}}^2 (1 + \|\mathbf{x}_t\|^2) + 3\varepsilon_{\mathrm{sys}}^2 \|\widehat{K}\|_{\mathrm{op}}^2 \bar{b}^2,$$

$$\leq 3L_{\mathrm{op}}^2 \varepsilon_{\mathrm{sys}}^2 \left( 2 + \|\mathbf{x}_{t+k}\|^2 + \|\mathbf{x}_t\|^2 + \|\widehat{K}\|_{\mathrm{op}}^2 \bar{b}^2 \right), \tag{I.19}$$

where the second-to-last inequality follows since we have assumed that the event $\mathcal{E}_{\mathrm{sys}}$ holds, and the last uses $L_{\mathrm{op}}^2 \geq 1$. Combining (I.18) and (I.19), with $\alpha_A, \Psi_\star \geq 1$ and $\Psi_M \geq \varepsilon_{\mathrm{sys}}$,

$$\sup_{h \in \mathcal{H}_{\mathrm{op}}} \|\phi(h, \mathbf{z})\|^2 + \sup_{h \in \mathcal{H}_{\mathrm{op}}} \|\delta(h, \mathbf{z})\|^2 \leq 6\alpha_A^2 L_{\mathrm{op}}^2 \Psi_M^2 \left( 2 + \|\mathbf{x}_t\|_2^2 + \|\mathbf{x}_{t+k}\|_2^2 + \Psi_\star^2 \|\widehat{K}\|_{\mathrm{op}}^2 \bar{b}^2 \right) =: \psi(\mathbf{z})^2,$$

where we use the simplification $\alpha_A \geq 1$, and the definitions of $\Psi_M$ and $\Psi_\star$, followed by $L_{\mathrm{op}} \geq 1$. This shows (I.14).

2. **Proof of Eq. (I.15)** Using the bounds $\|\widehat{M}_k\widehat{A}^k - M_k A^k\|_{\mathrm{op}}, \|\widehat{M}_k - M_k\|_{\mathrm{op}} \le \varepsilon_{\mathrm{sys}} \le 1$, and $\|M_k\|_{\mathrm{op}} \le \Psi_M$, $\|A^k\|_{\mathrm{op}} \le \alpha_A$, we have

$$
\begin{aligned}
&L_{\mathrm{op}}(\|\widehat{M}_k\|_{\mathrm{op}} + \|\widehat{M}_k\widehat{A}^k\|_{\mathrm{op}})(2 + \|f_\star(\mathbf{y}_t)\|_2 + \|f_\star(\mathbf{y}_{t+k})\|_2) \\
&= L_{\mathrm{op}}(\|\widehat{M}_k\|_{\mathrm{op}} + \|\widehat{M}_k\widehat{A}^k\|_{\mathrm{op}})(2 + \|\mathbf{x}_t\|_2 + \|\mathbf{x}_{t+k}\|_2) \\
&= L_{\mathrm{op}}(2\varepsilon_{\mathrm{sys}} + \|M_k\|_{\mathrm{op}} + \|M_k A^k\|_{\mathrm{op}})(2 + \|\mathbf{x}_t\|_2 + \|\mathbf{x}_{t+k}\|_2) \\
&\le L_{\mathrm{op}}(2\varepsilon_{\mathrm{sys}} + \Psi_M(1 + \alpha_A))(2 + \|\mathbf{x}_t\|_2 + \|\mathbf{x}_{t+k}\|_2).
\end{aligned}
$$

Since $\varepsilon_{\mathrm{sys}} \le 1 \le \Psi_M$, and $\alpha_A \ge 1$, the bound follows.

3. **Proof of Eq. (I.16)** By Lemmas I.1 and I.11 we have, for all $\delta \in (0, 1/e]$, and any $\tau \le t + k$,

$$
\mathbb{P}_{\widetilde{\pi}_t}\left[\|\mathbf{v}\|^2 \ge \sigma^2 \cdot (3kd_{\mathbf{u}} + 2)\ln\delta^{-1}\right] \le \delta, \quad \text{and} \quad \mathbb{P}_{\widetilde{\pi}_t}\left[\|\mathbf{x}_\tau\|^2 \ge 15d_{\mathbf{x}}\Psi_\Sigma + \bar{b}^2 \cdot \mathsf{dev}_x \ln\frac{2}{\delta}\right] \le \delta,
$$

and so by a union bound, with probability at least $1 - \delta$,

$$
\begin{aligned}
&\left(\ln\frac{5}{\delta}\right)^{-1} \cdot \left(\psi(\mathbf{z})^2 \vee \|\delta(h_\star, \mathbf{z}) - \mathbf{v}\|^2\right) \\
&\le \left(\ln\frac{5}{\delta}\right)^{-1} \cdot \left(2\psi(\mathbf{z})^2 + 2\|\mathbf{v}\|^2\right) \\
&\le \underbrace{\left(2\sigma^2 \cdot (3kd_{\mathbf{u}} + 2)\right)}_{\le 10kd_{\mathbf{u}}\sigma^2} + 6\alpha_A^2 L_{\mathrm{op}}^2\Psi_M^2\left(2 + 30d_{\mathbf{x}}\Psi_\Sigma + 2\bar{b}^2 \cdot \mathsf{dev}_x + \Psi_\star^2\|\widehat{K}\|_{\mathrm{op}}^2\bar{b}^2\right).
\end{aligned}
$$

Finally, since $\mathsf{dev}_x \ge \Psi_\star^2\|\widehat{K}\|_{\mathrm{op}}^2$ by definition (see Eq. (H.4)), and $\Psi_\Sigma \ge 1$, the above is at most

$$
\left(\ln\frac{5}{\delta}\right)^{-1} \cdot \left(\psi(\mathbf{z})^2 \vee \|\delta(f_\star, \mathbf{z}) - \mathbf{v}\|^2\right) \le 10kd_{\mathbf{u}}\sigma^2 + 6\alpha_A^2 L_{\mathrm{op}}^2\Psi_M^2\left(32d_{\mathbf{x}}\Psi_\Sigma + 3\bar{b}^2 \cdot \mathsf{dev}_x\right) := c_{w,\phi}/5.
$$

Finally, for $\delta \le 1/e$, we have that $\left(\ln\frac{5}{\delta}\right) \le 5\ln(1/\delta)$. This bound follows by the fact that $\ln(5/\delta) = \ln(5) + \ln(1/\delta) \le (\ln 5 + 1)\ln(1/\delta) \le 5\ln(1/\delta)$, for all $\delta \in (0, 1/e]$.

4. **Proof of Eq. (I.17)**. We bound

$$
\begin{aligned}
\mathbb{E}_{\widetilde{\pi}_t}\left[\sup_{h \in \mathscr{H}_{\mathrm{op}}}\|\delta(h, \mathbf{z})\|^2\right] &= \mathbb{E}_{\widetilde{\pi}_t}\left[\sup_{h \in \mathscr{H}_{\mathrm{op}}}\|\delta_{t,k}(h, \mathbf{y}_{0:t}, \mathbf{y}_{t+k})\|^2\right], \\
&\le 3L_{\mathrm{op}}^2\varepsilon_{\mathrm{sys}}^2\left(2 + \|\widehat{K}\|_{\mathrm{op}}^2\bar{b}^2 + \mathbb{E}_{\widetilde{\pi}_t}\left[\|\mathbf{x}_{t+k}\|^2 + \|\mathbf{x}_t\|^2\right]\right), \quad \text{(I.20)}
\end{aligned}
$$

where we use Eq. (I.19) in the last step. From Lemma I.1, we have

$$
3L_{\mathrm{op}}^2\varepsilon_{\mathrm{sys}}^2\left(2 + \|\widehat{K}\|_{\mathrm{op}}^2\bar{b}^2 + \mathbb{E}_{\widetilde{\pi}_t}\left[\|\mathbf{x}_{t+k}\|^2 + \|\mathbf{x}_t\|^2\right]\right) \le 3L_{\mathrm{op}}^2\varepsilon_{\mathrm{sys}}^2\left(2 + \|\widehat{K}\|_{\mathrm{op}}^2\bar{b}^2 + 6d_{\mathbf{x}}\Psi_\Sigma + 2\mathsf{dev}_x\bar{b}^2\right),
$$

Using the above two displays together with $\mathsf{dev}_x \ge \|\widehat{K}\|_{\mathrm{op}}^2$ and $\Psi_\Sigma \ge 1$ yields

$$
\mathbb{E}_{\widetilde{\pi}_t}\left[\sup_{h \in \mathscr{H}_{\mathrm{op}}}\|\delta(h, \mathbf{z})\|^2\right] \le 3L_{\mathrm{op}}^2\varepsilon_{\mathrm{sys}}^2\left(8d_{\mathbf{x}}\Psi_\Sigma + 3\mathsf{dev}_x\bar{b}^2\right) \le 24L_{\mathrm{op}}^2\varepsilon_{\mathrm{sys}}^2\left(d_{\mathbf{x}}\Psi_\Sigma + \mathsf{dev}_x\bar{b}^2\right).
$$

$\square$

**Lemma I.7.** Let $t \ge 0$, $k \in [\kappa]$. For $\hat{h}_{t,k}$ and $\phi_{t,k}$ as in (H.10) and (I.11), respectively, we have with probability at least $1 - 3\delta/2$,

$$
\mathbb{E}_{\widetilde{\pi}_t}\left[\left\|\phi_{t,k}(\hat{h}_{t,k}, \mathbf{y}_{0:t}, \mathbf{y}_{t+k}) - M_k(A^{k-1}\mathbf{w}_t + \cdots + \mathbf{w}_{t+k-1} + A^{k-1}B\mathbf{v}_t + \cdots + B\mathbf{v}_{t+k-1})\right\|^2\right] \le \varepsilon_w^2(\delta),
$$

where $\quad \varepsilon_w^2(\delta) = c_{\varepsilon_w}\left(\dfrac{c_{w,\phi}(\ln|\mathscr{F}| + d_{\mathbf{x}}^2)\ln^2(n/\delta)}{n} + L_{\mathrm{op}}^2\varepsilon_{\mathrm{sys}}^2\left(d_{\mathbf{x}}\Psi_\Sigma + \mathsf{dev}_x\bar{b}^2\right)\right),$  (I.21)

and where $c_{\varepsilon_w}$ is a sufficiently large constant, chosen to be at least 100 without loss of generality, and $c_{w,\phi}$ is defined in Eq. (I.13).

We denote the event of Lemma I.7 by $\mathcal{E}_{\phi;t,k}(\delta)$.

*Proof.* We will apply Corollary E.2. We verify that the conditions of the corollary hold one by one.

1. **Substitutions.** We apply Corollary E.2 with $\mathbf{e} = 0$, $\boldsymbol{z} \coloneqq (\mathbf{y}_{0:t}, \mathbf{y}_{t+k})$, $\mathbf{v} = [\mathbf{v}_t^\top, \ldots, \mathbf{v}_{t+k}^\top]^\top$, $\phi = \phi_{t,k}$, $\psi = \psi_{t,k}, \delta_\phi = \delta_{t,k}$, and $c = c_{w,\phi}$, where $\phi_{t,k}$, $\psi_{t,k}$, $\delta_{t,k}$, and $c_{w,\phi}$ are as in Lemma I.6. Moreover, we let $c_\psi$ be the constant implicit in Eq. (I.15). The dimension parameters are $d_\mathbf{x}, d_1 \leftarrow d_\mathbf{x}$.

2. **Realizability.** By our assumption on the function class $\mathscr{H}_{\mathrm{op}}$, there exists $f_\star \in \mathscr{H}_{\mathrm{op}}$ such that $f_\star(y) = x$, for all $y \in \operatorname{supp} q(\cdot \mid x)$. Therefore, by the system's dynamics and the definition of the policy $\widetilde{\pi}_t$, we have almost surely

$$\phi(h_\star, \boldsymbol{z}) = M_k(f_\star(\mathbf{y}_{t+k}) - A^k f_\star(\mathbf{y}_t) - A^{k-1} B \widehat{K} \hat{f}_t(\mathbf{y}_{0:t})),$$

$$= M_k(A^{k-1}\mathbf{w}_t + \cdots + \mathbf{w}_{t+k-1} + A^{k-1} B \mathbf{v}_t + \cdots + B\mathbf{v}_{t+k-1}),$$

$$= \mathbb{E}_{\widetilde{\pi}_t}\big[\mathbf{v} \mid A^{k-1}\mathbf{w}_t + \cdots + \mathbf{w}_{t+k-1} + A^{k-1} B \mathbf{v}_t + \cdots + B\mathbf{v}_{t+k-1}\big], \quad \text{(by Fact G.2)}$$

$$= \mathbb{E}_{\widetilde{\pi}_t}\left[\mathbf{v} \;\middle|\; \begin{array}{c} \sum_{j=1}^k (A^{j-1}\mathbf{w}_{t+k-j} + A^{j-1} B \mathbf{v}_{t+k-j}) \\ \mathbf{y}_{0:t} \end{array}\right], \tag{I.22}$$

$$= \mathbb{E}_{\widetilde{\pi}_t}\left[\mathbf{v} \;\middle|\; \begin{array}{c} A^k f_\star(\mathbf{y}_t) + A^{k-1} B \widehat{K} \hat{f}_t(\mathbf{y}_{0:t}) + \sum_{j=1}^k (A^{j-1}\mathbf{w}_{t+k-j} + A^{j-1} B \mathbf{v}_{t+k-j}) \\ \mathbf{y}_{0:t} \end{array}\right],$$
$$\tag{I.23}$$

$$= \mathbb{E}_{\widetilde{\pi}_t}\big[\mathbf{v} \mid \mathbf{y}_{0:t}, f_\star(\mathbf{y}_{t+k})\big], \tag{I.24}$$

$$= \mathbb{E}_{\widetilde{\pi}_t}\big[\mathbf{v} \mid \mathbf{y}_{0:t}, \mathbf{y}_{t+k}\big], \tag{I.25}$$

where (I.22) follows by the fact that $(\mathbf{v}_\tau)_{\tau \geq t}$ and $(\mathbf{w}_\tau)_{\tau \geq t}$ are independent of $\mathbf{y}_{0:t}$, (I.23) follows by the conditioning on $\mathbf{y}_{0:t}$ (which determines the term $A^k f_\star(\mathbf{y}_t) + A^{k-1} B \widehat{K} \hat{f}_t(\mathbf{y}_{0:t})$), and (I.24) uses the system's dynamics. Finally, (I.25) uses the realizability assumption. Thus, (I.25) ensures the realizability assumption in Corollary E.2 is satisfied.

3. **Conditions 1 & 2.** Lemma I.6 ensures that conditions 1 and 2 of Corollary E.2 are satisfied.

4. **Condition 3.** By the structure of $\mathscr{H}_{\mathrm{op}}$, condition 3 is satisfied with $L$ as in Assumption 5 and $bL = L_{\mathrm{op}}$. Examining $\widehat{\phi}_{t,k}$, we can take $X_1 = \widehat{M}_k$, and $X_2 = \widehat{A}^k$.

5. **Condition 4.** By Eq. (I.15), this holds for some $c_\psi \lesssim 1$.

Recall the notation $\operatorname{logs}(n, \delta) \lesssim \ln^2(n/\delta)$ defined in Corollary E.2. With the substitutions above, Corollary E.2 implies that with probability at least $1 - \frac{3\delta}{2}$:

$$\mathbb{E}\|\phi_{t,k}(\hat{h}, \mathbf{z}) - \phi_{t,k}(f_\star, \mathbf{z})\|^2 \leq \frac{12 c_{w,\phi}(\ln|\mathscr{F}| + d_\mathbf{x} \cdot d_\mathbf{x})\operatorname{logs}(c_\psi n, \delta)}{n} + 16\mathbb{E}\|\mathbf{e}\|^2 + 8\max_{h \in \mathscr{H}} \mathbb{E}\|\delta_{t,k}(h, \mathbf{z})\|^2$$

$$= \frac{12 c_{w,\phi}(\ln|\mathscr{F}| + d_\mathbf{x}^2)\operatorname{logs}(c_\psi n, \delta)}{n} + 8\max_{h \in \mathscr{H}} \mathbb{E}\|\delta_{t,k}(h, \mathbf{z})\|^2,$$

$$\lesssim \frac{c_{w,\phi}(\ln|\mathscr{F}| + d_\mathbf{x}^2)\ln^2(n/\delta)}{n} + L_{\mathrm{op}}^2 \varepsilon_{\mathrm{sys}}^2 \big(d_\mathbf{x} \Psi_\Sigma + \operatorname{dev}_x \bar{b}^2\big).$$
$$\text{(by Eq. (I.17))}$$

$\square$

### I.4.2 Regression Bound for $\phi$

Let $t \geq 0$ be fixed. Recall the various functions defined at the start of Appendix I.4.1. In addition, consider the following functions for $k \in [\kappa]$, $h \in \mathscr{H}_{\mathrm{op}}$:

$$\widehat{\phi}_t(h, \mathbf{y}_{0:t+1}) \coloneqq \widehat{\mathcal{M}}\big(h(\mathbf{y}_{t+1}) - \widehat{A}h(\mathbf{y}_t) - \widehat{B}\widehat{K}\hat{f}_t(\mathbf{y}_{0:t})\big),$$

$$\phi_t(h, \mathbf{y}_{0:t+1}) \coloneqq \mathcal{M}\big(h(\mathbf{y}_{t+1}) - A h(\mathbf{y}_t) - B \widehat{K}\hat{f}_t(\mathbf{y}_{0:t})\big),$$

$$\delta_t(h, \mathbf{y}_{0:t+1}) \coloneqq \widehat{\phi}_t(h, \mathbf{y}_{0:t+1}) - \phi_t(h, \mathbf{y}_{0:t+1}).$$

Further, for $(\hat{h}_{t,k})_{k\in\kappa}$ as in Eq. (H.10), define

$$\widehat{\phi}_t(\mathbf{y}_{0:t+\kappa}) \coloneqq [\widehat{\phi}_{t,1}(\hat{h}_{t,1}, \mathbf{y}_{0:t}, \mathbf{y}_{t+1})^\top, \ldots, \widehat{\phi}_{t,\kappa}(\hat{h}_{t,\kappa}, \mathbf{y}_{0:t}, \mathbf{y}_{t+\kappa})^\top]^\top,$$

$$\phi_t(\mathbf{y}_{0:t+\kappa}) \coloneqq [\phi_{t,1}(\hat{h}_{t,1}, \mathbf{y}_{0:t}, \mathbf{y}_{t+1})^\top, \ldots, \phi_{t,\kappa}(\hat{h}_{t,\kappa}, \mathbf{y}_{0:t}, \mathbf{y}_{t+\kappa})^\top]^\top,$$

$$\phi_t^\star(\mathbf{y}_{0:t+\kappa}) \coloneqq [\phi_{t,1}(f_\star, \mathbf{y}_{0:t}, \mathbf{y}_{t+1})^\top, \ldots, \phi_{t,\kappa}(f_\star, \mathbf{y}_{0:t}, \mathbf{y}_{t+\kappa})^\top]^\top.$$

Here, the first term uses estimated dynamics and estimates $\hat{h}$ of $f_\star$; the second term uses true dynamics and estimates $\hat{h}$; the third term uses true dynamics *and* the ground truth $f_\star$.

**Lemma I.8.** Let $\mathbf{v} \coloneqq \phi_t^\star(\mathbf{y}_{0:t+\kappa})$, $\mathbf{e} \coloneqq \phi_t^\star(\mathbf{y}_{0:t+\kappa}) - \widehat{\phi}_t(\mathbf{y}_{0:t+\kappa})$. Recall the function $\psi_{t,1}$ defined in Eq. (I.12). Then the following properties hold.

1. We have the bound

$$\sup_{h\in\mathscr{H}_{\mathrm{op}}} \|\phi_t(h, \mathbf{y}_{0:t+1})\|^2 + \sup_{h\in\mathscr{H}_{\mathrm{op}}} \|\delta_t(h, \mathbf{y}_{0:t+1})\|^2 \le \psi_{t,1}(\mathbf{y}_{0:t+1})^2. \tag{I.26}$$

2. We have the bound

$$L_{\mathrm{op}}(\|\widehat{\mathcal{M}}\|_{\mathrm{op}} + \|\widehat{\mathcal{M}\widehat{A}}\|_{\mathrm{op}})(2 + \|f_\star(\mathbf{y}_t)\|_2 + \|f_\star(\mathbf{y}_{t+1})\|_2) \lesssim \psi_{t,1}(\mathbf{y}_{0:t+1}). \tag{I.27}$$

3. For any $\delta \in (0, 1/e]$, we have

$$\mathbb{P}_{\widetilde{\pi}_t}[\psi_{t,1}(\mathbf{y}_{0:t+1})^2 \vee \|\mathbf{v} - \mathbf{e}\|^2 \le 2\kappa c_{w,\phi}(1 + \ln(\kappa))\ln(1/\delta)] \le \delta. \tag{I.28}$$

4. For any $\delta \in (0, 1/e]$, on the event $\bigcap_{k=1}^\kappa \mathcal{E}_{\phi;t,k}(\delta)$ (cf. Lemma I.7), we have that

$$\mathbb{E}_{\widetilde{\pi}_t}\|\mathbf{e}\|^2 \le 3\kappa\varepsilon_w(\delta)^2. \tag{I.29}$$

5. For any $\delta > 0$, we have the following bound (independent of $\delta$):

$$\sup_{h\in\mathscr{H}_{\mathrm{op}}} \mathbb{E}\|\delta_t(h, \mathbf{y}_{0:t+1})\|^2 \le \varepsilon_w(\delta)^2. \tag{I.30}$$

*Proof of Lemma I.8.* In what follows, let us suppress dependence on $\mathbf{y}$ and $\mathbf{z}_{t,k}$ when clear from context, where $\mathbf{z}_{t,k}$ is as in (I.10).

1. **Bounding** $\sup_{h\in\mathscr{H}_{\mathrm{op}}} \|\phi_t(h, \mathbf{y}_{0:t+1})\| + \sup_{h\in\mathscr{H}_{\mathrm{op}}} \|\delta_t(h, \mathbf{y}_{0:t+1})\| \le \psi_{t,1}(\mathbf{y}_{0:t+1})^2$.

   The bound in (I.26) actually follows from the same argument as in the proof of Lemma I.6 with $k = 1$ and $(\widehat{M}_k, M_k)$ replaced by $(\widehat{\mathcal{M}}, \mathcal{M})$ (using that $\Psi_M$ also bounds $\|\mathcal{M}\|_{\mathrm{op}}$, and $\varepsilon_{\mathrm{sys}}$ upper bounds $\|\mathcal{M} - \widehat{\mathcal{M}}\|_{\mathrm{op}}$) under $\mathcal{E}_{\mathrm{sys}}$.

2. **Establishing Eq. (I.27).** This is also analogous to the proof of Eq. (I.15) in Lemma I.6.

3a. **Bounding** $\|\mathbf{v} - \mathbf{e}\|^2$. We bound

$$\|\mathbf{v} - \mathbf{e}\|^2 = \left\|\widehat{\phi}(\mathbf{y}_{0:t+k})\right\|^2 = \sum_{k=1}^\kappa \left\|\widehat{\phi}_{t,k}(\hat{h}_{t,k}, \mathbf{y}_{0:t}, \mathbf{y}_{t+k})\right\|^2 \le \sum_{k=1}^\kappa \psi_{t,k}(\mathbf{z}_{t,k})^2,$$

where we use Eq. (I.14).

3b. **Establishing Eq. (I.28).** We have

$$\psi_{t,1}(\mathbf{y}_{0:t+1})^2 \vee \|\mathbf{v} - \mathbf{e}\|^2 \le \psi_{t,1}(\mathbf{y}_{0:t+1})^2 \vee \sum_{k=1}^\kappa \psi_{t,k}(\mathbf{z}_{t,k})^2,$$

$$= \sum_{k=1}^\kappa \psi_{t,k}(\mathbf{z}_{t,k})^2,$$

$$= \sum_{k=1}^\kappa 6\alpha_A^2 L_{\mathrm{op}}^2 \Psi_M^2 \left(2 + \|\mathbf{x}_t\|_2^2 + \|\mathbf{x}_{t+k}\|_2^2 + \Psi_\star^2 \|\widehat{K}\|_{\mathrm{op}}^2 \bar{b}^2\right).$$

Now, with probability $1 - (\kappa + 1)\delta$, we have that all $\|\mathbf{x}_{t+i}\|^2$ for $i \in \{0, 1, \ldots, \kappa\}$ simultaneously satisfy

$$\|\mathbf{x}_{t+i}\|^2 \leq 30 d_{\mathbf{x}} \Psi_\Sigma + 2\bar{b}^2 \cdot \mathsf{dev}_x \ln \frac{2}{\delta}$$

by Lemma I.1. Hence, with probability $1 - (\kappa + 1)\delta$, we have

$$
\begin{aligned}
&\psi_{t,1}(\mathbf{y}_{0:t+1})^2 \vee \|\mathbf{v} - \mathbf{e}\|^2 \\
&\leq \sum_{k=1}^{\kappa} \psi_{t,k}(\mathbf{z}_{t,k})^2 \\
&= \kappa 6 \alpha_A^2 L_{\mathrm{op}}^2 \Psi_M^2 \left( 2 + 60 d_{\mathbf{x}} \Psi_\Sigma + 4\bar{b}^2 \cdot \mathsf{dev}_x \ln \frac{2}{\delta} + \Psi_\star^2 \|\widehat{K}\|_{\mathrm{op}}^2 \bar{b}^2 \right) \\
&\leq \kappa 6 \alpha_A^2 L_{\mathrm{op}}^2 \Psi_M^2 \left( 62 d_{\mathbf{x}} \Psi_\Sigma + 5\bar{b}^2 \cdot \mathsf{dev}_x \right) \ln \frac{2}{\delta} \leq \frac{1}{2} \kappa c_{w,\phi} \ln \frac{2}{\delta}.
\end{aligned}
$$

where in the last line, we absorb various parameters into larger ones. Finally, replacing $\delta$ by $\delta/(\kappa+1)$ gives $(1/2)\cdot\ln(2(\kappa+1)/\delta) = (1/2)\cdot\ln(2/\delta)+\ln(2(\kappa+1)) \leq 2(1+\ln(\kappa))\ln(1/\delta)$ for $\delta \in (0, 1/e]$. This gives that,

$$\mathbb{P}\left[ \psi_{t,1}(\mathbf{y}_{0:t+1})^2 \vee \|\mathbf{v} - \mathbf{e}\|^2 \geq 2\kappa c_{w,\phi}(1 + \ln(\kappa))\ln(1/\delta) \right] \leq \delta.$$

4. **Establishing Eq. (I.29).** First, we bound

$$\|\phi_t(\mathbf{y}_{0:t+\kappa}) - \phi_t^\star(\mathbf{y}_{0:t+\kappa})\|_2^2 = \sum_{k=1}^{\kappa} \|\phi_{t,k}(\hat{h}_{t,k}) - \phi_{t,k}(h_\star)\|^2.$$

From Lemma I.7, we have on the event $\bigcap_{k=1}^{\kappa} \mathcal{E}_{\phi;t,k}(\delta)$ (recall the definition of the event $\mathcal{E}_{\phi;t,k}(\delta)$ from Lemma I.7) that

$$\mathbb{E}_{\widetilde{\pi}_k} \|\phi_t(\mathbf{y}_{0:t+\kappa}) - \widehat{\phi}_t(\mathbf{y}_{0:t+\kappa})\|^2 \leq \sum_{k=1}^{\kappa} \varepsilon_w(\delta)^2 = \kappa \varepsilon_w(\delta)^2. \tag{I.31}$$

Second, we note that

$$
\begin{aligned}
\|\phi_t(\mathbf{y}_{0:t+\kappa}) - \phi_t^\star(\mathbf{y}_{0:t+\kappa})\|_2^2 &= \sum_{k=1}^{\kappa} \|\phi_{t,k}(\hat{h}_{t,k}) - \phi_{t,k}(\hat{h}_{t,k})\|^2, \\
&= \sum_{k=1}^{\kappa} \|\delta_{t,1}(\hat{h}_{t,k})\|^2,
\end{aligned}
$$

so that by Eq. (I.17),

$$\mathbb{E}_{\widetilde{\pi}_k} \|\phi_t(\mathbf{y}_{0:t+\kappa}) - \phi_t^\star(\mathbf{y}_{0:t+\kappa})\|_2^2 \leq 24\kappa L_{\mathrm{op}}^2 \varepsilon_{\mathrm{sys}}^2 \left( d_{\mathbf{x}} \Psi_\Sigma + \mathsf{dev}_x \bar{b}^2 \right).$$

Hence, on $\mathcal{E}_{\mathrm{sys}} \cap \bigcap_{k=1}^{\kappa} \mathcal{E}_{\phi;t,k}(\delta)$, it holds that

$$
\begin{aligned}
\mathbb{E}\|\mathbf{e}\|^2 &\leq 2\mathbb{E}_{\widetilde{\pi}_k} \|\phi_t(\mathbf{y}_{0:t+\kappa}) - \phi_t^\star(\mathbf{y}_{0:t+\kappa})\|_2 + 2\mathbb{E}_{\widetilde{\pi}_k} \|\phi_t(\mathbf{y}_{0:t+\kappa}) - \widehat{\phi}_t(\mathbf{y}_{0:t+\kappa})\|^2 \\
&\leq 2\kappa \varepsilon_w(\delta)^2 + 48\kappa L_{\mathrm{op}}^2 \varepsilon_{\mathrm{sys}}^2 \left( d_{\mathbf{x}} \Psi_\Sigma + \mathsf{dev}_x \bar{b}^2 \right) \leq 3\kappa \varepsilon_w(\delta)^2,
\end{aligned}
$$

where the last line uses the definition of $\varepsilon_w(\delta)^2$ from Lemma I.7.

5. **Establishing Eq. (I.30).** By using an analogous proof to that of Eq. (I.17) (in particular, exploiting that $\varepsilon_{\mathrm{sys}}$ bounds the error of both $\widehat{\mathcal{M}}$ and $\widehat{M}_k$), we can show that

$$\mathbb{E}_{\widetilde{\pi}_t}\left[ \sup_{h \in \mathscr{H}_{\mathrm{op}}} \|\delta_t(h, \mathbf{y}_{0:t+1})\|^2 \right] \leq 24 L_{\mathrm{op}}^2 \varepsilon_{\mathrm{sys}}^2 \left( d_{\mathbf{x}} \Psi_\Sigma + \mathsf{dev}_x \bar{b}^2 \right).$$

The right-hand-side is crudely bounded by $\varepsilon_w(\delta)^2$ for any $\delta \in (0, 1)$.

$\square$

### I.4.3 Proof of Theorem 8

Again, we appeal to Corollary E.2. We verify one by one that the conditions require to apply the corollary hold.

1. **Substitutions.** We appeal to the corollary with $\mathbf{e} = \phi_t^\star(\mathbf{y}_{0:t+\kappa}) - \widehat{\phi}_t(\mathbf{y}_{0:t+\kappa})$, $\mathbf{z} \coloneqq \mathbf{y}_{0:t+1}$, $\mathbf{v} = \phi_t^\star(\mathbf{y}_{0:t+\kappa})$, $\phi = \phi_t$, $\psi = \psi_t$, $\delta_\phi = \delta_t$, and $c = \kappa(\ln \kappa + 1)c_{w,\phi}$, where $\phi_t$, $\psi_t$, $\delta_t$, and $c_{w,\phi}$ are as in Lemma I.8. We also take $d_1, d_{\mathbf{x}} \leftarrow d_{\mathbf{x}}$.

2. **Realizability.**

   By our assumption on the function class $\mathscr{H}_{\mathrm{op}}$, there exists $f_\star \in \mathscr{H}_{\mathrm{op}}$ such that $f_\star(y) = x$ for all $y \in \operatorname{supp} q(\cdot \mid x)$. Therefore, by the system's dynamics and the definition of the policy $\widetilde{\pi}_t$, we have

$$
\begin{aligned}
\phi(f_\star, \mathbf{z}) &= \mathcal{M}(\mathbf{w}_t + B\mathbf{v}_t), \\
&= \begin{bmatrix} M_1(\mathbf{w}_t + B\mathbf{v}_t) \\ \vdots \\ M_\kappa A^{\kappa-1}(\mathbf{w}_t + B\mathbf{v}_t) \end{bmatrix}, \\
&= \mathbb{E}_{\widetilde{\pi}_t}\left[ \begin{bmatrix} M_1(\mathbf{w}_t + B\mathbf{v}_t) \\ \vdots \\ M_\kappa(\sum_{j=1}^\kappa A^{j-1}\mathbf{w}_{t+\kappa-j} + A^{j-1}B\mathbf{v}_{t+\kappa-j}) \end{bmatrix} \middle| \mathbf{w}_t + B\mathbf{v}_t \right], \\
&= \mathbb{E}_{\widetilde{\pi}_t}\left[ \begin{bmatrix} M_1(\mathbf{w}_t + B\mathbf{v}_t) \\ \vdots \\ M_\kappa(\sum_{j=1}^\kappa A^{j-1}\mathbf{w}_{t+\kappa-j} + A^{j-1}B\mathbf{v}_{t+\kappa-j}) \end{bmatrix} \middle| \begin{matrix} \mathbf{w}_t + B\mathbf{v}_t, \\ \mathbf{y}_{0:t} \end{matrix} \right], &\text{(I.32)} \\
&= \mathbb{E}_{\widetilde{\pi}_t}\left[ \begin{bmatrix} M_1(\mathbf{w}_t + B\mathbf{v}_t) \\ \vdots \\ M_\kappa(\sum_{j=1}^\kappa A^{j-1}\mathbf{w}_{t+\kappa-j} + A^{j-1}B\mathbf{v}_{t+\kappa-1}) \end{bmatrix} \middle| \mathbf{y}_{0:t}, \mathbf{x}_{t+1} \right], &\text{(I.33)} \\
&= \mathbb{E}_{\widetilde{\pi}_t}\left[ \begin{bmatrix} \phi_{t,1}^\star(\mathbf{y}_{0:t}, \mathbf{y}_{t+1}) \\ \vdots \\ \phi_{t,\kappa}^\star(\mathbf{y}_{0:t}, \mathbf{y}_{t+\kappa}) \end{bmatrix} \middle| \mathbf{y}_{0:t}, \mathbf{y}_{t+1} \right], &\text{(I.34)}
\end{aligned}
$$

   where: (I.32) follows by the fact that $(\mathbf{v}_\tau)_{\tau \geq t}$ and $(\mathbf{w}_\tau)_{\tau \geq t}$ are independent of $\mathbf{y}_{0:t}$; (I.33) follows by the fact that $\mathbf{w}_t + B\mathbf{v}_t$ can recovered from $\mathbf{x}_{t+1}$ given $\mathbf{y}_{0:t}$ and vice-versa; and finally, (I.34) follows from the system's dynamics and the definition of $\widetilde{\pi}_t$. Thus, (I.34) ensures that the realizability assumption in Corollary E.2 is satisfied.

3. **Conditions 1 & 2.** Lemma I.8 ensures that conditions 1 and 2 of Corollary E.2 are satisfied.

4. **Condition 3.** By the structure of $\mathscr{H}_{\mathrm{op}}$, condition 3 is satisfied with $L$ as in Assumption 5 and $bL = L_{\mathrm{op}}$. Examining $\widehat{\phi}_t(h, \mathbf{y}_{0:t+1}) \coloneqq \widehat{\mathcal{M}}\big(h(\mathbf{y}_{t+1}) - \widehat{A}h(\mathbf{y}_t) - \widehat{B}\widehat{K}\hat{f}_t(\mathbf{y}_{0:t})\big)$, we can take $X_1 = \widehat{\mathcal{M}}$, and $X_2 = \widehat{A}$. The term $\mathcal{M}\widehat{B}\widehat{K}\hat{f}_t(\mathbf{y}_{0:t})$ does not depend on $h$, and thus corresponds to $\delta_0$.

5. **Condition 4.** By Eq. (I.27), this holds for some $c_\psi \lesssim 1$.

Recall $\mathtt{logs}(n, \delta) \lesssim \ln^2(n/\delta)$, defined in Corollary E.2. Corollary E.2 implies that with probability at least $1 - \frac{3}{2}\delta$,

$$
\begin{aligned}
\mathbb{E}\|\phi_t(\hat{h}, \mathbf{z}) - \phi_t(f_\star, \mathbf{z})\|^2 &\leq \frac{12\kappa(\ln\kappa+1)c_{w,\phi}(\ln|\mathscr{F}| + d_{\mathbf{x}} \cdot d_{\mathbf{x}})\mathtt{logs}(c_\psi n, \delta)}{n} + 16\mathbb{E}\|\mathbf{e}\|^2 + 8 \max_{h \in \mathscr{H}_{\mathrm{op}}} \mathbb{E}\|\delta_t(h, \mathbf{z})\|^2 \\
&\lesssim \frac{c_{w,\phi}\kappa(\ln\kappa+1)(\ln|\mathscr{F}| + d_{\mathbf{x}}^2)\ln^2(n/\delta)}{n} + \mathbb{E}\|\mathbf{e}\|^2 + \max_{h \in \mathscr{H}_{\mathrm{op}}} \mathbb{E}\|\delta_t(h, \mathbf{z})\|^2.
\end{aligned}
$$

Substituting in the bounds in Eqs. (I.29) and (I.30), which hold on the events $\bigcap_{k=1}^\kappa \mathcal{E}_{\phi;t,k}(\delta)$ (i.e., the intersection of the events from Lemma I.7), followed by the definition of $\varepsilon_w$ given in Eq. (I.21), the

expression above is bounded as

$$
\mathbb{E}\|\phi_t(\hat{h}, \mathbf{z}) - \phi_t(f_\star, \mathbf{z})\|^2
$$
$$
\lesssim \frac{c_{w,\phi}\kappa(\ln\kappa + 1)(\ln|\mathscr{F}| + d_{\mathbf{x}}^2)\ln^2(n/\delta)}{n} + \kappa\varepsilon_w(\delta)^2
$$
$$
\lesssim \kappa(1 + \ln(\kappa))\varepsilon_w(\delta)^2
$$
$$
\lesssim \kappa(1 + \ln(\kappa))\left(\frac{c_{w,\phi}(\ln|\mathscr{F}| + d_{\mathbf{x}}^2)\ln^2(n/\delta)}{n} + L_{\mathrm{op}}^2\varepsilon_{\mathrm{sys}}^2\left(d_{\mathbf{x}}\Psi_\Sigma + \mathsf{dev}_x\bar{b}^2\right)\right).
$$

Finally, let us account for the total failure probability. By Lemma I.7, we have $\mathbb{P}[\bigcap_{k=1}^\kappa \mathcal{E}_{\phi;t,k}(\delta)] \geq 1 - \frac{3\kappa}{2}\delta$, and the above display holds with another probability $1 - \frac{3}{2}\delta$. Hence, our failure probability is at most $\frac{3(\kappa+1)\delta}{2}$. Rescaling $\delta \leftarrow \frac{2\delta}{3(\kappa+1)}$, and noting that $\ln(c_1/\delta) \lesssim c_1\ln(1/\delta)$ for constants $c_1$, we find that with probability $1 - \delta$,

$$
\mathbb{E}\|\phi_t(\hat{h}, \mathbf{z}) - \phi_t(f_\star, \mathbf{z})\|^2 \quad \lesssim \kappa(1 + \ln(\kappa))\left(\frac{c_{w,\phi}(\ln|\mathscr{F}| + d_{\mathbf{x}}^2)\ln^2(n\kappa/\delta)}{n} + L_{\mathrm{op}}^2\varepsilon_{\mathrm{sys}}^2\left(d_{\mathbf{x}}\Psi_\Sigma + \mathsf{dev}_x\bar{b}^2\right)\right).
$$
$$
\lesssim \varepsilon_{\mathrm{noise}}^2(\delta).
$$

□

## I.5 Proof of Lemma H.3

*Proof of Lemma H.3.* Let $t \geq 0$ be fixed. To begin, consider a fixed $0 \leq \tau \leq t$, and let $(\hat{h}_\tau)$ and $(\phi_\tau)$ be as in Lemma I.8. For notational convenience, we define $\widetilde{\phi}_\tau := \phi_\tau - \mathcal{M}B\mathbf{v}_\tau$. From the definitions of $\tilde{f}_{\tau+1}$ and $\phi_\tau$, we have

$$
\tilde{f}_{\tau+1}(\mathbf{y}_{0:\tau+1}) := \widehat{A}\hat{f}_\tau(\mathbf{y}_{0:\tau}) + \hat{h}_\tau(\mathbf{y}_{\tau+1}) - \widehat{A}\hat{h}_\tau(\mathbf{y}_\tau),
$$
$$
= \widehat{A}\hat{f}_\tau(\mathbf{y}_{0:\tau}) + B\widehat{K}\hat{f}_\tau(\mathbf{y}_{0:\tau}) + B\mathbf{v}_\tau + ((\hat{h}_\tau(\mathbf{y}_{\tau+1}) - A\hat{h}_\tau(\mathbf{y}_\tau) - B\widehat{K}\hat{f}_\tau(\mathbf{y}_{0:\tau})) - B\mathbf{v}_\tau),
$$
$$
= \widehat{A}\hat{f}_\tau(\mathbf{y}_{0:\tau}) + B\widehat{K}\hat{f}_\tau(\mathbf{y}_{0:\tau}) + B\mathbf{v}_\tau + \mathcal{M}^\dagger\widetilde{\phi}_\tau(\hat{h}_\tau, \mathbf{y}_{0:\tau+1}),
$$

where we have used that $\mathcal{M}$ has full row rank by Assumption 10. This implies that

$$
\tilde{f}_{\tau+1}(\mathbf{y}_{0:\tau+1}) - f_\star(\mathbf{y}_{\tau+1}) = (\widehat{A} - A)\hat{f}_\tau(\mathbf{y}_{0:\tau}) + A(\hat{f}_\tau(\mathbf{y}_{0:\tau}) - f_\star(\mathbf{y}_{0:\tau})) + (\mathcal{M}^\dagger\widetilde{\phi}_\tau(\hat{h}_\tau, \mathbf{y}_{0:\tau+1}) - \mathbf{w}_\tau).
$$

Under the event $\mathscr{E}_{0:t}$, we have in particular that $\hat{f}_s = \tilde{f}_s$, for all $0 \leq s \leq \tau$. Thus, by induction we have

$$
\tilde{f}_{\tau+1}(\mathbf{y}_{0:\tau+1}) - f_\star(\mathbf{y}_{\tau+1}) = \sum_{s=0}^\tau A^{\tau-s}\left((\widehat{A} - A)\tilde{f}_s(\mathbf{y}_{0:s}) + (\mathcal{M}^\dagger\widetilde{\phi}_s(\hat{h}_s, \mathbf{y}_{0:s+1}) - \mathbf{w}_s)\right)
$$
$$
+ A^{\tau-1}(\hat{f}_{A,0}(\mathbf{y}_0) - Af_\star(\mathbf{y}_0)),
$$

with $\hat{f}_0 \equiv \tilde{f}_0 \equiv 0$. As a result, we have, for $\epsilon_0 := \|\hat{f}_{A,0}(\mathbf{y}_0) - Af_\star(\mathbf{y}_0)\|$ and $\varepsilon_{\mathrm{sys}}$ as in (H.7),

$$
\|\tilde{f}_{\tau+1}(\mathbf{y}_{0:\tau+1}) - f_\star(\mathbf{y}_\tau)\| \tag{I.35}
$$
$$
\leq \left\|\sum_{s=0}^\tau A^{\tau-s}(\widehat{A} - A)\tilde{f}_s(\mathbf{y}_{0:s})\right\| + \left\|\sum_{s=0}^\tau A^{\tau-s}(\mathcal{M}^\dagger\widetilde{\phi}_s(\hat{h}_s, \mathbf{y}_{0:s+1}) - \mathbf{w}_s)\right\| + \alpha_A\gamma_A^{\tau-1}\epsilon_0,
$$
$$
\leq \alpha_A\varepsilon_{\mathrm{sys}}\bar{b}(1 - \gamma_A)^{-1} + \alpha_A\gamma_A^{\tau-1}\epsilon_0 + \sum_{s=0}^\tau \|A^{\tau-s}\|_{\mathrm{op}}\|(\mathcal{M}^\dagger\widetilde{\phi}_s(\hat{h}_s, \mathbf{y}_{0:s+1}) - \mathbf{w}_s)\|,
$$
$$
\leq \alpha_A\varepsilon_{\mathrm{sys}}\bar{b}(1 - \gamma_A)^{-1} + \alpha_A\gamma_A^{\tau-1}\epsilon_0 + \alpha_A\sum_{s=0}^\tau \gamma_A^{\tau-s}\|(\mathcal{M}^\dagger\widetilde{\phi}_s(\hat{h}_s, \mathbf{y}_{0:s+1}) - \mathbf{w}_s)\|,
$$
$$
\leq \alpha_A\varepsilon_{\mathrm{sys}}\bar{b}(1 - \gamma_A)^{-1} + \alpha_A\gamma_A^{\tau-1}\epsilon_0 + \alpha_A\|\mathcal{M}^\dagger\|_{\mathrm{op}}\sum_{s=0}^\tau \gamma_A^{\tau-s}\|\widetilde{\phi}_s(\hat{h}_s, \mathbf{y}_{0:s+1}) - \mathcal{M}\mathbf{w}_s\|,
$$
$$
\leq \alpha_A\varepsilon_{\mathrm{sys}}\bar{b}(1 - \gamma_A)^{-1} + \alpha_A\gamma_A^{\tau-1}\epsilon_0 + \alpha_A\|\mathcal{M}^\dagger\|_{\mathrm{op}}\sqrt{\sum_{s=0}^\tau \gamma_A^{2(\tau-s)}\sum_{s=0}^\tau \|\widetilde{\phi}_s(\hat{h}_s, \mathbf{y}_{0:s+1}) - \mathcal{M}\mathbf{w}_s\|^2},
$$

$$
\tag{I.36}
$$

Taking the square on both sides of (I.36), then applying the expectation $\mathbb{E}_{\widehat{\pi}}$, we get

$$\mathbb{E}_{\widehat{\pi}}\left[\max_{0\le\tau\le t}\|\tilde{f}_{\tau+1}(\mathbf{y}_{0:\tau+1})-f_{\star}(\mathbf{y}_{\tau})\|^2\cdot\mathbb{I}\{\mathscr{E}_{0:t}\}\right] \tag{I.37}$$

$$\le 3\alpha_A^2\varepsilon_{\text{sys}}^2\bar{b}^2(1-\gamma_A)^{-2}+3\alpha_A^2\varepsilon_{\text{init}}^2+3\alpha_A^2(1-\gamma_A^2)^{-1}\sigma_{\min}(\mathcal{M})^{-2}\sum_{s=0}^t\mathbb{E}_{\widehat{\pi}}\left[\|\widetilde{\phi}_s(\hat{h}_s,\mathbf{y}_{0:s+1})-\mathcal{M}\mathbf{w}_s\|^2\right],$$

$$\le 3\alpha_A^2\varepsilon_{\text{sys}}^2\bar{b}^2(1-\gamma_A)^{-2}+3\alpha_A^2\varepsilon_{\text{init}}^2+3\alpha_A^2(1-\gamma_A^2)^{-1}\sigma_{\min}(\mathcal{M})^{-2}\varepsilon_{\text{noise}}^2 t, \tag{I.38}$$

where the last inequality follows by the fact that under the event $\mathcal{E}_{\text{noise}}$, we have

$$\mathbb{E}_{\widehat{\pi}}\left[\|\widetilde{\phi}_s(\hat{h}_s,\mathbf{y}_{0:s+1})-\mathcal{M}\mathbf{w}_s\|^2\right]\le\varepsilon_{\text{noise}}^2,\quad\text{for all }0\le s\le t.$$

Finally, we simplify Eq. (I.38) to

$$3\alpha_A^2(1-\gamma_A)^{-2}\left(\varepsilon_{\text{sys}}^2\bar{b}^2+\varepsilon_{\text{init}}^2+\sigma_{\min}(\mathcal{M})^{-2}\varepsilon_{\text{noise}}^2 t\right).$$

$\square$

## I.6 Proof of Theorem 9

*Proof of Theorem 9.* Define $\mathscr{E}_{0:t}''\coloneqq\mathscr{E}_{0:t}\wedge\mathscr{E}_{0:t}'$ and let $p_t\coloneqq\mathbb{P}_{\widehat{\pi}}[\mathscr{E}_{0:t}'']$. We will recursively prove a lower bound on $p_{t+1}$ in terms of $p_t$. From Lemma H.3 and Markov's inequality, for all $\tau\in[t+1]$,

$$\mathbb{P}_{\widehat{\pi}}\left[\max_{\tau\in[t+1]}\|\tilde{f}_{\tau}(\mathbf{y}_{0:\tau})-f_{\star}(\mathbf{y}_{\tau})\|\ge\varepsilon_{\text{dec},t}\sqrt{\eta}\,\Big|\,\mathscr{E}_{0:t}''\right] \tag{I.39}$$

$$=\mathbb{P}_{\widehat{\pi}}\left[\max_{\tau\in[t+1]}\|\tilde{f}_{\tau}(\mathbf{y}_{0:\tau})-f_{\star}(\mathbf{y}_{\tau})\|^2\ge\eta\varepsilon_{\text{dec},t}^2\,\Big|\,\mathscr{E}_{0:t}''\right]$$

$$\le\frac{1}{\eta\varepsilon_{\text{dec},t}^2}\mathbb{E}_{\widehat{\pi}}\left[\max_{\tau\in[t+1]}\|\tilde{f}_{\tau}(\mathbf{y}_{0:\tau})-f_{\star}(\mathbf{y}_{\tau})\|^2\,\Big|\,\mathscr{E}_{0:t}''\right]$$

$$=\frac{1}{\eta p_t\varepsilon_{\text{dec},t}^2}\mathbb{E}_{\widehat{\pi}}\left[\max_{\tau\in[t+1]}\|\tilde{f}_{\tau}(\mathbf{y}_{0:\tau})-f_{\star}(\mathbf{y}_{\tau})\|^2\cdot\mathbb{I}\{\mathscr{E}_{0:t}''\}\right]$$

$$\le\frac{1}{\eta p_t\varepsilon_{\text{dec},t}^2}\mathbb{E}_{\widehat{\pi}}\left[\max_{\tau\in[t+1]}\|\tilde{f}_{\tau}(\mathbf{y}_{0:\tau})-f_{\star}(\mathbf{y}_{\tau})\|^2\cdot\mathbb{I}\{\mathscr{E}_{0:t}\}\right],$$

$$\le p_t^{-1}\eta^{-1}. \tag{I.40}$$

On the other hand, we also have that under the event $\mathscr{E}_{0:t}$, since no clipping occurs, the dynamics satisfy

$$\mathbf{x}_{t+1}=(A+B\widehat{K})\mathbf{x}_t+B\mathbf{v}_t+\boldsymbol{\delta}_t+\mathbf{w}_t,\quad\text{where }\boldsymbol{\delta}_t\coloneqq\widehat{B}\widehat{K}\tilde{f}_t(\mathbf{y}_{0:t})-B\widehat{K}f_{\star}(\mathbf{y}_t).$$

Thus, by induction we obtain,

$$\mathbf{x}_{t+1}=(A+B\widehat{K})^t\mathbf{x}_0+\sum_{\tau=0}^t(A+B\widehat{K})^{t-\tau}(B\mathbf{v}_t+\boldsymbol{\delta}_t+\mathbf{w}_t),$$

By Jensen's inequality, we have for $\bar{\gamma}_{\infty}$ as in (H.8),

$$\|\mathbf{x}_{t+1}\|=\alpha_{\infty}\bar{\gamma}_{\infty}^t\|\mathbf{x}_0\|+\alpha_{\infty}\sum_{\tau=0}^t\bar{\gamma}_{\infty}^{t-\tau}\|\boldsymbol{\delta}_{\tau}\|_2+\left\|\sum_{\tau=0}^t(A+B\widehat{K})^{t-\tau}(B\mathbf{v}_{\tau}+\mathbf{w}_{\tau})\right\|,$$

$$=\alpha_{\infty}\bar{\gamma}_{\infty}^t\|\mathbf{x}_0\|+\alpha_{\infty}\sum_{\tau=0}^t\bar{\gamma}_{\infty}^{t-\tau}\|\boldsymbol{\delta}_{\tau}\|_2+\|\boldsymbol{z}_t\|, \tag{I.41}$$

where $\boldsymbol{z}_t\coloneqq\sum_{\tau=0}^t(A+B\widehat{K})^{t-\tau}(B\mathbf{v}_{\tau}+\mathbf{w}_{\tau})$. In this case, we have $\boldsymbol{z}_t\sim\mathcal{N}(0,\Sigma_z)$, where

$$\Sigma_z\coloneqq\sum_{\tau=0}^t(A+B\widehat{K})^{t-\tau}(\sigma^2BB^{\intercal}+\Sigma_w)((A+B\widehat{K})^{t-\tau})^{\intercal}\le\Sigma_{z,\infty}, \tag{I.42}$$

with $\Sigma_{z,\infty}$ is as in (H.22). Under the event $\mathscr{E}'_{0:t}$, (and since $\bar{\gamma}_\infty < 1$) we have

$$b_\eta \coloneqq \sqrt{2(d_{\mathbf{u}}\|\Sigma_0\|_{\mathrm{op}}\alpha_\infty^2 + d_{\mathbf{x}}\|\Sigma_{z,\infty}\|_{\mathrm{op}})\ln(2\eta)} \geq \sqrt{2\hat{\alpha}_\infty^2\|\mathbf{x}_0\|^2 + 2\|\boldsymbol{z}_t\|^2},$$
$$\geq \alpha_\infty \bar{\gamma}_\infty^t \|\mathbf{x}_0\| + \|\boldsymbol{z}_t\|. \tag{I.43}$$

On the other hand, by Hölder's inequality, we have

$$\alpha_\infty \sum_{\tau=0}^t \bar{\gamma}_\infty^{t-\tau}\|\boldsymbol{\delta}_\tau\| \leq \frac{\alpha_\infty}{1-\bar{\gamma}_\infty}\max_{0\leq\tau\leq t}\|\boldsymbol{\delta}_\tau\|,$$
$$\leq \frac{\alpha_\infty}{1-\bar{\gamma}_\infty}\left(\varepsilon_{\mathrm{sys}}\bar{b} + \max_{0\leq\tau\leq t}\|B\widehat{K}(\tilde{f}_\tau(\mathbf{y}_{0:\tau}) - f_\star(\mathbf{y}_\tau))\|\right),$$
$$\leq \frac{\alpha_\infty}{1-\bar{\gamma}_\infty}\left(\varepsilon_{\mathrm{sys}}\bar{b} + 2\Psi_\star\max_{0\leq\tau\leq t}\|\tilde{f}_\tau(\mathbf{y}_{0:\tau}) - f_\star(\mathbf{y}_\tau)\|\right),$$

where we have used that under $\mathcal{E}_{\mathrm{sys}}$, $\|B\widehat{K}\|_{\mathrm{op}} \leq \|A\|_{\mathrm{op}} + \|A + B\widehat{K}\|_{\mathrm{op}} \leq \Psi_\star + \alpha_\infty\bar{\gamma}_\infty \leq 2\Psi_\star$. From (I.40), it follows that

$$\mathbb{P}_{\widehat{\pi}}\left[\alpha_\infty\sum_{\tau=0}^t\bar{\gamma}_\infty^{t-\tau}\|\boldsymbol{\delta}_\tau\| \geq \frac{\alpha_\infty(\varepsilon_{\mathrm{sys}}\bar{b} + 2\Psi_\star\varepsilon_{\mathrm{dec},t}\sqrt{\eta})}{1-\bar{\gamma}_\infty} \ \middle|\ \mathscr{E}''_{0:t}\right] \leq p_t^{-1}\eta^{-1}. \tag{I.44}$$

Thus, by (I.41), (I.40), (I.43), and (I.44), we have

$$\mathbb{P}_{\widehat{\pi}}[\mathscr{E} \mid \mathscr{E}''_{0:t}] \leq \eta^{-1}p_t^{-1},$$

where

$$\mathscr{E} \coloneqq \left\{\begin{array}{c}\|\mathbf{x}_{t+1}\| \geq (1-\bar{\gamma}_\infty)^{-1}\alpha_\infty(\varepsilon_{\mathrm{sys}}\bar{b} + 2\Psi_\star\varepsilon_{\mathrm{dec},t}\sqrt{\eta}) + b_\eta,\\ \text{or}\ \ \|\tilde{f}_{t+1}(\mathbf{y}_{0:t+1})\| \geq \|\mathbf{x}_{t+1}\| + \varepsilon_{\mathrm{dec},t}\sqrt{\eta}\end{array}\right\}.$$

This implies that

$$\mathbb{P}_{\widehat{\pi}}\left[\|\tilde{f}_{t+1}(\mathbf{y}_{t+1})\| \geq \frac{\alpha_\infty\varepsilon_{\mathrm{sys}}\bar{b} + (2\Psi_\star\alpha_\infty + 1 - \bar{\gamma}_\infty)\varepsilon_{\mathrm{dec},t}\sqrt{\eta}}{1-\bar{\gamma}_\infty} + b_\eta \ \middle|\ \mathscr{E}''_{0:t}\right] \leq \eta^{-1}p_t^{-1}. \tag{I.45}$$

which we simplify to

$$\mathbb{P}_{\widehat{\pi}}\left[\|\tilde{f}_{t+1}(\mathbf{y}_{t+1})\| \geq \frac{\alpha_\infty\varepsilon_{\mathrm{sys}}\bar{b} + 3\Psi_\star\alpha_\infty\varepsilon_{\mathrm{dec},t}\sqrt{\eta}}{1-\bar{\gamma}_\infty} + b_\eta \ \middle|\ \mathscr{E}''_{0:t}\right] \leq \eta^{-1}p_t^{-1}. \tag{I.46}$$

On the other hand, by Lemma I.11, we have

$$\mathbb{P}_{\widehat{\pi}}\left[\alpha_\infty^2\|\mathbf{x}_0\|^2 + \|\boldsymbol{z}_{t+1}\|^2 \geq b_\eta 2^{-1/2}\right] \leq \eta^{-1}. \tag{I.47}$$

Thus, for

$$\bar{b} \geq (1-\bar{\gamma}_\infty)^{-1}(\alpha_\infty\varepsilon_{\mathrm{sys}}\bar{b} + 3\Psi_\star\alpha_\infty\varepsilon_{\mathrm{dec},t}\sqrt{\eta}) + b_\eta, \tag{I.48}$$

we have with (I.47), (I.46), and a union bound,

$$\mathbb{P}_{\widehat{\pi}}\left[\left\{\|\tilde{f}_{t+1}(\mathbf{y}_{t+1})\| \geq \bar{b}\right\} \vee \left\{\alpha_\infty^2\|\mathbf{x}_0\|^2 + \|\boldsymbol{z}_{t+1}\|^2 \geq b_\eta 2^{-1/2}\right\} \mid \mathscr{E}''_{0:t}\right] \leq 2\eta^{-1}p_t^{-1}.$$

This implies that

$$\mathbb{P}_{\widehat{\pi}}[\mathscr{E}_{t+1} \wedge \mathscr{E}'_{t+1} \mid \mathscr{E}''_{0:t}] \geq 1 - 2\eta^{-1}p_t^{-1}. \tag{I.49}$$

Therefore, we have

$$\mathbb{P}_{\widehat{\pi}}[\mathscr{E}''_{0:t+1}] = p_t \cdot \mathbb{P}_{\widehat{\pi}}[\mathscr{E}_{t+1} \wedge \mathscr{E}'_{t+1} \mid \mathscr{E}''_{0:t}] \geq p_t - 2\eta^{-1} = \mathbb{P}_{\widehat{\pi}}[\mathscr{E}''_{0:t}] - 2\eta^{-1}.$$

Now by induction on $t$ we get, for all $\tau \geq 1$,

$$\mathbb{P}_{\widehat{\pi}}[\mathscr{E}''_{0:\tau}] \geq \mathbb{P}_{\widehat{\pi}}[\mathscr{E}_{0:1} \wedge \mathscr{E}_{0:1'}] - 2\tau/\eta. \tag{I.50}$$

For the base case, by (I.47) and a union bound, it follows that

$$\mathbb{P}_{\widehat{\pi}}[\neg\mathscr{E}'_{0:1}] \leq 2\eta^{-1}, \tag{I.51}$$

and therefore, by (I.50), we get

$$\mathbb{P}_{\widehat{\pi}}[\mathscr{E}''_{0:\tau}] \geq \mathbb{P}_{\widehat{\pi}}[\mathscr{E}_{0:1}] - 2(\tau+1)/\eta.$$

Finally, as $\tilde{f}_0 \equiv 0$, we have $\mathbb{P}_{\widehat{\pi}}[\mathscr{E}_{0:1}] = \mathbb{P}_{\widehat{\pi}}[\mathscr{E}_1]$, which completes the proof. To get the stated value for $\bar{b}_\infty$, we rearrange Eq. (I.48) and recall that $\bar{\gamma}_\infty = \frac{1}{2}(1+\gamma_\infty)$. $\qquad\square$

## I.7 Proof of Theorem 10

*Proof.* Let $\mathscr{E}''_{0:T} \coloneqq \mathscr{E}_{0:T} \wedge \mathscr{E}'_{0:T}$. By Lemma H.3, under the events $\mathcal{E}_{\mathrm{sys}}$ and $\mathcal{E}_{\mathrm{noise}}$, we have

$$\mathbb{E}_{\widehat{\pi}}\left[\max_{0 \le t \le T} \|\hat{f}_t(\mathbf{y}_{0:t}) - f_\star(\mathbf{y}_t)\|^2 \cdot \mathbb{I}\{\mathscr{E}_{0:T}\}\right] \le \varepsilon_{\mathrm{dec},T}^2. \tag{I.52}$$

It follows that for all $0 \le t \le T$, we have

$$\mathbb{E}_{\widehat{\pi}}\left[\max_{0 \le t \le T} \|\hat{f}_t(\mathbf{y}_{0:t}) - f_\star(\mathbf{y}_t)\|^2\right] \le \mathbb{E}_{\widehat{\pi}}\left[\max_{0 \le t \le T} \|\hat{f}_t(\mathbf{y}_{0:t}) - f_\star(\mathbf{y}_t)\|^2 \cdot \mathbb{I}\{\mathscr{E}''_{0:T}\}\right]$$
$$+ \mathbb{E}_{\widehat{\pi}}\left[\max_{0 \le t \le T} \|\hat{f}_t(\mathbf{y}_{0:t}) - f_\star(\mathbf{y}_t)\|^2 \cdot \mathbb{I}\{\neg \mathscr{E}''_{0:T}\}\right],$$
$$\le \mathbb{E}_{\widehat{\pi}}\left[\max_{0 \le t \le T} \|\hat{f}_t(\mathbf{y}_{0:t}) - f_\star(\mathbf{y}_t)\|^2 \cdot \mathbb{I}\{\mathscr{E}_{0:T}\}\right]$$
$$+ \mathbb{E}_{\widehat{\pi}}\left[(2\bar{b} + 2\max_{0 \le t \le T} \|\mathbf{x}_t\|^2) \cdot \mathbb{I}\{\neg \mathscr{E}''_{0:T}\}\right],$$
$$\le \varepsilon_{\mathrm{dec},T}^2 + 2(1 - \mathbb{P}_{\widehat{\pi}}[\mathscr{E}''_{0:T}])\left(\bar{b} + \sqrt{\mathbb{E}_{\widehat{\pi}}\left[\max_{0 \le t \le T} \|\mathbf{x}_t\|^4\right]}\right), \tag{I.53}$$

where the last inequality follows by Cauchy Schwarz and Eq. (I.52). Now by Lemma I.1, we have that the random variable $\|\mathbf{x}_t\|^2$ is $c_{\mathbf{x}}$-concentrated for all $0 \le t \le T$ with

$$c_{\mathbf{x}} \coloneqq 30 d_{\mathbf{x}} \Psi_\Sigma + 2\bar{b}^2 \cdot \mathsf{dev}_x.$$

Therefore, by Lemma E.1, we have $\mathbb{E}_{\widehat{\pi}}\left[\|\mathbf{x}_t\|^4\right] \le 4c_{\mathbf{x}}^2$. Using this, we get that

$$\mathbb{E}_{\widehat{\pi}}\left[\max_{0 \le t \le T} \|\mathbf{x}_t\|^4\right] \le (T+1) \max_{0 \le t \le T} \mathbb{E}_{\widehat{\pi}}\left[\|\mathbf{x}_t\|^4\right] \le 8T c_{\mathbf{x}}^2.$$

Combining this with (I.53), and Theorem 9, we have

$$\mathbb{E}_{\widehat{\pi}}\left[\max_{0 \le t \le T} \|\hat{f}_t(\mathbf{y}_{0:t}) - f_\star(\mathbf{y}_t)\|^2\right] \le \varepsilon_{\mathrm{dec},t}^2 + (4T^{1/2} c_{\mathbf{x}} + 2\bar{b}^2)(1 - \mathbb{P}_{\widehat{\pi}}[\mathscr{E}''_{0:T}]).$$
$$\le \varepsilon_{\mathrm{dec},t}^2 + (4T^{1/2} c_{\mathbf{x}} + 2\bar{b}^2)(\tfrac{2(T+1)}{\eta} + 1 - \mathbb{P}_{\widehat{\pi}}[\{\tilde{f}_0(\mathbf{y}_0) = \hat{f}_0(\mathbf{y}_0)\} \wedge \mathscr{E}'_0]).$$

$\square$

## I.8 Proof of Lemma H.4

For the proof of Lemma H.4, we introduce the following functions and random vectors:

$$\varphi(\mathbf{y}_1) \coloneqq L_{\mathrm{op}}^2(1 \vee \|\mathbf{x}_1\|^2),$$
$$\mathbf{u} \coloneqq \mathbf{w}_0,$$
$$\mathbf{e} \coloneqq \mathbf{w}_0 - (\hat{h}_{\mathrm{ol},0}(\mathbf{y}_1^{(i)}) - \widehat{A}\hat{h}_{\mathrm{ol},0}(\mathbf{y}_0^{(i)}) - \widehat{B}\mathbf{v}_0).$$

Let us abbreviate $n \equiv n_{\mathrm{init}}$. Recall that we are analyzing the regression

$$\hat{h}_{\mathrm{ol},1} \in \operatorname*{arg\,min}_{h \in \mathscr{H}_{\mathrm{op}}} \sum_{i=1}^n \left\|h(\mathbf{y}_1^{(i)}) - \left(\hat{h}_{\mathrm{ol},0}(\mathbf{y}_1^{(i)}) - \widehat{A}\hat{h}_{\mathrm{ol},0}(\mathbf{y}_0^{(i)}) - \widehat{B}\mathbf{v}_0^{(i)}\right)\right\|^2,$$

where $\{(\mathbf{y}_\tau^{(i)}, \mathbf{x}_\tau^{(i)}, \mathbf{v}_\tau^{(i)})\}_{1 \le i \le n}$ are fresh i.i.d. trajectories generated by the policy $\pi_{\mathrm{ol}}$.

*Proof of Lemma H.4.* Our strategy will be to invoke Corollary E.1 with $\varphi$, $\mathbf{u}$, and $\mathbf{e}$ as above. We start by verifying the technical conditions of the corollary.

1. We directly verify from the structure of $\mathscr{H}_{\mathrm{op}}$ we may take $b = \Psi_\star^3$ and $L$ as in Assumption 5. Hence, $bL = L_{\mathrm{op}}$, and thus $\varphi(\mathbf{y}_1)$ satisifes the requisite conditions of the $\varphi$ function. In addition, we may take $d_{\mathbf{u}}, d_{\mathbf{x}} \leftarrow d_{\mathbf{x}}$.

2. **Concentration Property.** Next, we bound the concentration parameter $c$. Recall that all $h \in \mathcal{H}_{\mathrm{op}}$ have $\|h(\mathbf{y}_t)\| \le L_{\mathrm{op}} \max\{1, \|\mathbf{x}_t\|\}$. Hence, under the event $\mathcal{E}_{\mathrm{sys}}$, we have by Jensen's inequality, Cauchy-Schwarz, and the fact that $\hat{h}_{\mathrm{ol},0} \in \mathcal{H}_{\mathrm{op}}$,

$$
\begin{aligned}
\varphi(\mathbf{y}_1) \vee \|\mathbf{e} - \mathbf{u}\|^2 &\le \varphi(\mathbf{y}_1) + 5L_{\mathrm{op}}^2(1 + \|\mathbf{x}_1\|^2) + 5L_{\mathrm{op}}^2(\|A\|_{\mathrm{op}}^2 + \varepsilon_{\mathrm{sys}}^2)(1 + \|\mathbf{x}_0\|^2) \\
&\quad + 5(\|B\|_{\mathrm{op}}^2 + \varepsilon_{\mathrm{sys}}^2)\|\mathbf{v}_0\|^2, \\
&\le 6L_{\mathrm{op}}^2(1 + \|\mathbf{x}_1\|^2) + 5L_{\mathrm{op}}^2(\|A\|_{\mathrm{op}}^2 + \varepsilon_{\mathrm{sys}}^2)(1 + \|\mathbf{x}_0\|^2) \\
&\quad + 5(\|B\|_{\mathrm{op}}^2 + \varepsilon_{\mathrm{sys}}^2)\|\mathbf{v}_0\|^2, \\
&= 6L_{\mathrm{op}}^2 + 5L_{\mathrm{op}}^2(\|A\|_{\mathrm{op}}^2 + \varepsilon_{\mathrm{sys}}^2) + 5(\|B\|_{\mathrm{op}}^2 + \varepsilon_{\mathrm{sys}}^2)\|\mathbf{v}_0\|^2 \quad\quad \text{(I.54)} \\
&\quad + 6L_{\mathrm{op}}^2\|\mathbf{x}_1\|^2 + 5L_{\mathrm{op}}^2(\|A\|_{\mathrm{op}}^2 + \varepsilon_{\mathrm{sys}}^2)\|\mathbf{x}_0\|^2.
\end{aligned}
$$

Let us simplify the above. Assume $\varepsilon_{\mathrm{sys}}^2 \le \Psi_\star^2$ (where $\Psi_\star \ge \max\{1, \|A\|_{\mathrm{op}}, \|B\|_{\mathrm{op}}\}$). This lets us simplify the above by

$$
\begin{aligned}
\varphi(\mathbf{y}_1) \vee \|\mathbf{e} - \mathbf{u}\|^2 &\le L_{\mathrm{op}}^2 \Psi_\star^2(6 + 10 + 10\|\mathbf{v}_0\|^2 + 6\|\mathbf{x}_1\|^2 + 10\|\mathbf{x}_0\|^2) \\
&\le 16L_{\mathrm{op}}^2 \Psi_\star^2(1 + \|\mathbf{v}_0\|^2 + \|\mathbf{x}_1\|^2 + \|\mathbf{x}_0\|^2). \quad\quad \text{(I.55)}
\end{aligned}
$$

Since $\mathbf{x}_0 \sim \mathcal{N}(0, \Sigma_0)$, $\mathbf{v}_0 \sim \mathcal{N}(0, \sigma^2 I_{d_{\mathbf{u}}})$, and $\mathbf{x}_1 \sim \mathcal{N}(0, \sigma^2 BB^\top + A\Sigma_0 A^\top + \Sigma_w)$, we have by Lemma I.11 and the fact that $\ln 3 + 1 \le 3$, the following holds: For all $\delta \in (0, 1/e]$,

$$
\mathbb{P}_{\pi_{\mathrm{ol}}}\left[\varphi(\mathbf{y}_1) \vee \|\mathbf{e} - \mathbf{u}\|^2 \ge c_1 \ln \delta^{-1}\right] \le \delta, \quad\quad \text{(I.56)}
$$

where

$$
c_1 := 48L_{\mathrm{op}}^2 \Psi_\star^2(1 + 5d_{\mathbf{u}}\sigma^2 + 5d_{\mathbf{x}}(\|\Sigma_1\|_{\mathrm{op}} + \|\Sigma_0\|_{\mathrm{op}})), \quad\quad \text{(I.57)}
$$

$$
\Sigma_1 := \sigma^2 BB^\top + A\Sigma_0 A^\top + \Sigma_w. \quad\quad \text{(I.58)}
$$

3. **Bounding the error $\mathbf{e}$.** On the other hand, by Jensen's inequality and Cauchy-Schwarz, we can bound the error $\mathbf{e}$ by

$$
\|\mathbf{e}\|^2 \le 2\|\mathbf{w}_0 - \hat{h}_{\mathrm{ol},0}(\mathbf{y}_1^{(i)}) - A\hat{h}_{\mathrm{ol},0}(\mathbf{y}_0^{(i)}) - B\mathbf{v}_0\|^2 + 4\varepsilon_{\mathrm{sys}}^2 L_{\mathrm{op}}^2\left(1 + \|\mathbf{x}_0\|^2 + \|\mathbf{v}_0\|^2\right).
$$

Therefore, by Lemma I.9 and (H.24), we have with probablity at least $1 - \delta$ over the trajectories used to form $\hat{h}_{\mathrm{ol},0}$,

$$
\mathbb{E}_{\pi_{\mathrm{ol}}}[\|\mathbf{e}\|^2] \le 2\sigma_{\min}(\mathcal{M})^{-2}\varepsilon_{\mathrm{noise}}^2(\delta) + 4\varepsilon_{\mathrm{sys}}^2 L_{\mathrm{op}}^2(1 + d_{\mathbf{x}}\|\Sigma_0\|_{\mathrm{op}} + \sigma^2 d_{\mathbf{u}}). \quad\quad \text{(I.59)}
$$

4. **Realizability.** We have

$$
\begin{aligned}
\mathbb{E}_{\pi_{\mathrm{ol}}}[\mathbf{u} \mid \mathbf{y}_1] &= \mathbb{E}_{\pi_{\mathrm{ol}}}[\mathbf{w}_0 \mid \mathbf{x}_1], \\
&= \mathbb{E}_{\pi_{\mathrm{ol}}}[\mathbf{w}_0 \mid A\mathbf{x}_0 + B\mathbf{v}_0 + \mathbf{w}_0], \\
&= \Sigma_w(\Sigma_w + \sigma^2 BB^\top + A\Sigma_0 A^\top)^{-1}\mathbf{x}_1 =: h_\star(\mathbf{y}_1), \quad\quad \text{(I.60)}
\end{aligned}
$$

where the last inequality follows by Fact G.2. Therefore, by the definition of $\mathcal{H}_{\mathrm{op}}$ in (H.1) and the fact that $\|\Sigma_w(\Sigma_w + \sigma^2 BB^\top + A\Sigma_0 A^\top)^{-1}\|_{\mathrm{op}} \le 1 \le \Psi_\star^3$, we are guaranteed the existence of $h \in \mathcal{H}_{\mathrm{op}}$ such that $h(\mathbf{x}_1) = \mathbb{E}[\mathbf{u} \mid \mathbf{y}_1]$.

Applying Corollary E.1 with $c \leftarrow c_1$, $d_{\mathbf{u}}, d_{\mathbf{x}} \leftarrow d_{\mathbf{x}}$, and the above bound on $\mathbb{E}_{\pi_{\mathrm{ol}}}[\|\mathbf{e}\|^2]$, we obtain for $n \leftarrow n_{\mathrm{init}}$ that with probability $1 - \frac{3\delta}{2} - \delta = 1 - 5\delta/2$ (the second $\delta$ factor comes from the event used to bound $\mathbb{E}_{\pi_{\mathrm{ol}}}[\|\mathbf{e}\|^2]$),

$$
\mathbb{E}\|\hat{h}_{\mathrm{ol},1}(\mathbf{y}) - h_\star(\mathbf{y})\|^2 \lesssim
$$
$$
\frac{c_1(d_{\mathbf{x}}^2 + \ln|\mathscr{F}|)\ln^2 \frac{n_{\mathrm{init}}}{\delta}}{n_{\mathrm{init}}} + \sigma_{\min}(\mathcal{M})^{-2}\varepsilon_{\mathrm{noise}}^2(\delta) + \varepsilon_{\mathrm{sys}}^2 L_{\mathrm{op}}^2(1 + d_{\mathbf{x}}\|\Sigma_0\|_{\mathrm{op}} + \sigma^2 d_{\mathbf{u}}),
$$

as needed. Moreover, because the above bound suppresses constants, we can replace $c_1$ in Eq. (I.57) by $c_1 \leftarrow L_{\mathrm{op}}^2 \Psi_\star^2(1 + d_{\mathbf{u}}\sigma^2 + d_{\mathbf{x}}(\|\Sigma_1\|_{\mathrm{op}} + \|\Sigma_0\|_{\mathrm{op}}))$. Substituting in the definition of $h_\star$ concludes the proof. $\square$

## I.9 Proof of Lemma H.5

*Proof.* Since $\hat{h}_{\mathrm{ol},1} \in \mathscr{H}_{\mathrm{op}}$, we have $\|\hat{h}_{\mathrm{ol},1}(\mathbf{y}_1)\| \leq L_{\mathrm{op}}(1 \vee \|\mathbf{x}_1\|)$, where $\mathbf{x}_1 \sim \mathcal{N}(0, \Sigma_1)$ and $\Sigma_1 := \sigma^2 BB^\top + A\Sigma_0 A^\top + \Sigma_w$. Therefore, by Lemma I.11, we have, for all $\delta \in (0, 1/e]$,

$$\mathbb{P}_{\pi_{\mathrm{ol}}}\left[ \|\hat{h}_{\mathrm{ol},1}\|^2 \geq L_{\mathrm{op}}^2 (1 + (3d_{\mathbf{x}} + 2)\|\sigma^2 BB^\top + A\Sigma_0 A^\top + \Sigma_w\|_{\mathrm{op}}) \ln \delta^{-1} \right] \leq \delta. \tag{I.61}$$

Combining this with the fact that $\|\hat{h}_{\mathrm{ol},1}(\mathbf{y}_1)\| \leq L_{\mathrm{op}}(1 \vee \|\mathbf{x}_1\|)$ implies that $\varphi(\mathbf{y}_1) := L_{\mathrm{op}}^2(1 \vee \|\mathbf{x}_1\|^2)$ is $c$-concentrated with $c = L_{\mathrm{op}}^2(1 + (3d_{\mathbf{x}} + 2)\|\sigma^2 BB^\top + A\Sigma_0 A^\top + \Sigma_w\|_{\mathrm{op}})$. Thus, applying Proposition E.2 with

$$(\varphi(\mathbf{y}_1), \hat{h}(\mathbf{y}_1), h_\star(\mathbf{y}_1)) = (L_{\mathrm{op}}^2(1 \vee \|\mathbf{x}_1\|^2), \hat{h}_{\mathrm{ol},1}(\mathbf{y}_1), \Sigma_w \Sigma_1^{-1} \mathbf{x}_1)$$

and invoking Lemma H.4, we get for all $\delta \in (0, 1/e]$, with probability at least $1 - (3\kappa + 4)\delta$,

$$\|\widehat{\Sigma}_{\mathrm{cov}} - \Sigma_w \Sigma_1^{-1} \Sigma_w\|_{\mathrm{op}} \leq \varepsilon'_{\mathrm{cov}}, \tag{I.62}$$

for $\varepsilon'_{\mathrm{cov}}$ as in the lemma statement. By the triangle inequality, whenever the condition (H.30) that $\varepsilon'_{\mathrm{cov}} < \sigma_{\min}(\Sigma_{\mathrm{cov}})/2 \leq \|\Sigma_{\mathrm{cov}}\|_{\mathrm{op}}$ holds, this implies that

$$\|\widehat{\Sigma}_{\mathrm{cov}}\|_{\mathrm{op}} \leq 2\|\Sigma_w \Sigma_1^{-1} \Sigma_w\|_{\mathrm{op}}, \quad \text{and} \quad \sigma_{\min}(\widehat{\Sigma}_{\mathrm{cov}}) \geq \sigma_{\min}(\Sigma_{\mathrm{cov}})/2 \tag{I.63}$$

which shows the second inequality in (H.31). Furthermore, whenever (I.62) holds, Lemma I.12 and the condition Eq. (H.30) imply that

$$\|\Sigma_w^{-1} - \widehat{\Sigma}_{\mathrm{cov}}^{-1} \Sigma_w \Sigma_1^{-1}\|_{\mathrm{op}} \leq \frac{2\|\Sigma_w^{-1}\|_{\mathrm{op}} \varepsilon'_{\mathrm{cov}}}{\sigma_{\min}(\Sigma_w \Sigma_1^{-1} \Sigma_w)}. \tag{I.64}$$

By the triangle inequality, this implies that under $\mathcal{E}_{\mathrm{sys}}$,

$$\|\widehat{\Sigma}_w^{-1} - \widehat{\Sigma}_{\mathrm{cov}}^{-1} \Sigma_w \Sigma_1^{-1}\|_{\mathrm{op}} \leq \varepsilon_{\mathrm{sys}} + \frac{2\|\Sigma_w^{-1}\|_{\mathrm{op}} \varepsilon'_{\mathrm{cov}}}{\sigma_{\min}(\Sigma_w \Sigma_1^{-1} \Sigma_w)}.$$

This further implies that

$$\|I_{d_{\mathbf{x}}} - \widehat{\Sigma}_w \widehat{\Sigma}_{\mathrm{cov}}^{-1} \Sigma_w \Sigma_1^{-1}\|_{\mathrm{op}} \leq \|\widehat{\Sigma}_w\|_{\mathrm{op}} \left( \varepsilon_{\mathrm{sys}} + \frac{2\|\Sigma_w^{-1}\|_{\mathrm{op}} \varepsilon'_{\mathrm{cov}}}{\sigma_{\min}(\Sigma_w \Sigma_1^{-1} \Sigma_w)} \right). \tag{I.65}$$

Since $\varepsilon_{\mathrm{sys}} \leq 1$ by the definition of $\mathcal{E}_{\mathrm{sys}}$, we have that $\|\widehat{\Sigma}_w\|_{\mathrm{op}} \leq 2\Psi_\star$, leading to the result. This establishes the main inequality in (H.31). $\square$

## I.10 Proof of Theorem 11

For the proof of Theorem 11, we introduce the following functions and random vectors:

$$\begin{aligned}
\varphi(\mathbf{y}_0) &:= L_{\mathrm{op}}^2(1 \vee \|\mathbf{x}_0\|^2) \\
\mathbf{u} &:= \Sigma_w(\Sigma_w + \sigma^2 BB^\top + A\Sigma_0 A^\top)^{-1}(A\mathbf{x}_0 + \mathbf{v}_0 + \mathbf{w}_0) \\
\mathbf{e} &:= \mathbf{u} - \hat{h}_{\mathrm{ol},1}(\mathbf{y}_1).
\end{aligned}$$

Recall that we are analyzing the following regression problem, where for $n \equiv n_{\mathrm{init}}$:

$$\tilde{h}_{\mathrm{ol},0} \in \underset{h \in \mathscr{H}_{\mathrm{op}}}{\arg\min} \sum_{i=n+1}^{2n} \left\| h(\mathbf{y}_0^{(i)}) - \hat{h}_{\mathrm{ol},1}(\mathbf{y}_1^{(i)}) \right\|^2.$$

*Proof of Theorem 11.* Our strategy will be to invoke Corollary E.1 with $\varphi$, $\mathbf{u}$, and $\mathbf{e}$ as above. We verify the technical conditions of the corollary.

1. We directly verify from the structure of $\mathscr{H}_{\mathrm{op}}$ we may take $b = \Psi_\star^3$ and $L$ as in Assumption 5. Hence, $bL = L_{\mathrm{op}}$, and thus $\varphi(\mathbf{y}_1)$ satisifes the requisite conditions of the $\varphi$ function.

2. **Concentration property.** Now, under event $\mathcal{E}_{\text{sys}}$, by Jensen's inequality, Cauchy-Schwarz, and the fact that $\hat{h}_{\text{ol},1} \in \mathscr{H}_{\text{op}}$ (and hence satisfies $\|\hat{h}_{\text{ol},1}(\mathbf{y}_1)\| \leq L_{\text{op}} \max\{1, \|\mathbf{x}_1\|\}$), we have

$$\varphi(\mathbf{y}_0) \vee \|\mathbf{e} - \mathbf{u}\|^2 \leq \varphi(\mathbf{y}_0) \vee (L_{\text{op}}^2(1 + \|\mathbf{x}_1\|^2)),$$
$$\leq L_{\text{op}}^2(1 + \|\mathbf{x}_0\|^2 + \|\mathbf{x}_1\|^2).$$

Since $\mathbf{x}_0 \sim \mathcal{N}(0, \Sigma_0)$ and $\mathbf{x}_1 \sim \mathcal{N}(0, \Sigma_1)$, where $\Sigma_1 = \Sigma_w + \sigma^2 BB^\top + A\Sigma_0 A^\top$, we have by Lemma I.11 that for all $\delta \in (0, 1/e]$,

$$\mathbb{P}_{\pi_{\text{ol}}}\left[\varphi(\mathbf{y}_0) \vee \|\mathbf{e} - \mathbf{u}\|^2 \geq c_0' \ln \delta^{-1}\right] \leq \delta, \quad \text{where} \quad c_0' := 2L_{\text{op}}^2(1 + 5d_{\mathbf{x}}(\|\Sigma_0\|_{\text{op}} + \|\Sigma_1\|_{\text{op}})). \tag{I.66}$$

Moreover, since $\sigma \leq 1$ under Assumption 10, we have $\|\Sigma_1\| \leq 3\Psi_\star^3$, so that

$$c_0' \leq 32L_{\text{op}}^2\Psi_\star^3 d_{\mathbf{x}} =: c_0,$$

and hence

$$\mathbb{P}_{\pi_{\text{ol}}}\left[\varphi(\mathbf{y}_0) \vee \|\mathbf{e} - \mathbf{u}\|^2 \geq c_0 \ln \delta^{-1}\right] \leq \delta. \tag{I.67}$$

3. **Bounding the error e.** By Lemma H.4, we have with probablity at least $1 - 5\delta/2$ over $\hat{h}_{\text{ol},1}$,

$$\mathbb{E}_{\pi_{\text{ol}}}[\|\mathbf{e}\|^2] \leq \varepsilon_{\text{ol},1}^2. \tag{I.68}$$

4. **Realizability.** We have

$$\mathbb{E}_{\pi_{\text{ol}}}[\mathbf{u} \mid \mathbf{y}_0] = \mathbb{E}_{\pi_{\text{ol}}}[\mathbf{u} \mid \mathbf{x}_0],$$
$$= \mathbb{E}_{\pi_{\text{ol}}}[\Sigma_w(\Sigma_w + \sigma^2 BB^\top + A\Sigma_0 A^\top)^{-1}(A\mathbf{x}_0 + \mathbf{v}_0 + \mathbf{w}_0) \mid \mathbf{x}_0],$$
$$= \Sigma_w(\Sigma_w + \sigma^2 BB^\top + A\Sigma_0 A^\top)^{-1}A\mathbf{x}_0,$$
$$= \Sigma_w\Sigma_1^{-1}A\mathbf{x}_0 := h_\star(\mathbf{x}_0). \tag{I.69}$$

Therefore, by the definition of $\mathscr{H}_{\text{op}}$ in (H.1) and the fact that $\|\Sigma_w\Sigma_1^{-1}A\|_{\text{op}} \leq \Psi_\star^3$, we are guaranteed the existence of $h \in \mathscr{H}_{\text{op}}$ such that $h(\mathbf{x}_0) = \mathbb{E}[\mathbf{u} \mid \mathbf{y}_0]$.

Applying Corollary E.1 with $c \leftarrow c_0$, $d_{\mathbf{u}}, d_{\mathbf{x}} \leftarrow d_{\mathbf{x}}$, and the above bound on $\mathbb{E}_{\pi_{\text{ol}}}[\|\mathbf{e}\|^2]$, we obtain for $n \leftarrow n_{\text{init}}$ that with probability at least $1 - \frac{3\delta}{2} - \frac{5\delta}{2} = 1 - 4\delta$ (the second term comes from the event used to bound $\mathbb{E}_{\pi_{\text{ol}}}[\|\mathbf{e}\|^2]$),

$$\mathbb{E}\|\tilde{h}_{\text{ol},0}(\mathbf{y}_0) - h_\star(\mathbf{y})\|^2 \lesssim \frac{c_0(d_{\mathbf{x}}^2 + \ln|\mathscr{F}|)\ln(\frac{n_{\text{init}}}{\delta})^2}{n_{\text{init}}} + \varepsilon_{\text{ol},1}^2$$

as needed. In particular, recalling that $h_\star(\mathbf{y}_0) = \Sigma_w\Sigma_1^{-1}A\mathbf{x}_0$ in the above realizability discussion, we find that for an appropriate upper bound $\tilde{\varepsilon}_{\text{ol},1}^2$,

$$\mathbb{E}\|\tilde{h}_{\text{ol},0}(\mathbf{y}_0) - \Sigma_w\Sigma_1^{-1}A\mathbf{x}_0\|^2 \leq \tilde{\varepsilon}_{\text{ol},1}^2 \lesssim \frac{c_0(d_{\mathbf{x}}^2 + \ln|\mathscr{F}|)\ln(\frac{n_{\text{init}}}{\delta})^2}{n_{\text{init}}} + \varepsilon_{\text{ol},1}^2$$

This further implies that

$$\mathbb{E}_{\pi_{\text{ol}}}\left[\|\widehat{\Sigma}_w\widehat{\Sigma}_{\text{cov}}^{-1}\tilde{h}_{\text{ol},0}(\mathbf{y}_0) - A\mathbf{x}_0\|^2\right] \leq 2\|\widehat{\Sigma}_w^{-1}\|_{\text{op}}^2\|\widehat{\Sigma}_{\text{cov}}\|_{\text{op}}^2\tilde{\varepsilon}_{\text{ol},1}^2$$
$$+ 2\mathbb{E}_{\pi_{\text{ol}}}\left[\|(I_{d_{\mathbf{x}}} - \widehat{\Sigma}_w\widehat{\Sigma}_{\text{cov}}^{-1}\Sigma_w\Sigma_1^{-1})A\mathbf{x}_0\|^2\right],$$
$$\leq 2\|\widehat{\Sigma}_w^{-1}\|_{\text{op}}^2\|\widehat{\Sigma}_{\text{cov}}\|_{\text{op}}^2\tilde{\varepsilon}_{\text{ol},1}^2$$
$$+ 2d_{\mathbf{x}}\|(I_{d_{\mathbf{x}}} - \widehat{\Sigma}_w\widehat{\Sigma}_{\text{cov}}^{-1}\Sigma_w\Sigma_1^{-1})A\|_{\text{op}}^2\|\Sigma_0\|, \tag{I.70}$$

where the last inequality follows by Lemma I.9 since $\mathbf{x}_0 \sim \mathcal{N}(0, \Sigma_0)$. Thus, under the event $\mathcal{E}_{\text{sys}}$, we have by Lemma H.5 and a union bound, with probability at least $1 - (3\kappa + 9)\delta$,

$$\mathbb{E}_{\pi_{\text{ol}}}\left[\|\widehat{\Sigma}_w\widehat{\Sigma}_{\text{cov}}^{-1}\tilde{h}_{\text{ol},0}(\mathbf{y}_0) - A\mathbf{x}_0\|^2\right] \lesssim \|\Sigma_w^{-1}\|_{\text{op}}^2\|\Sigma_{\text{cov}}\|_{\text{op}}^2\left(\frac{c_0(d_{\mathbf{x}}^2 + \ln|\mathscr{F}|)\ln(\frac{n_{\text{init}}}{\delta})^2}{n_{\text{init}}} + \varepsilon_{\text{ol},1}^2\right)$$
$$+ d_{\mathbf{x}}\varepsilon_{\text{cov}}^2\|A\|_{\text{op}}^2\|\Sigma_0\|, \tag{I.71}$$

where we recall that $\Sigma_{\mathrm{cov}} = \Sigma_w \Sigma_1^{-1} \Sigma_w$ by definition. The desired bound (H.34) follows by the fact $\pi_{\mathrm{ol}}$ and $\widehat{\pi}$ match at round zero.

We now prove that Eq. (H.36) holds. By definition of $\tilde{f}_1$, we have

$$\tilde{f}_1(\mathbf{y}_{0:1}) = \widehat{\Sigma}_w \widehat{\Sigma}_{\mathrm{cov}}^{-1} \tilde{h}_{\mathrm{ol},0}(\mathbf{y}_0) + \hat{h}_0(\mathbf{y}_1) - \widehat{A}\hat{h}_0(\mathbf{y}_0), \tag{I.72}$$

and so since $\tilde{h}_{\mathrm{ol},0}$ and $\hat{h}_0$ are in $\mathscr{H}_{\mathrm{op}}$, we have by Jensen's inequality and Cauchy-Schwarz,

$$\|\tilde{f}_1(\mathbf{y}_{0:1})\|^2 \le 4\|\widehat{\Sigma}_w\|_{\mathrm{op}}^2 \|\widehat{\Sigma}_{\mathrm{cov}}^{-1}\|_{\mathrm{op}}^2 L_{\mathrm{op}}^2 (1 + \|\mathbf{x}_0\|^2)$$
$$+ 4L_{\mathrm{op}}^2(1 + \|\mathbf{x}_1\|^2) + 4\|A\|_{\mathrm{op}}^2(1 + \|\mathbf{x}_0\|^2) + 4\|A - \widehat{A}\|_{\mathrm{op}}^2(1 + \|\mathbf{x}_0\|^2). \tag{I.73}$$

Under $\mathcal{E}_{\mathrm{sys}}$ we have $\|\widehat{\Sigma}_w\|_{\mathrm{op}} \le 2\|\Sigma_w\|$ and $\|\widehat{A} - A\|_{\mathrm{op}} \le 1$, and the event of Lemma H.5 implies that $\|\widehat{\Sigma}_{\mathrm{cov}}^{-1}\|_{\mathrm{op}} \le 2\|\Sigma_{\mathrm{cov}}^{-1}\|_{\mathrm{op}}$. Hence, using that $L_{\mathrm{op}} \ge 1$, we can further upper bound by

$$\|\tilde{f}_1(\mathbf{y}_{0:1})\|^2 \le L_{\mathrm{op}}^2(64\|\Sigma_w\|_{\mathrm{op}}^2 \|\Sigma_{\mathrm{cov}}^{-1}\|_{\mathrm{op}}^2 + 4\|A\|_{\mathrm{op}}^2 + 4)(1 + \|\mathbf{x}_0\|^2) + 4L_{\mathrm{op}}^2(1 + \|\mathbf{x}_1\|^2).$$

Next, we note that $\|\Sigma_w\|_{\mathrm{op}} \le \Psi_\star$ and $\|\Sigma_{\mathrm{cov}}^{-1}\|_{\mathrm{op}} \le 3\Psi_\star^5$. Hence, we can further simplify this bound to

$$584 L_{\mathrm{op}}^2 \Psi_\star^{12}(2 + \|\mathbf{x}_0\|^2 + \|\mathbf{x}_1\|^2).$$

Since $\mathbf{x}_0 \sim \mathcal{N}(0, \Sigma_0)$ and $\mathbf{x}_1 \sim \mathcal{N}(0, \Sigma_1)$, we have, by Lemma I.11,

$$\mathbb{P}_{\widehat{\pi}}\left[\|\tilde{f}_1(\mathbf{y}_{0:1})\|^2 \ge 584 L_{\mathrm{op}} \Psi_\star^{12}(2 + (3d_{\mathbf{x}} + 2)(\|\Sigma_0\|_{\mathrm{op}} + \|\Sigma_1\|_{\mathrm{op}})) \ln(2\eta)\right] \le \eta^{-1},$$

so that in particular, we may take

$$\bar{b}_0^2 := 10^4 d_{\mathbf{x}} L_{\mathrm{op}}^2 \Psi_\star^{12}(1 + \|\Sigma_0\|_{\mathrm{op}} + \|\Sigma_1\|_{\mathrm{op}}). \tag{I.74}$$

This establishes Eq. (H.36). $\qquad\square$

## I.11 Supporting Results

**Lemma I.9.** Let $\boldsymbol{z} \sim \mathcal{N}(0, \Sigma)$, where $\Sigma \in \mathbb{R}^{m \times m}$ is a positive definite matrix. Then $\mathbb{E}[\|\boldsymbol{z}\|^2] \le m\|\Sigma\|_{\mathrm{op}}$.

*Proof.* Let $\boldsymbol{z}' := \Sigma^{-1/2}\boldsymbol{z}$ and note that $\boldsymbol{z}' \sim \mathcal{N}(0, I_m)$, and so $\|\boldsymbol{z}'\|^2 \sim \chi^2(m)$. As a result, we have

$$\mathbb{E}[\|\boldsymbol{z}\|^2] = \mathbb{E}[\|\Sigma^{1/2}\boldsymbol{z}'\|^2] \overset{(*)}{\le} \|\Sigma\|_{\mathrm{op}} \mathbb{E}[\|\boldsymbol{z}'\|^2] = m\|\Sigma\|_{\mathrm{op}},$$

where $(*)$ follows by Cauchy-Schwarz. $\qquad\square$

**Lemma I.10.** Let $a_0 > 0$ and $(a_1, c_1, z_1), \ldots, (a_s, c_s, z_s) \subset \mathbb{R}^3_{>0}$, where $(z_i)$ are (potentially dependent) non-negative random variables satisfying $\mathbb{P}[z_i \ge c_i \ln \delta^{-1}] \le \delta$, for all $i \in [s]$ and $\delta \in (0, 1/e]$. Then, for all $\delta \in (0, 1/e]$ we have

$$\delta \ge \mathbb{P}\left[a_0 + a_1 z_1 + \cdots + a_s z_s \ge \left(a_0 + \sum_{i=1}^s a_i c_i\right) \ln(s/\delta)\right],$$
$$\ge \mathbb{P}\left[a_0 + a_1 z_1 + \cdots + a_s z_s \ge (\ln s + 1)\left(a_0 + \sum_{i=1}^s a_i c_i\right) \ln \delta^{-1}\right]. \tag{I.75}$$

*Proof.* Let $c(\delta) := \left(a_0 + \sum_{i=1}^s a_i c_i\right) \ln \delta^{-1}$. Define $z_0 = c_0 = 1$. Since $\delta \in (0, e^{-1}]$, we have

$$\mathbb{P}[a_0 + a_1 z_1 + \cdots + a_s z_s \ge c(\delta)] = \mathbb{P}\left[\sum_{i=0}^s a_i(z_i - c_i \ln \delta^{-1}) \ge 0\right],$$
$$\le \mathbb{P}\left[\exists i \in [s] : z_i - c_i \ln \delta^{-1} \ge 0\right],$$
$$\le \sum_{i=1}^s \mathbb{P}\left[z_i - c_i \ln \delta^{-1} \ge 0\right],$$
$$\le s\delta. \quad \text{(by assumption)} \tag{I.76}$$

For any given $\delta \le 1/e$, by applying this result with $\delta' \coloneqq \delta/s$, we have for all $\delta \in (0, 1/(se)]$, $\delta' \le 1/e$, and so

$$c(\delta') = \left(a_0 + \sum_{i=1}^s a_i c_i\right) \ln(s/\delta) \le (\ln s + 1)\left(a_0 + \sum_{i=1}^s a_i c_i\right) \ln(1/\delta).$$

This together with (I.76) implies (I.75). $\qquad\square$

**Lemma I.11.** Let $a_0 > 0$ and $(a_1, c_1, \boldsymbol{z}_1), \ldots, (a_s, c_s, \boldsymbol{z}_s)$ be such that $(a_i, c_i) \subset \mathbb{R}_{>0}^2$ and $\boldsymbol{z}_i \in \mathbb{R}^{d_i}$ are random vectors satisfying $\boldsymbol{z}_i \sim \mathcal{N}(0, \Sigma_i)$ for $i \in [s]$. Then, for all $\delta \in (0, 1/e]$ we have

$$\delta \ge \mathbb{P}\left[a_0 + a_1\|\boldsymbol{z}_1\|^2 + \cdots + a_s\|\boldsymbol{z}_s\|^2 \ge \left(a_0 + \sum_{i=1}^s a_i\|\Sigma_i\|_{\mathrm{op}} \cdot (3d_i + 2)\right) \ln(s/\delta)\right],$$

$$\ge \mathbb{P}\left[a_0 + a_1\|\boldsymbol{z}_1\|^2 + \cdots + a_s\|\boldsymbol{z}_s\|^2 \ge (\ln s + 1)\left(a_0 + \sum_{i=1}^s a_i\|\Sigma_i\|_{\mathrm{op}} \cdot (3d_i + 2)\right) \ln \delta^{-1}\right]. \quad \text{(I.77)}$$

*Proof.* For $i \in [s]$, let $\boldsymbol{z}_i' \coloneqq \Sigma_i^{-1/2}\boldsymbol{z}_i$; in this case, $\boldsymbol{z}_i' \sim \mathcal{N}(0, I_{d_i})$. Thus, by Lemma 1 of [21], we have that

$$\delta \ge \mathbb{P}\left[\|\boldsymbol{z}_i'\|^2 \ge d_i + 2\sqrt{d_i \ln \delta^{-1}} + 2\ln \delta^{-1}\right]$$

$$\ge \mathbb{P}\left[\|\boldsymbol{z}_i\|^2 \ge \|\Sigma_i\|_{\mathrm{op}}\left(d_i + 2\sqrt{d_i \ln \delta^{-1}} + 2\ln \delta^{-1}\right)\right]$$

$$\ge \mathbb{P}\left[\|\boldsymbol{z}_i\|^2 \ge \|\Sigma_i\|_{\mathrm{op}}\left(d_i + 2\sqrt{d_i \ln \delta^{-1}} + 2\ln \delta^{-1}\right)\right]$$

$$\ge \mathbb{P}\left[\|\boldsymbol{z}_i\|^2 \ge \|\Sigma_i\|_{\mathrm{op}} \cdot (3d_i + 2)\ln \delta^{-1}\right], \quad \text{(I.78)}$$

where the last inequality follows by the fact that $\delta \in (0, 1/e]$. By (I.78) and Lemma I.10, we get (I.77). $\qquad\square$

**Lemma I.12.** Let $\varepsilon > 0$, and $M, N \in \mathbb{R}^{m \times m}$ be given. Suppose $N$ is non-singular and $\|M - N\|_{\mathrm{op}} \le \varepsilon$. Then if $\varepsilon < \sigma_{\min}(N)/2$, $M$ is non-singular and

$$\|I_m - M^{-1}N\|_{\mathrm{op}} \le \frac{2\varepsilon}{\sigma_{\min}(N)}. \quad \text{(I.79)}$$

*Proof.* We first bound the minimum singular value of $M$. Let $x \in \mathbb{R}^m$ be a unit-norm vector such that $\|Mx\| = \sigma_{\min}(M)$. Then, from the fact that $\|M - N\|_{\mathrm{op}} \le \varepsilon$, we have,

$$\varepsilon \ge \|Mx - Nx\|,$$
$$\ge \|Nx\| - \|Mx\|, \quad \text{(by the triangle inequality)}$$
$$= \sigma_{\min}(N) - \sigma_{\min}(M). \quad \text{(using that } \|x\| = 1\text{)}$$

In particular, the last inequality implies that

$$\sigma_{\min}(M) \ge \sigma_{\min}(N) - \varepsilon. \quad \text{(I.80)}$$

Thus, since $\varepsilon < \sigma_{\min}(N)$, the matrix $M$ is invertible. On the other hand, we have

$$\varepsilon \ge \|M - N\|_{\mathrm{op}},$$
$$\ge \sigma_{\min}(M) \cdot \|I_m - (M)^{-1}N\|_{\mathrm{op}},$$
$$\ge (\sigma_{\min}(N) - \varepsilon)\|I_m - M^{-1}N\|_{\mathrm{op}}. \quad \text{(by (I.80))}$$

The desired result follows by the fact that $\varepsilon < \sigma_{\min}(N)/2$. $\qquad\square$

## J  Main Theorem and Proof

We now state and prove the main guarantee for RichID-CE (Algorithm 1). To begin, we state the values for the algorithm's parameters $n_{\mathrm{id}}$ and $n_{\mathrm{op}}$:

$$n_{\mathrm{id}} = \Omega_\star\left(\lambda_{\mathcal{M}}^{-2} T^3 \kappa^5 (d_{\mathbf{x}} + d_{\mathbf{u}})^{16} \ln^{15}(1/\delta) \cdot \frac{\ln|\mathscr{F}|}{\varepsilon^6}\right), \quad \text{(J.1)}$$

$$n_{\mathrm{op}} = \Omega_\star\left(\lambda_{\mathcal{M}}^{-2} T^3 \kappa^3 (d_{\mathbf{x}} + d_{\mathbf{u}})^{12} \ln^{11}(1/\delta) \cdot \frac{\ln|\mathscr{F}|}{\varepsilon^6}\right). \quad \text{(J.2)}$$

We also recall from Section 2.1 that for the burn-in time, we use the choice

$$\kappa_0 := \left\lceil (1 - \gamma_\star)^{-1} \ln\left(84 \Psi_\star^5 \alpha_\star^4 d_\mathbf{x} (1 - \gamma_\star)^{-2} \ln(10^3 \cdot n_{\mathrm{id}})\right) \right\rceil.$$

Finally, we set $r_{\mathrm{id}} = \sqrt{\Psi_\star}$, and set $r_{\mathrm{op}} = \Omega_\star(1)$ to be a sufficiently large problem-dependent constant. The values for $\sigma^2$ and $\bar{b}$ are given in the following theorem.

**Theorem 1a.** *Let $\delta \in (0, 1/e]$ and $\varepsilon = \check{\mathcal{O}}(1)$ be given. Suppose we set $\bar{b}^2 = \Theta_\star((d_\mathbf{x} + d_\mathbf{u}) \ln(1/\delta))$, $\sigma^2 = \check{\mathcal{O}}(\varepsilon^2/\bar{b}^2 \wedge \lambda_\mathcal{M})$, and choose $n_{\mathrm{id}}$ and $n_{\mathrm{op}}$ as in (J.1) and (J.2). Then with probability at least $1 - \mathcal{O}(\kappa T \cdot \delta)$, Algorithm 1 produces a policy $\widehat{\pi}$ with*

$$J_T(\widehat{\pi}) - J_T(\pi_\infty) \le \varepsilon, \tag{J.3}$$

*and does so while using at most*

$$\mathcal{O}_\star\left( \lambda_\mathcal{M}^{-2} T^4 \kappa^5 (d_\mathbf{x} + d_\mathbf{u})^{16} \ln^{15}(1/\delta) \cdot \frac{\ln|\mathscr{F}|}{\varepsilon^6} \right)$$

*trajectories of length $\mathcal{O}_\star(T)$.*

## J.1  Proof of Theorem 1a

*Proof of Theorem 1a.* We first restate Theorem 3, which bounds the estimation error for the system parameter estimates produced by Phase II of Algorithm 1.

**Theorem 3** (Guarantee for Phase II). *If $n_{\mathrm{id}} = \Omega_\star\left( d_\mathbf{x}^2 d_\mathbf{u} \kappa (\ln|\mathscr{F}| + d_\mathbf{u} d_\mathbf{x} \kappa) \max\{1, \sigma_{\min}(\mathcal{C}_\kappa)^{-4}\} \right)$, then with probability at least $1 - 11\delta$ over Phases I and II,*

$$\|[\widehat{A}_{\mathrm{id}}; \widehat{B}_{\mathrm{id}}] - [A_{\mathrm{id}}; B_{\mathrm{id}}]\|_{\mathrm{op}} \vee \|\widehat{Q}_{\mathrm{id}} - Q_{\mathrm{id}}\|_{\mathrm{op}} \vee \|\widehat{\Sigma}_{w,\mathrm{id}} - \Sigma_{w,\mathrm{id}}\|_{\mathrm{op}} \le \varepsilon_{\mathrm{id}}, \tag{12}$$

*where $\varepsilon_{\mathrm{id}} \le \mathcal{O}_\star\left( n_{\mathrm{id}}^{-1/2} \ln^2(n_{\mathrm{id}}/\delta) \sqrt{d_\mathbf{x} d_\mathbf{u} \kappa (\ln|\mathscr{F}| + d_\mathbf{u} d_\mathbf{x} \kappa)} \right).$*

Going forward we condition on the event in Theorem 3, and define $f_{\star,\mathrm{id}} := S_{\mathrm{id}} f_{\star,\mathrm{id}}$ and $K_{\infty,\mathrm{id}} := K_\infty S_{\mathrm{id}}^{-1}$. We recall that whenever this event holds, we have

$$\|S_{\mathrm{id}}\|_{\mathrm{op}} \vee \|S_{\mathrm{id}}^{-1}\|_{\mathrm{op}} \le \Psi_\star^{1/2} \vee (1 - \gamma_\star)^{-1} (4 \Psi_\star^2 \alpha_\star^2) \sigma_{\min}^{-1}(\mathcal{C}_\kappa) = \mathcal{O}_\star(1),$$

as per Theorem 2. As a consequence, we have the following fact, which we will use heavily going forward: If we define $(\Psi_\star', \alpha_\star', \gamma_\star', \kappa_\star', L')$ to be the analogues of $(\Psi_\star, \alpha_\star, \gamma_\star, \kappa_\star, L)$ for $(A_{\mathrm{id}}, B_{\mathrm{id}}, Q_{\mathrm{id}}, R, \Sigma_{w,\mathrm{id}}, f_{\star,\mathrm{id}})$, we have $\Psi_\star' = \mathcal{O}_\star(\Psi_\star)$ and $L' = \mathcal{O}_\star(L)$, and we may take $\alpha_\star' = \mathcal{O}_\star(\alpha_\star)$, $\gamma_\star' \le \gamma_\star$, and $\kappa_\star' \le \kappa_\star$.

We first apply Lemma H.1, which implies that once $\varepsilon_{\mathrm{id}} = \check{\mathcal{O}}(1)$, we have

$$\|\widehat{K} - K_{\infty,\mathrm{id}}\|_{\mathrm{op}} = \mathcal{O}_\star(\varepsilon_{\mathrm{id}}). \tag{J.4}$$

Next, we invoke Theorem 4, associating $A \leftarrow A_{\mathrm{id}}$, $B \leftarrow B_{\mathrm{id}}$, $Q \leftarrow Q_{\mathrm{id}}$, $\Sigma_w \leftarrow \Sigma_{w,\mathrm{id}}$, $f_\star \leftarrow f_{\star,\mathrm{id}}$, and inflating the problem-dependent parameters by $\mathcal{O}_\star(1)$ accordingly. In particular, suppose that $\varepsilon_{\mathrm{id}}^2 \le \check{\mathcal{O}}((\ln|\mathscr{F}| + d_\mathbf{x}^2) n_{\mathrm{op}}^{-1})$ for a problem-dependent constant $c_{\mathrm{id}}' = \check{\mathcal{O}}(1)$, and suppose we set $\bar{b}^2 = \Theta_\star((d_\mathbf{x} + d_\mathbf{u}) \ln(n_{\mathrm{op}}))$ and $\sigma^2 = \check{\mathcal{O}}(\lambda_\mathcal{M}') = \check{\mathcal{O}}(\lambda_\mathcal{M})$. Then conditioned on the event of Theorem 3, we are guaranteed that for any $\delta \in (0, 1/e]$, with probability at least $1 - \mathcal{O}(\kappa T \delta)$,

$$\varepsilon_{\mathrm{op}}^2 := \mathbb{E}_{\widehat{\pi}}\left[ \max_{1 \le t \le T} \|\hat{f}_t(\mathbf{y}_{0:t}) - f_{\star,\mathrm{id}}(\mathbf{y}_t)\|_2^2 \right] \le \mathcal{O}_\star\left( \frac{\lambda_\mathcal{M}'^{-2}}{\sigma^4} \cdot T^3 \kappa^2 (d_\mathbf{x} + d_\mathbf{u})^4 \cdot \frac{(d_\mathbf{x}^2 + \ln|\mathscr{F}|) \ln^5(n_{\mathrm{op}}/\delta)}{n_{\mathrm{op}}} \right).$$

where $\lambda_\mathcal{M}'$ is the analogue of $\lambda_\mathcal{M}$ for the parameters $(A_{\mathrm{id}}, B_{\mathrm{id}}, \Sigma_{w,\mathrm{id}})$; note that to apply the theorem, we must set the radius of the class $\mathscr{H}_{\mathrm{op}}$ based on $\Psi_\star'$ rather than $\Psi_\star$, which leads to the value for this parameter passed into Algorithm 5 when it is invoked within Algorithm 1. Likewise, we must inflate $\bar{b}$ by $\Omega_\star(1)$. Lastly, we note that $\lambda_\mathcal{M}' \ge \Omega_\star(\lambda_\mathcal{M})$; which can be quickly verified.

Taking a union bound and simplifying the upper bounds slightly, we are guaranteed that with probability at least $1 - \mathcal{O}_\star(\kappa T \delta)$,

$$\varepsilon_{\mathrm{id}}^2 \le \kappa^2 (d_\mathbf{x} + d_\mathbf{u})^4 \ln^4(n_{\mathrm{id}}/\delta) \frac{\ln|\mathscr{F}|}{n_{\mathrm{id}}}$$

$$\varepsilon_{\mathrm{op}}^2 \le \mathcal{O}_\star\left(\frac{\lambda_\mathcal{M}^{-2}}{\sigma^4} T^3 \kappa^3 (d_\mathbf{x} + d_\mathbf{u})^6 \ln^5(n_{\mathrm{op}}/\delta) \cdot \frac{\ln|\mathscr{F}|}{n_{\mathrm{op}}}\right),$$

so long as the conditions on $n_{\mathrm{id}}$, $\varepsilon_{\mathrm{id}}$, $\bar{b}$, and $\sigma^2$ described so far hold. We next invoke Theorem 7 with $\tau = T$ (again, we use that changing the basis by $S_{\mathrm{id}}$ inflates problem-dependent constants by $\mathcal{O}_\star(1)$),[12] which implies that

$$J_T(\widehat{\pi}) - J_T(\pi_\infty) \le \mathcal{O}_\star(\bar{b} \cdot c_\mathbf{x} \cdot (c_\mathbf{x} \cdot \varepsilon_{\mathrm{id}} + \varepsilon_{\mathrm{op}} + \sigma)),$$

where $c_\mathbf{x}^2 = \max_{1 \le t \le T} \mathbb{E}_{\widehat{\pi}} \|\mathbf{x}_t\|^2$, so long as $\sigma^2 = \mathcal{O}_\star(1)$. From Lemma I.1, we have $c_\mathbf{x}^2 \le \mathcal{O}_\star(\bar{b}^2 + d_\mathbf{x}) = \mathcal{O}_\star(\bar{b}^2)$, so we may further simplify to

$$J_T(\widehat{\pi}) - J_T(\pi_\infty) \le \mathcal{O}_\star(\bar{b}^2 \cdot (\bar{b} \cdot \varepsilon_{\mathrm{id}} + \varepsilon_{\mathrm{op}} + \sigma)).$$

Hence, to ensure the regret is at most $\mathcal{O}_\star(\varepsilon)$, as a first step we choose $\sigma = \breve{\mathcal{O}}(\varepsilon/\bar{b}^2)$. This leads to

$$\varepsilon_{\mathrm{op}}^2 \le \mathcal{O}_\star\left(\lambda_\mathcal{M}^{-2} T^3 \kappa^3 (d_\mathbf{x} + d_\mathbf{u})^{10} \ln^9(1/\delta) \cdot \frac{\ln|\mathscr{F}|}{n_{\mathrm{op}}} \cdot \frac{1}{\varepsilon^4}\right).$$

We next choose $\varepsilon_{\mathrm{op}}^2 = \breve{\mathcal{O}}(\varepsilon^2/\bar{b}^4)$ which, per the inequality above, entails setting

$$n_{\mathrm{op}} = \Omega_\star\left(\lambda_\mathcal{M}^{-2} \kappa^3 T^3 (d_\mathbf{x} + d_\mathbf{u})^{12} \ln^{11}(1/\delta) \cdot \frac{\ln|\mathscr{F}|}{\varepsilon^6}\right).$$

Finally, we require that $\varepsilon_{\mathrm{id}} \le \breve{\mathcal{O}}(\varepsilon/\bar{b}^3)$, and we also require $\varepsilon_{\mathrm{id}}$ to satisfy the earlier constraint that $\varepsilon_{\mathrm{id}}^2 \le \breve{\mathcal{O}}((\ln|\mathscr{F}| + d_\mathbf{x}^2) n_{\mathrm{op}}^{-1})$. To satisfy the first constraint, it suffices to take

$$n_{\mathrm{id}} = \Omega_\star\left(\kappa^2 (d_\mathbf{x} + d_\mathbf{u})^6 \ln^6(1/\delta) \frac{\ln|\mathscr{F}|}{\varepsilon^2}\right).$$

For the second constraint, it suffices to take

$$n_{\mathrm{id}} = \Omega_\star\left(n_{\mathrm{op}} \cdot \kappa^2 (d_\mathbf{x} + d_\mathbf{u})^4 \ln^4(/\delta)\right)$$

$$= \Omega_\star\left(\lambda_\mathcal{M}^{-2} T^3 \kappa^5 (d_\mathbf{x} + d_\mathbf{u})^{16} \ln^{15}(1/\delta) \cdot \frac{\ln|\mathscr{F}|}{\varepsilon^6}\right).$$

Lastly, we observe that the algorithm uses $\mathcal{O}(n_{\mathrm{op}} \cdot T + n_{\mathrm{id}})$ trajectories in total, leading to the final calculation in the theorem statement. $\qquad\square$