[Reviews · NeurIPS 2020]

Review 1

Summary and Contributions: This paper provides a new algorithm, which learns a close to optimal policy where the sample complexity scales only with the dimension of the latent state space and the capacity of the decoder function class. The algorithm is oracle-efficient and accesses the decoder class only through calls to a least-squares regression oracle. The authors claim that this work provides the first provable sample complexity guarantee for continuous control with an unknown nonlinearity in the system model.

Strengths: The paper is very well written and clearly formulates motivation and contributions of the presented work. The technical contributions are sophisticated and well organized using appendices. The problem formulations are worthy and relevant. I also find them to be of interest to the general NIPS community. Furthermore and in my opinion, this work has a very good potential in contributing to the area of perception based control.

Weaknesses: There are some limitations due to linearity in the modeling but even this is well recognized and justified by the authors.

Correctness: Given the length of the appendices and the time I could spend on checking them, I would say it all looks fine.

Clarity: Yes, definitely well written and organized.

Relation to Prior Work: Yes.

Reproducibility: Yes

Additional Feedback: None.


Review 2

Summary and Contributions: The topic of this paper is adaptive linear quadratic (LQ) control in a setting where the dynamics are x’ = Ax + Bu + w, costs are quadratic c=x^TQx + u^TRu, and we only observe y = q(x), a high-dimensional non-linear function of the state. The authors assume that noise w is Gaussian, (A, B) controllable, A stable. They also assume a known function class F of decoders x=f(y) and decoding realizability (F contains the true decoder f* and f* is perfect). To justify the last assumption, their Theorem 5 shows that in the absence of realizability, when y is a noisy version of f*^{-1}(x), learning the decoder has exponential sample complexity. In the first phase, the algorithm learns a coarse decoder up to a linear transformation. The key insight is that in each trajectory, the last state x is a linear function of all previous inputs. Therefore, to find a decoding x, they train a regressor on (y_{H+1}, [u_1, …, u_H]) pairs, and reduce the output dimension to the dimension of x using PCA. The inputs u_h for this phase are Gaussian. In the second phase, they estimate dynamics (A, B) and cost parameters in the predicted decoded space. They compute the certainty-equivalent controller corresponding to this estimate. In the third phase, they improve the decoder by training on near-optimal trajectories, generated using a noisy version of the Phase 2 policy. Rather than having a single decoder, the algorithm now relies on a sequence of decoders f_t (one per timestep), where f_{t+1} is formed from f_t and h_t, and h_t is a residual decoder satisfying h_t(y_{t+1}) - Ah_t(y_t) = x_{t+1} - Ax_t. The main theorem (Thm 1) states that under appropriate assumptions, the algorithm learns an eps-optimal policy for horizon T using O((dx + du)^17 T^3 log|F| / eps^6) trajectories. The main contribution is a new algorithm and analysis of LQ control with hidden states and high-dimensional observations, which are a nonlinear function of the state.

Strengths: The paper applies a rigorous approach to adaptive control in systems with latent linearly-evolving states, and high dimensional observations which are a nonlinear function of the state. This is an important topic with applications in e.g. perception-based control. Most existing work in this setting is somewhat heuristic (other than LQG control).

Weaknesses: There are no experiments / numerical simulations of the algorithm. It would be more convincing if demonstrated even in a synthetic realizable setting. Some of the assumptions in the paper are quite strong (stability, realizability) and it is difficult to assess the extent to which the algorithm is practical. The decoding procedure in Phase 3 is quite elaborate. Assuming that the simple regression-based Phase 1 decoder is good enough to find a near-optimal controller, it is unclear why a similar procedure does not suffice on noisy near-optimal trajectories, and why data from the two phases cannot be combined. It would be useful to clarify this and/or demonstrate any problematic aspects in simulation.

Correctness: The claims seem correct, though I have not fully checked (and the 80-page appendix is a bit overwhelming). There are no experiments.

Clarity: Section 2.3 is difficult to parse and could be organized better. Otherwise, the main paper is well-written.

Relation to Prior Work: The discussion of related work seems adequate.

Reproducibility: Yes

Additional Feedback: In Algorithm 4 line 28, why is noise added to the optimal policy that is returned to the user? Most practical systems are only locally linear (and this is also the setting modeled by iLQR and embed-to-control approaches). How difficult is it to extend this algorithm to the locally-linear setting? Thanks to the authors for the rebuttal and the experiment. After reading the other reviews, my overall opinion is the same. I would not be opposed to acceptance if authors include more simulations in the final version.


Review 3

Summary and Contributions: This paper provides an algorithm for performing control in a high-dimensional observation space based on LQR. The authors provide a sample-complexity bound for learning a “decoder” into a latent space where LQR can succeed.

Strengths: The problem setting (performing LQR in a latent space with nonlinear observations) is generally difficult so this paper’s perhaps greatest strength is its significance. Given the wide variety of methods employed in model-based reinforcement learning, it is good to have a straightforward presentation of an algorithm that has guarantees, especially in a powerful model like a latent LQR. The algorithm has a sample complexity polynomial in the dimensionality of the latent space and control space. The argument is generally well structured but as someone not very familiar with the surrounding literature, I cannot comment on the novelty of the result.

Weaknesses: The paper lacks any empirical evaluation. Although the theoretical argument is strong, even a simple synthetic experiment would go a long way in strengthening its case (perhaps evaluating on a 2-d pointmass system with image observations). With such an experiment this paper merits a higher score in my opinion. As for space constraints, Phase III could be tightened. For example, the noninvertible case could be explained with a sketch and the details relegated to the supplement. ------------------------------------------------------------------------------------------- In the rebuttal, the authors present the additional empirical experiment with convincing enough results. I've updated my score and I advocate this paper's acceptance due to the difficulty of the problem setting and the paper's rigorous results. It is a clear starting point for research into other interesting control settings.

Correctness: I skimmed the proofs in the supplement and found no obvious mistakes. The argument in the paper looks correct.

Clarity: The writing and presentation of the problem setting and algorithm is clear and compelling. The authors divide the algorithm into three phases, each of which needs some number of samples and each phase clearly outlines its goals and highlights its achieved result. Some minor comments: The paper uses “decoder” in place of what in traditional autoencoding literature is called the “encoder” (an encoder usually maps a high dimensional state to a low dimensional code where the f function in this paper maps y to x). The paper would be more clear to people familiar with autoencoding literature if they swap the terms. It seems like the $h$ in 2.3 is different from the $h$ in 2.1. One should be changed to make that explicitly clear.

Relation to Prior Work: The paper does a sufficient job of drawing comparisons to prior work. However, in the abstract when it is stated “We introduce a new problem setting for continuous control called the LQR with 2 Rich Observations, or RichLQR”, it is certainly not the case that this paper is introducing this as a novel problem. It has been extensively studied in the E2C line of work and related papers.

Reproducibility: Yes

Additional Feedback: Minor writing comments: 230, 235, 240, 242, 257: gaussian -> Gaussian 213: propogate -> propagate


Review 4

Summary and Contributions: This paper suggests a novel problem setting, RichLQR, whose observation is high, complex information while the environment is in a low-dimensional state and described with a linear equation. Based on the perfect decodability assumption, the authors suggest RichID-CE solve the problem. The decoder is realized with a trainable neural network and is iteratively updated. The proposed algorithm is the first to achieve polynomial sample complexity.

Strengths: RichLQR problem is expected to be practical and useful. Adopting a neural network as a decoder is a novel idea and a reasonable choice. Phase 1 -> 2 -> 3 of RichID-CE makes sense. Especially, iteratively refining the decoder to follow a near-optimal trajectory in phase 3 is a novel approach.

Weaknesses: I am not much familiar with control theory and related works, so I will mainly focus on questions for the neural network decoder. In Theorem 1, RichID-CE achieves polynomial-in-dimension sample complexity with T^3 log|F|. However, if F is a family of neural networks, isn’t the search space (capacity) |F| increases exponentially? (much faster than T?) During phase 3, is the sequence of decoders f_t has any convergence promise? Can you give more detailed explanation of how parameterized neural network decoder f_t+1 updated for each parameter?

Correctness: I cannot say whether the mathematical derivations are correct.

Clarity: The paper is well organized and describes each part in detail.

Relation to Prior Work: This work directly expands the rich observation case, and related works are discussed, mentioned, and compared.

Reproducibility: Yes

Additional Feedback: The simplest decoder network would be a deep convolutional neural network (CNN). An example of the network architecture, training objective, and update rule during phase 1, 2, and 3 would help much. ---- I've read the rebuttal and updated my score.

[Author Response · NeurIPS 2020]

We thank all reviewers for their detailed feedback. We will be sure to address all questions and incorporate all
suggestions in the final version of the paper. Please see individual responses below.

**Reviewer #1.** Thank you for your positive comments!

**Reviewer #2.** *"The decoding procedure in Phase 3 is quite elaborate..."*. The regression procedures in phases 1 and
2 only allow us to approximate $K_\infty$, representing the solution of the Riccati equations for $(A, B, Q)$. This does not
immediately allow us to approximate the optimal controller, which is a function of the latent state $\mathbf{x}_t$ and is given by
$\pi_\star(\mathbf{y}_t) = K_\infty \mathbf{x}_t$. To approximate $\pi_\star(\mathbf{y}_t)$, we need to learn a decoder $\hat{f}_t(\mathbf{y}_t) \approx \mathbf{x}_t$. The decoder learned during the first
phase is only guaranteed to be accurate on the state distribution generated by taking random actions from the start state.
There is no guarantee that this decoder will be accurate on the state distribution induced by a near-optimal policy, so we
need to learn a new decoder at each step with new data generated using $\widehat{\pi}$. We emphasize that this is simply an instance
of a common technical issue in statistical learning; In general, given a function $\hat{f}$ such that $\mathbb{E}_P \|\hat{f}(x) - f^\star(x)\|^2 \leq \varepsilon$
for a distribution $P$, we have no guarantee that $\mathbb{E}_Q \|\hat{f}(x) - f^\star(x)\|^2 \leq \varepsilon$ for a different distribution $Q$ unless we put
strong structural assumptions on either $P/Q$ or the function class $\mathcal{F}$. Since we do not make such assumptions, we solve
this problem by re-learning on the new distribution.

"*In Algorithm 4 line 28, why is noise added to the optimal policy...?*" This is closely related to the point above: Since
we train on a distribution in which random noise is injected, our decoders are only guaranteed to have low error on this
distribution. However, since the noise decays with $O(\varepsilon)$, the resulting controller is still $\varepsilon$-suboptimal.

"*Most practical systems are only locally linear... How difficult is it to extend this algorithm to the locally-linear
setting?*" Extending our algorithm to the locally linear setting is a very exciting direction for future research, but we are
not yet aware of sample complexity guarantees for locally linear control even when the state is fully observed, let alone
for the more challenging nonlinear-observation setup we consider.

**Reviewer #3.** *"The paper lacks any empirical evaluation...With such an experiment this paper
merits a higher score in my opinion"* We believe that our paper represents a substantial theoretical
contribution and stands on its own merits even without experiments. Nonetheless, we have
performed some basic validation experiments, which we can include in the appendix, focusing in
Phase 1 and 2 of the algorithm for simplicity. We considered a 2-d Newtonian dynamical system
with unit process noise, where $A \in \mathbb{R}^{4\times4}$ upper triangular matrix and $B \in \mathbb{R}^{4\times2}$. We slightly
dampened the dynamics to ensure that $\rho(A) < 1$. The final system is 2-controllable. Observations
come in pairs of images, one for position information and one for velocity information. Each image
contains one green pixel representing either a vector of position or velocity. Greyscale noise is

Figure 1: Green
dot = position or
velocity vector.

added to the rest of the pixels (see Figure 1). We model the function $h$ using a 3-layer convolutional
neural network with Leaky ReLu nonlinearity. After executing phases 1 and 2 of our algorithm, we
successfully recover the systems' dynamics, i.e. the matrices $A$ and $B$, up to a similarity transform.
In our preliminary experiments, using $n_{\mathrm{id}} = 30000$, we can recover the system matrices up to element-wise absolute
error of $< 0.07$.

**Reviewer #4.** *"The paper uses "decoder" in place of what in traditional autoencoding..."* We will add a note on
this to avoid any confusion. *"...it is certainly not the case that this paper is introducing this as a novel problem..."*
Our main claim is that we introduce the theoretical problem of developing *finite-sample bounds* for this setting, which
we believe is true. We will update the abstract to be more precise.

*"... if $F$ is a family of neural networks, isn't the search space (capacity) $|F|$ increases exponentially? (much faster
than $T$?)"* Our sample complexity depends *logarithmically* in $|\mathcal{F}|$ and polynomially in $T$, and so even if $|\mathcal{F}|$ were to
grow exponentially in $T$, this would not be an issue—the sample complexity would remain polynomial in $T$. More
broadly, we emphasize that the $\log|\mathcal{F}|$ factor in our theorem arises from a standard generalization bound for the square
loss, and can trivially be replaced by more standard learning-theoretic quantities such as the Rademacher complexity or
covering numbers of $\mathcal{F}$ (indeed, if you look at the appendix, our intermediate results are already in terms of covering
numbers). In particular, this means that we can appeal to modern Rademacher complexity bounds for deep neural
networks, such as Bartlett et al. (2017) or Golowich et al. (2018). We will make this clear in the final version.

*"During phase 3, is the sequence of decoders $f_t$ has any convergence promise? Can you give more detailed expla-
nation of how parameterized neural network decoder updated for each parameter?"* The objective function used to
learn the (parameterized neural network) decoder $\hat{f}_t$ is an average square-loss see (20) and (21). Any out-of-the-box
neural network architecture / training algorithm (e.g., SGD) can be used to minimize the objective. As long as the
objective is approximately minimized, Theorem 4 ensures that the decoders $\hat{f}_t$ will have low decoding error, and thus
lead to an approximately optimal controller. The sequence $(\hat{f}_t)$ is not guaranteed to converge to a fixed decoder, but this
is not required for our theoretical guarantees to hold.

[Meta-Review · NeurIPS 2020]

The authors were unhappy about one review, but it was very low-confidence anyway. Except for this one, the reviews were positive, and the author feedback increased their confidence (and at least one score). Reviewers appreciated the rigorous approach to an important problem, establishing finite-sample bounds. One issue pointed out by two reviewers, namely the lack of any basic validation experiments, was addressed in the author feedback.